



# Sensitivity of the Variability of Mineral Aerosol Simulations to Meteorological Forcing Datasets

Molly B. Smith[1,2], Natalie M. Mahowald[1], Samuel Albani[1,3], Aaron Perry[1], Remi Losno[4], Zihan Qu[4,5],

Beatrice Marticorena[5], David A. Ridley[6], Colette L. Heald[6]

[1] Department of Earth and Atmospheric Sciences, Cornell University, Ithaca, NY 14850, USA.
[2] Department of Atmospheric and Environmental Sciences, University at Albany, State University of New York, Albany, NY 12222, USA.
[3] Institute for Geophysics and Meteorology, University of Cologne, Cologne, Germany.
[4] Institut de Physique du Globe de Paris, University of Paris Diderot, USPC, UMR CNRS 7154, Paris France.
[5] LISA, Universites Paris Est-Paris Diderot-Paris 7, UMR CNRS 7583, Creteil, France.
[6] Department of Civil and Environmental Engineering, Massachusetts Institute of Technology, MA 02139, USA.

*Correspondence to*: Molly B. Smith (mbs244@cornell.edu)

**Abstract.** Interannual variability in desert dust is widely observed and simulated, yet the sensitivity of these desert dust simulations to a particular meteorological dataset, as well as a particular model construction, is not well known. Here we use version 4 of the Community Atmospheric Model (CAM4) with the Community Earth System Model (CESM) to simulate dust forced by three different reanalysis meteorological datasets for the period 1990-2005. We then contrast the results of these simulations with dust simulated using online winds dynamically generated from sea surface temperatures, as well as with simulations conducted using other modeling frameworks but the same meteorological forcings, in order to determine the sensitivity of climate model output to the specific reanalysis dataset used. For the seven cases considered in our study, the different model configurations are able to simulate the annual mean of the global dust cycle, seasonality and interannual variability approximately equally well at the limited observational sites available. Overall, aerosol dust source strength has remained fairly constant during the time period from 1990 to 2005, although there is strong seasonal and some interannual variability simulated in the models and seen in the observations over this time period. The models predict similar seasonal variability and timing as one another, and obtain the best comparison to observations at the seasonal scale. Interannual variability comparisons to observations, as well as comparisons between models, suggest that interannual variability in dust is still difficult to simulate accurately, with averaged correlation coefficients of 0.1 to 0.6. Because of the large variability, at least one year of observations at most sites are needed to correctly observe the mean, but in some regions, particularly the remote oceans of the Southern Hemisphere, where interannual variability appears to be larger, 2-3 years of data are needed.

## 1. Introduction

Mineral aerosols, or desert dust, are soil particles suspended in the atmosphere, and are intimately connected with



many earth system processes (e.g. Shao et al., 2011). Mineral aerosols both scatter and absorb incoming solar radiation and outgoing long wave radiation (Tegen and Lacis, 1996;Sokolik and Toon, 1996;Miller and Tegen, 1998a;Dufresne et al., 2002) and thus impact directly the radiative budget of the Earth (e.g. Balkanski et al., 2007;Reddy et al., 2005;Myhre et al., 2013).  In addition, dust can interact with water and ice clouds, indirectly impacting climate by changing cloud properties or

lifetimes (e.g. Rosenfeld et al., 2001;DeMott et al., 2003;Mahowald and Kiehl, 2003;Atkinson et al., 2013;Cziczo et al., 2013).  Deposited desert dust can impact snow albedo (e.g. Painter et al., 2007), as well as provide important micronutrients to land and ocean ecosystems (e.g. Swap et al., 1992;Okin et al., 2004;Jickells et al., 2005).  In addition, dust is widely variable in space and time, with 4x fluctuations regionally on decadal time scales (Prospero and Lamb, 2003), and globally equally large fluctuations on glacial-interglacial time scales (Kohfeld and Harrison, 2003).  The importance of interannual

variability of dust in changing climate (Yoshioka et al., 2007;Evan et al., 2009;Mahowald et al., 2010) and biogeochemistry (Aumont et al., 2008;Doney et al., 2009) have previously been simulated and shown to potentially be large. Previous studies have shown some skill in simulations of the annual mean and seasonal cycle (Huneeus et al., 2011), and some studies have sought to consider the causes of interannual variability in dust, such as changes in precipitation or winds or land use (e.g. Tegen and Miller, 1998;Mahowald et al., 2002;Miller and Tegen, 1998b;Mahowald et al., 2003;Ginoux et al., 2004;Ridley et

al., 2014).   The question we address here, using primarily sensitivity studies, is how sensitive our simulation of interannual variability is to the meteorology, or alternatively, the modeling framework used.

While previous modeling studies have evaluated the annual mean and seasonal cycle coherence across dust models (Huneeus et al., 2011) or contrasted specific models (Luo et al., 2003), prior studies exploring the ability of models to simulate interannual variability and the role of different meteorological mechanisms have used only one model (e.g. Miller

and Tegen, 1998b;Mahowald et al., 2002;Mahowald et al., 2003;Ginoux et al., 2004;Ridley et al., 2014). It is well established in the dust literature that meteorology plays a central role in driving changes in dust, primarily from changes in precipitation, winds or vegetation cover (e.g. Petit et al., 1999; Marticorena and Bergametti, 1996; Mahowald et al., 2002; Prospero and Lamb, 2004; Engelstadter and Washington, 2007; McGee et al., 2010).  Here in this study, we use three different reanalysis winds, online dynamic winds, and different modeling frameworks to try to understand how robust

interannual variability in simulated dust is across 1990-2005.  Our emphasis is on conducting sensitivity studies comparing the importance of different model meteorology and frameworks, but we do include some comparison to observations. We also emphasize the variability within one year (seasonal cycle) compared to the variability across years (interannual variability).  The period between 1990-2005 has more available observational and reanalysis data, but not as much variability as previous periods (e.g. dry 1980s versus wet 1960s in the Sahel region; Prospero and Lamb, 2003).  Model

results are compared to limited available deposition, in situ concentration and AOD data, in order to evaluate the models' ability to simulate the spatial and temporal variability observed in the dust cycle.





## 2. Methodology

### 2.1 Model Description

#### 2.1.1 CAM4 dust model

5  For the bulk of the analysis in this study, simulations using the Community Earth System Model (CESM) were conducted with the Community Atmospheric Model (CAM4) (Neale et al., 2013; Hurrell et al., 2013). This model is capable of simulations based on prognostic dynamic meteorology, as conducted for long climate simulations, or simulations forced to follow specific meteorological events, which allows comparison to specific observational data or field campaigns (Neale et al., 2013). The CESM model includes four main global climate model (GCM) components: atmosphere, land,

10 ocean, and sea ice, all linked by a flux coupler. However, only the land and atmosphere components were prognostic for these simulations, with prescribed ocean and sea ice being used instead.

  Dust is entrained into the atmosphere in dry, unvegetated regions with easily erodible soils, when the winds are strong (Marticorena and Bergametti, 1995), using the Dust Entrainment and Deposition module (Zender et al., 2003a). There is a dust source when the leaf area index is sufficiently low ($<0.3$ $m^2/m^2$) and the soil moisture modifies the threshold wind

15 velocity, as described in more detail elsewhere (Zender et al., 2003a; Mahowald et al., 2006). For most of the simulations used here the CAM4 is used with the bulk aerosol module (BAM), This module includes four size bins: 0.1 to 1.0 µm, 1.0 to 2.5 µm, 2.5 to 5.0 µm, and 5.0 to 10.0 µm (Zender et al., 2003a; Mahowald et al., 2006), with the source size distributions described in Albani et al. (2014), following Kok (2011). The mass fraction of dust contained in each of the four bins is allowed to change over time as aerosols are transported and deposited out of the atmosphere (Zender et al., 2003a;

20 Mahowald et al., 2006). Transport occurs through the CAM4 tracer advection scheme (Neale et al., 2010). Dry deposition includes both gravitational and turbulent settling (Seinfeld and Pandis, 1998;Zender et al., 2003a), while wet deposition includes both convective and stratiform precipitation, incorporating prescribed solubility and parameterized scavenging coefficients (Mahowald et al., 2006;Albani et al., 2014). Emission over different soil types is parameterized by a geomorphic soil erodibility coefficient (Zender et al., 2003b), following the preferential source ideas of Ginoux et al. (2001). Finally,

25 regional soil erodibility is optimized using the methodology described in Albani et al. (2014), by applying scale factors to existing soil erodibility parameters for macro regions, in order to best match available data (Albani et al., 2014). For example, in most versions of the model the approach reduced dust over Central Asia and the Atacama Desert, and increased dust over Argentina (Albani et al., 2014).

30 **2.1.2 Meteorological forcing datasets**

  In the CAM model, the reanalysis forcing can be used to nudge the model close to specific weather patterns so that events can be simulated (Lamarque et al., 2011) following the methodology developed in the Model of Atmospheric Chemistry and Transport (MATCH) (Rasch et al., 1997; Mahowald et al., 1997). Large scale meteorology is input into the





model, while the subgrid scale mixing and precipitation are deduced within the model, allowing the model flexibility in terms of which input meteorology is used (Rasch et al., 1997;Mahowald et al., 1997). While this approach has the advantage of increased versatility with the input reanalysis dataset, it does allow inconsistencies to arise between the transport model and the reanalysis meteorology.  The approach used in MATCH and CAM was developed to minimize these inconsistencies

(e.g. Mahowald et al., 1995; Mahowald et al., 1997).  Previous studies have shown that the MATCH/CAM framework can reproduce the reanalysis precipitation to a very high degree at the sub-daily to monthly to annual time scales (Mahowald et al., 1997;Mahowald, 1996).

Three different reanalysis meteorological datasets were used to simulate dust entrainment, transport and removal in the CAM4 simulations: MERRA (Modern Era-Retrospective Analysis for Research and Applications version 1; Rienecker et

al., 2011), NCEP (National Centers for Environmental Prediction; Kistler et al., 2001), and ECMWF (European Center for Medium-Range Weather Forecasts; Dee et al., 2011).  Within the meteorological literature there are many studies looking in detail at the specific characteristics of the reanalysis datasets and contrasting these datasets to available observations (Trenberth and Guillemot, 1998;Trenberth et al., 2000; Treberth et al., 2011;Trenberth and Fasullo, 2013).  Reanalysis should not themselves be considered  observations, but are the closest representation we have to observed meteorology, that

can drive chemical transport models, and thus represent an important resource.  Here in this paper we supplement the meteorological analysis of different reanalysis datasets by contrasting how they impact dust emission, transport and deposition.

A fourth simulation was also conducted using AMIP-type protocol (Atmospheric Model Intercomparison Project; Gates et al., 1999) parameters, in which winds are calculated within the atmospheric model using forcings from monthly sea

surface temperatures.  All the CAM4 and CAM5 simulations were conducted using a ~2x2° horizontal resolution.  Not all of the input reanalysis data was available in the correct input format for the entire satellite era, so most of the analysis was conducted over a time period of 15 years when all the input data was accessible.  There are parts of the analysis for which data outside this time period was used, but this is always indicated in the results.  A summary of the model simulations and time periods of data availability are shown (Table 1). The first year of each simulation was neglected to allow for spinup.

### 2.1.3 Other dust models

In this paper, we also include sensitivity studies, where additional models are used, to contrast the importance of meteorological dataset with model construction.  We briefly describe here the other models used in the study, with an emphasis on contrasting the differences in the model construction, but refer the interested reader to the specific model

descriptions elsewhere.

An AMIP-style simulation was also conducted using the CAM5 model for comparison with the CAM4 AMIP simulation.  For AMIP simulations, the monthly mean sea surface temperatures are used to force the model online meteorology, but no atmospheric fields are used to constrain the model, in contrast to the reanalysis-driven simulations described above. Because in this case there is no inconsistency between the atmospheric model and observations, it is often




considered a more robust way to simulate water vapor and thus chemistry (e.g. Hess and Mahowald, 2008;Trenberth and Guillemot, 1998;Trenberth et al., 2000). However, AMIP simulations cannot simulate exact weather events, but only interannual variability. While the dust generation module in the CAM5 is identical to the CAM4, there are significant differences in the physics, especially, as well as the aerosol formulation (Albani et al., 2014). The CAM5 model includes

new planetary boundary layer, radiation, and moist convective parameterizations (Hurrell et al., 2013), in addition to a modal aerosol module (Liu et al., 2011;Ma et al., 2013). Similar to CAM4, the dust simulations in the CAM5 were evaluated and tuned as in Albani et al. (2014). CAM5 assumes all aerosols are internally mixed (all aerosols in the same size are assumed to be mixed together for radiative forcing calculations), in contrast to CAM4, where aerosols are assumed to be externally mixed. CAM5 also allows aerosol indirect cloud effects to be calculated (e.g. Wang et al., 2011), although these are not

used here.

Dust simulations using the GEOS-Chem model (version v9-01-03; http://www.geos-chem.org/) from Ridley et al. (2014) were also considered here (Table 1). The model includes similar processes as the CAM4 model, including externally mixed aerosols, but the GEOS-Chem model is forced here only by MERRA-1 winds (referred to as GCHEM(MERRA) here). GCHEM (MERRA) uses the DEAD dust module (Zender et al., 2003a), and employs the dust source function derived

from (Ginoux et al., 2001). More details on the dust simulations from the GCHEM (MERRA) model can be found in (Ridley et al., 2013;Ridley et al., 2014). Notice that the version used here for these comparisons does not include source function derived from Koven and Fung (2008) or the vegetation phenology and interannual variability, as included in the study focusing on African emissions (Ridley et al., 2013;Ridley et al., 2014). The horizontal resolutions of the model simulations were all 2x2.5°, and were interpolated onto the CAM grid for analysis in the paper. This model uses a similar dust

entrainment scheme, but has different size distribution, and deposition mechanisms, as well as different boundary layer, and moist convection physics.

It should also be noted that there is a difference between how the GCHEM (MERRA) model and CAM models incorporate reanalysis meteorology. The GCHEM (MERRA) model reads in all meteorological parameters from the reanalysis datasets, including turbulent and moist convective mixing, as well as the hydrology. This has the advantage that

the chemical transport model does not have to rederive the hydrological cycle or mixing, which can be difficult (e.g. Mahowald et al., 1995;Rasch et al., 1997). On the other hand, it makes the model less flexible, as it can only be simulated using reanalysis products with mixing parameters output at the right frequency, in contrast to the MATCH or CAM framework (e.g. Rasch et al., 1997;Mahowald et al., 1997). This means that although both the CAM4 (MERRA) and GCHEM (MERRA) models are forced by MERRA winds, the surface winds and transport may still be slightly different.

Finally, simulations using the Model of Atmospheric Transport and Chemistry (MATCH) (Rasch et al., 1997) with NCEP reanalysis data (Mahowald et al., 1997;Kalnay et al., 1996) are included. These simulations use a similar entrainment and deposition scheme (Zender et al., 2003a), with a simple wet removal scavenging coefficient (Luo et al., 2003), and have been extensively compared against observations (Luo et al., 2003; Luo et al., 2004; Mahowald et al., 2003). In contrast to the CAM models discussed earlier, there is no vegetation phenology included. Instead, the preferential source term from





Ginoux et al. (2001) was used. The horizontal resolution of the MATCH (NCEP) model used here is 1.8x1.8°, and the results were interpolated to the CAM grid before comparison to the other models.

Note that only the GCHEM (MERRA) model simulations are independently simulated: the others all come from the same group, and thus are likely to have similar attributes (Knutti et al., 2013), although there are differences, such as in dust
vertical distributions and transport in relation to vertical mixing (shown in Figure 5 and Albani et al., 2014), and the strength of the Sahel source (e.g. Scanza et al., 2015). Comparisons of different meteorological reanalysis from an energy or water budget perspective suggest different strengths and weaknesses of the reanalyses (Treberth et al., 2011;Trenberth and Fasullo, 2013).

For two of the model simulations used here (CAM4 (MERRA) and CAM4 (NCEP)), additional aerosol species were
available for analysis (Sea salts, black carbon (BC), organic carbon (OC), and sulfate ($SO_4$)), and we use these to contrast the correlations between dust aerosols and other aerosols for a sensitivity study.

**2.2 Observational Data Description**

For completeness, we compare standard annual means from each simulation to available data (e.g. Ginoux et al., 2001;Huneeus et al., 2011) in order to show that the mean dust cycle is reasonable in our models. In this study, we use the
annual mean surface concentration, aerosol optical depth, and deposition compilations from Albani et al. (2014).

For the variability studies, data was chosen for overlap with model runs and availability of longer datasets to evaluate interannual variability. In situ observations from the University of Miami network (Prospero and Nees, 1986;Prospero et al., 1996;Arimoto et al., 1990;Arimoto et al., 1997) were used here, as compiled in (Luo et al., 2003;Mahowald et al., 2003) and updated at some sites (Prospero and Lamb, 2003). In addition in situ observations from 3 sites in North Africa from the
AMMA (African Monsoon Multidisciplinary Analysis) campaign were included (Marticorena et al., 2010). Details on the individual sites and locations are included in Table 2.

Sun photometer-derived aerosol optical depth is included from the AERONET (AErosol RObotic NETwork) database (Holben et al., 2000). Only sites where more than 50% of the modeled aerosol optical depth was from dust are included in this comparison (as filtered using the MATCH model, described below, in Mahowald et al., 2007), and only sites with more
than 18 months of data are used here to estimate variability (Table 2).

Two sites in the Southern Hemisphere are included (bottom of Table 2): surface concentration data from Rio Gallegos (Zihan, 2016) and deposition data from Kerguelen (Heimburger et al., 2012). Because of the limited datasets in the Southern Hemisphere, we will consider these sites separately in Section 3.3.

For evaluation of the models' precipitation, we use the CPC Merged Analysis of Precipitation (CMAP)
(http://www.esrl.noaa.gov/psd/data/gridded/data.cmap.html) (Xie and Arkin, 1997). This is a combination of in situ and satellite observations, as well as models, to present the best estimate of precipitation.





### 2.3 Analysis Methods

For comparison of the variability in the monthly mean modeled and observational values, we define variability similarly to previous studies (Mahowald et al., 2003):

$$\text{Variability} = \text{Standard deviation (values)} / \text{Mean (values)} \qquad (1)$$

We report the variability at three different time scales: i) using all monthly means, across years, ii) using the climatological monthly means to evaluate seasonal variability, and iii) using annual means to evaluate interannual variability.

Models and observational time series of in situ concentrations and AOD (Table 2) were also correlated, using rank correlations to assess the ability of the models to simulate variability (similar to Mahowald et al., 2003). Rank correlations

are used in order to reduce the importance of individual, extremely high data points, which can dominate regular correlations (Wilks, 2006).  Similar to the variability, three sets of correlations are reported: i) correlations of the monthly mean values across multiple years, ii) correlations of the climatological monthly means, which test the ability of the models to simulate the seasonal cycle, and iii) correlations of the annual means, which test the ability of the models to simulate interannual variability.  Notice that the first correlation (monthly mean correlation) includes elements from the second (seasonal cycle)

and the third (interannual variability).

We also analyze the observations and model output for trends, by calculating the least squares fit slope, as well as the standard deviation in the slope.

In order to understand how similar model results are, we also calculate the variability on the three time scales (monthly mean, seasonal and interannual variability) at each grid box and compare different models.  We also correlate the

time series of models at individual grid boxes across model simulations, again on the three different time scales.

To show the sensitivity to meteorology, we correlate the 3 CAM4-reanalysis simulations (CAM4-RE; which will give us 3 different correlation coefficients: CAM4-MERRA vs. CAM4-NCEP; CAM4-MERRA vs. CAM4-ERAI; and CAM4-NCEP vs. CAM4-ERAI) and then average at each grid point the three different correlation coefficients to find the average correlation. Similar results are conducted using the AMIP simulations (CAM4-AMIP vs. CAM5-AMIP), and for the model

simulations using the exact same meteorology (CAM4-MERRA and GCHEM-MERRA; and CAM4-NCEP and MATCH-NCEP).

Finally, we use the model values to estimate the number of monthly mean observations required to correctly estimate the climatological annual mean value over 1990-2005.  To do this, we assume that we would like to have a 95% chance to be within one standard deviation of the climatological mean. 1000 Monte-Carlo simulations were conducted, and each time we

chose randomly from the modeled monthly mean values at each grid point, and for each number of observations (between 1 and 50) we calculated the percentage of the time that the mean is within one standard deviation of the climatological mean of the 1990-2005 simulation.  At every gridbox, the number of observations that would meet the 95% criteria is then calculated, providing an estimate of the number of months of observations required.





Note that the modeled monthly mean values are not at all Gaussian distributed, and thus normal methods for determining the number of observations would not work (e.g. Wilks, 2006). For the correlation coefficients, we use rank correlations, which works with non-gaussian data, but for mean and standard deviation analysis described above, we use these metrics, as they are standard in the climate model community (Taylor, 2001; Gleckler et al., 2008), even though they

are statistically only valid with Gaussian distributions.

### 3. Results and Discussion

### 3.1 Comparison of Model and Observational variability

The annual mean distribution of the model simulations included here are evaluated elsewhere in more detail (Luo et al., 2003;Huneeus et al., 2011;Albani et al., 2014;Ridley et al., 2013;Ridley et al., 2014), but for completeness we repeat

comparisons of annual mean surface concentration, AOD, and deposition between available observations and the model simulations in the online supplement (Table 3). Concentrations vary over several orders of magnitude spatially, and the models are able to simulate these variations. In addition, the models can be shown to be mostly accurate in simulating the observed dust AOD and deposition. Most of the model versions presented here do an equally good job when compared against the observations (Table 3). Recognize that the CAM4 and CAM5 simulations were tuned against these same

observations (Albani et al., 2014), while the MATCH (NCEP) and GCHEM (MERRA) models were previously compared to similar observational syntheses (Luo et al., 2003;Huneeus et al., 2011;Ridley et al., 2013;Ridley et al., 2014).

The mean source flux and globally averaged AOD of the three CAM4-RE is 2400 +/- 26% Tg/yr and 0.026 +/- 30%, respectively, while for all the models included here the mean emission flux and AOD is 2400 +/- 26% and 0.025 +/- 40%, respectively. These ranges are similar to previous studies (Huneeus et al., 2011; Reddy et al., 2005). Note that using a

similar model (CAM4-RE) and similar methodology to constrain the dust AOD, based on a combination of surface concentration, deposition and AOD in dust regions (Albani et al., 2014) obtains an uncertainty just due to meteorology of 30%. A more recent estimate, based more extensively on remote sensing data with limited information from 4 different models, but without using deposition or surface concentration data, finds a higher AOD of 0.033+/- 0.006 than found here (Ridley et al., 2016). Thus there are strong sensitivities of the deduced AOD to assumptions about how to include different

data, as well as the details of the models and methodology used.

Since this study is focused on an inter-comparison of different model simulations, rather than an evaluation of a specific model, we conduct limited comparison to observations. We focus on the highest quality data, coming from in situ concentrations and sun photometry data (e.g. Prospero and Lamb, 2003;Holben et al., 2000; Table 2), and ignore satellite based measurements (e.g. Torres et al., 2002;Evan et al., 2006) and dust visibility data (Mahowald et al., 2007) which are

more difficult to interpret, both because they are not always only dust aerosols, and because they can have larger errors and thus be more difficult to compare for interannual variability (Torres et al., 2002;Evan et al., 2006;Mahowald et al., 2007).





Some previous studies have included comparison of these data in terms of interannual variability for specific model simulation evaluation (e.g. Mahowald et al., 2003;Ridley et al., 2014). The vertical distribution of the aerosol can also vary depending on the meteorology used (e.g. Albani et al., 2014), which may introduce some additional variability and discrepancy for in-situ ground based measurements.

5       We consider separately here the seasonal cycle in the surface concentrations and the AOD produced by each simulation at fifteen specific sites (Figures 1a-1i and 2a-2f), compared with the interannual variability at those sites (Figure 3a-3i and 4a-4f), which when combined give the monthly mean variability (Figure S1 and S2). Notice that each time series in these figures is divided by the mean value across the time period, so that the seasonal variability (Figures 1 and 2) can be contrasted with the interannual variability (Figures 3 and 4) without concerns about differences in scale. The amount of

variability (represented here by the standard deviation of the monthly mean over the annual mean; Eq. 1 in Section 2.3) in surface concentration simulated by the range of the models encompasses that of the observations at the different sites (Table S1; Figure 5a), which is between 0.75 (Mbour) and 2.0 (Bermuda). Note that as seen previously (e.g. Tegen and Miller, 1998; Mahowald et al., 2003), the largest variabilities in both surface concentration and AOD are seen downwind from the source regions, in ocean regions that intermittently experience events (e.g. Bermuda, where there was a specific event in the

Atlantic described by Ginoux et al., 2001). The models simulate the stronger variability at Bermuda (for example), and the smaller variability over the source regions (i.e. Mbour). Much, but not all of this variability in the observations comes from the seasonal cycle, both in the models and in the observations in these stations (Table S2; Figure 5c). The fraction of the monthly mean variability that is due to the seasonal cycle varies from 0.5 (Izana) to 0.94 (Midway) (Figure 5c; Table S2).

      The variability in the AOD (Table S1; Figure 2 and 4; Figure 5b), tends to be smaller than in the surface

concentrations, which is likely to be due to both the fact that for the AODs, we are limited to regions close to source regions, and the fact that AODs are more integrated quantities as seen in previous studies (Mahowald et al., 2003). Notice that again, close to the source regions, the seasonal variability in AODs is the dominant portion of the variability in both the models and observations (Figure 5d; Table S2).

      Next we evaluate the ability of the models to simulate the high and low monthly means using rank correlations. Most

of the models have relatively good correlation coefficients (Figure 5e and 5f), which are statistically significant at most of the stations (Table S3). The exceptions are at Izana and Ilorin for all of the models, and the CAM4 (AMIP) simulation, which is not statistically significantly correlated at most of the stations (Table S3; Figure 5e and 5f). For the CAM4, forcing with only SSTs substantially degrades the ability of the dust model to simulate the seasonal cycle, while in CAM5, the seasonal cycle is better simulated. Most of the ability of the models to simulate the variability of the observations comes

from the seasonal cycle (Figure 5e,f vs. Figure 5g, h). There is very little skill in the models' ability to simulate interannual variability at any of the stations (Figure 5i,j).

      In model intercomparisons it has been observed that the model mean often does a better job than the individual model simulations (e.g. Flato et al., 2013). We evaluate this in the case of dust using the average of the CAM4 reanalysis models (CAM4 (MERRA), CAM4 (NCEP) and CAM4 (ERAI)). At the observational sites considered here, we do not see a large



increase in the correlation coefficients for the model average versus individual models for either the seasonal cycle (Table S3) or the interannual variability (Table S4).

Overall this section supports previous studies (Prospero, 1996;Mahowald et al., 2003;Ginoux et al., 2004;Marticorena et al., 2010) suggesting that at the limited observational stations (Table 2) seasonal variability is larger than interannual

variability (Table S2 and S3; Figure 5c and 5d), and that the model can simulate seasonal variability better than interannual variability in both surface concentrations and AOD (Figure 1 vs. 3; Figure 2 vs. 4; Figure 5e vs. 5g and Figure 5f vs. 5h). Taken overall, the models driven by reanalysis winds (all except CAM4 (AMIP) and CAM5 (AMIP)) compare roughly similarly against the available observations for both seasonal cycle and interannual variability (Figure 5).

New in this section is the evaluation of the relative ability of reanalysis driven models versus sea surface temperature

forced models (CAM4 reanalysis versus CAM4 (AMIP)), which suggest a degradation in the ability of the models driven only by sea surface temperature to simulate both seasonal and interannual variability, although this is dependent on which model version (CAM4 vs. CAM5; Table S3 and S4; Figure 5). This correlation potentially provides insight into how much of the variability in dust is driven by sea surface temperatures. There are also significant differences between models driven by the same meteorology, for example (CAM4 (MERRA) versus GCHEM (MERRA) and CAM4(NCEP) and

MATCH(NCEP), highlighting the importance of model formulation as well as meteorology.

### 3.2 Comparison to trends in data in the North Atlantic

Recent studies have highlighted the importance of fluctuations in rainfall in the Sahel for driving decadal scale variability in North Atlantic dust concentrations as seen in Barbados (e.g. Prospero and Lamb, 2003), although the importance of land use, winds and vegetation changes have been noted as well (e.g. Marticorena and Bergametti, 1996;

M'bourou et al., 1997; Mahowald et al., 2002; Mahowald et al., 2007). Since 2000, it has been noted that the Sahel precipitation no longer anti-correlates with dust at Barbados, suggesting a different mechanism may have become important. (Prospero, 2006; Mahowald et al., 2009). Ridley et al. (2014) proposes the hypothesis that the observed decrease in dust from 1982 to 2008 at Barbados is controlled by source wind strength over source regions in North Africa. For this argument, they use model evaluation with the GCHEM (MERRA) model (a different version of which is included also here), as well as

analysis of ERAI and NCEP reanalysis winds and other observations (Ridley et al., 2014). Indeed, station data in the North African region, especially the Sahel, support the idea that winds decreased in this region between the late 1970s and 2003 (Mahowald et al., 2007), and there is also an observed widespread decrease in surface winds across many land regions (McVicar et al., 2012). Of course, there are many issues with the observation of surface winds due to small scale effects of buildings or topography (e.g. discussed in McVicar et al., 2012), so it is unclear how robust trends in observed surface winds

are. Note that data correlations between visibility and winds suggest that both precipitation (Prospero and Lamb, 2003) and winds (Engelstaedter and Washington, 2007) are important for changes in dust near the source regions of North Africa (Mahowald et al., 2007).

Here we can consider whether the hypothesis put forward in (Ridley et al., 2014), that the decrease in winds in the





surface region is responsible for the decrease in surface concentration at Barbados, is consistent with the simulated trends in the multiple models included in this analysis. For this part of the paper, we use the full time period of our models, although for some models only some of the 1982-2008 time period is available (Table 1). For simplicity we consider only annual averages. As shown in (Ridley et al., 2014), there is a statistically significant downward trend in the data at Barbados (Table

4). All the model versions considered here simulate a downward trend in the data at Barbados, although for some models this is not a statistically significant trend (CAM4 (NCEP); CAM5 (AMIP)). Only one model simulates the slope within one standard deviation of the observed value: the CAM4 (MERRA) (Table 4). The GCHEM (MERRA) overpredicts the magnitude of the negative slope as shown by Ridley et al. (2014), while the other model versions underpredict the magnitude of the slope (Table 4).

Only some of the models see a statistically significant decrease in source strength in North Africa (Table 4), and some models predict an increase over this time period. However, a look at the trends in surface concentration across the models show that all the models see a decrease in surface concentrations that extends from the Sahel area of North Africa across the Tropical North Atlantic to Barbados (Figure 6), supporting the idea that the source strength is decreasing. While individual models might simulate downward or upward trends elsewhere, this is the only region that sees a consistent model signal

across this time period (Figure 6). If we focus on the Sahel (western) area of North Africa, indeed, most of the models simulate a decrease in the source (exceptions are CAM4 (ERAI) and CAM5 (AMIP)) (Table 4). Since the visibility data in North Africa also suggests a decrease across this time period in the western Sahel (Mahowald et al., 2007), these support the idea that the decrease in the source is the cause of the decrease in Barbados surface concentrations. In the CAM4 models, the strongest correlation in IAV of the source occurs with surface winds (Table S5), and indeed in all the models there is a

decrease in surface winds over this time period over the source regions, as seen in (Ridley et al., 2014) (Table S6).

There is also a correlation between Sahel precipitation and Barbados concentrations (Prospero and Lamb, 2003), or precipitation and visibility in the Sahel (e.g. Mbourou et al., 1997;Mahowald et al., 2007), so the other driver could be precipitation. In some of the CAM4 model simulations, the IAV of the source strength does feature significant correlations with both LAI and precipitation, but in general those same cases feature even stronger correlations with surface winds (Table

S6). Although the quality of the surface wind data precludes us from evaluating the reanalysis for surface winds, as discussed in McVicar et al. (2012), we can evaluate the precipitation, which is commonly done (e.g. Trenberth and Guillemot, 1998). When we do, we see that the reanalysis precipitation datasets are not capturing either interannual variability in precipitation nor the slope in the precipitation, compared to the CMAP precipitation compilation (more details in Section 2.2) (Table 4). Since precipitation and winds, especially gustiness in moist convection (Engelstaedter and

Washington, 2007), are likely related to each other, an error in the IAV of precipitation may be indicative of an error in the IAV of winds.

Overall, the model simulations conducted here support the hypothesis of Ridley et al. (2014), although the quality of the reanalysis data forcing our simulations does not allow us to be conclusive about our results.



### 3.3 Southern Hemisphere variability

Most of the available dust observations come from the Northern Hemisphere (Table 2). Here we consider two sets of data from the Southern Hemisphere, one of surface concentrations in Rio Gallegos (Argentina/Patagonia, 52S, 69W) (Zihan, 2016) and one of the deposition at Kergulen Island (49S, 70E), which is likely to be influenced by both South American and

South African dust sources (Heimburger et al., 2012) (Figure 7). The surface concentration data at Rio Gallegos suggests a monthly mean variability of 0.7 (Table 5), which is at the lower edge, but in the same range as the observations in the Northern Hemisphere (values between 0.7 and 2.0; Figure 5a; Table S1). The models, however, tend to predict too large a variability at this site (between 2 to 7). It is unclear why the models overpredict the variability, but it may be due to issues with the actual location of the sources in the model compared to the observations, or the strength of the North-South gradient

in concentrations in the models versus observations (e.g. Gaiero, 2007;Gasso et al., 2010). In addition, the way the tuning in Albani et al. (2014) was conducted will increase the interannual variability of the dust cycle in the CAM4 and CAM5 simulations in the Southern Hemisphere (Table 5). These model simulations had too small an emission value from some regions and too large from others, and thus the model source strengths were tuned in order to broadly match observations (Albani et al., 2014). In particular, in Argentina, the dust source strength had to be tuned up very strongly in order to obtain

a climatological dust that matched observations. This means that there were very few grid boxes and time periods with active sources, increasing the temporal variability in the model. If we had instead changed the wind threshold in our model formulation in order to tune the source strength (e.g. Tegen and Miller, 1998), we presumably would not have increased our variability as much, highlighting the importance of details in the model formulation for model results. We will explore the ramifications for variability predictions in Section 3.4 and 3.5. Further downstream of the sources, the amount of variability

in deposition observed at Kerguelen is 0.71, which is close to that simulated by several models (CAM4 (MERRA); CAM4 (NCEP), GCHEM (MERRRA) and MATCH (NCEP)), although the other models overpredict variability (Table 5).

At both Rio Gallegos and at Kerguelen, the fraction of the variability due to the seasonal cycle (0.2 and 0.45, respectively, Table 5) is much smaller than that observed in the Northern Hemisphere at the stations available (values between 0.7 and 0.9; Table S1; Figure 5). The models are able to simulate the reduced fraction of variability due to the

seasonal cycle at these sites (Table 5). Because of the limited data and length of data, we cannot be sure, but the observations presented here are consistent with a stronger role of interannual variability, compared to seasonal variability, in dust sources in the Southern Hemisphere than in the Northern Hemisphere, as simulated by the models.

### 3.4 Spatial analysis of model simulations of variability and correlations

Consistent with previous studies (e.g. Tegen and Miller, 1998;Mahowald et al., 2003; Mahowald et al., 2011), the
largest variability (standard deviation over the mean, described in equation 1, Section 2.3) in modeled dust concentrations occurs not in the main dust source areas or outflow regions, but rather adjacent to these regions, where intermittent dust events occur (Figure 8a). Some of the highest variability occurs over ocean regions, especially in the Southern Hemisphere.



Most of the monthly variability in the North Hemisphere can be attributed to the seasonal cycle, rather than interannual variability (Figure 8b). The surface concentration and deposition variability have similar patterns, with perhaps a slightly higher variability in deposition across most regions (Figure 8a vs. 8c), and similar importance of the seasonal cycle (Figure 8b vs. 8d.); AOD instead tends to have less variability and a more important seasonal cycle (Figure 8e and f). In all

variables (surface concentration, deposition or AOD), the southern hemisphere has much larger variability, and much of this extra variability is due to interannual variability (Figure 8). As discussed in Section 3.3, for the CAM4 and CAM5 model simulations, some of this enhanced southern hemisphere variability could be due to the tuning of the source areas, because the dust sources were not consistently active enough (Albani et al., 2014). If we consider only the models in which the Southern Hemisphere dust sources did not have to be tuned up so much (GCHEM (MERRA) and MATCH (NCEP)), then

the monthly mean variability in the South Atlantic is similar to the North Atlantic (Figure 9a), but there still tends to be a smaller proportion of the variability from seasonal variability, and thus a more important role for interannual variability, in the South Atlantic, Indian or Pacific oceans than in the Northern Hemisphere oceans (Figure 9b). Indeed the limited observational data supports strong interannual variability in the Southern Hemisphere, although not as strong as some of the model versions (Section 3.3).

All of the models included here, except the GCHEM (MERRA) and MATCH (NCEP) simulations, include seasonal and interannual variability in vegetation as a control on sources (CAM4 and CAM5) (Albani et al., 2014;Ridley et al., 2014). It appears that in many locations the model simulations including time varying LAI have larger variability (Figure S3; contrast f vs. others, especially in the Southern Oceans). However, over the North Atlantic, for example, one simulation without time varying LAI (MATCH (NCEP)) has greater variability than the CAM4 and CAM5 simulations, which have

time varying LAI (Figure S3f vs. others). In the North Pacific, the MATCH (NCEP) simulation has more variability than the CAM4 (ERAI) simulation, which is consistent with the MATCH (NCEP) simulation being able to simulate the seasonal cycle in transport to the Pacific (Luo et al., 2003) without having a seasonal cycle in LAI, in contrast to other models (e.g. (Tegen et al., 2002) which is based on ERAI winds). This highlights how the details of model construction and meteorology can change model behavior, making it difficult to broadly generalize about what model requirements (e.g. time varying LAI)

are needed to match observed seasonal or interannual variability.

        Next we consider how similar the temporal variability is in the model simulations covering the same time period. If two model simulations are temporally correlated, it implies the timing of the monthly mean variability in the models is similar. Of course, to obtain the fraction of the variability that is similar, the correlation coefficient needs to be squared (if we assume a Gaussian distribution in the model output), which means that statistically significant moderate correlation of 0.3

to 0.5 only implies that 10-25% of the variability is similar. However, the correlation is a useful way to consider how similar the simulations are in their variability. The correlations between the model simulations used here, for the surface concentration, suggest that the models simulate similar variability over most of the globe (Figure 10), especially for the cases with reanalysis driven simulations (Figure 10a, b, d, e), but the simulations with time varying-SSTs (AMIP) were more different (Figure 10c and f; Figure S4). This suggests that sea surface temperature forcings are not the only important driver



for the dust cycle, which is consistent with the statistically significant but low correlations between such ocean-forced phenomenon as NAO and El Nino and dust (e.g. Moulin and Chiapello, 2004;Mahowald et al., 2003). Notice that simulations with the same model but different winds (CAM4 (MERRA) vs. CAM4 (NCEP); Figure 10a) had similar correlation coefficients as using different winds and model framework (e.g. CAM4 (MERRA) vs. MATCH (NCEP); Figure 10e) or different models with the same winds (e.g. CAM4 (MERRA) vs. GCHEM (MERRA); Figure 10d). This suggests that both model framework and winds contribute to variability, and perhaps in a similar, but not additive magnitude. The correlations tend to be lower near the southwestern US, and in the Southern hemisphere (Figure 10).

The correlation coefficients averaged over the CAM4-reanalysis models (Figure 11a, c and f) suggest that the surface concentration and deposition are similarly correlated, but that AOD has smoother and higher correlations. This is consistent with AOD being a more integrated quantity (Figure 8) (Huneeus et al., 2011; Albani et al., 2014). The lowest correlations occur over remote ocean regions, especially in the Southern Hemisphere, for all the variables. Note also that the correlation coefficients between the simulations using the same modeling framework but different meteorology (Figure 11a, c and f) had similar correlation coefficients, with a similar spatial structure, as the correlations between simulations using different modeling frameworks but the same meteorology (Figure 11b, d, and e). This suggests that meteorology and the modeling framework are equally important for simulating variability.

The magnitude and the spatial structure of the correlation in the seasonal cycle is similar for all the variables as that seen using all monthly means (Figure 12a, c, e vs. 11a, c, e), suggesting that most of the correlation between the model versions is due to the correlation in the seasonal cycle (Figure 12). This is made clearer when considering just the correlation of the annual means, showing that the models simulations are much less similar in their interannual variability (Figure 12b, d and f). Note however, that the spatial structure in the correlations for IAV is very different than considering either the monthly mean or seasonal cycle (Figure 12b, d, f vs. 12 a, c, e). Surprisingly in some remote ocean regions, the models are simulating similar interannual variability (notably, parts of the North and South Atlantic) (Figure 12). There are some statistically significant trends in the model simulations over the 1990-2005 time period, as seen in Barbados (Section 3.2) and the nearby North Atlantic and some parts of North Africa (Figure 6), which may be responsible for this coherence. A comparison of other aerosols in two of the CAM4-reanalysis based simulations available here (Figure S5), is consistent with the idea that dust is highly variable, and that there is some correlation between the models driven by different meteorology far from the sources as well as close, but that interannual variability can be quite different for transport of aerosols (Figure S5). We will next discuss regional averages to understand how similarly ocean basin averages are simulated, to see if these IAV correlations in some regions are large enough to provide coherent basin estimates.

## 3.5 Modelled temporal trends in different regions

As discussed in the cases of the Tropical North Atlantic and South America (Sections 3.2 and 3.3), as well as in previous studies (Tegen and Miller, 1998), some of the variability in dust comes from the source regions. Looking across the model simulations considered here, we see strong IAV in many of the source regions, with the strongest IAV in the smallest



source regions, as seen previously (Tegen and Miller, 1998;Mahowald et al., 2003) (Figure 13). Only the western Sahel source (Section 3.2) is simulated to have a statistically significant trend in all the model simulations (Table 4, also seen in Figure 6 and 13). There are strong increases in some of the model simulations of the Australian source, consistent with the observed increase in drought over this time period (Cai et al., 2014), although we do not know of dust observations verifying

this. For example, visibility data extending through 2003 does not support an increase in dust source strength in Australia (Mahowald et al., 2007). Previous studies have shown that there are decreases in dust from South America over this time period in some models (Doney et al., 2009), but these are not shown in all the model versions or supported by robust observational data (Doney et al., 2009). Overall, although some models simulate an increase in the global dust source, other models simulate a decrease, suggesting no clear trends from the modeling of the global dust cycle over this time period

(Figure 13).

For most of the sources considered here, there are moderate correlations in time (0.4-0.8) in the annual mean source strength, except for the Sahel region (0.13), suggesting they simulate similar interannual variability (Table 6; Figure 13). Although the emphasis of this paper is on the temporal variability, there are significant differences in the climatological mean source strengths in the models used here (Table 6), highlighting the uncertainties in dust sources. The larger sources

(North Arica, Sahel, and the global average) have mean source strengths varying by 25% while the smaller sources vary up to 160%, just due to differences in the meteorology, using the same CAM4 model and observational constraints (Albani et al., 2014).

Focusing on the drivers of the CAM4 modeled variation in sources suggests that LAI has the strongest correlation with IAV in sources for several source regions (Australia, South Africa and South America (Argentina), while surface winds

have the highest correlations for East Asia, Middle East, North Africa and the western Sahel (Table 7). It is reassuring that the model winds have high correlations with IAV source strength in the model for East Asia, for there is a strong correlation in the current climate observations between winds and sources in this region (e.g. Sun et al., 2001), as well as speculation that past climate variability in winds from East Asia is sensitive to synoptic scale wind events (e.g. Roe, 2008;McGee et al., 2010). Soil moisture has the highest correlation for North America. Notice that IAV in LAI is likely to be a strong function

of soil moisture, and precipitation (Table 7). There are trends across this relatively short time period of a few of the source regions and variables, averaged across all the models (Table S7), but longer records tend to suggest oscillation in dust related variables: for example the downward trend in dust in the Sahel from the 1980s followed a strong upward trend between the 1960s and the 1980s (Prospero and Lamb, 2003), and indeed there may have been even longer trends in dust from North Africa (Mulitza et al., 2010). Thus, trends in the short time period considered here (1982-2008) may not necessarily be

representative of the longer term trends.

The current generation of earth system models have a great deal of difficulty in simulating not only precipitation (as discussed in Section 3.2) but also LAI (e.g. Sitch et al., 2015;Mahowald et al., 2016), and thus this strong dependence on difficult-to-simulate variables may decrease our ability to simulate interannual variability. Note that using satellite retrieved vegetation (e.g. Zhu et al., 2013), as done in some models (e.g. Ridley et al., 2014), may remove model-derived





uncertainties, but the satellite retrieved vegetation has large uncertainties, especially in regions with low LAI, and does not do a good job of detecting brown vegetation, which is very important in resisting dust entrainment (e.g. Okin et al., 2001). This suggests that simulation of the surface conditions (e.g. vegetation, soil moisture, surface winds) in the source regions is likely to limit our ability to accurately simulate IAV in dust.

We consider next the variability in deposition and AOD across different ocean regions, and how similar these are across the models. The mean deposition from the different model simulations can vary widely (Table 7), as seen in the source strengths (Table 6) and previous studies (e.g. Huneeus et al., 2011). Downwind of the large source regions (e.g. North Atlantic, Central North Atlantic, Equatorial Atlantic and Northern and Equatorial Indian Ocean) the standard deviations between CAM4-RE climatology mean are the lowest 7-25%, and they are largest in the remote ocean regions, like

the Southern Ocean (100%). The standard deviation between models is slightly larger if all models are included (Table 7). Similarly, for the AOD in the ocean basins, the difference in model simulations is smallest close to the source regions (20%) and largest in the remote ocean regions (100%) (Table 8).

Here we emphasize the temporal variability, and the comparison of the CAM4-RE model simulations suggest in many basins there are consistent signals, with correlations in net deposition above 0.3 in most basins, with the exception of

North Indian and South Atlantic (Table 7). Interestingly, AOD is consistent (r>0.3) in different basins, with North Central Pacific, North Atlantic Equatorial Pacific having the lowest correlations. This highlights the disconnect between AOD and deposition, which can make diagnosing deposition variability from AOD difficult, as noted previously (e.g. Mahowald et al., 2003). Note even in the basin with the most consistent simulations (0.68 in the North Atlantic), the variability in deposition simulated similarly in the models represents only 50% of the variability (assuming Gaussian distributions).

**3.6 Implication of modelled variability for sampling**

The large variability in dust implies that it maybe difficult to constrain the dust cycle from limited observations. While we have satellite data, we can only use that data to constrain dust, when it is the dominant aerosol, which occurs only in limited regions just downwind of major sources like North Africa (Mahowald et al., 2007). Over much of the ocean, we only have individual daily averaged values from cruise data (Baker et al., 2006;Buck et al., 2006;Sholkovitz et al., 2012).

Previous model studies have shown that over much of the ocean, modeled daily averaged dust concentrations will tend to underestimate the annual average, and only be within a factor of 10 to 2 of the true modeled average (Mahowald et al., 2008). Here we consider how many monthly means are required to obtain an estimate of the annual mean that is within one standard deviation of the true model annual mean value (Figure 15). Over most of the globe, the number of monthly mean observations is 8-12, or almost a full seasonal cycle. But in regions with large variability (Figure 7), longer time periods of

more than 2 full years are required to obtain a mean and standard deviation including the true mean (Figure 16). Note that here we assume that there is no trend or significant change in the dust cycle. Characterizations of changes in dust show that there are interesting trends over the longer term, which requires longer records (Prospero and Lamb, 2003;Ridley et al., 2014).





## 4. Conclusions

Simulations of monthly mean variability in 7 different model simulations are compared to better understand how robust the variability is in models for the period 1990-2005. Although the emphasis of this paper was not on evaluation of specific models, the models were compared with in situ concentration (Prospero and Lamb, 2003;Marticorena et al., 2010) and

AERONET AOD (Holben et al., 2000) observations. The models considered here were 4 versions of the CAM4, the GCHEM (MERRA), MATCH (NCEP) and CAM5-AMIP (Table 1) (Albani et al., 2014;Luo et al., 2003;Ridley et al., 2014). Here we ignore the possible effects from land use and land cover change, which is hypothesized to represent 25% of dust sources currently (Ginoux et al., 2012).

The model simulations do roughly similarly well compared to observations when driven by reanalysis meteorology,

but less well when driven by sea surface temperatures with meteorology being prognostically calculated (Figure 5). The models' ability to simulate the observations is strongest for the seasonal cycle, and the models are less able to simulate the interannual variability, similar to previous studies (Mahowald et al., 2003;Ginoux et al., 2004) (Figure 5). Surface concentration and deposition have similar variability structures, while AOD tends to have less variability (Figure 8). There is more variability, especially interannual variability, in parts of the Southern Hemisphere (Figure 8). Some of this was

artificially (potentially) enhanced in the simulations considered here because of the way that the CAM4 and CAM5 were tuned in the Southern Hemisphere (Figure 8 vs. Figure 9). But the limited observations suggest that in the Southern Hemisphere there is a larger fraction of interannual variability than seasonal variability compared with the Northern Hemisphere, consistent with the smaller sources in that region (e.g. Tegen and Miller, 1998).

Modeling interannual variability is sensitive to both meteorology as well as model construction, and thus drawing

firm conclusions about how best to capture observed variability based on only one model is likely to be difficult. Our results that model construction as well as meteorology is important is consistent with general circulation model studies which suggest that modeling group tend to have models which behave similarly (Knutti et al., 2013). These studies also complement reanalysis-based studies of the energy and water cycle, showing that there remain issues with the reanalysis datasets (Treberth et al., 2011; Trenberth and Fasullo, 2013).

Here we considered the hypothesis from (Ridley et al., 2014) that Barbados surface concentrations were decreasing over the period 1980-2008 due to decrease in winds in North Africa and thus a decreasing source, which follows previous studies in highlighting the importance of winds on short (Engelstaedter and Washington, 2007; Sun et al., 2001) and long time scales in some source regions (Roe, 2008;McGee et al., 2010). Consistent with that hypothesis, most of the model versions considered here can simulate a decreasing concentration at Barbados, and the models trace this back to a decrease in

dust source in the western Sahel region of North Africa, linked to a decrease in surface winds. This is consistent with the meteorological station data in this region, which show both a correlation between dust sources and wind, as well as a decrease in winds over this time period (Mahowald et al., 2007). Basic meteorological principals suggest associations between winds and precipitation, and the reanalysis models still cannot do a good job simulating mean or variability in





precipitation, suggesting that it will be difficult to improve the dust source, transport and deposition variability without improvements in the reanalyses. Of course, the time period considered here is relatively short, so it is unclear whether other drives might be more important on longer time scales (e.g. Prospero and Lamb, 2003;Mulitza et al., 2010;Mahowald et al., 2010).

Because of the strong variability in dust, model simulations suggest that observations need to be made for around 1 year in many regions, but in remote regions, especially in the Southern Hemisphere, observations need to be made for more than 2 years in order to sample the modeled variability and correctly capture the annual mean concentrations (Figure 16). Of course, this assumes there is no long-term variability. Long time records of dust concentrations represent some of our most important data records to characterize variability in desert dust (e.g. Prospero and Lamb, 2003).

**Acknowledgements**

We acknowledge the support of NSF 0932946 and 1003509 and DOE DE-SC0006735, as well as the assistance of Computational and Information Systems Laboratory of the National Center for Atmospheric Research, whose resources were used for these simulations. DAR and CLH acknowledge support from NASA (NN14AP38G). We thank the PIs of the AERONET stations for access to the AERONET data (Table 2). CMAP Precipitation data provided by the

NOAA/OAR/ESRL PSD, Boulder, Colorado, USA, from their Web site at http://www.esrl.noaa.gov/psd/

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





**Table 1: Description of model simulations considered here.**

| Case name | Model base | Meteorology | Retuned sources | Years available | Citation |
|---|---|---|---|---|---|
| CAM4 (MERRA) | CAM4 | MERRA | Y | 1980-2008 | (Albani et al., 2014) |
| CAM4 (NCEP) | CAM4 | NCEP | Y | 1989-2006 | (Albani et al., 2014) |
| CAM4 (ERAI) | CAM4 | ERA-Interim | Y | 1989-2008 | (Albani et al., 2014) |
| CAM4 (AMIP) | CAM4 | Online/AMIP | Y | 1980-2006 | (Albani et al., 2014) |
| GCHEM (MERRA) | GEOS-CHEM | MERRA | N | 1982-2008 | (Ridley et al., 2014) |
| MATCH (NCEP) | MATCH | NCEP | N | 1982-2008 | (Luo et al., 2003) |
| CAM5 (AMIP) | CAM5 | Online/AMIP | Y | 1990-2008 | (Albani et al., 2014) |
| | | | | | |





**Table 2: Description of observational sites included here.**

| Site | Latitude °N | Longitude °E | Years | Citation or PI for Aeronet |
|---|---|---|---|---|
| **Surface Concentration** | | | | |
| Banizoumbou | 14 | 3 | 2006-2013 | (Marticorena et al., 2010) |
| Barbados | 13 | -59 | 1979-2008 | (Prospero and Lamb, 2003) |
| Bermuda | 32 | -65 | 1989-1997 | (Arimoto et al., 1995) |
| Cinzana | 13 | -6 | 2006-2013 | (Marticorena et al., 2010) |
| Izana | 28 | -16 | 1989-1998 | (Arimoto et al., 1995) |
| Macehead | 53 | -10 | 1989-1994 | (Arimoto et al., 1995) |
| Mbour | 14 | -17 | 2006-2013 | (Marticorena et al., 2010) |
| Miami | 26 | -80 | 1974-1999 | (Prospero, 1999) |
| Midway | 28 | -177 | 1982-2000 | (Prospero and Savoie, 1989) |
| | | | | |
| **AOD** | | | | |
| Bahrain | 26 | 50 | 1998-2006 | B. Holben |
| Dahkla | 23 | -16 | 2002-2003 | H.Benchekroun B. Holben |
| Dalanzadgad | 43 | 104 | 1997-2012 | B. Holben |
| Dhabi | 24 | 54 | 2003-2008 | B. Holben |
| Ilorin | 8 | 4 | 1998-2009 | R. Pinker |
| Sede Boker | 30 | 34 | 1998-2010 | A. Karnieli |
| | | | | |
| **Southern Hemisphere observations** | | | | |
| Rio Gallegos Surface concentrations | -52 | -69 | 2011-2014 | (Zihan, 2016) |
| Kerguelen deposition | -49 | 70 | 2008-2010 | (Heimburger et al., 2012) |




**Table 3: Annual average spatial comparison to observations for different cases (described in Table 1 and Methods). Correlations which are statistical significant at the 95 percentile are in bold.**

| Case | Surface Concentration Correlation (log space) | Deposition (log space) | AOD |
|------|-----------------------------------------------|------------------------|-----|
| CAM4 (MERRA) | **0.73 (0.89)** | **0.94 (0.84)** | **0.73** |
| CAM4 (NCEP) | **0.67 (0.86)** | **0.79 (0.84)** | **0.67** |
| CAM4 (ERAI) | **0.48 (0.81)** | **0.58 (0.76)** | **0.87** |
| CAM4 (AMIP) | **0.79 (0.78)** | **0.73 (0.84)** | **0.41** |
| GCHEM (MERRA) | **0.73 (0.90)** | **0.63 (0.84)** | **0.43** |
| MATCH (NCEP) | **0.83 (0.84)** | **0.43 (0.81)** | 0.32 |
| CAM5 (AMIP) | **0.79 (0.89)** | **0.59 (0.85)** | **0.70** |
|  |  |  |  |





**Table 4: Slope of the normalized annual mean values from 1982 to 2008 (or time period available, shown in Table 1) for the Western Sahel (13 to 22°N and -20 to 13°E) and North Africa (0 to 35°N and -20 to 40°E) and model (Figure 4) (statistically significant values are in bold, standard deviation of slope in parenthesis for Barbados surf. conc. slope) in the first four columns. Values are normalized by dividing by the mean, so that slopes represent relative change per year. The last column is the correlation of interannual variability in precipitation in each model compared to observations.**

| | Slope Barbados surf. conc. | Slope Sahel source | Slope North African source | Slope Sahel Precip. | Correl. With obs. Sahel precip. |
|---|---|---|---|---|---|
| CAM4 (MERRA) | **-0.014 (0.0054)** | **-0.0065** | -0.0005 | 0.0035 | **0.43** |
| CAM4 (NCEP) | -0.0017 (0.016) | **-0.0169** | **-0.0074** | **0.0425** | 0.29 |
| CAM4 (ERAI) | **-0.0058 (0.0051)** | -0.0006 | 0.002 | 0.0008 | **0.81** |
| CAM4 (AMIP) | **-0.0079 (0.0035)** | **-0.0061** | **-0.0037** | **0.0186** | **0.42** |
| GCHEM (MERRA) | **-0.025 (0.0047)** | **-0.021** | **-0.0072** | 0.0035 | **0.43** |
| MATCH (NCEP) | **-0.0087 (0.0059)** | **-0.0047** | 0.027 | **0.0425** | 0.29 |
| CAM5 (AMIP) | -0.01 (0.01) | **0.0027** | 0.029 | | |
| Obs | **-0.016 (0.006)** | | | **0.0089** | |





**Table 5: Variability in Southern Hemisphere**
**Values for the monthly variability (monthly average standard deviation divided by mean) and the fraction of the variability from the seasonal cycle (climatological monthly average standard deviation, divided by monthly mean standard deviation) for the surface concentration in the model cases and data from Rio Gallego and deposition data from Kerguelen (locations listed in Table 1).**

| Model/ Observations | Rio Gallego Surface concentrations | | Kerguelen deposition | |
|---|---|---|---|---|
| | Monthly variability | Fraction variability from seasonal cycle | Monthly variability | Fraction variability from seasonal cycle |
| CAM4 (MERRA) | 5.67 | 0.33 | 0.64 | 0.41 |
| CAM4 (NCEP) | 5.89 | 0.29 | 0.82 | 0.33 |
| CAM4 (ERAI) | 7.52 | 0.30 | 4.39 | 0.43 |
| CAM4 (AMIP) | 8.98 | 0.24 | 4.47 | 0.22 |
| GCHEM (MERRA) | 1.94 | 0.43 | 0.70 | 0.47 |
| MATCH (NCEP) | 2.18 | 0.42 | 0.62 | 0.59 |
| CAM5 (AMIP) | 2.01 | 0.71 | 1.94 | 0.56 |
| Obs | 0.70 | 0.20 | 0.71 | 0.45 |



**Table 6: Regional sources of dust. For each source region, the averaged correlation across time between annual mean source strengths for the CAM-RE cases is shown in the second column. The following columns show the climatological mean source strength (Tg/year) for the mean of the 3 CAM4-RE simulations and the mean of the 7 simulations included in this study. The +/- % standard deviation is also shown, and represents the standard deviations across the models included in the averaging. The regions are defined as follows: Australia: 35 to 25°S, 130 to 150°E; East Asia: 35 to 50°N, 70 to 112°E; Middle East: 10 to 45°N, 40 to 70°E; North Africa: 10 to 35°N, 40°W to 40°E; Sahel (western): 13 to 22°N, 40°W to 13°E; South Africa: 35 to 20°S, 15 to 40°E; South America (Argentina): 55 to 35°S, 285 to 310°E.**

|  | Avg. IAV Temporal Correlation | Mean CAM4-RE | Mean all |
|---|---|---|---|
| Australia | 0.73 | 25 +/-70% | 38 +/- 90% |
| East Asia | 0.58 | 230+/- 70% | 230 +/-70% |
| Middle East | 0.40 | 570+/-40% | 510 +/- 40% |
| North Africa | 0.47 | 1370 +/-23% | 1490 +/- 40% |
| North America | 0.78 | 72 +/-160% | 70 +/-130% |
| Sahel (western) | 0.13 | 460 +/-27% | 520 +/- 40% |
| South Africa | 0.46 | 9 +/- 90% | 8 +/- 70% |
| South America | 0.68 | 14 +/-160% | 34 +/- 130% |
| Globe | 0.49 | 2400 +/-26% | 2500 +/- 40% |



**Table 7: Correlations in meteorological variables and mobilization in different regions for IAV. Time series are correlated for the annual average over 1990-2005 in each region (only including gridboxes which are active at any time in that model simulation). Values shown are the averages of the correlations across the CAM4-Reanalysis models (CAM4 (MERRA), CAM4 (NCEP) and CAM4 (ERAI)). The regions are defined as in Table 6.**

| | Precipitation. | Soil moisture | Leaf Area Index | Sfc. Wind |
|---|---|---|---|---|
| Australia | -0.59 | -0.61 | -0.72 | 0.10 |
| East Asia | 0.06 | -0.08 | -0.32 | 0.67 |
| Middle East | -0.32 | -0.33 | -0.28 | 0.36 |
| North Africa | -0.27 | -0.26 | -0.20 | 0.51 |
| North America | -0.57 | -0.64 | -0.53 | 0.32 |
| Sahel (western) | -0.28 | -0.26 | -0.42 | 0.81 |
| South Africa | -0.37 | -0.38 | -0.55 | 0.22 |
| Globe | -0.36 | -0.29 | -0.46 | 0.33 |



**Table 8: Deposition into ocean basins. For each ocean region, the averaged correlation across time between annual mean deposition fluxes for the CAM-RE cases is shown in the second column. The following columns show the climatological mean deposition flux (Tg/year) for each model simulation. Regions are defined as the ocean gridboxes (not including sea ice or land boxes) in the following latitude and longitude areas as from (Gregg et al., 2003): North Atlantic (>30°N; 270 to 30°E); North**

5  **Pacific (>30°N; 120 to 270°E); North Central Atlantic (10 to 30°N, 270 to 30°E); North Central Pacific (10 to 30N; 120 to 270°E); North Indian (10 to 30°N; 30 to 120°E); Equatorial Atlantic (-10 to 10°N; 300 to 30°E); Equatorial Pacific (-10 to 10°N; 120 to 285°E); Equatorial Indian (-10 to 10°N; 30-120°E); South Atlantic (-30 to -10°N; 30 to 300°E); South Pacific (-30 to -10°N; 120 to 295°E); South Indian (-30 to -30°N, 30 to 120°E); Antarctic (<-30°N).**

| | Correlation | Mean CAM4-Re | Mean all |
|---|---|---|---|
| North Atlantic | 0.68 | 54 +/-7% | 61 +/-17% |
| North Pacific | 0.50 | 30 +/- 30% | 30 +/- 60% |
| North Central Atlantic | 0.66 | 91 +/- 8% | 120 +/- 30% |
| North Central Pacific | 0.33 | 20 +/- 90% | 20 +/- 90% |
| North Indian | 0.10 | 100 +/- 30% | 110 +/- 40% |
| Equatorial Atlantic | 0.39 | 77 +/- 20% | 80 +/- 35% |
| Equatorial Pacific | 0.33 | 4 +/- 100% | 6 +/- 130% |
| Equatorial Indian | 0.31 | 23 +/- 25% | 21 +/-23% |
| South Atlantic | 0.16 | 3+/-60% | 3 +/- 90% |
| South Pacific | 0.62 | 1 +/- 40% | 3 +/- 60% |
| South Indian | 0.38 | 2 +/- 40% | 4 +/- 75% |
| Antarctic | 0.35 | 15 +/- 100% | 30 +/- 100% |
| Global | 0.45 | 450 +/- 10% | 510 +/- 24% |





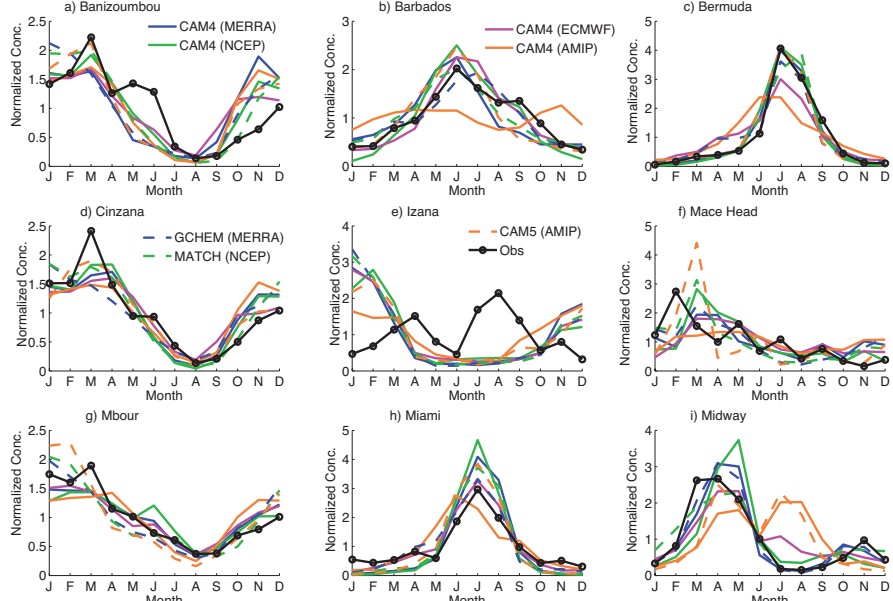

**Figure 1: Climatological monthly average time series of dust concentration divided by the mean of each time series (unitless) at stations a) Banizoumbou, b) Barbados, c) Bermuda, d) Cinzana, e) Izana, f) Macehead, g) Mbour, h) Miami, i) Midway. Observations are in black and are from the locations and citations in Table 2. Different colors and line styles indicate the different model versions: CAM4 (MERRA) (blue solid), CAM4 (NCEP) (green solid), CAM4 (ERAI) (pink solid), CAM4 (AMIP) (orange solid), GCHEM (MERRA) (blue dashed), MATCH (NCEP) (green dashed), CAM5 (AMIP) (orange dashed). The monthly means are divided by the long term annual mean to allow comparison with interannual variability, since they are similarly normalized (Figure 3).**





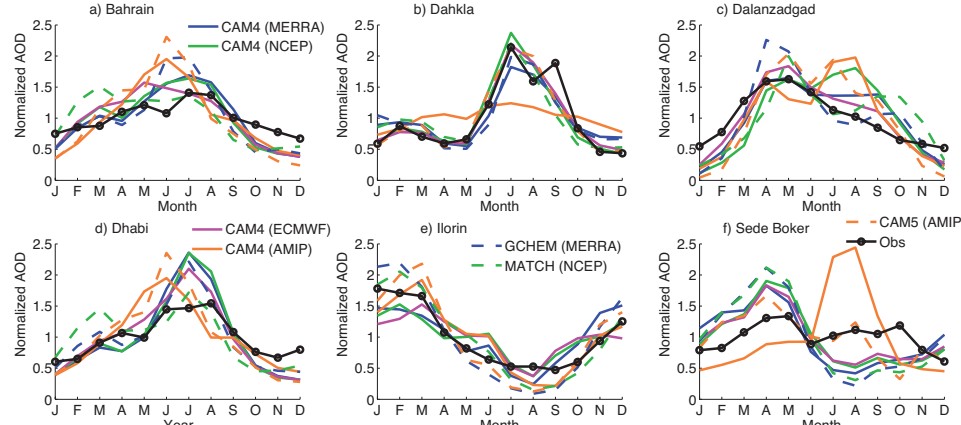

**Figure 2: Climatological monthly average aerosol optical depth (AOD) for model simulations (based on dust only), compared with AERONET observations for: a) Bahrain b) Dahkla c) Dalanzadgad, d) Dhabi, e) Ilorin and f) Sede Boker for each of the different model versions (Colors and information are the same as in Figure 1, but for the AOD). Observational data from AERONET stations (citations and locations listed in Table 2). The monthly means are divided by the long term annual mean to allow comparison with interannual variability, since they are similarly normalized (Figure 4).**





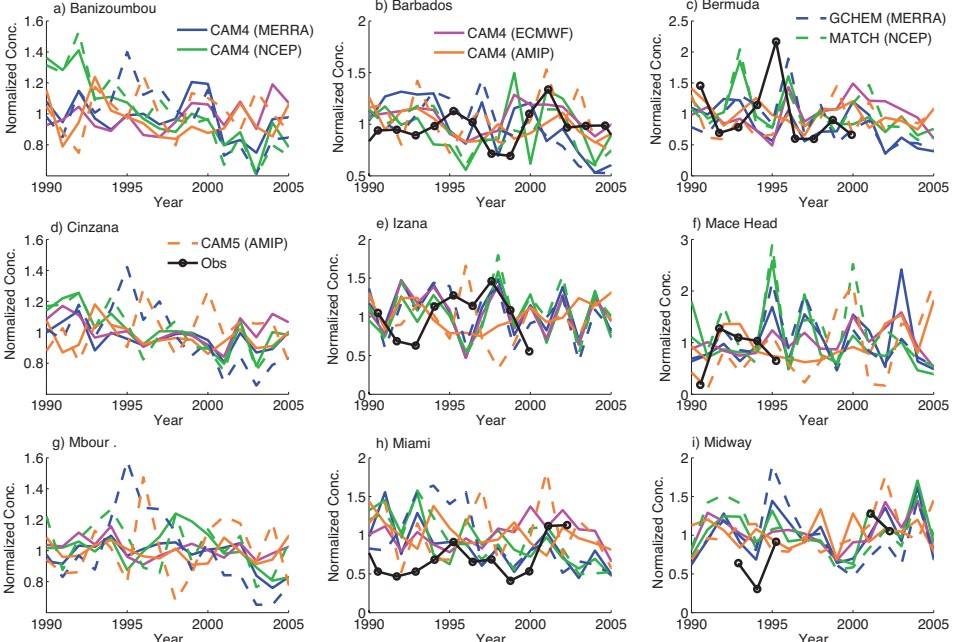

**Figure 3: Time series of annual mean dust concentration divided by the long term mean for each time series (unitless) at stations in Table 2: a) Banizoumbou, b) Barbados, c) Bermuda, d) Cinzana, e) Izana, f) Macehead, g) Mbour, h) Miami, i) Midway. Observations are in black: if no observations shown, observations are from a different time period, and only used for variability and seasonal cycle calculations. Different colors and line styles indicate the different model versions: CAM4 (MERRA) (blue solid), CAM4 (NCEP) (green solid), CAM4 (ERAI) (pink solid), CAM4 (AMIP) (orange solid), GCHEM (MERRA) (blue dashed), MATCH (NCEP) (green dashed), CAM5 (AMIP) (orange dashed). The annual means are divided by the long term mean to allow comparison with seasonal variability, since they are similarly normalized (Figure 1).**





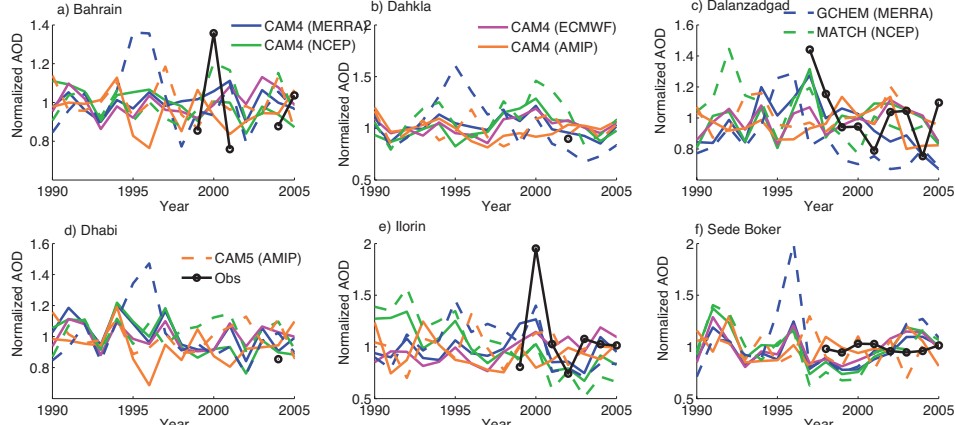

**Figure 4: Timeseries of annual average AOD for model simulations (based on dust only), compared with AERONET observations in Table 2 for: a) Bahrain b) Dahkla c) Dalanzadgad, d) Dhabi, e) Ilorin and f) Sede Boker for each of the different model versions (Colors are the same as in Figure 3). Observational data from AERONET stations (citations listed in Table 2). The annual means**
5 **are divided by the long term mean to allow comparison with seasonal variability, since they are similarly normalized (Figure 2).**





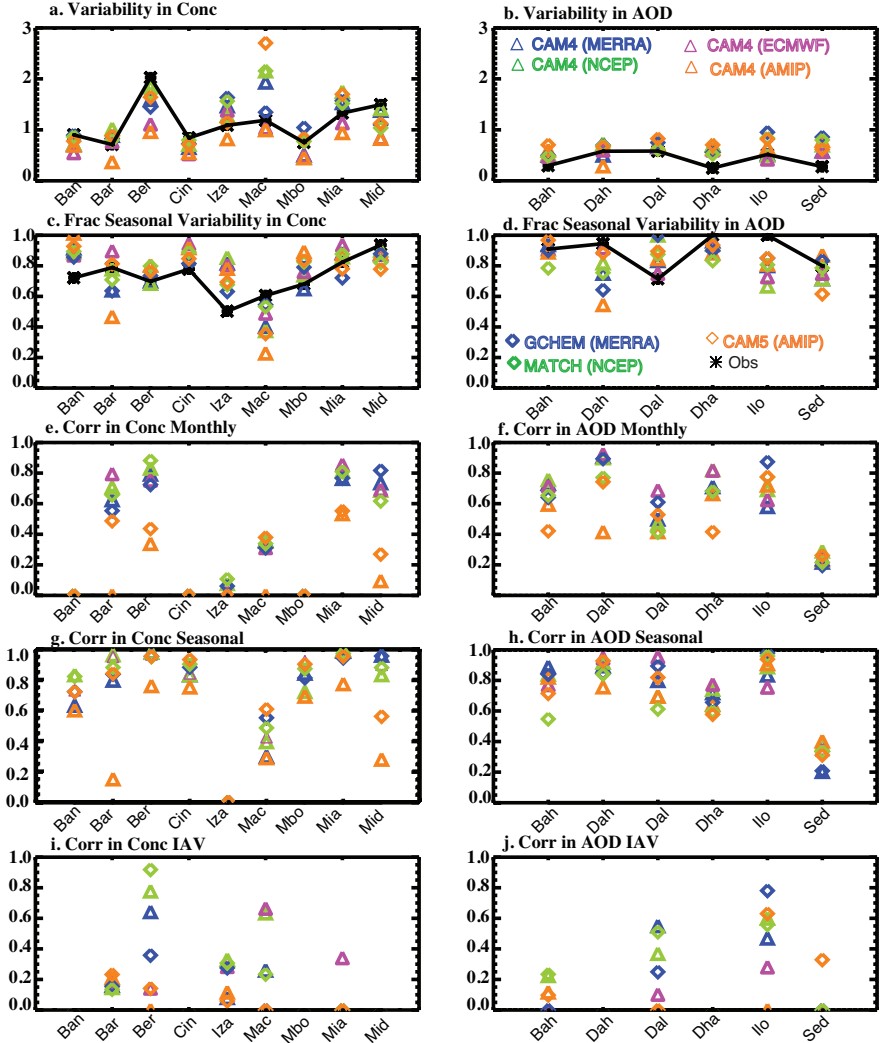

**Figure 5: Model and observed variability for (a) the concentration and (b) AOD from the observation sites listed in Table 2 and shown in Figures 1-4. Variability is defined as the standard deviation over the mean using monthly means. The fraction of the total variability (across all monthly means) from the seasonal cycle (12 month climatology) is shown for (c) concentration and (d) AOD. Model results are in color, while observations are in black. Correlation coefficients for monthly mean time series for (e) concentration and (f) AOD between the models and observations at the stations. The correlation coefficients between model and observed values for the seasonal cycle (g and h) and the interannual variability (i and j) for concentration and AOD. Observations are described in Table 2. Concentration stations are abbreviated: Ban: Banizoumbou; Bar: Barbados; Ber: Bermuda; Cin: Cinzana; Iza: Izana; Mac: Macehead; Mbo: Mbour; Mia: Miami; Mid: Midway. AOD stations abbreviated: Bah: Bahrain; Dah: Dahkla; Dal: Dalanzadgad; Dha: Dhabi; Ilo: Ilorin and Sed: Sede Boker**



**Figure 6: Slope of the trend in the relative concentrations (linear regression of concentration onto time) of the modeled annual mean surface concentrations (normalized by the mean) in units of fraction change per decade for a) CAM4 (MERRA), b. CAM4 (NCEP), c. CAM4 (ERAI), d. CAM4 (AMIP), e. GCHEM (MERRA), f. MATCH (NCEP), g. CAM5 (AMIP) and the mean slope across the models (h). Only slopes that are larger in magnitude than one sigma from the regression are plotted. Positive slopes imply increasing concentrations.**





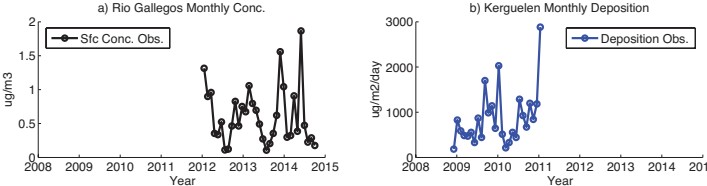

**Figure 7:** **Monthly mean surface concentration observations at the Rio Gallegos site in Argentina (a) in µg/m$^3$ (Zihan, 2016) and monthly mean deposition fluxes in µg/m$^2$/day from Kerguelen (Heimburger et al., 2012) (b).**





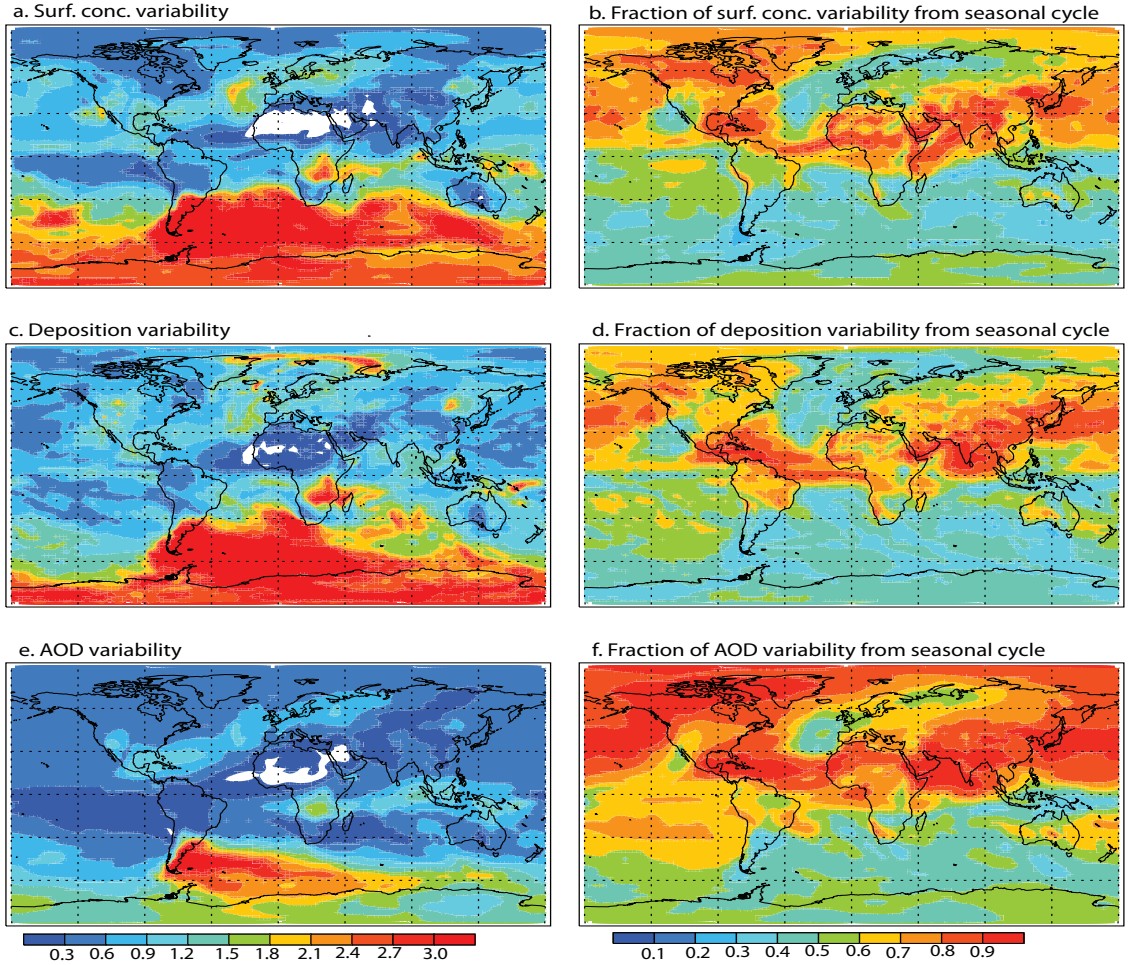

**Figure 8: Spatial plot of the modeled variability in the model simulations at each grid box (mean of CAM4 (MERRA), CAM4 (NCEP) and CAM4 (ERAI)), where variability is unitless and is the standard deviation divided by the mean of the monthly mean between 1990 and 2005 for a) surface concentration, c) deposition and e) AOD.  The fraction of the monthly mean variability which comes from the seasonal variability (calculated using the 12 climatological monthly means) is shown in the right hand panel for b) surface concentration, d) deposition and f) AOD.**





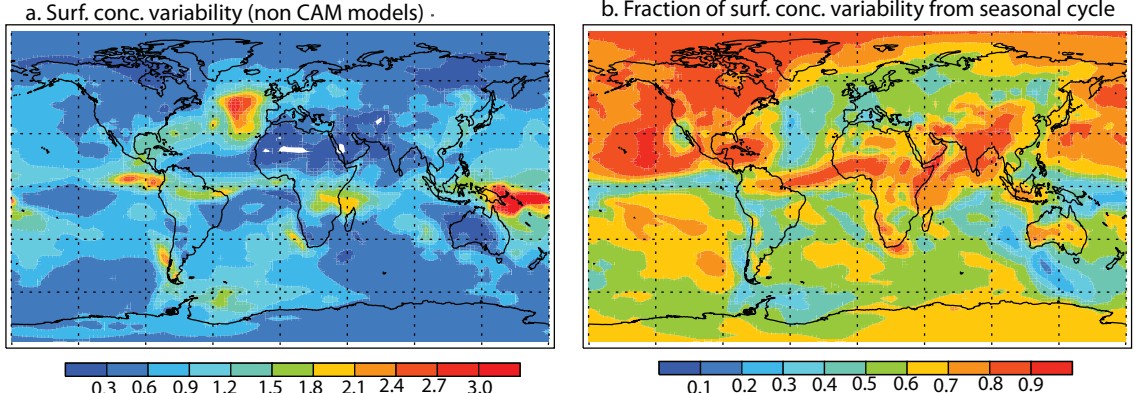

**Figure 9: Spatial plot of the modeled variability (as in Figure 8) in the non-CAM4 or CAM5 models (mean of GCHEM (MERRA) and MATCH (NCEP)), where variability is the standard deviation divided by the mean of the monthly mean between 1990 and 2005 for a) surface concentration, and b) the fraction of the monthly mean variability which comes from seasonal variability (calculated using the 12 climatological monthly means).**





**Figure 10: Spatial plot of the temporal rank correlation of the monthly mean modeled surface concentration (12 months for 16 years) in the CAM4 (MERRA) case compared to each of the other model simulations: a) CAM4 (NCEP), b) CAM4 (ERAI), c) CAM4 (AMIP), d) GCHEM (MERRA), e) MATCH (NCEP) and f) CAM5 (AMIP). At each grid-box the 192 months (12 months times 16 years) time series are correlated between the two model versions, and the color indicates the value of the correlation.**





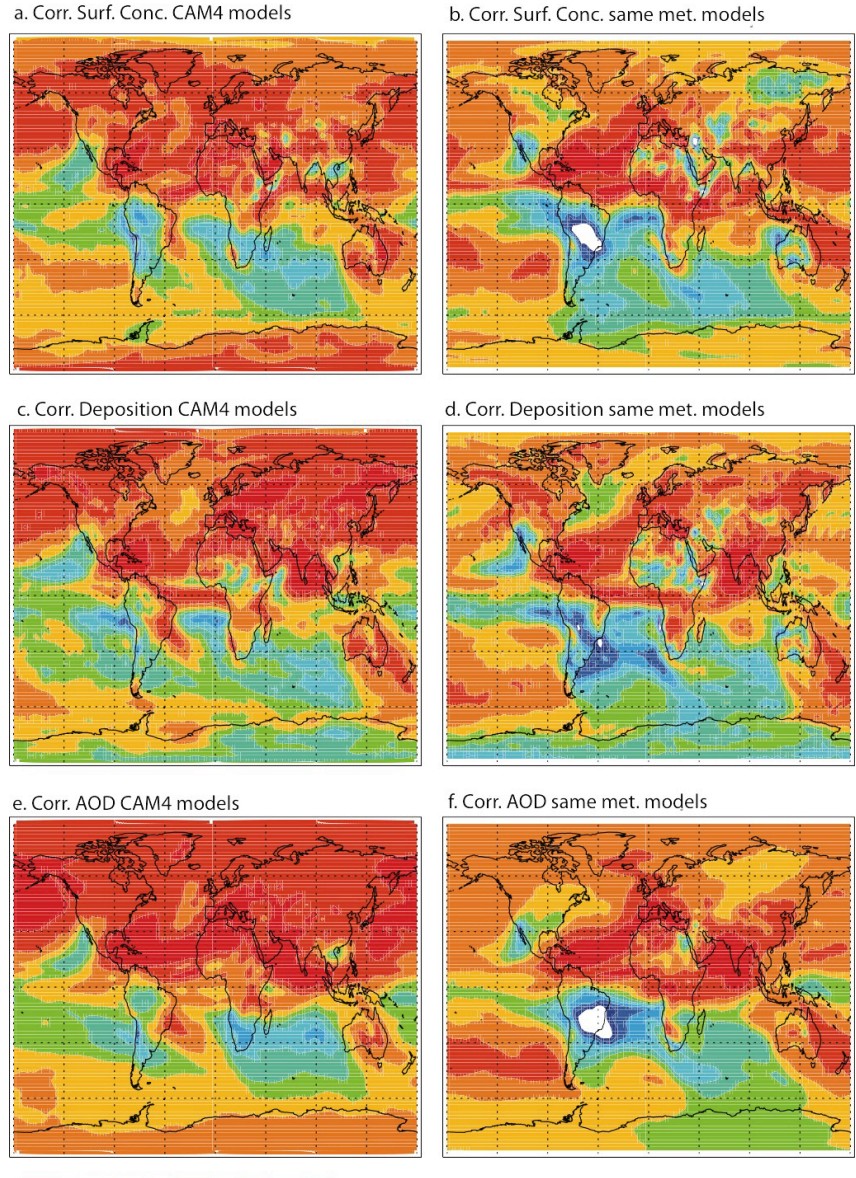

**Figure 11: Spatial plot of the temporal rank correlation of the monthly mean modeled values for the average correlation between the CAM4-Reanalysis models (CAM4 (MERRA); CAM4 (NCEP) and CAM4 (ERAI); left hand column: a, c and e) and the models driven by the same meteorology (average of CAM4 (MERRA) vs. GCHEM (MERRA) and CAM4 (NCEP) vs. MATCH (NDEP); right hand column: b, d and f) for surface concentration (a and b), deposition (c and d) and AOD (e and f). At each grid-box the 192 months (12 months times 16 years) time series are correlated between the models, and the color indicates the value of**





the correlation.

a. Corr. of surf. conc. seasonal cycle

b. Corr. of surf. conc. IAV

c. Corr. of deposition seasonal cycle

d. Corr. of deposition IAV

e. Corr. of AOD seasonal cycle

f. Corr. of AOD IAV

0.1  0.2  0.3  0.4  0.5  0.6  0.7  0.8  0.9

**Figure 12: Spatial plot of the temporal rank correlation of the modeled values for the average correlation between the CAM4-
Reanalysis models (CAM4 (MERRA); CAM4 (NCEP) and CAM4 (ERAI)) for the seasonal cycle (1990-2005 averaged 12 months;
left hand column: a, c and e) and the interannual variability using the annual mean for each year (right hand column: b, d and f)**





for surface concentration (a and b), deposition (c and d) and AOD (e and f).

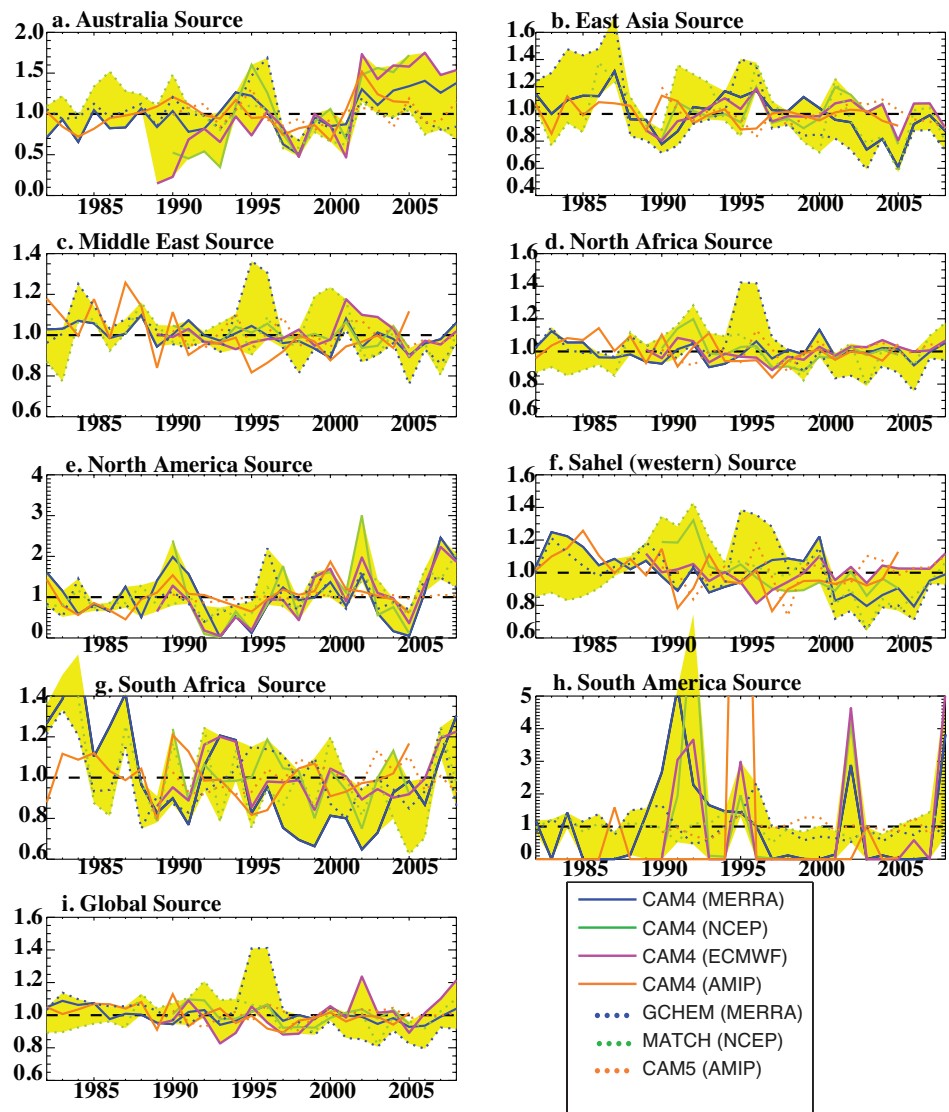

**Figure 13: Time series of the annual mean source strength in different regions as simulated in the different model versions (different colors, as in legend). The regions are defined as: Australia (130 to 150°E, -35 to -25°N), East Asia (80 to 112°E, 35 to 50°N), Middle East (40 to 70°E, 10 to 45°N), North Africa (-20 to 40°E, 10 to 35°N), North America (235 to 265°E, 25 to 40°N), Sahel (western) (-20 to 13°E, 13 to 22°N), South Africa (15 to 40°E, -35 to -20°N), South America (285 to 310°E, -50 to -30°N). All time series are normalized by the climatological mean (Table S8) in order to focus on interannual variability. The yellow highlighted area is the area encompassed by the 5 reanalysis-based simulations (CAM4 (MERRA), CAM4 (NCEP), CAM4 (ERAI), GCHEM (MERRA), MATCH (NCEP)).**





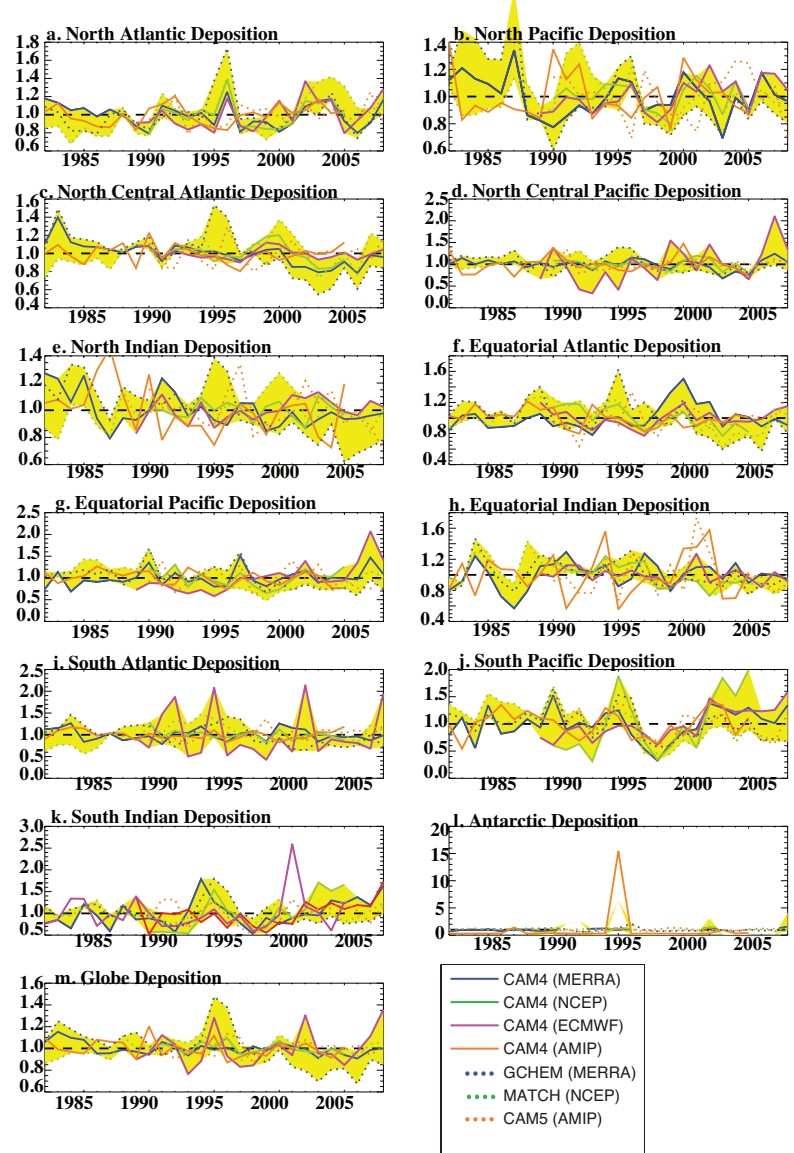

**Figure 14: Time series of the annual mean deposition flux in different ocean regions as simulated in the different model versions (different colors, as in legend). All time series are normalized by the climatological mean in order to focus on interannual variability (shown in Table S9). The yellow highlighted area is the area encompassed by the 5 reanalysis-based simulations (CAM4 (MERRA), CAM4 (NCEP), CAM4 (ERAI), GCHEM (MERRA), MATCH (NCEP)). The ocean basin areas are defined as over ocean in the regions in Table S8 from (Gregg et al., 2003).**





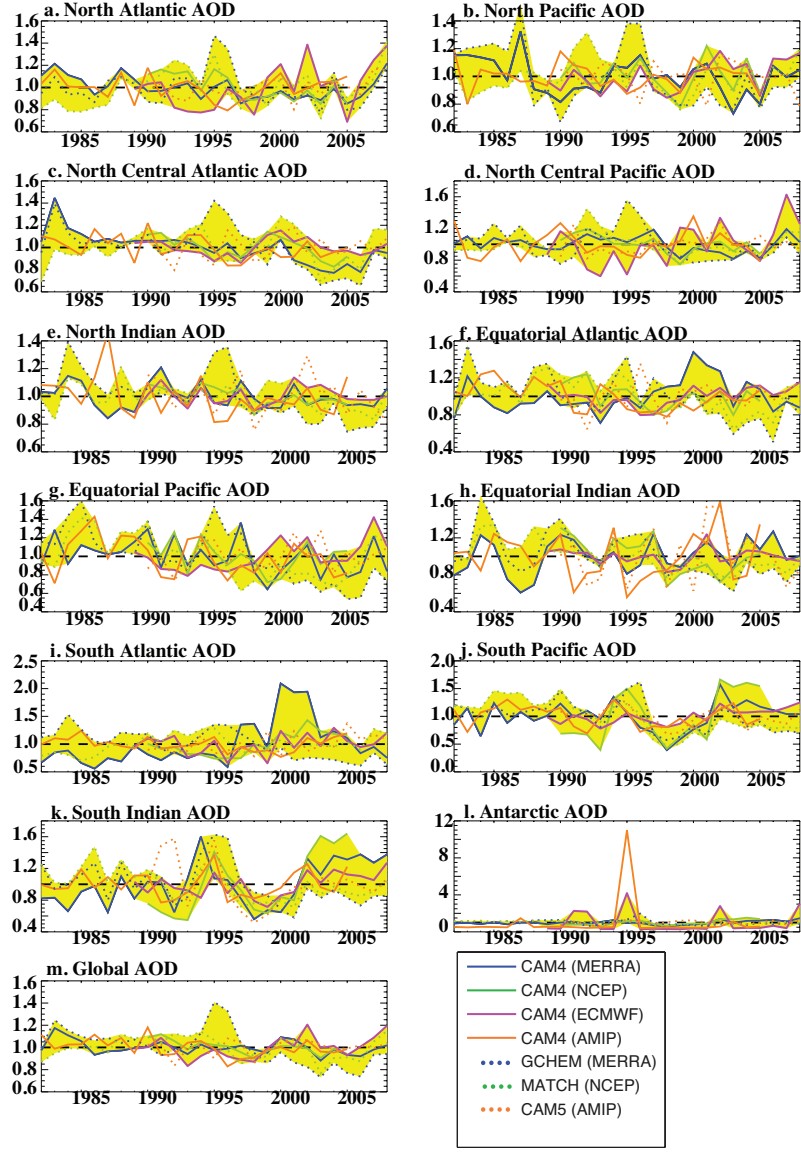

**Figure 15: Time series of the annual mean Aerosol Optical Depth (AOD) in different ocean regions as simulated in the different model versions (different colors, as in legend). All time series are normalized by the climatological mean (Table S10) in order to focus on interannual variability. The yellow highlighted area is the area encompassed by the 5 reanalysis-based simulations (CAM4 (MERRA), CAM4 (NCEP), CAM4 (ERAI), GCHEM (MERRA), MATCH (NCEP)). The ocean basin areas are defined as over ocean in the regions in Table S8 from (Gregg et al., 2003Gregg et al., 2003).**





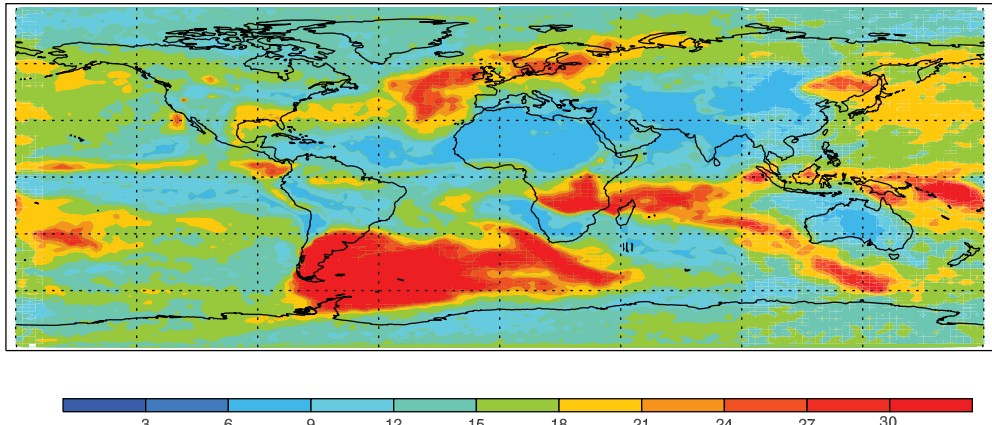

**Figure 16: Monte Carlo-based estimation of the number of monthly mean observations required to obtain a 95% chance of obtaining a mean and standard deviation consistent with the true model mean between 1990-2005 at each gridbox, based on model simulations using the CAM4 (MERRA) model. More details on methods in Section 2.3.**

