# Peer review of "Sensitivity of the Interannual Variability of Mineral Aerosol Simulations to Meteorological Forcing Dataset"

_Atmospheric Chemistry and Physics, 2016_

## Referee Comment (RC1) · Anonymous Referee #2 · 26 Sep 2016

General comments:

The manuscript addresses the highly relevant issue of the representation of the variability of mineral dust in state-of-the-art global model simulations. The study is pursued using a set of simulations including runs with the global model CAM4/5 driven by different boundary conditions and two further other global models. The article is generally well written, and the number and quality of tables and figures is good.

What disappoints is the simplistic and superficial analysis of different correlation parameters without relating them in detail to the varying surface characteristics and meteorological drivers in specific geographic regions, and to global atmospheric circulation patterns (NAO, ENSO, etc.). In the presentation of results, there is no discrimination

in the contribution of different desert regions to local dust conditions, and explanations of processes behind the correlations are missing. As an example, the link between precipitation and dust emission in the Sahel is left to the reader's interpretation. The contribution of Asian dust sources to the northern hemispheric dust cycle seems underrepresented in the discussion. A more detailed description and explanation of the figures would be desirable.

Specific comments:

1. Page 2: The Introduction lacks a brief overview of what is already known about source strength, seasonal and inter-annual variability of dust emissions, as well as their main drivers for important dust source regions. What is expected to be seen in the model results? An outline of the paper at the end of the Introduction would be helpful.

2. Page 3, line 18: Could you provide the scale factors here?

3.Page 6, lines: 9-11: I might have overseen this but I could not find the results in the manuscript. So, please omit the paragraph or present the other species in the text. 4.Page 6, lines: 23-24: You say that only sites where more than 50% of the modelled AOD was from dust are included in this comparison. To me it seems that due to this filtering the comparison is not independent. Would not it be better to compare observed coarse-mode AOD with modelled dust AOD or to use the angstrom coefficient to filter dust events? There is no further description of the filtering below. Which wavelength has the AOD used? For an overview it would be good to have the location of observation sites on a map.

5. Page 7, lines 26-28: I do not understand why these two records have been chosen, which have a much shorter and inconsistent time coverage, while considerably less but still several AERONET observations are available in the southern hemisphere.

6. Page 9, lines 5-23: The analysis is too general. At least it should be clearly described which desert contributes to the dust load at the individual sites, and then the variability and relevant changing atmospheric and/or surface conditions have to be evaluated respectively. I do not see why for the AODs you are limited to regions close to source regions (line 20).

7. Page 10, lines 12-13: "[. . .] insight into how much of the variability in dust is driven by SSTs." Can this be quantified any further?

8. Page 10, line 17: Here and elsewhere in the text, even though it is obvious, please mention the dust source affecting the geographic region under consideration.

9. Page 10, lines 29-32: Could also vegetation-related changes in surface roughness explain the trend in the Sahelian dust emissions as discussed by Cowie et al. (GRL, 2013), and would these changes be considered by any of the models used? What exactly are the mechanisms behind these correlations?

10. Page 11, line 13: Correct: "[. . .] supporting the idea that the source strength decreased within the time period 1990 – 2005."

11. Page 11, lines 29-31: Here, you may include a study by, e.g., Marsham et al. (GRL, 2011) who showed that coarse-resolution models usually fail to reproduce dust emission events related to moist convection.

12. Page 12, lines 1-27: The database for the analysis in the southern hemisphere is too small to be statistically significant. Problematic is also the shorter and, in particular, inconsistent time coverage of the measurements. Moreover, the two sites are affected by mineral dust from different sources, which is not discussed at all. You could easily broaden your database by using AERONET sun photometer observations.

13. Page 15, lines 15-17: The 'global average' is not a larger source. Could you please specify the smaller sources meant?

Technical corrections:

1. Page 2, line 8: Do you mean "4-fold fluctuations"? 2. Page 5, line 26: Wording: "[. . .] it [the model] can only be operated using reanalysis datasets, [. . .]". 3. Page 7, Eq. 1: A proper equation would look more scientific. 4. Page 11: IAV needs to be resolved. 5. Page 18: Correct: "whether other drivers".

---

## Referee Comment (RC2) · Anonymous Referee #1 · 30 Sep 2016

The authors compare several model frameworks, including different forcing data, different forcing methods and different models, to estimate the sensitivity in the seasonal and interannual variability of dust emission and transport. They find that the correlation in variability is roughly similar between simulations using different meteorological forcings and different models, while simulations forced by sea surface temperature only do not perform as well.

Considering the high uncertainty in dust simulations, investigating the sensitivity to different forcing data sets and models is an important contribution to the field of dust modelling. However, the manuscript presents a lot of information that is not necessarily relevant and that dilutes the actual results of the study. In addition, several parts of the

methodology and results are unclear, which make the interpretation further confusing.

The manuscript thus likely need to be rewritten before it can be considered for publication in Atmospheric Chemistry and Physics.

General comments

A. The model description is often confusing and strongly needs to be reorganized and clarified.

B. The discussion of the results jumps between a large number of figures and tables split between the manuscript and the supplement, which need to be reduced.

C. A significant part of the results further appears not to be significant, thus the discussion needs to be more focused to emphasize the actual contribution of the manuscript.

Specific comments

1. Introduction

p.2 l.7-15: this part is disconnected from the previous one and needs to be separated into a new paragraph; the different space and time scales and their relevance are confusing and need some hierarchy; what are "4x fluctuations"? p.2 l.15: this is partially redundant with the following lines and should be reorganized. p.2 l.21: this is obvious and could be more specific (interannual variability of dust emissions?). p.2 l.28-29: the purpose of the sentence is unclear. p.2 l.31: a short paragraph to describe the contents of the paper would be helpful at the end of the introduction.

2. Methodology

p.3 l.5-10: the meaning of "model" is confusing here between CESM, CAM, GCM... A description of the nature of these "models" would be helpful. p.3 l.12: "when the winds are strong" needs to be clarified. p.3 l.15: "as described in more detail elsewhere" needs to be rephrased. p.3 l.25-27: the purpose of the sentence is unclear. p.3 l.31-p.4 l.7: the methodology is unclear. Which variables are nudged? What about the

wind? What is the MATCH/CAM approach? Is MATCH used here? p.4 l.8-11: some details about the data are needed e.g. which reanalysis is used (NCEP and ECMWF are forecast centres). p.4 l.11-13: the purpose of the sentence is unclear; discuss conclusions of these studies? p.4 l.18-20: more details are needed, e.g. how the model is initialized and what is computed in the atmospheric model (CAM?) exactly (winds cannot be computed alone). p.4 l.20-22: this is unclear; what is the "correct input format"? p.4 l.31 onwards: the paragraph should be moved and merged with the description of AMIP above. p.4 l.34: why is there "no inconsistency between the atmospheric model and observations"? p.5 l.9-10: the purpose of the sentence is unclear. p.5 l.11-21: the contrast with CAM needs to be better emphasized. p.5 l.30 onwards: this should probably appear earlier, as MATCH is referred to in the previous section. p.6 l.9-11: this appears disconnected from the other paragraphs. p.7 l.5-7: the definitions are confusing. p.8 l.1-4: this is very confusing; does it mean that the method is wrong?

3. Results and Discussion

p.8 l.8: does it mean that the simulations are taken from previous studies? p.8 l.11: should it be S3? p.8 l.24-25: the sentence is vague. p.9 l.1: which data? The purpose of the sentence is unclear. p.9 l.5 onwards: there is a lot of information presented in many figures and tables split between the main manuscript and the supporting information, which makes the discussion very difficult to follow; the information needs to be presented in a concise way and the discussion better structured. p.10 l.17-32: this should be moved to the introduction. p.11: the whole discussion is confusing (is there a trend or not?) and its purpose is unclear, as it is not related neither to the monthly nor to the interannual variability. p.12 l.7-19: CAM4 exhibits much higher monthly variability than the other models. p.12 l.19-21: models driven by MERRA and NCEP are clearly better than ERAI and AMIP. p.12 l.25-27: the statement is weak. p.13 l.1-25: the results do not appear robust. p.13 l.28-31: this should be moved to the methods. p.13 l.33-p.14 l.2: are the AMIP simulations meaningful then? p.15 l.8-10: is Figure 13

relevant then? p.16 l.28: should this be Figure 16? Are Figure 14 and 15 needed at all? p.16 l.28-30: isn't it obvious that a whole year of data is required to estimate the annual mean?

4. Conclusions

p.17 l.11-12: this again requires the set-up of AMIP simulations to be better explained and asks if they are meaningful. p.17 l.25 onwards: the results do not appear that clear in section 3.

---

## Author Comment (AC1) · 23 Dec 2016

We would like to thank Referee #2 for the suggestions for improvements, as we think these will improve the quality of the manuscript.  We try to incorporate all the comments of the reviewer, as discussed below, and try to reconcile these comments with the sometimes contradictory comments by the other reviewer. Reviewer comments are in black, responses in blue.

Reviewer 2 feedback:

General comments

The manuscript addresses the highly relevant issue of the representation of the variability of mineral dust in state-of-the-art global model simulations. The study is pursued using a set of simulations including runs with the global model CAM4/5 driven by different boundary conditions and two further other global models. The article is generally well written, and the number and quality of tables and figures is good.

What disappoints is the simplistic and superficial analysis of different correlation parameters without relating them in detail to the varying surface characteristics and meteorological drivers in specific geographic regions, and to global atmospheric circulation patterns (NAO, ENSO, etc.). In the presentation of results, there is no discrimination in the contribution of different desert regions to local dust conditions, and explanations of processes behind the correlations are missing.

The main focus of this paper is to address the question of how sensitive the simulation of dust interannual variability is to the reanalyzed meteorology used, or alternatively, the modeling framework used, and to examine long term trends in global dust variability. Other studies have examined the mechanics by which meteorology drives changes in dust mobilization, transportation, and deposition, and we are not trying to repeat that interesting work here.

We think some of these comments are due to a misunderstanding of the point of the paper, so we add more justification of the goals of the paper in the introduction and change the title of the paper to focus on IAV, and the sensitivity to the reanalyses dataset (not to meteorology). However, we have expanded our analysis to discuss the relation of dust interannual variability to NAO and ENSO.

Note that the comments of this reviewer to have more detail, contradicts the comments of the other reviewer who wants less details, so we have focused the paper more on the coherence of model simulations of IAV in surface concentrations in order to accommodate the other reviewer. Expanding this paper further would make it unwieldy, and we encourage the reviewer or other authors to conduct further studies to look at whether specific mechanisms are driving the coherence of the model results driven by different datasets.

The new introduction section should explain this better:

"It is well established in the dust literature that meteorology and surface conditions play central roles in driving changes in dust emissions, primarily from changes in precipitation, winds, surface roughness or vegetation cover on daily to interannual to geological time scales (e.g. Westphal et al., 1987; Petit et al., 1999; Marticorena and Bergametti, 1996; Mahowald et al.,

2002; Prospero and Lamb, 2004; Engelstadter and Washington, 2007; Engelstaedter et al., 2003; Roe, 2008; McGee et al., 2010; Knippertz and Todd, 2012; Cowie et al., 2013; Yu et al., 2015), and we do not seek to repeat or review that work here. Rather, we address the question of how sensitive our simulation of interannual variability is to the meteorology, or alternatively, the modeling framework used. While previous modeling studies have evaluated the annual mean and seasonal cycle coherence across dust models (Huneeus et al., 2011) or contrasted specific models (Luo et al., 2003), prior studies exploring the ability of models to simulate interannual variability and the role of different meteorological mechanisms have used only one model (e.g. Miller and Tegen, 1998b;Mahowald et al., 2002;Mahowald et al., 2003;Ginoux et al., 2004;Ridley et al., 2014). Thus an open question remains of how robust our simulations of inter-annual variability are, and how sensitive they are to the meteorological dataset versus the modeling framework. Characterizing how well IAV is simulated in models allows a better understanding of how much we should trust model output in studies directed at understanding the role of dust IAV in contributing to IAV in total aerosol AOD variability (e.g. Streets et al., 2009) or in dust impacts on ocean biogeochemistry IAV (e.g. Doney et al., 2009). A related question is how much observational data do we need in order to correctly characterize the mean dust amount, based on how much interannual variability we think exists in different locations."

As an example, the link between precipitation and dust emission in the Sahel is left to the reader's interpretation.

There is substantial discussion of the role of precipitation and dust emission in both the introduction and the Sahel source trend section, and we add more of this discussion.

The contribution of Asian dust sources to the northern hemispheric dust cycle seems underrepresented in the discussion.

The model results shown here are compared to the available observations, and seem to support the dust strength used here for the Asian sources (we include Figures S1-S6, which were previously excluded to streamline the paper to show this better). Perhaps the reviewer means that we were unable to find observations isolating the impacts of these regions, so that we did not discuss them as in depth as the North African sources, which is true, and an unfortunate fact that these sources are difficult to isolate due to other aerosols and a lack of data. There are so many other aerosols that these areas are difficult to obtain high quality information to discuss. Note also that our study focuses on the time period 1990-2005, and there is more data covering other time periods.

A more detailed description and explanation of the figures would be desirable.

We have tried to both simplify the number of figures as well as improve the figure captions, especially as noted below in detail by the reviewer.

Specific comments:

1. Page 2: The Introduction lacks a brief overview of what is already known about source strength, seasonal and inter-annual variability of dust emissions, as well as their main drivers for important dust source regions.

As indicated above, we have some details about important mechanisms to the introductory paragraph above, although perhaps not in the detail the reviewer requests, because our goal is not to repeat these studies NOR to detail all causes of IAV, but rather to see how robust model results are for IAV.

What is expected to be seen in the model results? An outline of the paper at the end of the Introduction would be helpful.

These are good suggestions, and we have added the requested information at the end of introduction:

"In order to simplify the paper, we will focus on IAV in surface concentrations, and provide information on how deposition and aerosol optical depth contrast with surface concentration variability. Section 2 describes the methods used in the study, including briefly describing the models, data and comparison metrics. Section 3 describes the results of the study, starting with comparison to observations, comparison between different model simulations, and the implication for observational needs."

2. Page 3, line 18: Could you provide the scale factors here?

The scale factors are different for each model and are really much more detail than needed here. The important factor is where the model was enhanced substantially for the rest of the discussion, which comes in the following sentence. More details are provided in Albani et al., 2014:

"For example, in most versions of the model the tuning reduced dust source strength over Central Asia and the Atacama Desert, and increased dust source strength over Argentina (Albani et al., 2014), which becomes important when analyzing dust IAV, as discussed in the results sections (Section 3)."

3.Page 6, lines: 9-11: I might have overseen this but I could not find the results in the manuscript. So, please omit the paragraph or present the other species in the text.

We move this paragraph to be within the CAM4 discussion, in response to the other reviewer, and include a reference to the section where it is used:

"For two of the model simulations used here (CAM4 (MERRA) and CAM4 (NCEP)), additional aerosol species were available for analysis (Sea salts, black carbon (BC), organic carbon (OC), and sulfate (SO4)), and we use these to contrast the correlations between dust aerosols and other aerosols for a sensitivity study, referenced in Section 3.3."

4.Page 6, lines: 23-24: You say that only sites where more than 50% of the modelled AOD was from dust are included in this comparison. To me it seems that due to this filtering the comparison is not independent. Would not it be better to compare observed coarse-mode AOD with modelled dust AOD or to use the angstrom coefficient to filter dust events? There is no further description of the filtering below. Which wave- length has the AOD used? For an overview it would be good to have the location of observation sites on a map.

We add the location of the detailed observing sites to the maps for the appropriate variable (Figures S1, S3 and S5). We cannot use the coarse/fine mode in the AERONET observations because this could also be seasalts, especially in coastal regions, so we chose to use model results for this calculation. It is unfortunate that there is so little data across the globe to consider IAV, and one of the focus points of this paper is to try to motivate for more data collection, especially long term sites that can be clearly identified to be dust sites. Notice that, as we say in the text in several places, our goal is not to use every observation (especially low quality observations like visibility or AOD, which can be multiple aerosols), but rather to show the differences in model simulations, that all do about equally against observations, but still show substantial differences. We hope the new introduction makes this point clearer.

We add in an explanation:

"Because both seasalts and dust occur in the coarse mode, we cannot use remote sensing measurements of the coarse versus fine mode to identify dust dominated stations."

5. Page 7, lines 26-28: I do not understand why these two records have been chosen, which have a much shorter and inconsistent time coverage, while considerably less but still several AERONET observations are available in the southern hemisphere.

The problem we encountered with the AERONET sites in the southern hemisphere is that, there is substantial seasalts in the Southern Hemisphere, so that coarse mode is often dominate by that aerosol. We did find one station, where there is a dominant contribution from dust in the coarse mode, at least, and thus we have added an Australian site (Tinga Tingana), which has a large dust percentage, to our analysis, from the coarse mode. Since this is likely to have substantial other aerosols, we include it only in the Southern Hemisphere analysis in the online suplement. Much more data is needed to better resolve the dust cycle and variability in the Southern Hemisphere.

6. Page 9, lines 5-23: The analysis is too general. At least it should be clearly described which desert contributes to the dust load at the individual sites, and then the variability and relevant changing atmospheric and/or surface conditions have to be evaluated respectively. I do not see why for the AODs you are limited to regions close to source regions (line 20).

With this paper, we are not focusing so much on regional dust sources and their contribution at individual sites as we are on the modeled interannual variability of the dust once it is in the atmosphere, and whether it is consistent across models. As we try to explain in the introduction, the reviewer seems to misunderstand the goal of this paper. Other papers have examined the observed causes of variability in dust sources at different sites, and we are not trying to recreate that work, but to examine differences between dust simulations caused by the use of different reanalyses and see what fraction of the IAV is robust.

7. Page 10, lines 12-13: "[. . .] insight into how much of the variability in dust is driven by SSTs." Can this be quantified any further?

The results are discussed in more detail later. We use the metrics described later in the Section (2.3) to do this more quantatively.

8. Page 10, line 17: Here and elsewhere in the text, even though it is obvious, please mention the

dust source affecting the geographic region under consideration.

This is a good idea. We do not include it in this section, but rather lower, where we discuss different source regions and downwind regions.

"The sources which dominate the surface concentration, deposition and AOD in different downwind regions were not diagnosed in this study, but were previously shown for the mean in related model simulations (see Albani et al., 2014; Figure S1), and suggest source-receptor type relationships consistent with previous studies (e.g. Tanaka and Chiba, 2006;Mahowald, 2007). Those studies suggest that, as expected, North African source dominant North Atlantic sources, and East Asian and Central Asian sources dominate the North Pacific. The Central Asian sources are important for the North Indian Ocean. The Southern Hemisphere sources tend to dominate the regions just downwind of the sources (Figure S1, Albani et al., 2014). These results are consistent with the available source provenance data, which were used to 'tune' the CAM4 simulations (see Albani et al., 2014 for more details)."

9. Page 10, lines 29-32: Could also vegetation-related changes in surface roughness explain the trend in the Sahelian dust emissions as discussed by Cowie et al. (GRL, 2013), and would these changes be considered by any of the models used? What exactly are the mechanisms behind these correlations?

Thank you for bringing this paper to our attention: this is a very important paper that we had not seen. We modify the text throughout the paper to include this hypothesis, which links the other hypotheses in a novel manner. We are unable to check this hypothesis and whether it is operating our model simulations, because we did not archive surface roughness, but bring up this point. As we discuss elsewhere, this paper has too much material already, so we cannot add more information about the specific mechanisms, which is also not the point of this paper, and is better explored in other papers (like the Cowie et al., 2013 paper). In addition, Ridley et al. (2014) tested the potential role for vegetation changes on the trend in dust and found it was not a factor (at least in that simulation).

10. Page 11, line 13: Correct: "[...] supporting the idea that the source strength decreased within the time period 1990 – 2005."

The clarification has been added.

11. Page 11, lines 29-31: Here, you may include a study by, e.g., Marsham et al. (GRL, 2011) who showed that coarse-resolution models usually fail to reproduce dust emission events related to moist convection.

Yes, this is a good point. we add the point that all of our model simulations may miss small scale features like dust devils or moist convection. We add this in the introduction:

"Note that the global model simulations used here may miss small scale feature that may be important for IAV, such as dust devils or moist convective events (e.g. Renno et al., 2000; Marsham et al., 2011)."

12. Page 12, lines 1-27: The database for the analysis in the southern hemisphere is too small to

be statistically significant. Problematic is also the shorter and, in particular, inconsistent time coverage of the measurements. Moreover, the two sites are affected by mineral dust from different sources, which is not discussed at all. You could easily broaden your database by using AERONET sun photometer observations.

We have added one additional AERONET observations in the Southern Hemisphere. As discussed above, the rest of the AERONET sites fail even if we focus on coarse mode, because they have potentially too much sea salt influence. We need more observations for the southern hemisphere that can be robustly thought of as dust.

13. Page 15, lines 15-17: The 'global average' is not a larger source. Could you please specify the smaller sources meant?

We have removed the reference to the global average and added examples of smaller sources.

Technical corrections:

1. Page 2, line 8: Do you mean "4-fold fluctuations"?

we mean 4-fold fluctuations. We clarify the sentence:

"Dust is widely variable in space and time, with 4-fold fluctuations in surface concentration observed across a large region on decadal time scales (Prospero and Lamb, 2003),"

2. Page 5, line 26: Wording: "[. . .] it [the model] can only be operated using reanalysis datasets, [. . .]".

The sentence has been reworded.

3. Page 7, Eq. 1: A proper equation would look more scientific.

The equation has been formatted.

4. Page 11: IAV needs to be resolved.

We change the definition the IAV to be more clear, we hope.

5. Page 18: Correct: "whether other drivers".

The typo has been corrected and clarified.

[revised manuscript text omitted]

**Table 4: Slope of the normalized annual mean values from 1982 to 2008 (or time period available, shown in Table 1) for the Western Sahel (13 to 22°N and -20 to 13°E) and North Africa (0 to 35°N and -20 to 40°E) and model (Figure 4) (statistically significant values are in bold, standard deviation of slope in parenthesis for Barbados surf. conc. slope) in the first four columns. Values are normalized by dividing by the mean, so that slopes represent relative change per year. The last column is the correlation of interannual variability in precipitation in each model compared to observations.**

| | Slope Barbados surf. conc. | Slope Sahel source | Slope North African source | Slope Sahel Precip. | Correl. With obs. Sahel precip. |
|---|---|---|---|---|---|
| CAM4 (MERRA) | **-0.014 (0.0054)** | **-0.0065** | -0.0005 | 0.0035 | **0.43** |
| CAM4 (NCEP) | -0.0017 (0.016) | **-0.0169** | **-0.0074** | **0.0425** | 0.29 |
| CAM4 (ERAI) | **-0.0058 (0.0051)** | -0.0006 | 0.002 | 0.0008 | **0.81** |
| CAM4 (AMIP) | **-0.0079 (0.0035)** | **-0.0061** | **-0.0037** | **0.0186** | **0.42** |
| GCHEM (MERRA) | **-0.025 (0.0047)** | **-0.021** | **-0.0072** | 0.0035 | **0.43** |
| MATCH (NCEP) | **-0.0087 (0.0059)** | **-0.0047** | 0.027 | **0.0425** | 0.29 |
| CAM5 (AMIP) | -0.01 (0.01) | **0.0027** | 0.029 | | |
| Obs | **-0.016 (0.006)** | | | **0.0089** | |

**Table 5: Variability in Southern Hemisphere**
**Values for the IAV variability (annual average standard deviation divided by mean) and the ratio of the variability from the seasonal cycle over the IAV (for the surface concentration in the model cases and data from Rio Gallegos; deposition data from Kerguelen; and coarse mode AOD from Tinga Tingana (locations listed in Table 1).**

| Model/ Observations | Rio Gallegos Surface concentrations | | Kerguelen deposition | | Tinga Tingana AOD | |
|---|---|---|---|---|---|---|
| | IAV variability | Ratio variability from seasonal cycle over IAV | IAV variability | Ratio variability from seasonal cycle over IAV | IAV variability | Ratio variability from seasonal cycle over IAV |
| CAM4 (MERRA) | 1.78 | 1.06 | 0.22 | 1.21 | 0.11 | 3.61 |
| CAM4 (NCEP) | 2.13 | 0.79 | 0.27 | 1.02 | 0.23 | 2.21 |
| CAM4 (ERAI) | 2.66 | 0.84 | 1.42 | 1.32 | 0.21 | 1.81 |
| CAM4 (AMIP) | 3.86 | 0.55 | 2.06 | 0.48 | 0.17 | 2.68 |
| GCHEM (MERRA) | 0.67 | 1.26 | 0.32 | 1.03 | 0.25 | 1.87 |
| MATCH (NCEP) | 0.68 | 1.37 | 0.17 | 2.16 | 0.21 | 2.55 |
| CAM5 (AMIP) | 0.42 | 3.42 | 0.46 | 2.37 | 0.14 | 3.25 |
| Obs | 0.10 | 1.39 | 0.08 | 4.01 | 0.42 | 1.48 |

Formatted ... [456]
Formatted ... [369]
Formatted ... [372]
Formatted ... [373]
Formatted Table ... [370]
Formatted ... [371]
Formatted ... [374]
Formatted ... [375]
Formatted ... [377]
Formatted ... [378]
Formatted ... [379]
Formatted ... [380]
Formatted ... [381]
Formatted ... [382]
Formatted ... [376]
Formatted ... [383]
Formatted ... [384]
Formatted ... [386]
Formatted ... [387]
Formatted ... [388]
Formatted ... [389]
Formatted ... [390]
Formatted ... [391]
Formatted ... [385]
Formatted ... [392]
Formatted ... [393]
Formatted ... [395]
Formatted ... [396]
Formatted ... [397]
Formatted ... [398]
Formatted ... [399]
Formatted ... [400]
Formatted ... [394]
Formatted ... [401]
Formatted ... [402]
Formatted ... [404]
Formatted ... [405]
Formatted ... [406]
Formatted ... [407]
Formatted ... [408]
Formatted ... [409]
Formatted ... [403]
Formatted ... [410]
Formatted ... [411]
Formatted ... [413]
Formatted ... [414]
Formatted ... [415]
Formatted ... [416]
Formatted ... [417]
Formatted ... [418]
Formatted ... [412]
Formatted ... [419]
Formatted ... [420]
Formatted ... [422]
Formatted ... [423]
Formatted ... [424]
Formatted ... [425]
Formatted ... [426]
Formatted ... [427]
Formatted ... [421]
Formatted ... [428]
Formatted ... [431]
Formatted ... [432]
Formatted ... [433]
Formatted ... [434]
Formatted ... [435]
Formatted ... [436]

[revised manuscript text omitted]
 (0.42), with a strong anticorrelation over the Antarctic Ocean (-0.63). The NAO also featured a strong Antarctic anticorrelation (-0.42), and had the highest correlation coefficient over the South Indian Ocean (0.49). Again, this is similar to previous studies (e.g. Mahowald, 2003).The correlations tend to be lower near the southwestern US, and in the Southern hemisphere (Figure 10).

The correlation coefficients averaged over the CAM4-reanalysis models (Figure 11a, c and f) suggest that the surface concentration and deposition are similarly correlated, but that AOD has smoother and higher correlations. This is consistent with AOD being a more integrated quantity (Figure 8) (Huneeus et al., 2011; Albani et al., 2014). The lowest correlations occur over remote ocean regions, especially in the Southern Hemisphere, for all the variables. Note also that the correlation coefficients between the simulations using the same modeling framework but different meteorology (Figure 11a, c and f) had similar correlation coefficients, with a similar spatial structure, as the correlations between simulations using different modeling frameworks but the same meteorology (Figure 11b, d, and e). This suggests that meteorology and the modeling framework are equally important for simulating variability.

| Page 15: [9] Deleted | Cornell University | 11/15/16 2:23:00 PM |

The magnitude and the spatial structure of the correlation in the seasonal cycle is similar for all the variables as that seen using all monthly means (Figure 12a, c, e vs. 11a, c, e), suggesting that most of the correlation between the model versions is due to the correlation in the seasonal cycle (Figure 12). This is made clearer when considering just

the correlation of the annual means, showing that the models simulations are much less similar in their interannual variability (Figure 12b, d and f). Note however, that the spatial structure in the correlations for IAV is very different than considering either the monthly mean or seasonal cycle (Figure 12b, d, f vs. 12 a, c, e).

| Page 15: [10] Moved to page 15 (Move #2)Cornell University | 11/15/16 2:25:00 PM |
|---|---|

A comparison of other aerosols in two of the CAM4-reanalysis based simulations available here (Figure S5), is consistent with the idea that dust is highly variable, and that there is some correlation between the models driven by different meteorology far from the sources as well as close, but that interannual variability can be quite different for transport of aerosols (Figure S5). We will next discuss regional averages to understand how similarly ocean basin averages are simulated, to see if these IAV correlations in some regions are large enough to provide coherent basin estimates.

| Page 15: [11] Deleted | Cornell University | 12/3/16 3:04:00 PM |
|---|---|---|

, but that interannual variability can be quite different for transport of aerosols

| Page 15: [12] Deleted | Smith, Molly B | 11/24/16 2:03:00 PM |
|---|---|---|

| Page 30: [13] Formatted | Smith, Molly B | 11/24/16 2:05:00 PM |
|---|---|---|

Font:12 pt

| Page 30: [14] Formatted | Smith, Molly B | 11/24/16 2:12:00 PM |
|---|---|---|

None, Space Before: 0 pt, Line spacing: single, Don't keep with next, Don't keep lines together

| Page 30: [15] Formatted Table | Smith, Molly B | 11/24/16 2:12:00 PM |
|---|---|---|

Formatted Table

| Page 30: [16] Formatted | Smith, Molly B | 11/24/16 2:05:00 PM |
|---|---|---|

Font:12 pt, Not Bold, Not Italic, Font color: Auto

| Page 30: [17] Formatted | Smith, Molly B | 11/24/16 2:05:00 PM |
|---|---|---|

Font:12 pt

| Page 30: [17] Formatted | Smith, Molly B | 11/24/16 2:05:00 PM |
|---|---|---|

Font:12 pt

| Page 30: [18] Formatted | Smith, Molly B | 11/24/16 2:05:00 PM |
|---|---|---|

Font:12 pt

| Page 30: [18] Formatted | Smith, Molly B | 11/24/16 2:05:00 PM |
|---|---|---|

Font:12 pt

| Page 30: [19] Formatted | Smith, Molly B | 11/24/16 2:05:00 PM |
|---|---|---|

Font:12 pt

| Page 30: [19] Formatted | Smith, Molly B | 11/24/16 2:05:00 PM |
|---|---|---|

Font:12 pt

| Page 30: [20] Formatted | Smith, Molly B | 11/24/16 2:05:00 PM |
|---|---|---|

Font:12 pt

| Page 30: [20] Formatted | Smith, Molly B | 11/24/16 2:05:00 PM |
|---|---|---|

Font:12 pt

| Page 30: [21] Formatted | Smith, Molly B | 11/24/16 2:05:00 PM |
|---|---|---|

Font:12 pt

| Page 30: [21] Formatted | Smith, Molly B | 11/24/16 2:05:00 PM |
|---|---|---|

Font:12 pt

| Page 30: [22] Formatted | Smith, Molly B | 11/24/16 2:05:00 PM |
|---|---|---|

Font:12 pt

| Page 30: [23] Formatted | Smith, Molly B | 11/24/16 2:12:00 PM |
|---|---|---|

None, Space Before:  0 pt, Line spacing:  single, Don't keep with next, Don't keep lines together

| Page 30: [24] Formatted | Smith, Molly B | 11/24/16 2:05:00 PM |
|---|---|---|

Font:12 pt, Not Bold, Not Italic, Font color: Auto

| Page 30: [25] Formatted | Smith, Molly B | 11/24/16 2:05:00 PM |
|---|---|---|

Font:12 pt

| Page 30: [25] Formatted | Smith, Molly B | 11/24/16 2:05:00 PM |
|---|---|---|

Font:12 pt

| Page 30: [26] Formatted | Smith, Molly B | 11/24/16 2:05:00 PM |
|---|---|---|

Font:12 pt

| Page 30: [26] Formatted | Smith, Molly B | 11/24/16 2:05:00 PM |
|---|---|---|

Font:12 pt

| Page 30: [27] Formatted | Smith, Molly B | 11/24/16 2:05:00 PM |
|---|---|---|

Font:12 pt

| Page 30: [27] Formatted | Smith, Molly B | 11/24/16 2:05:00 PM |
|---|---|---|

Font:12 pt

| Page 30: [28] Formatted | Smith, Molly B | 11/24/16 2:05:00 PM |
|---|---|---|

Font:12 pt

| Page 30: [28] Formatted | Smith, Molly B | 11/24/16 2:05:00 PM |
|---|---|---|

Font:12 pt

| Page 30: [29] Formatted | Smith, Molly B | 11/24/16 2:05:00 PM |
|---|---|---|

Font:12 pt

| Page 30: [29] Formatted | Smith, Molly B | 11/24/16 2:05:00 PM |
|---|---|---|

Font:12 pt

| Page 30: [30] Formatted | Smith, Molly B | 11/24/16 2:05:00 PM |
|---|---|---|

Font:12 pt

| Page 30: [31] Formatted | Smith, Molly B | 11/24/16 2:12:00 PM |
|---|---|---|

None, Space Before:  0 pt, Line spacing:  single, Don't keep with next, Don't keep lines together

| Page 30: [32] Formatted | Smith, Molly B | 11/24/16 2:05:00 PM |
|---|---|---|

Font:12 pt, Not Bold, Not Italic, Font color: Auto

| Page 30: [33] Formatted | Smith, Molly B | 11/24/16 2:05:00 PM |
|---|---|---|

Font:12 pt

| Page 30: [33] Formatted | Smith, Molly B | 11/24/16 2:05:00 PM |
|---|---|---|

Font:12 pt

| Page 30: [34] Formatted | Smith, Molly B | 11/24/16 2:05:00 PM |
|---|---|---|

Font:12 pt

| Page 30: [34] Formatted | Smith, Molly B | 11/24/16 2:05:00 PM |
|---|---|---|

Font:12 pt

| Page 30: [35] Formatted | Smith, Molly B | 11/24/16 2:05:00 PM |
|---|---|---|

Font:12 pt

| Page 30: [35] Formatted | Smith, Molly B | 11/24/16 2:05:00 PM |
|---|---|---|

Font:12 pt

| Page 30: [36] Formatted | Smith, Molly B | 11/24/16 2:05:00 PM |
|---|---|---|

Font:12 pt

| Page 30: [36] Formatted | Smith, Molly B | 11/24/16 2:05:00 PM |
|---|---|---|

Font:12 pt

| Page 30: [37] Formatted | Smith, Molly B | 11/24/16 2:05:00 PM |
|---|---|---|

Font:12 pt

| Page 30: [37] Formatted | Smith, Molly B | 11/24/16 2:05:00 PM |
|---|---|---|

Font:12 pt

| Page 30: [38] Formatted | Smith, Molly B | 11/24/16 2:05:00 PM |
|---|---|---|

Font:12 pt

| Page 30: [39] Formatted | Smith, Molly B | 11/24/16 2:12:00 PM |
|---|---|---|

None, Space Before:  0 pt, Line spacing:  single, Don't keep with next, Don't keep lines together

| Page 30: [40] Formatted | Smith, Molly B | 11/24/16 2:05:00 PM |
|---|---|---|

Font:12 pt, Not Bold, Not Italic, Font color: Auto

| Page 30: [41] Formatted | Smith, Molly B | 11/24/16 2:05:00 PM |
|---|---|---|

Font:12 pt

| Page 30: [41] Formatted | Smith, Molly B | 11/24/16 2:05:00 PM |
|---|---|---|

Font:12 pt

| Page 30: [42] Formatted | Smith, Molly B | 11/24/16 2:05:00 PM |
|---|---|---|

Font:12 pt

| Page 30: [42] Formatted | Smith, Molly B | 11/24/16 2:05:00 PM |
|---|---|---|

Font:12 pt

| Page 30: [43] Formatted | Smith, Molly B | 11/24/16 2:05:00 PM |
|---|---|---|

Font:12 pt

| Page 30: [43] Formatted | Smith, Molly B | 11/24/16 2:05:00 PM |

Font:12 pt

| Page 30: [44] Formatted | Smith, Molly B | 11/24/16 2:05:00 PM |

Font:12 pt

| Page 30: [44] Formatted | Smith, Molly B | 11/24/16 2:05:00 PM |

Font:12 pt

| Page 30: [45] Formatted | Smith, Molly B | 11/24/16 2:05:00 PM |

Font:12 pt

| Page 30: [45] Formatted | Smith, Molly B | 11/24/16 2:05:00 PM |

Font:12 pt

| Page 30: [46] Formatted | Smith, Molly B | 11/24/16 2:05:00 PM |

Font:12 pt

| Page 30: [47] Formatted | Smith, Molly B | 11/24/16 2:12:00 PM |

None, Space Before:  0 pt, Line spacing:  single, Don't keep with next, Don't keep lines together

| Page 30: [48] Formatted | Smith, Molly B | 11/24/16 2:05:00 PM |

Font:12 pt, Not Bold, Not Italic, Font color: Auto

| Page 30: [49] Formatted | Smith, Molly B | 11/24/16 2:05:00 PM |

Font:12 pt

| Page 30: [49] Formatted | Smith, Molly B | 11/24/16 2:05:00 PM |

Font:12 pt

| Page 30: [50] Formatted | Smith, Molly B | 11/24/16 2:05:00 PM |

Font:12 pt

| Page 30: [50] Formatted | Smith, Molly B | 11/24/16 2:05:00 PM |

Font:12 pt

| Page 30: [51] Formatted | Smith, Molly B | 11/24/16 2:05:00 PM |

Font:12 pt

| Page 30: [51] Formatted | Smith, Molly B | 11/24/16 2:05:00 PM |

Font:12 pt

| Page 30: [52] Formatted | Smith, Molly B | 11/24/16 2:05:00 PM |

Font:12 pt

| Page 30: [52] Formatted | Smith, Molly B | 11/24/16 2:05:00 PM |

Font:12 pt

| Page 30: [53] Formatted | Smith, Molly B | 11/24/16 2:05:00 PM |

Font:12 pt

| Page 30: [53] Formatted | Smith, Molly B | 11/24/16 2:05:00 PM |

Font:12 pt

| Page 30: [54] Formatted | Smith, Molly B | 11/24/16 2:05:00 PM |
|---|---|---|

Font:12 pt

| Page 30: [55] Formatted | Smith, Molly B | 11/24/16 2:12:00 PM |
|---|---|---|

None, Space Before:  0 pt, Line spacing:  single, Don't keep with next, Don't keep lines together

| Page 30: [56] Formatted | Smith, Molly B | 11/24/16 2:05:00 PM |
|---|---|---|

Font:12 pt, Not Bold, Not Italic, Font color: Auto

| Page 30: [57] Formatted | Smith, Molly B | 11/24/16 2:05:00 PM |
|---|---|---|

Font:12 pt

| Page 30: [57] Formatted | Smith, Molly B | 11/24/16 2:05:00 PM |
|---|---|---|

Font:12 pt

| Page 30: [58] Formatted | Smith, Molly B | 11/24/16 2:05:00 PM |
|---|---|---|

Font:12 pt

| Page 30: [58] Formatted | Smith, Molly B | 11/24/16 2:05:00 PM |
|---|---|---|

Font:12 pt

| Page 30: [59] Formatted | Smith, Molly B | 11/24/16 2:05:00 PM |
|---|---|---|

Font:12 pt

| Page 30: [59] Formatted | Smith, Molly B | 11/24/16 2:05:00 PM |
|---|---|---|

Font:12 pt

| Page 30: [60] Formatted | Smith, Molly B | 11/24/16 2:05:00 PM |
|---|---|---|

Font:12 pt

| Page 30: [60] Formatted | Smith, Molly B | 11/24/16 2:05:00 PM |
|---|---|---|

Font:12 pt

| Page 30: [61] Formatted | Smith, Molly B | 11/24/16 2:05:00 PM |
|---|---|---|

Font:12 pt

| Page 30: [61] Formatted | Smith, Molly B | 11/24/16 2:05:00 PM |
|---|---|---|

Font:12 pt

| Page 30: [62] Formatted | Smith, Molly B | 11/24/16 2:05:00 PM |
|---|---|---|

Font:12 pt

| Page 30: [63] Formatted | Smith, Molly B | 11/24/16 2:12:00 PM |
|---|---|---|

None, Space Before:  0 pt, Line spacing:  single, Don't keep with next, Don't keep lines together

| Page 30: [64] Formatted | Smith, Molly B | 11/24/16 2:05:00 PM |
|---|---|---|

Font:12 pt, Not Bold, Not Italic, Font color: Auto

| Page 30: [65] Formatted | Smith, Molly B | 11/24/16 2:05:00 PM |
|---|---|---|

Font:12 pt

| Page 30: [65] Formatted | Smith, Molly B | 11/24/16 2:05:00 PM |
|---|---|---|

Font:12 pt

| Page 30: [66] Formatted | Smith, Molly B | 11/24/16 2:05:00 PM |
|---|---|---|

Font:12 pt

| Page 30: [66] Formatted | Smith, Molly B | 11/24/16 2:05:00 PM |

Font:12 pt

| Page 30: [67] Formatted | Smith, Molly B | 11/24/16 2:05:00 PM |

Font:12 pt

| Page 30: [67] Formatted | Smith, Molly B | 11/24/16 2:05:00 PM |

Font:12 pt

| Page 30: [68] Formatted | Smith, Molly B | 11/24/16 2:05:00 PM |

Font:12 pt

| Page 30: [68] Formatted | Smith, Molly B | 11/24/16 2:05:00 PM |

Font:12 pt

| Page 30: [69] Formatted | Smith, Molly B | 11/24/16 2:05:00 PM |

Font:12 pt

| Page 30: [69] Formatted | Smith, Molly B | 11/24/16 2:05:00 PM |

Font:12 pt

| Page 30: [70] Formatted | Smith, Molly B | 11/24/16 2:05:00 PM |

Font:12 pt

| Page 30: [71] Formatted | Smith, Molly B | 11/24/16 2:12:00 PM |

None, Space Before:  0 pt, Line spacing:  single, Don't keep with next, Don't keep lines together

| Page 30: [72] Formatted | Smith, Molly B | 11/24/16 2:05:00 PM |

Font:12 pt, Not Bold, Not Italic, Font color: Auto

| Page 30: [73] Formatted | Smith, Molly B | 11/24/16 2:05:00 PM |

Font:12 pt

| Page 30: [73] Formatted | Smith, Molly B | 11/24/16 2:05:00 PM |

Font:12 pt

| Page 30: [74] Formatted | Smith, Molly B | 11/24/16 2:05:00 PM |

Font:12 pt

| Page 30: [74] Formatted | Smith, Molly B | 11/24/16 2:05:00 PM |

Font:12 pt

| Page 30: [75] Formatted | Smith, Molly B | 11/24/16 2:05:00 PM |

Font:12 pt

| Page 30: [75] Formatted | Smith, Molly B | 11/24/16 2:05:00 PM |

Font:12 pt

| Page 30: [76] Formatted | Smith, Molly B | 11/24/16 2:05:00 PM |

Font:12 pt

| Page 30: [76] Formatted | Smith, Molly B | 11/24/16 2:05:00 PM |

Font:12 pt

| Page 30: [77] Formatted | Smith, Molly B | 11/24/16 2:05:00 PM |
|---|---|---|

Font:12 pt

| Page 30: [77] Formatted | Smith, Molly B | 11/24/16 2:05:00 PM |
|---|---|---|

Font:12 pt

| Page 30: [78] Deleted | Smith, Molly B | 11/24/16 2:07:00 PM |
|---|---|---|

|  |  |  |  |  |
|---|---|---|---|---|

| Page 30: [79] Formatted | Smith, Molly B | 11/24/16 2:05:00 PM |
|---|---|---|

Font:12 pt

| Page 30: [80] Formatted | Smith, Molly B | 11/24/16 2:05:00 PM |
|---|---|---|

Line spacing:  single

| Page 30: [81] Formatted | Smith, Molly B | 11/24/16 2:05:00 PM |
|---|---|---|

Font:12 pt

| Page 31: [82] Formatted | Smith, Molly B | 11/24/16 2:05:00 PM |
|---|---|---|

Font:12 pt

| Page 31: [83] Formatted | Smith, Molly B | 11/24/16 2:12:00 PM |
|---|---|---|

None, Space Before:  0 pt, Line spacing:  single, Don't keep with next, Don't keep lines together

| Page 31: [84] Formatted Table | Smith, Molly B | 11/24/16 2:12:00 PM |
|---|---|---|

Formatted Table

| Page 31: [85] Formatted | Smith, Molly B | 11/24/16 2:05:00 PM |
|---|---|---|

Font:12 pt, Not Bold, Not Italic, Font color: Auto

| Page 31: [86] Formatted | Smith, Molly B | 11/24/16 2:05:00 PM |
|---|---|---|

Font:12 pt

| Page 31: [86] Formatted | Smith, Molly B | 11/24/16 2:05:00 PM |
|---|---|---|

Font:12 pt

| Page 31: [86] Formatted | Smith, Molly B | 11/24/16 2:05:00 PM |
|---|---|---|

Font:12 pt

| Page 31: [86] Formatted | Smith, Molly B | 11/24/16 2:05:00 PM |
|---|---|---|

Font:12 pt

| Page 31: [87] Formatted | Smith, Molly B | 11/24/16 2:05:00 PM |
|---|---|---|

Font:12 pt

| Page 31: [87] Formatted | Smith, Molly B | 11/24/16 2:05:00 PM |
|---|---|---|

Font:12 pt

| Page 31: [87] Formatted | Smith, Molly B | 11/24/16 2:05:00 PM |
|---|---|---|

Font:12 pt

| Page 31: [87] Formatted | Smith, Molly B | 11/24/16 2:05:00 PM |
|---|---|---|

Font:12 pt

| Page 31: [88] Formatted | Smith, Molly B | 11/24/16 2:05:00 PM |
|---|---|---|

Font:12 pt

| Page 31: [88] Formatted | Smith, Molly B | 11/24/16 2:05:00 PM |
|---|---|---|

Font:12 pt

| Page 31: [89] Formatted | Smith, Molly B | 11/24/16 2:05:00 PM |
|---|---|---|

Font:12 pt

| Page 31: [89] Formatted | Smith, Molly B | 11/24/16 2:05:00 PM |
|---|---|---|

Font:12 pt

| Page 31: [90] Formatted | Smith, Molly B | 11/24/16 2:05:00 PM |
|---|---|---|

Font:12 pt

| Page 31: [91] Formatted | Smith, Molly B | 11/24/16 2:12:00 PM |
|---|---|---|

None, Space Before:  0 pt, Line spacing:  single, Don't keep with next, Don't keep lines together

| Page 31: [92] Formatted | Smith, Molly B | 11/24/16 2:05:00 PM |
|---|---|---|

Font:12 pt, Not Italic, Font color: Auto

| Page 31: [93] Formatted | Smith, Molly B | 11/24/16 2:05:00 PM |
|---|---|---|

Font:12 pt

| Page 31: [94] Formatted | Smith, Molly B | 11/24/16 2:12:00 PM |
|---|---|---|

Line spacing:  single

| Page 31: [95] Formatted | Smith, Molly B | 11/24/16 2:05:00 PM |
|---|---|---|

Font:12 pt

| Page 31: [96] Formatted | Smith, Molly B | 11/24/16 2:12:00 PM |
|---|---|---|

None, Space Before:  0 pt, Line spacing:  single, Don't keep with next, Don't keep lines together

| Page 31: [97] Formatted | Smith, Molly B | 11/24/16 2:05:00 PM |
|---|---|---|

Font:12 pt, Not Bold, Not Italic, Font color: Auto

| Page 31: [98] Formatted | Smith, Molly B | 11/24/16 2:05:00 PM |
|---|---|---|

Font:12 pt

| Page 31: [98] Formatted | Smith, Molly B | 11/24/16 2:05:00 PM |
|---|---|---|

Font:12 pt

| Page 31: [99] Formatted | Smith, Molly B | 11/24/16 2:05:00 PM |
|---|---|---|

Font:12 pt

| Page 31: [99] Formatted | Smith, Molly B | 11/24/16 2:05:00 PM |
|---|---|---|

Font:12 pt

| Page 31: [100] Formatted | Smith, Molly B | 11/24/16 2:05:00 PM |
|---|---|---|

Font:12 pt

| Page 31: [100] Formatted | Smith, Molly B | 11/24/16 2:05:00 PM |
|---|---|---|

Font:12 pt

| Page 31: [101] Formatted | Smith, Molly B | 11/24/16 2:05:00 PM |
|---|---|---|

Font:12 pt

| Page 31: [101] Formatted | Smith, Molly B | 11/24/16 2:05:00 PM |
| --- | --- | --- |

Font:12 pt

| Page 31: [102] Formatted | Smith, Molly B | 11/24/16 2:05:00 PM |
| --- | --- | --- |

Font:12 pt

| Page 31: [103] Formatted | Smith, Molly B | 11/24/16 2:12:00 PM |
| --- | --- | --- |

None, Space Before:  0 pt, Line spacing:  single, Don't keep with next, Don't keep lines together

| Page 31: [104] Formatted | Smith, Molly B | 11/24/16 2:05:00 PM |
| --- | --- | --- |

Font:12 pt, Not Italic, Font color: Auto

| Page 31: [105] Formatted | Smith, Molly B | 11/24/16 2:05:00 PM |
| --- | --- | --- |

Font:12 pt

| Page 31: [105] Formatted | Smith, Molly B | 11/24/16 2:05:00 PM |
| --- | --- | --- |

Font:12 pt

| Page 31: [106] Formatted | Smith, Molly B | 11/24/16 2:05:00 PM |
| --- | --- | --- |

Font:12 pt

| Page 31: [106] Formatted | Smith, Molly B | 11/24/16 2:05:00 PM |
| --- | --- | --- |

Font:12 pt

| Page 31: [107] Formatted | Smith, Molly B | 11/24/16 2:05:00 PM |
| --- | --- | --- |

Font:12 pt

| Page 31: [107] Formatted | Smith, Molly B | 11/24/16 2:05:00 PM |
| --- | --- | --- |

Font:12 pt

| Page 31: [108] Formatted | Smith, Molly B | 11/24/16 2:05:00 PM |
| --- | --- | --- |

Font:12 pt

| Page 31: [108] Formatted | Smith, Molly B | 11/24/16 2:05:00 PM |
| --- | --- | --- |

Font:12 pt

| Page 31: [109] Formatted | Smith, Molly B | 11/24/16 2:05:00 PM |
| --- | --- | --- |

Font:12 pt

| Page 31: [110] Formatted | Smith, Molly B | 11/24/16 2:12:00 PM |
| --- | --- | --- |

None, Space Before:  0 pt, Line spacing:  single, Don't keep with next, Don't keep lines together

| Page 31: [111] Formatted | Smith, Molly B | 11/24/16 2:05:00 PM |
| --- | --- | --- |

Font:12 pt, Not Bold, Not Italic, Font color: Auto

| Page 31: [112] Formatted | Smith, Molly B | 11/24/16 2:05:00 PM |
| --- | --- | --- |

Font:12 pt

| Page 31: [112] Formatted | Smith, Molly B | 11/24/16 2:05:00 PM |
| --- | --- | --- |

Font:12 pt

| Page 31: [113] Formatted | Smith, Molly B | 11/24/16 2:05:00 PM |
| --- | --- | --- |

Font:12 pt

| Page 31: [113] Formatted | Smith, Molly B | 11/24/16 2:05:00 PM |
| --- | --- | --- |

Font:12 pt

| Page 31: [114] Formatted | Smith, Molly B | 11/24/16 2:05:00 PM |
|---|---|---|

Font:12 pt

| Page 31: [114] Formatted | Smith, Molly B | 11/24/16 2:05:00 PM |
|---|---|---|

Font:12 pt

| Page 31: [115] Formatted | Smith, Molly B | 11/24/16 2:05:00 PM |
|---|---|---|

Font:12 pt

| Page 31: [115] Formatted | Smith, Molly B | 11/24/16 2:05:00 PM |
|---|---|---|

Font:12 pt

| Page 31: [116] Formatted | Smith, Molly B | 11/24/16 2:05:00 PM |
|---|---|---|

Font:12 pt

| Page 31: [117] Formatted | Smith, Molly B | 11/24/16 2:12:00 PM |
|---|---|---|

None, Space Before:  0 pt, Line spacing:  single, Don't keep with next, Don't keep lines together

| Page 31: [118] Formatted | Smith, Molly B | 11/24/16 2:05:00 PM |
|---|---|---|

Font:12 pt, Not Bold, Not Italic, Font color: Auto

| Page 31: [119] Formatted | Smith, Molly B | 11/24/16 2:05:00 PM |
|---|---|---|

Font:12 pt

| Page 31: [119] Formatted | Smith, Molly B | 11/24/16 2:05:00 PM |
|---|---|---|

Font:12 pt

| Page 31: [120] Formatted | Smith, Molly B | 11/24/16 2:05:00 PM |
|---|---|---|

Font:12 pt

| Page 31: [120] Formatted | Smith, Molly B | 11/24/16 2:05:00 PM |
|---|---|---|

Font:12 pt

| Page 31: [121] Formatted | Smith, Molly B | 11/24/16 2:05:00 PM |
|---|---|---|

Font:12 pt

| Page 31: [121] Formatted | Smith, Molly B | 11/24/16 2:05:00 PM |
|---|---|---|

Font:12 pt

| Page 31: [122] Formatted | Smith, Molly B | 11/24/16 2:05:00 PM |
|---|---|---|

Font:12 pt

| Page 31: [122] Formatted | Smith, Molly B | 11/24/16 2:05:00 PM |
|---|---|---|

Font:12 pt

| Page 31: [123] Formatted | Smith, Molly B | 11/24/16 2:05:00 PM |
|---|---|---|

Font:12 pt

| Page 31: [124] Formatted | Smith, Molly B | 11/24/16 2:12:00 PM |
|---|---|---|

None, Space Before:  0 pt, Line spacing:  single, Don't keep with next, Don't keep lines together

| Page 31: [125] Formatted | Smith, Molly B | 11/24/16 2:05:00 PM |
|---|---|---|

Font:12 pt, Not Bold, Not Italic, Font color: Auto

| Page 31: [126] Formatted | Smith, Molly B | 11/24/16 2:05:00 PM |
|---|---|---|

Font:12 pt

| Page 31: [126] Formatted | Smith, Molly B | 11/24/16 2:05:00 PM |
|---|---|---|

Font:12 pt

| Page 31: [127] Formatted | Smith, Molly B | 11/24/16 2:05:00 PM |
|---|---|---|

Font:12 pt

| Page 31: [127] Formatted | Smith, Molly B | 11/24/16 2:05:00 PM |
|---|---|---|

Font:12 pt

| Page 31: [128] Formatted | Smith, Molly B | 11/24/16 2:05:00 PM |
|---|---|---|

Font:12 pt

| Page 31: [128] Formatted | Smith, Molly B | 11/24/16 2:05:00 PM |
|---|---|---|

Font:12 pt

| Page 31: [129] Formatted | Smith, Molly B | 11/24/16 2:05:00 PM |
|---|---|---|

Font:12 pt

| Page 31: [129] Formatted | Smith, Molly B | 11/24/16 2:05:00 PM |
|---|---|---|

Font:12 pt

| Page 31: [130] Formatted | Smith, Molly B | 11/24/16 2:05:00 PM |
|---|---|---|

Font:12 pt

| Page 31: [131] Formatted | Smith, Molly B | 11/24/16 2:12:00 PM |
|---|---|---|

None, Space Before:  0 pt, Line spacing:  single, Don't keep with next, Don't keep lines together

| Page 31: [132] Formatted | Smith, Molly B | 11/24/16 2:05:00 PM |
|---|---|---|

Font:12 pt, Not Bold, Not Italic, Font color: Auto

| Page 31: [133] Formatted | Smith, Molly B | 11/24/16 2:05:00 PM |
|---|---|---|

Font:12 pt

| Page 31: [133] Formatted | Smith, Molly B | 11/24/16 2:05:00 PM |
|---|---|---|

Font:12 pt

| Page 31: [134] Formatted | Smith, Molly B | 11/24/16 2:05:00 PM |
|---|---|---|

Font:12 pt

| Page 31: [134] Formatted | Smith, Molly B | 11/24/16 2:05:00 PM |
|---|---|---|

Font:12 pt

| Page 31: [135] Formatted | Smith, Molly B | 11/24/16 2:05:00 PM |
|---|---|---|

Font:12 pt

| Page 31: [135] Formatted | Smith, Molly B | 11/24/16 2:05:00 PM |
|---|---|---|

Font:12 pt

| Page 31: [136] Formatted | Smith, Molly B | 11/24/16 2:05:00 PM |
|---|---|---|

Font:12 pt

| Page 31: [136] Formatted | Smith, Molly B | 11/24/16 2:05:00 PM |
|---|---|---|

Font:12 pt

| Page 31: [137] Formatted | Smith, Molly B | 11/24/16 2:05:00 PM |

Font:12 pt

| Page 31: [138] Formatted | Smith, Molly B | 11/24/16 2:12:00 PM |

None, Space Before:  0 pt, Line spacing:  single, Don't keep with next, Don't keep lines together

| Page 31: [139] Formatted | Smith, Molly B | 11/24/16 2:05:00 PM |

Font:12 pt, Not Bold, Not Italic, Font color: Auto

| Page 31: [140] Formatted | Smith, Molly B | 11/24/16 2:05:00 PM |

Font:12 pt

| Page 31: [140] Formatted | Smith, Molly B | 11/24/16 2:05:00 PM |

Font:12 pt

| Page 31: [141] Formatted | Smith, Molly B | 11/24/16 2:05:00 PM |

Font:12 pt

| Page 31: [141] Formatted | Smith, Molly B | 11/24/16 2:05:00 PM |

Font:12 pt

| Page 31: [142] Formatted | Smith, Molly B | 11/24/16 2:05:00 PM |

Font:12 pt

| Page 31: [142] Formatted | Smith, Molly B | 11/24/16 2:05:00 PM |

Font:12 pt

| Page 31: [143] Formatted | Smith, Molly B | 11/24/16 2:05:00 PM |

Font:12 pt

| Page 31: [143] Formatted | Smith, Molly B | 11/24/16 2:05:00 PM |

Font:12 pt

| Page 31: [144] Formatted | Smith, Molly B | 11/24/16 2:05:00 PM |

Font:12 pt

| Page 31: [145] Formatted | Smith, Molly B | 11/24/16 2:12:00 PM |

None, Space Before:  0 pt, Line spacing:  single, Don't keep with next, Don't keep lines together

| Page 31: [146] Formatted | Smith, Molly B | 11/24/16 2:05:00 PM |

Font:12 pt, Not Bold, Not Italic, Font color: Auto

| Page 31: [147] Formatted | Smith, Molly B | 11/24/16 2:05:00 PM |

Font:12 pt

| Page 31: [147] Formatted | Smith, Molly B | 11/24/16 2:05:00 PM |

Font:12 pt

| Page 31: [148] Formatted | Smith, Molly B | 11/24/16 2:05:00 PM |

Font:12 pt

| Page 31: [148] Formatted | Smith, Molly B | 11/24/16 2:05:00 PM |

Font:12 pt

| Page 31: [149] Formatted | Smith, Molly B | 11/24/16 2:05:00 PM |
|---|---|---|

Font:12 pt

| Page 31: [149] Formatted | Smith, Molly B | 11/24/16 2:05:00 PM |
|---|---|---|

Font:12 pt

| Page 31: [150] Formatted | Smith, Molly B | 11/24/16 2:05:00 PM |
|---|---|---|

Font:12 pt

| Page 31: [150] Formatted | Smith, Molly B | 11/24/16 2:05:00 PM |
|---|---|---|

Font:12 pt

| Page 31: [151] Formatted | Smith, Molly B | 11/24/16 2:05:00 PM |
|---|---|---|

Font:12 pt

| Page 31: [152] Formatted | Smith, Molly B | 11/24/16 2:12:00 PM |
|---|---|---|

None, Space Before:  0 pt, Line spacing:  single, Don't keep with next, Don't keep lines together

| Page 31: [153] Formatted | Smith, Molly B | 11/24/16 2:05:00 PM |
|---|---|---|

Font:12 pt, Not Bold, Not Italic, Font color: Auto

| Page 31: [154] Formatted | Smith, Molly B | 11/24/16 2:05:00 PM |
|---|---|---|

Font:12 pt

| Page 31: [154] Formatted | Smith, Molly B | 11/24/16 2:05:00 PM |
|---|---|---|

Font:12 pt

| Page 31: [155] Formatted | Smith, Molly B | 11/24/16 2:05:00 PM |
|---|---|---|

Font:12 pt

| Page 31: [155] Formatted | Smith, Molly B | 11/24/16 2:05:00 PM |
|---|---|---|

Font:12 pt

| Page 31: [156] Formatted | Smith, Molly B | 11/24/16 2:05:00 PM |
|---|---|---|

Font:12 pt

| Page 31: [156] Formatted | Smith, Molly B | 11/24/16 2:05:00 PM |
|---|---|---|

Font:12 pt

| Page 31: [157] Formatted | Smith, Molly B | 11/24/16 2:05:00 PM |
|---|---|---|

Font:12 pt

| Page 31: [157] Formatted | Smith, Molly B | 11/24/16 2:05:00 PM |
|---|---|---|

Font:12 pt

| Page 31: [158] Formatted | Smith, Molly B | 11/24/16 2:05:00 PM |
|---|---|---|

Font:12 pt

| Page 31: [159] Formatted | Smith, Molly B | 11/24/16 2:12:00 PM |
|---|---|---|

Line spacing:  single

| Page 31: [160] Formatted | Smith, Molly B | 11/24/16 2:12:00 PM |
|---|---|---|

Space Before:  0 pt, Line spacing:  single

| Page 31: [161] Formatted | Smith, Molly B | 11/24/16 2:05:00 PM |
|---|---|---|

Font:12 pt

| Page 31: [162] Formatted | Smith, Molly B | 11/24/16 2:12:00 PM |
|---|---|---|

None, Space Before:  0 pt, Line spacing:  single, Don't keep with next, Don't keep lines together

| Page 31: [163] Formatted | Smith, Molly B | 11/24/16 2:05:00 PM |
|---|---|---|

Font:12 pt, Not Italic, Font color: Auto

| Page 31: [164] Formatted | Smith, Molly B | 11/24/16 2:05:00 PM |
|---|---|---|

Font:12 pt

| Page 31: [165] Formatted | Smith, Molly B | 11/24/16 2:12:00 PM |
|---|---|---|

Line spacing:  single

| Page 31: [166] Formatted | Smith, Molly B | 11/24/16 2:05:00 PM |
|---|---|---|

Font:12 pt

| Page 31: [167] Formatted | Smith, Molly B | 11/24/16 2:12:00 PM |
|---|---|---|

None, Space Before:  0 pt, Line spacing:  single, Don't keep with next, Don't keep lines together

| Page 31: [168] Formatted | Smith, Molly B | 11/24/16 2:05:00 PM |
|---|---|---|

Font:12 pt, Not Bold, Not Italic, Font color: Auto

| Page 31: [169] Formatted | Smith, Molly B | 11/24/16 2:05:00 PM |
|---|---|---|

Font:12 pt

| Page 31: [169] Formatted | Smith, Molly B | 11/24/16 2:05:00 PM |
|---|---|---|

Font:12 pt

| Page 31: [170] Formatted | Smith, Molly B | 11/24/16 2:05:00 PM |
|---|---|---|

Font:12 pt

| Page 31: [170] Formatted | Smith, Molly B | 11/24/16 2:05:00 PM |
|---|---|---|

Font:12 pt

| Page 31: [171] Formatted | Smith, Molly B | 11/24/16 2:05:00 PM |
|---|---|---|

Font:12 pt

| Page 31: [171] Formatted | Smith, Molly B | 11/24/16 2:05:00 PM |
|---|---|---|

Font:12 pt

| Page 31: [172] Formatted | Smith, Molly B | 11/24/16 2:05:00 PM |
|---|---|---|

Font:12 pt

| Page 31: [172] Formatted | Smith, Molly B | 11/24/16 2:05:00 PM |
|---|---|---|

Font:12 pt

| Page 31: [173] Formatted | Smith, Molly B | 12/1/16 2:33:00 PM |
|---|---|---|

Font:12 pt

| Page 31: [174] Formatted | Smith, Molly B | 11/24/16 2:12:00 PM |
|---|---|---|

Line spacing:  single

| Page 31: [175] Deleted | Cornell University | 11/15/16 2:45:00 PM |
|---|---|---|

| Dalanzadgad | 43 | 104 | 1997-2012 | B. Holben |
|---|---|---|---|---|

| Page 31: [176] Formatted | Smith, Molly B | 12/1/16 2:33:00 PM |
|---|---|---|

Font:12 pt

| Page 31: [177] Formatted | Smith, Molly B | 11/24/16 2:12:00 PM |
|---|---|---|

Line spacing:  single

| Page 31: [178] Formatted | Smith, Molly B | 12/1/16 2:33:00 PM |
|---|---|---|

Font:12 pt

| Page 31: [179] Formatted | Smith, Molly B | 12/1/16 2:33:00 PM |
|---|---|---|

Font:12 pt

| Page 31: [180] Formatted | Smith, Molly B | 11/24/16 2:12:00 PM |
|---|---|---|

Line spacing:  single

| Page 31: [181] Formatted | Smith, Molly B | 12/1/16 2:33:00 PM |
|---|---|---|

Font:12 pt

| Page 31: [182] Formatted | Smith, Molly B | 11/24/16 2:12:00 PM |
|---|---|---|

Line spacing:  single

| Page 31: [183] Formatted | Smith, Molly B | 12/1/16 2:33:00 PM |
|---|---|---|

Font:12 pt

| Page 31: [184] Formatted | Smith, Molly B | 11/24/16 2:12:00 PM |
|---|---|---|

Line spacing:  single

| Page 31: [185] Formatted | Smith, Molly B | 12/1/16 2:33:00 PM |
|---|---|---|

Font:12 pt

| Page 31: [186] Formatted | Smith, Molly B | 12/1/16 2:32:00 PM |
|---|---|---|

Line spacing:  single

| Page 31: [187] Formatted | Smith, Molly B | 12/1/16 2:32:00 PM |
|---|---|---|

Space Before:  0 pt, Line spacing:  single

| Page 31: [188] Formatted | Smith, Molly B | 12/1/16 2:33:00 PM |
|---|---|---|

Font:12 pt, Not Italic, Font color: Auto

| Page 31: [189] Formatted | Smith, Molly B | 11/24/16 2:05:00 PM |
|---|---|---|

Font:12 pt

| Page 31: [190] Formatted | Smith, Molly B | 11/24/16 2:12:00 PM |
|---|---|---|

Line spacing:  single

| Page 31: [191] Formatted | Smith, Molly B | 12/1/16 2:33:00 PM |
|---|---|---|

Font:12 pt

| Page 31: [192] Formatted | Smith, Molly B | 11/24/16 2:12:00 PM |
|---|---|---|

Line spacing:  single

| Page 31: [193] Deleted | Cornell University | 11/15/16 2:45:00 PM |
|---|---|---|

Rio Gallegos
Surface concentrations

| Page 31: [194] Formatted | Smith, Molly B | 11/24/16 2:12:00 PM |
|---|---|---|

Line spacing:  single

| Page 31: [195] Formatted | Smith, Molly B | 12/1/16 2:33:00 PM |
|---|---|---|

Font:12 pt

| Page 31: [196] Formatted | Smith, Molly B | 11/24/16 2:05:00 PM |
|---|---|---|

Font:12 pt

| Page 31: [197] Formatted | Smith, Molly B | 11/24/16 2:05:00 PM |
|---|---|---|

Font:12 pt

| Page 31: [198] Formatted | Smith, Molly B | 11/24/16 2:05:00 PM |
|---|---|---|

Font:12 pt

| Page 31: [199] Formatted | Smith, Molly B | 11/24/16 2:05:00 PM |
|---|---|---|

Font:12 pt

| Page 31: [200] Formatted | Smith, Molly B | 11/24/16 2:05:00 PM |
|---|---|---|

Font:12 pt

| Page 32: [201] Formatted | Smith, Molly B | 11/24/16 2:13:00 PM |
|---|---|---|

None, Space Before:  0 pt, Line spacing:  single, Don't keep with next, Don't keep lines together

| Page 32: [202] Formatted Table | Smith, Molly B | 11/24/16 2:13:00 PM |
|---|---|---|

Formatted Table

| Page 32: [203] Formatted | Smith, Molly B | 11/24/16 2:05:00 PM |
|---|---|---|

Font:12 pt, Not Bold, Not Italic, Font color: Auto

| Page 32: [204] Formatted | Smith, Molly B | 11/24/16 2:05:00 PM |
|---|---|---|

Font:12 pt

| Page 32: [204] Formatted | Smith, Molly B | 11/24/16 2:05:00 PM |
|---|---|---|

Font:12 pt

| Page 32: [205] Formatted | Smith, Molly B | 11/24/16 2:05:00 PM |
|---|---|---|

Font:12 pt

| Page 32: [205] Formatted | Smith, Molly B | 11/24/16 2:05:00 PM |
|---|---|---|

Font:12 pt

| Page 32: [206] Formatted | Smith, Molly B | 11/24/16 2:05:00 PM |
|---|---|---|

Font:12 pt

| Page 32: [206] Formatted | Smith, Molly B | 11/24/16 2:05:00 PM |
|---|---|---|

Font:12 pt

| Page 32: [207] Formatted | Smith, Molly B | 11/24/16 2:05:00 PM |
|---|---|---|

Font:12 pt

| Page 32: [207] Formatted | Smith, Molly B | 11/24/16 2:05:00 PM |
|---|---|---|

Font:12 pt

| Page 32: [208] Formatted | Smith, Molly B | 11/24/16 2:05:00 PM |
|---|---|---|

Font:12 pt

| **Page 32: [209] Formatted** | **Smith, Molly B** | **11/24/16 2:13:00 PM** |

None, Space Before:  0 pt, Line spacing:  single, Don't keep with next, Don't keep lines together

| **Page 32: [210] Formatted** | **Smith, Molly B** | **11/24/16 2:05:00 PM** |

Font:12 pt, Not Bold, Not Italic, Font color: Auto

| **Page 32: [211] Formatted** | **Smith, Molly B** | **11/24/16 2:05:00 PM** |

Font:12 pt

| **Page 32: [211] Formatted** | **Smith, Molly B** | **11/24/16 2:05:00 PM** |

Font:12 pt

| **Page 32: [212] Formatted** | **Smith, Molly B** | **11/24/16 2:05:00 PM** |

Font:12 pt

| **Page 32: [212] Formatted** | **Smith, Molly B** | **11/24/16 2:05:00 PM** |

Font:12 pt

| **Page 32: [213] Formatted** | **Smith, Molly B** | **11/24/16 2:05:00 PM** |

Font:12 pt

| **Page 32: [213] Formatted** | **Smith, Molly B** | **11/24/16 2:05:00 PM** |

Font:12 pt

| **Page 32: [214] Formatted** | **Smith, Molly B** | **11/24/16 2:05:00 PM** |

Font:12 pt

| **Page 32: [215] Formatted** | **Smith, Molly B** | **11/24/16 2:13:00 PM** |

None, Space Before:  0 pt, Line spacing:  single, Don't keep with next, Don't keep lines together

| **Page 32: [216] Formatted** | **Smith, Molly B** | **11/24/16 2:05:00 PM** |

Font:12 pt, Not Bold, Not Italic, Font color: Auto

| **Page 32: [217] Formatted** | **Smith, Molly B** | **11/24/16 2:05:00 PM** |

Font:12 pt

| **Page 32: [217] Formatted** | **Smith, Molly B** | **11/24/16 2:05:00 PM** |

Font:12 pt

| **Page 32: [218] Formatted** | **Smith, Molly B** | **11/24/16 2:05:00 PM** |

Font:12 pt

| **Page 32: [218] Formatted** | **Smith, Molly B** | **11/24/16 2:05:00 PM** |

Font:12 pt

| **Page 32: [219] Formatted** | **Smith, Molly B** | **11/24/16 2:05:00 PM** |

Font:12 pt

| **Page 32: [219] Formatted** | **Smith, Molly B** | **11/24/16 2:05:00 PM** |

Font:12 pt

| **Page 32: [220] Formatted** | **Smith, Molly B** | **11/24/16 2:05:00 PM** |

Font:12 pt

| Page 32: [221] Formatted | Smith, Molly B | 11/24/16 2:13:00 PM |
|---|---|---|

None, Space Before:  0 pt, Line spacing:  single, Don't keep with next, Don't keep lines together

| Page 32: [222] Formatted | Smith, Molly B | 11/24/16 2:05:00 PM |
|---|---|---|

Font:12 pt, Not Italic, Font color: Auto

| Page 32: [223] Formatted | Smith, Molly B | 11/24/16 2:05:00 PM |
|---|---|---|

Font:12 pt

| Page 32: [223] Formatted | Smith, Molly B | 11/24/16 2:05:00 PM |
|---|---|---|

Font:12 pt

| Page 32: [224] Formatted | Smith, Molly B | 11/24/16 2:05:00 PM |
|---|---|---|

Font:12 pt

| Page 32: [224] Formatted | Smith, Molly B | 11/24/16 2:05:00 PM |
|---|---|---|

Font:12 pt

| Page 32: [225] Formatted | Smith, Molly B | 11/24/16 2:05:00 PM |
|---|---|---|

Font:12 pt

| Page 32: [225] Formatted | Smith, Molly B | 11/24/16 2:05:00 PM |
|---|---|---|

Font:12 pt

| Page 32: [226] Formatted | Smith, Molly B | 11/24/16 2:05:00 PM |
|---|---|---|

Font:12 pt

| Page 32: [227] Formatted | Smith, Molly B | 11/24/16 2:13:00 PM |
|---|---|---|

None, Space Before:  0 pt, Line spacing:  single, Don't keep with next, Don't keep lines together

| Page 32: [228] Formatted | Smith, Molly B | 11/24/16 2:05:00 PM |
|---|---|---|

Font:12 pt, Not Bold, Not Italic, Font color: Auto

| Page 32: [229] Formatted | Smith, Molly B | 11/24/16 2:05:00 PM |
|---|---|---|

Font:12 pt

| Page 32: [229] Formatted | Smith, Molly B | 11/24/16 2:05:00 PM |
|---|---|---|

Font:12 pt

| Page 32: [230] Formatted | Smith, Molly B | 11/24/16 2:05:00 PM |
|---|---|---|

Font:12 pt

| Page 32: [230] Formatted | Smith, Molly B | 11/24/16 2:05:00 PM |
|---|---|---|

Font:12 pt

| Page 32: [231] Formatted | Smith, Molly B | 11/24/16 2:05:00 PM |
|---|---|---|

Font:12 pt

| Page 32: [231] Formatted | Smith, Molly B | 11/24/16 2:05:00 PM |
|---|---|---|

Font:12 pt

| Page 32: [232] Formatted | Smith, Molly B | 11/24/16 2:05:00 PM |
|---|---|---|

Font:12 pt

| Page 32: [233] Formatted | Smith, Molly B | 11/24/16 2:13:00 PM |
|---|---|---|

None, Space Before:  0 pt, Line spacing:  single, Don't keep with next, Don't keep lines together

| Page 32: [234] Formatted | Smith, Molly B | 11/24/16 2:05:00 PM |
|---|---|---|

Font:12 pt, Not Bold, Not Italic, Font color: Auto

| Page 32: [235] Formatted | Smith, Molly B | 11/24/16 2:05:00 PM |
|---|---|---|

Font:12 pt

| Page 32: [236] Formatted | Smith, Molly B | 11/24/16 2:13:00 PM |
|---|---|---|

Line spacing:  single

| Page 32: [237] Formatted | Smith, Molly B | 11/24/16 2:05:00 PM |
|---|---|---|

Font:12 pt, Not Italic, Font color: Auto

| Page 32: [238] Formatted | Smith, Molly B | 11/24/16 2:05:00 PM |
|---|---|---|

Font:12 pt

| Page 32: [238] Formatted | Smith, Molly B | 11/24/16 2:05:00 PM |
|---|---|---|

Font:12 pt

| Page 32: [239] Formatted | Smith, Molly B | 11/24/16 2:05:00 PM |
|---|---|---|

Font:12 pt

| Page 32: [239] Formatted | Smith, Molly B | 11/24/16 2:05:00 PM |
|---|---|---|

Font:12 pt

| Page 32: [240] Formatted | Smith, Molly B | 11/24/16 2:05:00 PM |
|---|---|---|

Font:12 pt

| Page 32: [241] Formatted | Smith, Molly B | 11/24/16 2:13:00 PM |
|---|---|---|

None, Space Before:  0 pt, Line spacing:  single, Don't keep with next, Don't keep lines together

| Page 32: [242] Formatted | Smith, Molly B | 11/24/16 2:05:00 PM |
|---|---|---|

Font:12 pt, Not Bold, Not Italic, Font color: Auto

| Page 32: [243] Formatted | Smith, Molly B | 11/24/16 2:05:00 PM |
|---|---|---|

Font:12 pt

| Page 32: [243] Formatted | Smith, Molly B | 11/24/16 2:05:00 PM |
|---|---|---|

Font:12 pt

| Page 32: [244] Formatted | Smith, Molly B | 11/24/16 2:05:00 PM |
|---|---|---|

Font:12 pt

| Page 32: [244] Formatted | Smith, Molly B | 11/24/16 2:05:00 PM |
|---|---|---|

Font:12 pt

| Page 32: [245] Formatted | Smith, Molly B | 11/24/16 2:05:00 PM |
|---|---|---|

Font:12 pt

| Page 32: [245] Formatted | Smith, Molly B | 11/24/16 2:05:00 PM |
|---|---|---|

Font:12 pt

| Page 32: [246] Formatted | Smith, Molly B | 11/24/16 2:05:00 PM |
|---|---|---|

Font:12 pt

| Page 32: [247] Formatted | Smith, Molly B | 11/24/16 2:13:00 PM |
|---|---|---|

None, Space Before:  0 pt, Line spacing:  single, Don't keep with next, Don't keep lines together

| Page 32: [248] Formatted | Smith, Molly B | 11/24/16 2:05:00 PM |
|---|---|---|

Font:12 pt, Not Bold, Not Italic, Font color: Auto

| Page 32: [249] Formatted | Smith, Molly B | 11/24/16 2:05:00 PM |
|---|---|---|

Font:12 pt

| Page 32: [249] Formatted | Smith, Molly B | 11/24/16 2:05:00 PM |
|---|---|---|

Font:12 pt

| Page 32: [250] Formatted | Smith, Molly B | 11/24/16 2:05:00 PM |
|---|---|---|

Font:12 pt

| Page 32: [250] Formatted | Smith, Molly B | 11/24/16 2:05:00 PM |
|---|---|---|

Font:12 pt

| Page 32: [251] Formatted | Smith, Molly B | 11/24/16 2:05:00 PM |
|---|---|---|

Font:12 pt

| Page 32: [251] Formatted | Smith, Molly B | 11/24/16 2:05:00 PM |
|---|---|---|

Font:12 pt

| Page 32: [252] Formatted | Smith, Molly B | 11/24/16 2:05:00 PM |
|---|---|---|

Font:12 pt

| Page 32: [253] Formatted | Smith, Molly B | 11/24/16 2:13:00 PM |
|---|---|---|

Line spacing:  single

| Page 32: [254] Formatted | Smith, Molly B | 11/24/16 2:05:00 PM |
|---|---|---|

Font:12 pt

| Page 32: [255] Formatted | Smith, Molly B | 11/24/16 2:05:00 PM |
|---|---|---|

Line spacing:  single

| Page 33: [256] Formatted | Smith, Molly B | 11/24/16 2:05:00 PM |
|---|---|---|

Font:12 pt

| Page 33: [256] Formatted | Smith, Molly B | 11/24/16 2:05:00 PM |
|---|---|---|

Font:12 pt

| Page 33: [256] Formatted | Smith, Molly B | 11/24/16 2:05:00 PM |
|---|---|---|

Font:12 pt

| Page 33: [256] Formatted | Smith, Molly B | 11/24/16 2:05:00 PM |
|---|---|---|

Font:12 pt

| Page 33: [256] Formatted | Smith, Molly B | 11/24/16 2:05:00 PM |
|---|---|---|

Font:12 pt

| Page 33: [256] Formatted | Smith, Molly B | 11/24/16 2:05:00 PM |
|---|---|---|

Font:12 pt

| Page 33: [256] Formatted | Smith, Molly B | 11/24/16 2:05:00 PM |
|---|---|---|

Font:12 pt

| Page 33: [256] Formatted | Smith, Molly B | 11/24/16 2:05:00 PM |
|---|---|---|

Font:12 pt

| Page 33: [257] Formatted | Smith, Molly B | 11/24/16 2:13:00 PM |
|---|---|---|

Line spacing:  single

| Page 33: [258] Formatted Table | Smith, Molly B | 11/24/16 2:13:00 PM |
|---|---|---|

Formatted Table

| Page 33: [259] Formatted | Smith, Molly B | 11/24/16 2:13:00 PM |
|---|---|---|

Space Before:  0 pt, Line spacing:  single

| Page 33: [260] Formatted | Smith, Molly B | 11/24/16 2:05:00 PM |
|---|---|---|

Font:12 pt, Not Bold, Not Italic

| Page 33: [261] Formatted | Smith, Molly B | 11/24/16 2:05:00 PM |
|---|---|---|

Font:12 pt

| Page 33: [261] Formatted | Smith, Molly B | 11/24/16 2:05:00 PM |
|---|---|---|

Font:12 pt

| Page 33: [262] Formatted | Smith, Molly B | 11/24/16 2:05:00 PM |
|---|---|---|

Font:12 pt

| Page 33: [262] Formatted | Smith, Molly B | 11/24/16 2:05:00 PM |
|---|---|---|

Font:12 pt

| Page 33: [263] Formatted | Smith, Molly B | 11/24/16 2:05:00 PM |
|---|---|---|

Font:12 pt

| Page 33: [263] Formatted | Smith, Molly B | 11/24/16 2:05:00 PM |
|---|---|---|

Font:12 pt

| Page 33: [264] Formatted | Smith, Molly B | 11/24/16 2:05:00 PM |
|---|---|---|

Font:12 pt

| Page 33: [264] Formatted | Smith, Molly B | 11/24/16 2:05:00 PM |
|---|---|---|

Font:12 pt

| Page 33: [265] Formatted | Smith, Molly B | 11/24/16 2:05:00 PM |
|---|---|---|

Font:12 pt

| Page 33: [265] Formatted | Smith, Molly B | 11/24/16 2:05:00 PM |
|---|---|---|

Font:12 pt

| Page 33: [266] Formatted | Smith, Molly B | 11/24/16 2:05:00 PM |
|---|---|---|

Font:12 pt

| Page 33: [266] Formatted | Smith, Molly B | 11/24/16 2:05:00 PM |
|---|---|---|

Font:12 pt

| Page 33: [267] Formatted | Smith, Molly B | 11/24/16 2:05:00 PM |
|---|---|---|

Font:12 pt

| Page 33: [268] Formatted | Smith, Molly B | 11/24/16 2:13:00 PM |

None, Space Before:  0 pt, Line spacing:  single, Don't keep with next, Don't keep lines together

| Page 33: [269] Formatted | Smith, Molly B | 11/24/16 2:05:00 PM |

Font:12 pt, Not Bold, Not Italic

| Page 33: [270] Formatted | Smith, Molly B | 11/24/16 2:05:00 PM |

Font:12 pt

| Page 33: [271] Formatted | Smith, Molly B | 11/24/16 2:13:00 PM |

Space Before:  0 pt, Line spacing:  single

| Page 33: [272] Formatted | Smith, Molly B | 11/24/16 2:05:00 PM |

Font:12 pt, Not Bold, Not Italic, Font color: Auto

| Page 33: [273] Formatted | Smith, Molly B | 11/24/16 2:05:00 PM |

Font:12 pt

| Page 33: [273] Formatted | Smith, Molly B | 11/24/16 2:05:00 PM |

Font:12 pt

| Page 33: [274] Formatted | Smith, Molly B | 11/24/16 2:05:00 PM |

Font:12 pt

| Page 33: [274] Formatted | Smith, Molly B | 11/24/16 2:05:00 PM |

Font:12 pt

| Page 33: [275] Formatted | Smith, Molly B | 11/24/16 2:05:00 PM |

Font:12 pt

| Page 33: [276] Formatted | Smith, Molly B | 11/24/16 2:13:00 PM |

None, Space Before:  0 pt, Line spacing:  single, Don't keep with next, Don't keep lines together

| Page 33: [277] Formatted | Smith, Molly B | 11/24/16 2:05:00 PM |

Font:12 pt, Not Bold, Not Italic, Font color: Auto

| Page 33: [278] Formatted | Smith, Molly B | 11/24/16 2:05:00 PM |

Font:12 pt

| Page 33: [279] Formatted | Smith, Molly B | 11/24/16 2:13:00 PM |

Space Before:  0 pt, Line spacing:  single

| Page 33: [280] Formatted | Smith, Molly B | 11/24/16 2:05:00 PM |

Font:12 pt, Not Italic, Font color: Auto

| Page 33: [281] Formatted | Smith, Molly B | 11/24/16 2:05:00 PM |

Font:12 pt

| Page 33: [281] Formatted | Smith, Molly B | 11/24/16 2:05:00 PM |

Font:12 pt

| Page 33: [282] Formatted | Smith, Molly B | 11/24/16 2:05:00 PM |

Font:12 pt

| Page 33: [282] Formatted | Smith, Molly B | 11/24/16 2:05:00 PM |

Font:12 pt

| Page 33: [283] Formatted | Smith, Molly B | 11/24/16 2:05:00 PM |

Font:12 pt

| Page 33: [283] Formatted | Smith, Molly B | 11/24/16 2:05:00 PM |

Font:12 pt

| Page 33: [284] Formatted | Smith, Molly B | 11/24/16 2:05:00 PM |

Font:12 pt

| Page 33: [285] Formatted | Smith, Molly B | 11/24/16 2:13:00 PM |

None, Space Before:  0 pt, Line spacing:  single, Don't keep with next, Don't keep lines together

| Page 33: [286] Formatted | Smith, Molly B | 11/24/16 2:05:00 PM |

Font:12 pt, Not Italic

| Page 33: [287] Formatted | Smith, Molly B | 11/24/16 2:05:00 PM |

Font:12 pt

| Page 33: [288] Formatted | Smith, Molly B | 11/24/16 2:13:00 PM |

None, Space Before:  0 pt, Line spacing:  single, Don't keep with next, Don't keep lines together

| Page 33: [289] Formatted | Smith, Molly B | 11/24/16 2:05:00 PM |

Font:12 pt, Not Bold, Not Italic, Font color: Auto

| Page 33: [290] Formatted | Smith, Molly B | 11/24/16 2:05:00 PM |

Font:12 pt

| Page 33: [290] Formatted | Smith, Molly B | 11/24/16 2:05:00 PM |

Font:12 pt

| Page 33: [291] Formatted | Smith, Molly B | 11/24/16 2:05:00 PM |

Font:12 pt

| Page 33: [291] Formatted | Smith, Molly B | 11/24/16 2:05:00 PM |

Font:12 pt

| Page 33: [292] Formatted | Smith, Molly B | 11/24/16 2:05:00 PM |

Font:12 pt

| Page 33: [293] Formatted | Smith, Molly B | 11/24/16 2:13:00 PM |

Space Before:  0 pt, Line spacing:  single

| Page 33: [294] Formatted | Smith, Molly B | 11/24/16 2:05:00 PM |

Font:12 pt, Not Italic, Font color: Auto

| Page 33: [295] Formatted | Smith, Molly B | 11/24/16 2:05:00 PM |

Font:12 pt

| Page 33: [295] Formatted | Smith, Molly B | 11/24/16 2:05:00 PM |

Font:12 pt

| Page 33: [296] Formatted | Smith, Molly B | 11/24/16 2:05:00 PM |

Font:12 pt

| Page 33: [296] Formatted | Smith, Molly B | 11/24/16 2:05:00 PM |

Font:12 pt

| Page 33: [297] Formatted | Smith, Molly B | 11/24/16 2:05:00 PM |

Font:12 pt

| Page 33: [298] Formatted | Smith, Molly B | 11/24/16 2:13:00 PM |

None, Space Before:  0 pt, Line spacing:  single, Don't keep with next, Don't keep lines together

| Page 33: [299] Formatted | Smith, Molly B | 11/24/16 2:05:00 PM |

Font:12 pt, Not Bold, Not Italic, Font color: Auto

| Page 33: [300] Formatted | Smith, Molly B | 11/24/16 2:05:00 PM |

Font:12 pt

| Page 33: [301] Formatted | Smith, Molly B | 11/24/16 2:13:00 PM |

Space Before:  0 pt, Line spacing:  single

| Page 33: [302] Formatted | Smith, Molly B | 11/24/16 2:05:00 PM |

Font:12 pt, Not Italic, Font color: Auto

| Page 33: [303] Formatted | Smith, Molly B | 11/24/16 2:05:00 PM |

Font:12 pt

| Page 33: [303] Formatted | Smith, Molly B | 11/24/16 2:05:00 PM |

Font:12 pt

| Page 33: [304] Formatted | Smith, Molly B | 11/24/16 2:05:00 PM |

Font:12 pt

| Page 33: [305] Formatted | Smith, Molly B | 11/24/16 2:13:00 PM |

None, Space Before:  0 pt, Line spacing:  single, Don't keep with next, Don't keep lines together

| Page 33: [306] Formatted | Smith, Molly B | 11/24/16 2:05:00 PM |

Font:12 pt, Not Bold, Not Italic, Font color: Auto

| Page 33: [307] Formatted | Smith, Molly B | 11/24/16 2:05:00 PM |

Font:12 pt

| Page 33: [307] Formatted | Smith, Molly B | 11/24/16 2:05:00 PM |

Font:12 pt

| Page 33: [308] Formatted | Smith, Molly B | 11/24/16 2:05:00 PM |

Font:12 pt

| Page 33: [309] Formatted | Smith, Molly B | 11/24/16 2:13:00 PM |

Space Before:  0 pt, Line spacing:  single

| Page 33: [310] Formatted | Smith, Molly B | 11/24/16 2:05:00 PM |

Font:12 pt, Not Italic

| Page 33: [311] Formatted | Smith, Molly B | 11/24/16 2:05:00 PM |

Font:12 pt

| Page 33: [312] Formatted | Smith, Molly B | 11/24/16 2:13:00 PM |

None, Space Before: 0 pt, Line spacing: single, Don't keep with next, Don't keep lines together

| Page 33: [313] Formatted | Smith, Molly B | 11/24/16 2:05:00 PM |
|---|---|---|

Font:12 pt, Not Bold, Not Italic, Font color: Auto

| Page 33: [314] Formatted | Smith, Molly B | 11/24/16 2:05:00 PM |
|---|---|---|

Font:12 pt

| Page 33: [315] Formatted | Smith, Molly B | 11/24/16 2:13:00 PM |
|---|---|---|

Space Before: 0 pt, Line spacing: single

| Page 33: [316] Formatted | Smith, Molly B | 11/24/16 2:05:00 PM |
|---|---|---|

Font:12 pt, Not Italic, Font color: Auto

| Page 33: [317] Formatted | Smith, Molly B | 11/24/16 2:05:00 PM |
|---|---|---|

Font:12 pt

| Page 33: [317] Formatted | Smith, Molly B | 11/24/16 2:05:00 PM |
|---|---|---|

Font:12 pt

| Page 33: [318] Formatted | Smith, Molly B | 11/24/16 2:05:00 PM |
|---|---|---|

Font:12 pt

| Page 33: [318] Formatted | Smith, Molly B | 11/24/16 2:05:00 PM |
|---|---|---|

Font:12 pt

| Page 33: [319] Formatted | Smith, Molly B | 11/24/16 2:05:00 PM |
|---|---|---|

Font:12 pt

| Page 33: [319] Formatted | Smith, Molly B | 11/24/16 2:05:00 PM |
|---|---|---|

Font:12 pt

| Page 33: [320] Formatted | Smith, Molly B | 11/24/16 2:05:00 PM |
|---|---|---|

Font:12 pt

| Page 33: [320] Formatted | Smith, Molly B | 11/24/16 2:05:00 PM |
|---|---|---|

Font:12 pt

| Page 33: [321] Formatted | Smith, Molly B | 11/24/16 2:05:00 PM |
|---|---|---|

Font:12 pt

| Page 33: [322] Formatted | Smith, Molly B | 11/24/16 2:13:00 PM |
|---|---|---|

None, Space Before: 0 pt, Line spacing: single, Don't keep with next, Don't keep lines together

| Page 33: [323] Formatted | Smith, Molly B | 11/24/16 2:05:00 PM |
|---|---|---|

Font:12 pt, Not Bold, Not Italic, Font color: Auto

| Page 33: [324] Formatted | Smith, Molly B | 11/24/16 2:05:00 PM |
|---|---|---|

Font:12 pt

| Page 33: [325] Formatted | Smith, Molly B | 11/24/16 2:13:00 PM |
|---|---|---|

Space Before: 0 pt, Line spacing: single

| Page 33: [326] Formatted | Smith, Molly B | 11/24/16 2:05:00 PM |
|---|---|---|

Font:12 pt, Not Italic, Font color: Auto

| Page 33: [327] Formatted | Smith, Molly B | 11/24/16 2:05:00 PM |

Font:12 pt

| Page 33: [327] Formatted | Smith, Molly B | 11/24/16 2:05:00 PM |

Font:12 pt

| Page 33: [328] Formatted | Smith, Molly B | 11/24/16 2:05:00 PM |

Font:12 pt

| Page 33: [328] Formatted | Smith, Molly B | 11/24/16 2:05:00 PM |

Font:12 pt

| Page 33: [329] Formatted | Smith, Molly B | 11/24/16 2:05:00 PM |

Font:12 pt

| Page 33: [329] Formatted | Smith, Molly B | 11/24/16 2:05:00 PM |

Font:12 pt

| Page 33: [330] Formatted | Smith, Molly B | 11/24/16 2:05:00 PM |

Font:12 pt

| Page 33: [331] Formatted | Smith, Molly B | 11/24/16 2:13:00 PM |

None, Space Before:  0 pt, Line spacing:  single, Don't keep with next, Don't keep lines together

| Page 33: [332] Formatted | Smith, Molly B | 11/24/16 2:05:00 PM |

Font:12 pt, Not Italic

| Page 33: [333] Formatted | Smith, Molly B | 11/24/16 2:05:00 PM |

Font:12 pt

| Page 33: [334] Formatted | Smith, Molly B | 11/24/16 2:13:00 PM |

None, Space Before:  0 pt, Line spacing:  single, Don't keep with next, Don't keep lines together

| Page 33: [335] Formatted | Smith, Molly B | 11/24/16 2:05:00 PM |

Font:12 pt, Not Bold, Not Italic, Font color: Auto

| Page 33: [336] Formatted | Smith, Molly B | 11/24/16 2:05:00 PM |

Font:12 pt

| Page 33: [337] Formatted | Smith, Molly B | 11/24/16 2:13:00 PM |

Space Before:  0 pt, Line spacing:  single

| Page 33: [338] Formatted | Smith, Molly B | 11/24/16 2:05:00 PM |

Font:12 pt, Not Italic, Font color: Auto

| Page 33: [339] Formatted | Smith, Molly B | 11/24/16 2:05:00 PM |

Font:12 pt

| Page 33: [339] Formatted | Smith, Molly B | 11/24/16 2:05:00 PM |

Font:12 pt

| Page 33: [340] Formatted | Smith, Molly B | 11/24/16 2:05:00 PM |

Font:12 pt

| Page 33: [340] Formatted | Smith, Molly B | 11/24/16 2:05:00 PM |

Font:12 pt

| Page 33: [341] Formatted | Smith, Molly B | 11/24/16 2:05:00 PM |

Font:12 pt

| Page 33: [341] Formatted | Smith, Molly B | 11/24/16 2:05:00 PM |

Font:12 pt

| Page 33: [342] Formatted | Smith, Molly B | 11/24/16 2:05:00 PM |

Font:12 pt

| Page 33: [343] Formatted | Smith, Molly B | 11/24/16 2:13:00 PM |

None, Space Before:  0 pt, Line spacing:  single, Don't keep with next, Don't keep lines together

| Page 33: [344] Formatted | Smith, Molly B | 11/24/16 2:05:00 PM |

Font:12 pt, Not Italic

| Page 33: [345] Formatted | Smith, Molly B | 11/24/16 2:05:00 PM |

Font:12 pt

| Page 33: [346] Formatted | Smith, Molly B | 11/24/16 2:13:00 PM |

None, Space Before:  0 pt, Line spacing:  single, Don't keep with next, Don't keep lines together

| Page 33: [347] Formatted | Smith, Molly B | 11/24/16 2:05:00 PM |

Font:12 pt, Not Bold, Not Italic, Font color: Auto

| Page 33: [348] Formatted | Smith, Molly B | 11/24/16 2:05:00 PM |

Font:12 pt

| Page 33: [349] Formatted | Smith, Molly B | 11/24/16 2:13:00 PM |

Space Before:  0 pt, Line spacing:  single

| Page 33: [350] Formatted | Smith, Molly B | 11/24/16 2:05:00 PM |

Font:12 pt, Not Bold, Not Italic, Font color: Auto

| Page 33: [351] Formatted | Smith, Molly B | 11/24/16 2:05:00 PM |

Font:12 pt

| Page 33: [351] Formatted | Smith, Molly B | 11/24/16 2:05:00 PM |

Font:12 pt

| Page 33: [352] Formatted | Smith, Molly B | 11/24/16 2:05:00 PM |

Font:12 pt

| Page 33: [352] Formatted | Smith, Molly B | 11/24/16 2:05:00 PM |

Font:12 pt

| Page 33: [353] Formatted | Smith, Molly B | 11/24/16 2:05:00 PM |

Font:12 pt

| Page 33: [354] Formatted | Smith, Molly B | 11/24/16 2:13:00 PM |

Line spacing:  single

| Page 33: [355] Formatted | Smith, Molly B | 11/24/16 2:05:00 PM |

Font:12 pt

| Page 33: [356] Formatted | Smith, Molly B | 11/24/16 2:13:00 PM |
|---|---|---|

None, Space Before:  0 pt, Line spacing:  single, Don't keep with next, Don't keep lines together

| Page 33: [357] Formatted | Smith, Molly B | 11/24/16 2:05:00 PM |
|---|---|---|

Font:12 pt, Not Bold, Not Italic, Font color: Auto

| Page 33: [358] Formatted | Smith, Molly B | 11/24/16 2:05:00 PM |
|---|---|---|

Font:12 pt

| Page 33: [359] Formatted | Smith, Molly B | 11/24/16 2:13:00 PM |
|---|---|---|

Space Before:  0 pt, Line spacing:  single

| Page 33: [360] Formatted | Smith, Molly B | 11/24/16 2:05:00 PM |
|---|---|---|

Font:12 pt, Not Italic, Font color: Auto

| Page 33: [361] Formatted | Smith, Molly B | 11/24/16 2:05:00 PM |
|---|---|---|

Font:12 pt

| Page 33: [362] Formatted | Smith, Molly B | 11/24/16 2:13:00 PM |
|---|---|---|

Line spacing:  single

| Page 33: [363] Formatted | Smith, Molly B | 11/24/16 2:13:00 PM |
|---|---|---|

None, Space Before:  0 pt, Line spacing:  single, Don't keep with next, Don't keep lines together

| Page 33: [364] Formatted | Smith, Molly B | 11/24/16 2:05:00 PM |
|---|---|---|

Font:12 pt, Not Italic, Font color: Auto

| Page 33: [365] Formatted | Smith, Molly B | 11/24/16 2:05:00 PM |
|---|---|---|

Font:12 pt

| Page 33: [365] Formatted | Smith, Molly B | 11/24/16 2:05:00 PM |
|---|---|---|

Font:12 pt

| Page 33: [366] Formatted | Smith, Molly B | 11/24/16 2:05:00 PM |
|---|---|---|

Font:12 pt

| Page 33: [367] Formatted | Smith, Molly B | 11/24/16 2:13:00 PM |
|---|---|---|

Line spacing:  single

| Page 33: [368] Formatted | Smith, Molly B | 11/24/16 2:05:00 PM |
|---|---|---|

Font:12 pt

| Page 34: [369] Formatted | Smith, Molly B | 11/24/16 2:14:00 PM |
|---|---|---|

Line spacing:  single

| Page 34: [370] Formatted Table | Smith, Molly B | 11/24/16 2:14:00 PM |
|---|---|---|

Formatted Table

| Page 34: [371] Formatted | Smith, Molly B | 11/24/16 2:05:00 PM |
|---|---|---|

Font:12 pt, Not Italic

| Page 34: [372] Formatted | Smith, Molly B | 11/24/16 2:05:00 PM |
|---|---|---|

Font:12 pt

| Page 34: [372] Formatted | Smith, Molly B | 11/24/16 2:05:00 PM |
|---|---|---|

Font:12 pt

| Page 34: [373] Formatted | Smith, Molly B | 11/24/16 2:05:00 PM |
|---|---|---|

Font:12 pt

| Page 34: [373] Formatted | Smith, Molly B | 11/24/16 2:05:00 PM |
|---|---|---|

Font:12 pt

| Page 34: [374] Formatted | Smith, Molly B | 11/24/16 2:05:00 PM |
|---|---|---|

Font:12 pt

| Page 34: [375] Formatted | Smith, Molly B | 11/24/16 2:14:00 PM |
|---|---|---|

Line spacing:  single

| Page 34: [376] Formatted | Smith, Molly B | 11/24/16 2:05:00 PM |
|---|---|---|

Font:12 pt, Not Italic, Font color: Auto

| Page 34: [377] Formatted | Smith, Molly B | 11/24/16 2:05:00 PM |
|---|---|---|

Font:12 pt

| Page 34: [377] Formatted | Smith, Molly B | 11/24/16 2:05:00 PM |
|---|---|---|

Font:12 pt

| Page 34: [378] Formatted | Smith, Molly B | 11/24/16 2:05:00 PM |
|---|---|---|

Font:12 pt

| Page 34: [378] Formatted | Smith, Molly B | 11/24/16 2:05:00 PM |
|---|---|---|

Font:12 pt

| Page 34: [379] Formatted | Smith, Molly B | 11/24/16 2:05:00 PM |
|---|---|---|

Font:12 pt

| Page 34: [379] Formatted | Smith, Molly B | 11/24/16 2:05:00 PM |
|---|---|---|

Font:12 pt

| Page 34: [380] Formatted | Smith, Molly B | 11/24/16 2:05:00 PM |
|---|---|---|

Font:12 pt

| Page 34: [380] Formatted | Smith, Molly B | 11/24/16 2:05:00 PM |
|---|---|---|

Font:12 pt

| Page 34: [381] Formatted | Smith, Molly B | 11/24/16 2:05:00 PM |
|---|---|---|

Font:12 pt

| Page 34: [381] Formatted | Smith, Molly B | 11/24/16 2:05:00 PM |
|---|---|---|

Font:12 pt

| Page 34: [382] Formatted | Smith, Molly B | 11/24/16 2:05:00 PM |
|---|---|---|

Font:12 pt

| Page 34: [382] Formatted | Smith, Molly B | 11/24/16 2:05:00 PM |
|---|---|---|

Font:12 pt

| Page 34: [383] Formatted | Smith, Molly B | 11/24/16 2:05:00 PM |
|---|---|---|

Font:12 pt

| Page 34: [384] Formatted | Smith, Molly B | 11/24/16 2:14:00 PM |
|---|---|---|

Line spacing:  single

| Page 34: [385] Formatted | Smith, Molly B | 11/24/16 2:05:00 PM |
|---|---|---|

Font:12 pt, Not Italic, Font color: Auto

| Page 34: [386] Formatted | Smith, Molly B | 11/24/16 2:05:00 PM |
|---|---|---|

Font:12 pt

| Page 34: [386] Formatted | Smith, Molly B | 11/24/16 2:05:00 PM |
|---|---|---|

Font:12 pt

| Page 34: [387] Formatted | Smith, Molly B | 11/24/16 2:05:00 PM |
|---|---|---|

Font:12 pt

| Page 34: [387] Formatted | Smith, Molly B | 11/24/16 2:05:00 PM |
|---|---|---|

Font:12 pt

| Page 34: [388] Formatted | Smith, Molly B | 11/24/16 2:05:00 PM |
|---|---|---|

Font:12 pt

| Page 34: [388] Formatted | Smith, Molly B | 11/24/16 2:05:00 PM |
|---|---|---|

Font:12 pt

| Page 34: [389] Formatted | Smith, Molly B | 11/24/16 2:05:00 PM |
|---|---|---|

Font:12 pt

| Page 34: [389] Formatted | Smith, Molly B | 11/24/16 2:05:00 PM |
|---|---|---|

Font:12 pt

| Page 34: [390] Formatted | Smith, Molly B | 11/24/16 2:05:00 PM |
|---|---|---|

Font:12 pt

| Page 34: [390] Formatted | Smith, Molly B | 11/24/16 2:05:00 PM |
|---|---|---|

Font:12 pt

| Page 34: [391] Formatted | Smith, Molly B | 11/24/16 2:05:00 PM |
|---|---|---|

Font:12 pt

| Page 34: [391] Formatted | Smith, Molly B | 11/24/16 2:05:00 PM |
|---|---|---|

Font:12 pt

| Page 34: [392] Formatted | Smith, Molly B | 11/24/16 2:05:00 PM |
|---|---|---|

Font:12 pt

| Page 34: [393] Formatted | Smith, Molly B | 11/24/16 2:14:00 PM |
|---|---|---|

Line spacing:  single

| Page 34: [394] Formatted | Smith, Molly B | 11/24/16 2:05:00 PM |
|---|---|---|

Font:12 pt, Not Italic, Font color: Auto

| Page 34: [395] Formatted | Smith, Molly B | 11/24/16 2:05:00 PM |
|---|---|---|

Font:12 pt

| Page 34: [395] Formatted | Smith, Molly B | 11/24/16 2:05:00 PM |
|---|---|---|

Font:12 pt

| Page 34: [396] Formatted | Smith, Molly B | 11/24/16 2:05:00 PM |
|---|---|---|

Font:12 pt

| Page 34: [396] Formatted | Smith, Molly B | 11/24/16 2:05:00 PM |
|---|---|---|

Font:12 pt

| Page 34: [397] Formatted | Smith, Molly B | 11/24/16 2:05:00 PM |
|---|---|---|

Font:12 pt

| Page 34: [397] Formatted | Smith, Molly B | 11/24/16 2:05:00 PM |
|---|---|---|

Font:12 pt

| Page 34: [398] Formatted | Smith, Molly B | 11/24/16 2:05:00 PM |
|---|---|---|

Font:12 pt

| Page 34: [398] Formatted | Smith, Molly B | 11/24/16 2:05:00 PM |
|---|---|---|

Font:12 pt

| Page 34: [399] Formatted | Smith, Molly B | 11/24/16 2:05:00 PM |
|---|---|---|

Font:12 pt

| Page 34: [399] Formatted | Smith, Molly B | 11/24/16 2:05:00 PM |
|---|---|---|

Font:12 pt

| Page 34: [400] Formatted | Smith, Molly B | 11/24/16 2:05:00 PM |
|---|---|---|

Font:12 pt

| Page 34: [400] Formatted | Smith, Molly B | 11/24/16 2:05:00 PM |
|---|---|---|

Font:12 pt

| Page 34: [401] Formatted | Smith, Molly B | 11/24/16 2:05:00 PM |
|---|---|---|

Font:12 pt

| Page 34: [402] Formatted | Smith, Molly B | 11/24/16 2:14:00 PM |
|---|---|---|

Line spacing:  single

| Page 34: [403] Formatted | Smith, Molly B | 11/24/16 2:05:00 PM |
|---|---|---|

Font:12 pt, Not Italic, Font color: Auto

| Page 34: [404] Formatted | Smith, Molly B | 11/24/16 2:05:00 PM |
|---|---|---|

Font:12 pt

| Page 34: [404] Formatted | Smith, Molly B | 11/24/16 2:05:00 PM |
|---|---|---|

Font:12 pt

| Page 34: [405] Formatted | Smith, Molly B | 11/24/16 2:05:00 PM |
|---|---|---|

Font:12 pt

| Page 34: [405] Formatted | Smith, Molly B | 11/24/16 2:05:00 PM |
|---|---|---|

Font:12 pt

| Page 34: [406] Formatted | Smith, Molly B | 11/24/16 2:05:00 PM |
|---|---|---|

Font:12 pt

| Page 34: [406] Formatted | Smith, Molly B | 11/24/16 2:05:00 PM |
|---|---|---|

Font:12 pt

| Page 34: [407] Formatted | Smith, Molly B | 11/24/16 2:05:00 PM |
|---|---|---|

Font:12 pt

| Page 34: [407] Formatted | Smith, Molly B | 11/24/16 2:05:00 PM |
|---|---|---|

Font:12 pt

| Page 34: [408] Formatted | Smith, Molly B | 11/24/16 2:05:00 PM |
|---|---|---|

Font:12 pt

| Page 34: [408] Formatted | Smith, Molly B | 11/24/16 2:05:00 PM |
|---|---|---|

Font:12 pt

| Page 34: [409] Formatted | Smith, Molly B | 11/24/16 2:05:00 PM |
|---|---|---|

Font:12 pt

| Page 34: [409] Formatted | Smith, Molly B | 11/24/16 2:05:00 PM |
|---|---|---|

Font:12 pt

| Page 34: [410] Formatted | Smith, Molly B | 11/24/16 2:05:00 PM |
|---|---|---|

Font:12 pt

| Page 34: [411] Formatted | Smith, Molly B | 11/24/16 2:14:00 PM |
|---|---|---|

Line spacing:  single

| Page 34: [412] Formatted | Smith, Molly B | 11/24/16 2:05:00 PM |
|---|---|---|

Font:12 pt, Not Italic, Font color: Auto

| Page 34: [413] Formatted | Smith, Molly B | 11/24/16 2:05:00 PM |
|---|---|---|

Font:12 pt

| Page 34: [413] Formatted | Smith, Molly B | 11/24/16 2:05:00 PM |
|---|---|---|

Font:12 pt

| Page 34: [414] Formatted | Smith, Molly B | 11/24/16 2:05:00 PM |
|---|---|---|

Font:12 pt

| Page 34: [414] Formatted | Smith, Molly B | 11/24/16 2:05:00 PM |
|---|---|---|

Font:12 pt

| Page 34: [415] Formatted | Smith, Molly B | 11/24/16 2:05:00 PM |
|---|---|---|

Font:12 pt

| Page 34: [415] Formatted | Smith, Molly B | 11/24/16 2:05:00 PM |
|---|---|---|

Font:12 pt

| Page 34: [416] Formatted | Smith, Molly B | 11/24/16 2:05:00 PM |
|---|---|---|

Font:12 pt

| Page 34: [416] Formatted | Smith, Molly B | 11/24/16 2:05:00 PM |
|---|---|---|

Font:12 pt

| Page 34: [417] Formatted | Smith, Molly B | 11/24/16 2:05:00 PM |
|---|---|---|

Font:12 pt

| Page 34: [417] Formatted | Smith, Molly B | 11/24/16 2:05:00 PM |
|---|---|---|

Font:12 pt

| Page 34: [418] Formatted | Smith, Molly B | 11/24/16 2:05:00 PM |
|---|---|---|

Font:12 pt

| Page 34: [418] Formatted | Smith, Molly B | 11/24/16 2:05:00 PM |
|---|---|---|

Font:12 pt

| Page 34: [419] Formatted | Smith, Molly B | 11/24/16 2:05:00 PM |
|---|---|---|

Font:12 pt

| Page 34: [420] Formatted | Smith, Molly B | 11/24/16 2:14:00 PM |
|---|---|---|

Line spacing:  single

| Page 34: [421] Formatted | Smith, Molly B | 11/24/16 2:05:00 PM |
|---|---|---|

Font:12 pt, Not Italic, Font color: Auto

| Page 34: [422] Formatted | Smith, Molly B | 11/24/16 2:05:00 PM |
|---|---|---|

Font:12 pt

| Page 34: [422] Formatted | Smith, Molly B | 11/24/16 2:05:00 PM |
|---|---|---|

Font:12 pt

| Page 34: [423] Formatted | Smith, Molly B | 11/24/16 2:05:00 PM |
|---|---|---|

Font:12 pt

| Page 34: [423] Formatted | Smith, Molly B | 11/24/16 2:05:00 PM |
|---|---|---|

Font:12 pt

| Page 34: [424] Formatted | Smith, Molly B | 11/24/16 2:05:00 PM |
|---|---|---|

Font:12 pt

| Page 34: [424] Formatted | Smith, Molly B | 11/24/16 2:05:00 PM |
|---|---|---|

Font:12 pt

| Page 34: [425] Formatted | Smith, Molly B | 11/24/16 2:05:00 PM |
|---|---|---|

Font:12 pt

| Page 34: [425] Formatted | Smith, Molly B | 11/24/16 2:05:00 PM |
|---|---|---|

Font:12 pt

| Page 34: [426] Formatted | Smith, Molly B | 11/24/16 2:05:00 PM |
|---|---|---|

Font:12 pt

| Page 34: [426] Formatted | Smith, Molly B | 11/24/16 2:05:00 PM |
|---|---|---|

Font:12 pt

| Page 34: [427] Formatted | Smith, Molly B | 11/24/16 2:05:00 PM |
|---|---|---|

Font:12 pt

| Page 34: [427] Formatted | Smith, Molly B | 11/24/16 2:05:00 PM |
|---|---|---|

Font:12 pt

| Page 34: [428] Formatted | Smith, Molly B | 11/24/16 2:05:00 PM |
|---|---|---|

Font:12 pt

| Page 34: [429] Formatted | Smith, Molly B | 11/24/16 2:14:00 PM |
|---|---|---|

Line spacing:  single

| Page 34: [430] Formatted | Smith, Molly B | 11/24/16 2:05:00 PM |
|---|---|---|

Font:12 pt, Not Italic, Font color: Auto

| Page 34: [431] Formatted | Smith, Molly B | 11/24/16 2:05:00 PM |
|---|---|---|

Font:12 pt

| Page 34: [431] Formatted | Smith, Molly B | 11/24/16 2:05:00 PM |
|---|---|---|

Font:12 pt

| Page 34: [432] Formatted | Smith, Molly B | 11/24/16 2:05:00 PM |
|---|---|---|

Font:12 pt

| Page 34: [432] Formatted | Smith, Molly B | 11/24/16 2:05:00 PM |
|---|---|---|

Font:12 pt

| Page 34: [433] Formatted | Smith, Molly B | 11/24/16 2:05:00 PM |
|---|---|---|

Font:12 pt

| Page 34: [433] Formatted | Smith, Molly B | 11/24/16 2:05:00 PM |
|---|---|---|

Font:12 pt

| Page 34: [434] Formatted | Smith, Molly B | 11/24/16 2:05:00 PM |
|---|---|---|

Font:12 pt

| Page 34: [434] Formatted | Smith, Molly B | 11/24/16 2:05:00 PM |
|---|---|---|

Font:12 pt

| Page 34: [435] Formatted | Smith, Molly B | 11/24/16 2:05:00 PM |
|---|---|---|

Font:12 pt

| Page 34: [435] Formatted | Smith, Molly B | 11/24/16 2:05:00 PM |
|---|---|---|

Font:12 pt

| Page 34: [436] Formatted | Smith, Molly B | 11/24/16 2:05:00 PM |
|---|---|---|

Font:12 pt

| Page 34: [436] Formatted | Smith, Molly B | 11/24/16 2:05:00 PM |
|---|---|---|

Font:12 pt

| Page 34: [437] Formatted | Smith, Molly B | 11/24/16 2:05:00 PM |
|---|---|---|

Font:12 pt

| Page 34: [438] Formatted | Smith, Molly B | 11/24/16 2:14:00 PM |
|---|---|---|

Line spacing:  single

| Page 34: [439] Formatted | Smith, Molly B | 11/24/16 2:05:00 PM |
|---|---|---|

Font:12 pt, Not Italic, Font color: Auto

| Page 34: [440] Formatted | Smith, Molly B | 11/24/16 2:05:00 PM |
|---|---|---|

Font:12 pt

| Page 34: [440] Formatted | Smith, Molly B | 11/24/16 2:05:00 PM |

Font:12 pt

| Page 34: [441] Formatted | Smith, Molly B | 11/24/16 2:05:00 PM |

Font:12 pt

| Page 34: [441] Formatted | Smith, Molly B | 11/24/16 2:05:00 PM |

Font:12 pt

| Page 34: [442] Formatted | Smith, Molly B | 11/24/16 2:05:00 PM |

Font:12 pt

| Page 34: [442] Formatted | Smith, Molly B | 11/24/16 2:05:00 PM |

Font:12 pt

| Page 34: [443] Formatted | Smith, Molly B | 11/24/16 2:05:00 PM |

Font:12 pt

| Page 34: [443] Formatted | Smith, Molly B | 11/24/16 2:05:00 PM |

Font:12 pt

| Page 34: [444] Formatted | Smith, Molly B | 11/24/16 2:05:00 PM |

Font:12 pt

| Page 34: [444] Formatted | Smith, Molly B | 11/24/16 2:05:00 PM |

Font:12 pt

| Page 34: [445] Formatted | Smith, Molly B | 11/24/16 2:05:00 PM |

Font:12 pt

| Page 34: [445] Formatted | Smith, Molly B | 11/24/16 2:05:00 PM |

Font:12 pt

| Page 34: [446] Formatted | Smith, Molly B | 11/24/16 2:05:00 PM |

Font:12 pt

| Page 34: [447] Formatted | Smith, Molly B | 11/24/16 2:14:00 PM |

Line spacing:  single

| Page 34: [448] Formatted | Smith, Molly B | 11/24/16 2:05:00 PM |

Font:12 pt, Not Italic, Font color: Auto

| Page 34: [449] Formatted | Smith, Molly B | 11/24/16 2:05:00 PM |

Font:12 pt

| Page 34: [449] Formatted | Smith, Molly B | 11/24/16 2:05:00 PM |

Font:12 pt

| Page 34: [450] Formatted | Smith, Molly B | 11/24/16 2:05:00 PM |

Font:12 pt

| Page 34: [450] Formatted | Smith, Molly B | 11/24/16 2:05:00 PM |

Font:12 pt

| Page 34: [451] Formatted | Smith, Molly B | 11/24/16 2:05:00 PM |
|---|---|---|

Font:12 pt

| Page 34: [451] Formatted | Smith, Molly B | 11/24/16 2:05:00 PM |
|---|---|---|

Font:12 pt

| Page 34: [452] Formatted | Smith, Molly B | 11/24/16 2:05:00 PM |
|---|---|---|

Font:12 pt

| Page 34: [452] Formatted | Smith, Molly B | 11/24/16 2:05:00 PM |
|---|---|---|

Font:12 pt

| Page 34: [453] Formatted | Smith, Molly B | 11/24/16 2:05:00 PM |
|---|---|---|

Font:12 pt

| Page 34: [453] Formatted | Smith, Molly B | 11/24/16 2:05:00 PM |
|---|---|---|

Font:12 pt

| Page 34: [454] Formatted | Smith, Molly B | 11/24/16 2:05:00 PM |
|---|---|---|

Font:12 pt

| Page 34: [454] Formatted | Smith, Molly B | 11/24/16 2:05:00 PM |
|---|---|---|

Font:12 pt

| Page 34: [455] Deleted | Cornell University | 11/14/16 5:26:00 PM |
|---|---|---|

**Table 5: Variability in Southern Hemisphere**
**Values for the monthly variability (monthly average standard deviation divided by mean) and the fraction of the variability from the seasonal cycle (climatological monthly average standard deviation, divided by monthly mean standard deviation) for the surface concentration in the model cases and data from Rio Gallego and deposition data from Kerguelen (locations listed in Table 1).**

| Model/ Observations | Rio Gallego Surface concentrations | | Kerguelen deposition | |
|---|---|---|---|---|
| | Monthly variability | Fraction variability from seasonal cycle | Monthly variability | Fraction variability from seasonal cycle |
| CAM4 (MERRA) | 5.67 | 0.33 | 0.64 | 0.41 |
| CAM4 (NCEP) | 5.89 | 0.29 | 0.82 | 0.33 |
| CAM4 (ERAI) | 7.52 | 0.30 | 4.39 | 0.43 |
| CAM4 (AMIP) | 8.98 | 0.24 | 4.47 | 0.22 |
| GCHEM (MERRA) | 1.94 | 0.43 | 0.70 | 0.47 |
| MATCH (NCEP) | 2.18 | 0.42 | 0.62 | 0.59 |
| CAM5 (AMIP) | 2.01 | 0.71 | 1.94 | 0.56 |
| Obs | 0.70 | 0.20 | 0.71 | 0.45 |

| Page 34: [456] Formatted | Smith, Molly B | 11/24/16 2:05:00 PM |
|---|---|---|

Font:12 pt

| Page 34: [457] Formatted | Smith, Molly B | 11/24/16 2:05:00 PM |
|---|---|---|

Font:12 pt, Not Italic

| Page 34: [457] Formatted | Smith, Molly B | 11/24/16 2:05:00 PM |
|---|---|---|

Font:12 pt, Not Italic

| Page 34: [457] Formatted | Smith, Molly B | 11/24/16 2:05:00 PM |
|---|---|---|

Font:12 pt, Not Italic

| Page 34: [457] Formatted | Smith, Molly B | 11/24/16 2:05:00 PM |
|---|---|---|

Font:12 pt, Not Italic

| Page 34: [457] Formatted | Smith, Molly B | 11/24/16 2:05:00 PM |
|---|---|---|

Font:12 pt, Not Italic

| Page 34: [457] Formatted | Smith, Molly B | 11/24/16 2:05:00 PM |
|---|---|---|

Font:12 pt, Not Italic

| Page 34: [457] Formatted | Smith, Molly B | 11/24/16 2:05:00 PM |
|---|---|---|

Font:12 pt, Not Italic

| Page 34: [457] Formatted | Smith, Molly B | 11/24/16 2:05:00 PM |
|---|---|---|

Font:12 pt, Not Italic

| Page 34: [457] Formatted | Smith, Molly B | 11/24/16 2:05:00 PM |
|---|---|---|

Font:12 pt, Not Italic

| Page 34: [457] Formatted | Smith, Molly B | 11/24/16 2:05:00 PM |
|---|---|---|

Font:12 pt, Not Italic

| Page 34: [457] Formatted | Smith, Molly B | 11/24/16 2:05:00 PM |
|---|---|---|

Font:12 pt, Not Italic

| Page 34: [457] Formatted | Smith, Molly B | 11/24/16 2:05:00 PM |
|---|---|---|

Font:12 pt, Not Italic

| Page 34: [457] Formatted | Smith, Molly B | 11/24/16 2:05:00 PM |
|---|---|---|

Font:12 pt, Not Italic

| Page 34: [457] Formatted | Smith, Molly B | 11/24/16 2:05:00 PM |
|---|---|---|

Font:12 pt, Not Italic

| Page 34: [457] Formatted | Smith, Molly B | 11/24/16 2:05:00 PM |
|---|---|---|

Font:12 pt, Not Italic

| Page 34: [457] Formatted | Smith, Molly B | 11/24/16 2:05:00 PM |
|---|---|---|

Font:12 pt, Not Italic

| Page 34: [457] Formatted | Smith, Molly B | 11/24/16 2:05:00 PM |
|---|---|---|

Font:12 pt, Not Italic

| Page 34: [457] Formatted | Smith, Molly B | 11/24/16 2:05:00 PM |
|---|---|---|

Font:12 pt, Not Italic

| Page 34: [457] Formatted | Smith, Molly B | 11/24/16 2:05:00 PM |
|---|---|---|

Font:12 pt, Not Italic

| Page 34: [457] Formatted | Smith, Molly B | 11/24/16 2:05:00 PM |
|---|---|---|

Font:12 pt, Not Italic

| Page 34: [457] Formatted | Smith, Molly B | 11/24/16 2:05:00 PM |
|---|---|---|

Font:12 pt, Not Italic

| Page 34: [457] Formatted | Smith, Molly B | 11/24/16 2:05:00 PM |
|---|---|---|

Font:12 pt, Not Italic

| Page 34: [457] Formatted | Smith, Molly B | 11/24/16 2:05:00 PM |
|---|---|---|

Font:12 pt, Not Italic

| Page 34: [457] Formatted | Smith, Molly B | 11/24/16 2:05:00 PM |
|---|---|---|

Font:12 pt, Not Italic

| Page 34: [457] Formatted | Smith, Molly B | 11/24/16 2:05:00 PM |
|---|---|---|

Font:12 pt, Not Italic

| Page 34: [457] Formatted | Smith, Molly B | 11/24/16 2:05:00 PM |
|---|---|---|

Font:12 pt, Not Italic

| Page 34: [457] Formatted | Smith, Molly B | 11/24/16 2:05:00 PM |
|---|---|---|

Font:12 pt, Not Italic

| Page 34: [457] Formatted | Smith, Molly B | 11/24/16 2:05:00 PM |
|---|---|---|

Font:12 pt, Not Italic

| Page 34: [457] Formatted | Smith, Molly B | 11/24/16 2:05:00 PM |
|---|---|---|

Font:12 pt, Not Italic

| Page 34: [457] Formatted | Smith, Molly B | 11/24/16 2:05:00 PM |
|---|---|---|

Font:12 pt, Not Italic

| Page 34: [457] Formatted | Smith, Molly B | 11/24/16 2:05:00 PM |
|---|---|---|

Font:12 pt, Not Italic

| Page 34: [457] Formatted | Smith, Molly B | 11/24/16 2:05:00 PM |
|---|---|---|

Font:12 pt, Not Italic

| Page 34: [457] Formatted | Smith, Molly B | 11/24/16 2:05:00 PM |
|---|---|---|

Font:12 pt, Not Italic

| Page 34: [457] Formatted | Smith, Molly B | 11/24/16 2:05:00 PM |
|---|---|---|

Font:12 pt, Not Italic

| Page 34: [457] Formatted | Smith, Molly B | 11/24/16 2:05:00 PM |
|---|---|---|

Font:12 pt, Not Italic

| Page 34: [457] Formatted | Smith, Molly B | 11/24/16 2:05:00 PM |
|---|---|---|

Font:12 pt, Not Italic

| Page 34: [457] Formatted | Smith, Molly B | 11/24/16 2:05:00 PM |
|---|---|---|

Font:12 pt, Not Italic

| Page 34: [457] Formatted | Smith, Molly B | 11/24/16 2:05:00 PM |
|---|---|---|

Font:12 pt, Not Italic

| Page 34: [457] Formatted | Smith, Molly B | 11/24/16 2:05:00 PM |
|---|---|---|

Font:12 pt, Not Italic

| Page 34: [457] Formatted | Smith, Molly B | 11/24/16 2:05:00 PM |
|---|---|---|

Font:12 pt, Not Italic

| Page 34: [457] Formatted | Smith, Molly B | 11/24/16 2:05:00 PM |
|---|---|---|

Font:12 pt, Not Italic

| Page 34: [457] Formatted | Smith, Molly B | 11/24/16 2:05:00 PM |
|---|---|---|

Font:12 pt, Not Italic

| Page 34: [457] Formatted | Smith, Molly B | 11/24/16 2:05:00 PM |
|---|---|---|

Font:12 pt, Not Italic

| Page 34: [457] Formatted | Smith, Molly B | 11/24/16 2:05:00 PM |
|---|---|---|

Font:12 pt, Not Italic

| Page 34: [457] Formatted | Smith, Molly B | 11/24/16 2:05:00 PM |
|---|---|---|

Font:12 pt, Not Italic

| Page 34: [457] Formatted | Smith, Molly B | 11/24/16 2:05:00 PM |
|---|---|---|

Font:12 pt, Not Italic

| Page 34: [457] Formatted | Smith, Molly B | 11/24/16 2:05:00 PM |
|---|---|---|

Font:12 pt, Not Italic

| Page 34: [457] Formatted | Smith, Molly B | 11/24/16 2:05:00 PM |
|---|---|---|

Font:12 pt, Not Italic

| Page 34: [457] Formatted | Smith, Molly B | 11/24/16 2:05:00 PM |
|---|---|---|

Font:12 pt, Not Italic

| Page 34: [457] Formatted | Smith, Molly B | 11/24/16 2:05:00 PM |
|---|---|---|

Font:12 pt, Not Italic

| Page 34: [457] Formatted | Smith, Molly B | 11/24/16 2:05:00 PM |
|---|---|---|

Font:12 pt, Not Italic

| Page 34: [457] Formatted | Smith, Molly B | 11/24/16 2:05:00 PM |
|---|---|---|

Font:12 pt, Not Italic

| Page 34: [457] Formatted | Smith, Molly B | 11/24/16 2:05:00 PM |
|---|---|---|

Font:12 pt, Not Italic

| Page 34: [457] Formatted | Smith, Molly B | 11/24/16 2:05:00 PM |
|---|---|---|

Font:12 pt, Not Italic

| Page 34: [457] Formatted | Smith, Molly B | 11/24/16 2:05:00 PM |
|---|---|---|

Font:12 pt, Not Italic

| Page 34: [457] Formatted | Smith, Molly B | 11/24/16 2:05:00 PM |
|---|---|---|

Font:12 pt, Not Italic

| Page 34: [457] Formatted | Smith, Molly B | 11/24/16 2:05:00 PM |
|---|---|---|

Font:12 pt, Not Italic

| Page 34: [457] Formatted | Smith, Molly B | 11/24/16 2:05:00 PM |
|---|---|---|

Font:12 pt, Not Italic

| Page 34: [457] Formatted | Smith, Molly B | 11/24/16 2:05:00 PM |
|---|---|---|

Font:12 pt, Not Italic

| Page 34: [457] Formatted | Smith, Molly B | 11/24/16 2:05:00 PM |
|---|---|---|

Font:12 pt, Not Italic

| Page 34: [457] Formatted | Smith, Molly B | 11/24/16 2:05:00 PM |
|---|---|---|

Font:12 pt, Not Italic

| Page 34: [457] Formatted | Smith, Molly B | 11/24/16 2:05:00 PM |
|---|---|---|

Font:12 pt, Not Italic

| Page 34: [457] Formatted | Smith, Molly B | 11/24/16 2:05:00 PM |
|---|---|---|

Font:12 pt, Not Italic

| Page 34: [457] Formatted | Smith, Molly B | 11/24/16 2:05:00 PM |
|---|---|---|

Font:12 pt, Not Italic

| Page 34: [457] Formatted | Smith, Molly B | 11/24/16 2:05:00 PM |
|---|---|---|

Font:12 pt, Not Italic

| Page 34: [457] Formatted | Smith, Molly B | 11/24/16 2:05:00 PM |
|---|---|---|

Font:12 pt, Not Italic

| Page 34: [457] Formatted | Smith, Molly B | 11/24/16 2:05:00 PM |
|---|---|---|

Font:12 pt, Not Italic

| Page 34: [457] Formatted | Smith, Molly B | 11/24/16 2:05:00 PM |
|---|---|---|

Font:12 pt, Not Italic

| Page 34: [457] Formatted | Smith, Molly B | 11/24/16 2:05:00 PM |
|---|---|---|

Font:12 pt, Not Italic

| Page 34: [457] Formatted | Smith, Molly B | 11/24/16 2:05:00 PM |
|---|---|---|

Font:12 pt, Not Italic

| Page 34: [457] Formatted | Smith, Molly B | 11/24/16 2:05:00 PM |
|---|---|---|

Font:12 pt, Not Italic

| Page 34: [457] Formatted | Smith, Molly B | 11/24/16 2:05:00 PM |
|---|---|---|

Font:12 pt, Not Italic

| Page 34: [457] Formatted | Smith, Molly B | 11/24/16 2:05:00 PM |
|---|---|---|

Font:12 pt, Not Italic

| Page 34: [457] Formatted | Smith, Molly B | 11/24/16 2:05:00 PM |
|---|---|---|

Font:12 pt, Not Italic

| Page 34: [457] Formatted | Smith, Molly B | 11/24/16 2:05:00 PM |
|---|---|---|

Font:12 pt, Not Italic

| Page 34: [457] Formatted | Smith, Molly B | 11/24/16 2:05:00 PM |
|---|---|---|

Font:12 pt, Not Italic

| Page 34: [457] Formatted | Smith, Molly B | 11/24/16 2:05:00 PM |
|---|---|---|

Font:12 pt, Not Italic

| Page 34: [457] Formatted | Smith, Molly B | 11/24/16 2:05:00 PM |
|---|---|---|

Font:12 pt, Not Italic

| Page 34: [457] Formatted | Smith, Molly B | 11/24/16 2:05:00 PM |
|---|---|---|

Font:12 pt, Not Italic

| Page 34: [457] Formatted | Smith, Molly B | 11/24/16 2:05:00 PM |
|---|---|---|

Font:12 pt, Not Italic

| Page 34: [457] Formatted | Smith, Molly B | 11/24/16 2:05:00 PM |
|---|---|---|

Font:12 pt, Not Italic

| Page 34: [457] Formatted | Smith, Molly B | 11/24/16 2:05:00 PM |
|---|---|---|

Font:12 pt, Not Italic

| Page 34: [457] Formatted | Smith, Molly B | 11/24/16 2:05:00 PM |
|---|---|---|

Font:12 pt, Not Italic

| Page 34: [457] Formatted | Smith, Molly B | 11/24/16 2:05:00 PM |
|---|---|---|

Font:12 pt, Not Italic

| Page 34: [457] Formatted | Smith, Molly B | 11/24/16 2:05:00 PM |
|---|---|---|

Font:12 pt, Not Italic

| Page 34: [457] Formatted | Smith, Molly B | 11/24/16 2:05:00 PM |
|---|---|---|

Font:12 pt, Not Italic

| Page 34: [457] Formatted | Smith, Molly B | 11/24/16 2:05:00 PM |
|---|---|---|

Font:12 pt, Not Italic

| Page 34: [457] Formatted | Smith, Molly B | 11/24/16 2:05:00 PM |
|---|---|---|

Font:12 pt, Not Italic

| Page 34: [458] Formatted | Smith, Molly B | 11/24/16 2:05:00 PM |
|---|---|---|

Font:12 pt

| Page 35: [459] Moved to page 36 (Move #3)Cornell University | 12/3/16 9:08:00 AM |
|---|---|

**Table 6: Surface concentration over ocean basins. For each ocean region, the averaged correlation across time between annual mean deposition fluxes for the CAM-RE cases is shown in the second column. The third column shows the annual mean correlation with NAO, while the third column shows the annual mean correlation with the El Nino/Southern Oscillation climate index. Regions are defined as the ocean gridboxes (not including sea ice or land boxes) in the following latitude and longitude areas as from (Gregg et al., 2003): North Atlantic (>30°N; 270 to 30°E); North Pacific (>30°N; 120 to 270°E); North Central Atlantic (10 to 30°N, 270 to 30°E); North Central Pacific (10 to 30N; 120 to 270°E); North Indian (10 to 30°N; 30 to 120°E); Equatorial Atlantic (-10 to 10°N; 300 to 30°E); Equatorial Pacific (-10 to 10°N; 120 to 285°E); Equatorial Indian (-10 to 10°N; 30-120°E); South Atlantic (-30 to -10°N; 30 to 300°E); South Pacific (-30 to -10°N; 120 to 295°E); South Indian (-30 to -30°N, 30 to 120°E); Antarctic (<-30°N).**

| | CAM4-RE across model Correlation | NAO correlation | El Nino Correlation |
|---|---|---|---|
| North Atlantic | 0.66 | 0.10 | 0.45 |
| North Pacific | 0.51 | 0.19 | 0.62 |

| | | | |
|---|---|---|---|
| North Central Atlantic | 0.75 | 0.04 | -0.10 |
| North Central Pacific | 0.46 | -0.19 | 0.01 |
| North Indian | 0.30 | 0.13 | 0.38 |
| Equatorial Atlantic | 0.59 | -0.02 | -0.31 |
| Equatorial Pacific | 0.19 | -0.12 | 0.42 |
| Equatorial Indian | 0.31 | -0.18 | -0.15 |
| South Atlantic | 0.11 | -0.22 | -0.42 |
| South Pacific | 0.65 | 0.03 | 0.03 |
| South Indian | 0.46 | 0.29 | 0.16 |
| Antarctic | 0.28 | -0.42 | -0.63 |
| Global | 0.42 | 0.01 | -0.03 |

| Page 35: [460] Formatted | Smith, Molly B | 11/24/16 2:05:00 PM |
|---|---|---|

Font:12 pt

| Page 35: [460] Formatted | Smith, Molly B | 11/24/16 2:05:00 PM |
|---|---|---|

Font:12 pt

| Page 35: [460] Formatted | Smith, Molly B | 11/24/16 2:05:00 PM |
|---|---|---|

Font:12 pt

| Page 35: [460] Formatted | Smith, Molly B | 11/24/16 2:05:00 PM |
|---|---|---|

Font:12 pt

| Page 35: [460] Formatted | Smith, Molly B | 11/24/16 2:05:00 PM |
|---|---|---|

Font:12 pt

| Page 35: [460] Formatted | Smith, Molly B | 11/24/16 2:05:00 PM |
|---|---|---|

Font:12 pt

| Page 35: [460] Formatted | Smith, Molly B | 11/24/16 2:05:00 PM |
|---|---|---|

Font:12 pt

| Page 35: [460] Formatted | Smith, Molly B | 11/24/16 2:05:00 PM |
|---|---|---|

Font:12 pt

| Page 35: [460] Formatted | Smith, Molly B | 11/24/16 2:05:00 PM |
|---|---|---|

Font:12 pt

| Page 35: [460] Formatted | Smith, Molly B | 11/24/16 2:05:00 PM |
|---|---|---|

Font:12 pt

| Page 35: [460] Formatted | Smith, Molly B | 11/24/16 2:05:00 PM |
|---|---|---|

Font:12 pt

| Page 35: [460] Formatted | Smith, Molly B | 11/24/16 2:05:00 PM |
|---|---|---|

Font:12 pt

| Page 35: [460] Formatted | Smith, Molly B | 11/24/16 2:05:00 PM |
|---|---|---|

Font:12 pt

| Page 35: [460] Formatted | Smith, Molly B | 11/24/16 2:05:00 PM |
|---|---|---|

Font:12 pt

| Page 35: [460] Formatted | Smith, Molly B | 11/24/16 2:05:00 PM |
|---|---|---|

Font:12 pt

| Page 35: [460] Formatted | Smith, Molly B | 11/24/16 2:05:00 PM |
|---|---|---|

Font:12 pt

| Page 35: [460] Formatted | Smith, Molly B | 11/24/16 2:05:00 PM |
|---|---|---|

Font:12 pt

| Page 35: [460] Formatted | Smith, Molly B | 11/24/16 2:05:00 PM |
|---|---|---|

Font:12 pt

| Page 35: [460] Formatted | Smith, Molly B | 11/24/16 2:05:00 PM |
|---|---|---|

Font:12 pt

| Page 35: [460] Formatted | Smith, Molly B | 11/24/16 2:05:00 PM |
|---|---|---|

Font:12 pt

| Page 35: [460] Formatted | Smith, Molly B | 11/24/16 2:05:00 PM |
|---|---|---|

Font:12 pt

| Page 35: [460] Formatted | Smith, Molly B | 11/24/16 2:05:00 PM |
|---|---|---|

Font:12 pt

| Page 35: [460] Formatted | Smith, Molly B | 11/24/16 2:05:00 PM |
|---|---|---|

Font:12 pt

| Page 35: [460] Formatted | Smith, Molly B | 11/24/16 2:05:00 PM |
|---|---|---|

Font:12 pt

| Page 35: [460] Formatted | Smith, Molly B | 11/24/16 2:05:00 PM |
|---|---|---|

Font:12 pt

| Page 35: [460] Formatted | Smith, Molly B | 11/24/16 2:05:00 PM |
|---|---|---|

Font:12 pt

| Page 35: [460] Formatted | Smith, Molly B | 11/24/16 2:05:00 PM |
|---|---|---|

Font:12 pt

| Page 35: [460] Formatted | Smith, Molly B | 11/24/16 2:05:00 PM |
|---|---|---|

Font:12 pt

| Page 35: [460] Formatted | Smith, Molly B | 11/24/16 2:05:00 PM |
|---|---|---|

Font:12 pt

| Page 35: [460] Formatted | Smith, Molly B | 11/24/16 2:05:00 PM |
|---|---|---|

Font:12 pt

| Page 35: [460] Formatted | Smith, Molly B | 11/24/16 2:05:00 PM |
|---|---|---|

Font:12 pt

| Page 35: [460] Formatted | Smith, Molly B | 11/24/16 2:05:00 PM |
|---|---|---|

Font:12 pt

| Page 35: [460] Formatted | Smith, Molly B | 11/24/16 2:05:00 PM |
|---|---|---|

Font:12 pt

| Page 35: [460] Formatted | Smith, Molly B | 11/24/16 2:05:00 PM |
|---|---|---|

Font:12 pt

| Page 35: [460] Formatted | Smith, Molly B | 11/24/16 2:05:00 PM |
|---|---|---|

Font:12 pt

| Page 35: [460] Formatted | Smith, Molly B | 11/24/16 2:05:00 PM |
|---|---|---|

Font:12 pt

| Page 35: [460] Formatted | Smith, Molly B | 11/24/16 2:05:00 PM |
|---|---|---|

Font:12 pt

| Page 35: [460] Formatted | Smith, Molly B | 11/24/16 2:05:00 PM |
|---|---|---|

Font:12 pt

| Page 35: [460] Formatted | Smith, Molly B | 11/24/16 2:05:00 PM |
|---|---|---|

Font:12 pt

| Page 35: [460] Formatted | Smith, Molly B | 11/24/16 2:05:00 PM |
|---|---|---|

Font:12 pt

| Page 35: [460] Formatted | Smith, Molly B | 11/24/16 2:05:00 PM |
|---|---|---|

Font:12 pt

| Page 35: [461] Deleted | Cornell University | 12/3/16 12:16:00 PM |
|---|---|---|

| | CAM4-RE across model Correlation | NAO correlation | El Nino Correlation |
|---|---|---|---|
| North Atlantic | 0.66 | 0.10 | 0.45 |
| North Pacific | 0.51 | 0.19 | 0.62 |
| North Central Atlantic | 0.75 | 0.04 | -0.10 |
| North Central Pacific | 0.46 | -0.19 | 0.01 |
| North Indian | 0.30 | 0.13 | 0.38 |
| Equatorial Atlantic | 0.59 | -0.02 | -0.31 |
| Equatorial Pacific | 0.19 | -0.12 | 0.42 |
| Equatorial Indian | 0.31 | -0.18 | -0.15 |
| South Atlantic | 0.11 | -0.22 | -0.42 |
| South Pacific | 0.65 | 0.03 | 0.03 |
| South Indian | 0.46 | 0.29 | 0.16 |
| Antarctic | 0.28 | -0.42 | -0.63 |
| Global | 0.42 | 0.01 | -0.03 |

| Page 35: [462] Formatted | Smith, Molly B | 11/24/16 2:05:00 PM |
|---|---|---|

Font:12 pt, Not Italic

| Page 35: [462] Formatted | Smith, Molly B | 11/24/16 2:05:00 PM |
|---|---|---|

Font:12 pt, Not Italic

| Page 35: [462] Formatted | Smith, Molly B | 11/24/16 2:05:00 PM |
|---|---|---|

Font:12 pt, Not Italic

| Page 35: [462] Formatted | Smith, Molly B | 11/24/16 2:05:00 PM |
|---|---|---|

Font:12 pt, Not Italic

| Page 35: [462] Formatted | Smith, Molly B | 11/24/16 2:05:00 PM |

Font:12 pt, Not Italic

| Page 35: [462] Formatted | Smith, Molly B | 11/24/16 2:05:00 PM |

Font:12 pt, Not Italic

| Page 35: [462] Formatted | Smith, Molly B | 11/24/16 2:05:00 PM |

Font:12 pt, Not Italic

| Page 35: [462] Formatted | Smith, Molly B | 11/24/16 2:05:00 PM |

Font:12 pt, Not Italic

| Page 35: [462] Formatted | Smith, Molly B | 11/24/16 2:05:00 PM |

Font:12 pt, Not Italic

| Page 35: [462] Formatted | Smith, Molly B | 11/24/16 2:05:00 PM |

Font:12 pt, Not Italic

| Page 35: [462] Formatted | Smith, Molly B | 11/24/16 2:05:00 PM |

Font:12 pt, Not Italic

| Page 35: [462] Formatted | Smith, Molly B | 11/24/16 2:05:00 PM |

Font:12 pt, Not Italic

| Page 35: [462] Formatted | Smith, Molly B | 11/24/16 2:05:00 PM |

Font:12 pt, Not Italic

| Page 35: [462] Formatted | Smith, Molly B | 11/24/16 2:05:00 PM |

Font:12 pt, Not Italic

| Page 35: [462] Formatted | Smith, Molly B | 11/24/16 2:05:00 PM |

Font:12 pt, Not Italic

| Page 35: [462] Formatted | Smith, Molly B | 11/24/16 2:05:00 PM |

Font:12 pt, Not Italic

| Page 35: [462] Formatted | Smith, Molly B | 11/24/16 2:05:00 PM |

Font:12 pt, Not Italic

| Page 35: [462] Formatted | Smith, Molly B | 11/24/16 2:05:00 PM |

Font:12 pt, Not Italic

| Page 35: [462] Formatted | Smith, Molly B | 11/24/16 2:05:00 PM |

Font:12 pt, Not Italic

| Page 35: [462] Formatted | Smith, Molly B | 11/24/16 2:05:00 PM |

Font:12 pt, Not Italic

| Page 35: [462] Formatted | Smith, Molly B | 11/24/16 2:05:00 PM |

Font:12 pt, Not Italic

| Page 35: [462] Formatted | Smith, Molly B | 11/24/16 2:05:00 PM |
|---|---|---|

Font:12 pt, Not Italic

| Page 35: [462] Formatted | Smith, Molly B | 11/24/16 2:05:00 PM |
|---|---|---|

Font:12 pt, Not Italic

| Page 35: [462] Formatted | Smith, Molly B | 11/24/16 2:05:00 PM |
|---|---|---|

Font:12 pt, Not Italic

| Page 35: [462] Formatted | Smith, Molly B | 11/24/16 2:05:00 PM |
|---|---|---|

Font:12 pt, Not Italic

| Page 35: [462] Formatted | Smith, Molly B | 11/24/16 2:05:00 PM |
|---|---|---|

Font:12 pt, Not Italic

| Page 35: [462] Formatted | Smith, Molly B | 11/24/16 2:05:00 PM |
|---|---|---|

Font:12 pt, Not Italic

| Page 35: [462] Formatted | Smith, Molly B | 11/24/16 2:05:00 PM |
|---|---|---|

Font:12 pt, Not Italic

| Page 35: [462] Formatted | Smith, Molly B | 11/24/16 2:05:00 PM |
|---|---|---|

Font:12 pt, Not Italic

| Page 35: [462] Formatted | Smith, Molly B | 11/24/16 2:05:00 PM |
|---|---|---|

Font:12 pt, Not Italic

| Page 35: [462] Formatted | Smith, Molly B | 11/24/16 2:05:00 PM |
|---|---|---|

Font:12 pt, Not Italic

| Page 35: [462] Formatted | Smith, Molly B | 11/24/16 2:05:00 PM |
|---|---|---|

Font:12 pt, Not Italic

| Page 35: [462] Formatted | Smith, Molly B | 11/24/16 2:05:00 PM |
|---|---|---|

Font:12 pt, Not Italic

| Page 35: [462] Formatted | Smith, Molly B | 11/24/16 2:05:00 PM |
|---|---|---|

Font:12 pt, Not Italic

| Page 35: [462] Formatted | Smith, Molly B | 11/24/16 2:05:00 PM |
|---|---|---|

Font:12 pt, Not Italic

| Page 35: [462] Formatted | Smith, Molly B | 11/24/16 2:05:00 PM |
|---|---|---|

Font:12 pt, Not Italic

| Page 35: [462] Formatted | Smith, Molly B | 11/24/16 2:05:00 PM |
|---|---|---|

Font:12 pt, Not Italic

| Page 35: [462] Formatted | Smith, Molly B | 11/24/16 2:05:00 PM |
|---|---|---|

Font:12 pt, Not Italic

| Page 35: [462] Formatted | Smith, Molly B | 11/24/16 2:05:00 PM |
|---|---|---|

Font:12 pt, Not Italic

| Page 35: [462] Formatted | Smith, Molly B | 11/24/16 2:05:00 PM |
|---|---|---|

Font:12 pt, Not Italic

| Page 35: [462] Formatted | Smith, Molly B | 11/24/16 2:05:00 PM |
|---|---|---|

Font:12 pt, Not Italic

| Page 35: [462] Formatted | Smith, Molly B | 11/24/16 2:05:00 PM |
|---|---|---|

Font:12 pt, Not Italic

| Page 35: [462] Formatted | Smith, Molly B | 11/24/16 2:05:00 PM |
|---|---|---|

Font:12 pt, Not Italic

| Page 35: [462] Formatted | Smith, Molly B | 11/24/16 2:05:00 PM |
|---|---|---|

Font:12 pt, Not Italic

| Page 35: [462] Formatted | Smith, Molly B | 11/24/16 2:05:00 PM |
|---|---|---|

Font:12 pt, Not Italic

| Page 35: [462] Formatted | Smith, Molly B | 11/24/16 2:05:00 PM |
|---|---|---|

Font:12 pt, Not Italic

| Page 35: [462] Formatted | Smith, Molly B | 11/24/16 2:05:00 PM |
|---|---|---|

Font:12 pt, Not Italic

| Page 35: [462] Formatted | Smith, Molly B | 11/24/16 2:05:00 PM |
|---|---|---|

Font:12 pt, Not Italic

| Page 35: [462] Formatted | Smith, Molly B | 11/24/16 2:05:00 PM |
|---|---|---|

Font:12 pt, Not Italic

| Page 35: [462] Formatted | Smith, Molly B | 11/24/16 2:05:00 PM |
|---|---|---|

Font:12 pt, Not Italic

| Page 35: [462] Formatted | Smith, Molly B | 11/24/16 2:05:00 PM |
|---|---|---|

Font:12 pt, Not Italic

| Page 35: [462] Formatted | Smith, Molly B | 11/24/16 2:05:00 PM |
|---|---|---|

Font:12 pt, Not Italic

| Page 35: [462] Formatted | Smith, Molly B | 11/24/16 2:05:00 PM |
|---|---|---|

Font:12 pt, Not Italic

| Page 35: [462] Formatted | Smith, Molly B | 11/24/16 2:05:00 PM |
|---|---|---|

Font:12 pt, Not Italic

| Page 35: [462] Formatted | Smith, Molly B | 11/24/16 2:05:00 PM |
|---|---|---|

Font:12 pt, Not Italic

| Page 35: [462] Formatted | Smith, Molly B | 11/24/16 2:05:00 PM |
|---|---|---|

Font:12 pt, Not Italic

| Page 35: [462] Formatted | Smith, Molly B | 11/24/16 2:05:00 PM |
|---|---|---|

Font:12 pt, Not Italic

| Page 35: [462] Formatted | Smith, Molly B | 11/24/16 2:05:00 PM |
|---|---|---|

Font:12 pt, Not Italic

| Page 35: [462] Formatted | Smith, Molly B | 11/24/16 2:05:00 PM |
|---|---|---|

Font:12 pt, Not Italic

| Page 35: [462] Formatted | Smith, Molly B | 11/24/16 2:05:00 PM |
|---|---|---|

Font:12 pt, Not Italic

| Page 35: [462] Formatted | Smith, Molly B | 11/24/16 2:05:00 PM |
|---|---|---|

Font:12 pt, Not Italic

| Page 35: [462] Formatted | Smith, Molly B | 11/24/16 2:05:00 PM |
|---|---|---|

Font:12 pt, Not Italic

| Page 35: [462] Formatted | Smith, Molly B | 11/24/16 2:05:00 PM |
|---|---|---|

Font:12 pt, Not Italic

| Page 35: [462] Formatted | Smith, Molly B | 11/24/16 2:05:00 PM |
|---|---|---|

Font:12 pt, Not Italic

| Page 35: [462] Formatted | Smith, Molly B | 11/24/16 2:05:00 PM |
|---|---|---|

Font:12 pt, Not Italic

| Page 35: [462] Formatted | Smith, Molly B | 11/24/16 2:05:00 PM |
|---|---|---|

Font:12 pt, Not Italic

| Page 35: [462] Formatted | Smith, Molly B | 11/24/16 2:05:00 PM |
|---|---|---|

Font:12 pt, Not Italic

| Page 35: [462] Formatted | Smith, Molly B | 11/24/16 2:05:00 PM |
|---|---|---|

Font:12 pt, Not Italic

| Page 35: [462] Formatted | Smith, Molly B | 11/24/16 2:05:00 PM |
|---|---|---|

Font:12 pt, Not Italic

| Page 35: [462] Formatted | Smith, Molly B | 11/24/16 2:05:00 PM |
|---|---|---|

Font:12 pt, Not Italic

| Page 35: [462] Formatted | Smith, Molly B | 11/24/16 2:05:00 PM |
|---|---|---|

Font:12 pt, Not Italic

| Page 35: [462] Formatted | Smith, Molly B | 11/24/16 2:05:00 PM |
|---|---|---|

Font:12 pt, Not Italic

| Page 35: [462] Formatted | Smith, Molly B | 11/24/16 2:05:00 PM |
|---|---|---|

Font:12 pt, Not Italic

| Page 35: [462] Formatted | Smith, Molly B | 11/24/16 2:05:00 PM |
|---|---|---|

Font:12 pt, Not Italic

| Page 35: [462] Formatted | Smith, Molly B | 11/24/16 2:05:00 PM |
|---|---|---|

Font:12 pt, Not Italic

| Page 35: [462] Formatted | Smith, Molly B | 11/24/16 2:05:00 PM |
|---|---|---|

Font:12 pt, Not Italic

| Page 35: [462] Formatted | Smith, Molly B | 11/24/16 2:05:00 PM |
|---|---|---|

Font:12 pt, Not Italic

| Page 35: [462] Formatted | Smith, Molly B | 11/24/16 2:05:00 PM |
|---|---|---|

Font:12 pt, Not Italic

| Page 35: [462] Formatted | Smith, Molly B | 11/24/16 2:05:00 PM |
|---|---|---|

Font:12 pt, Not Italic

| Page 35: [462] Formatted | Smith, Molly B | 11/24/16 2:05:00 PM |
|---|---|---|

Font:12 pt, Not Italic

| Page 35: [462] Formatted | Smith, Molly B | 11/24/16 2:05:00 PM |
|---|---|---|

Font:12 pt, Not Italic

| Page 35: [462] Formatted | Smith, Molly B | 11/24/16 2:05:00 PM |
|---|---|---|

Font:12 pt, Not Italic

| Page 35: [462] Formatted | Smith, Molly B | 11/24/16 2:05:00 PM |
|---|---|---|

Font:12 pt, Not Italic

| Page 35: [462] Formatted | Smith, Molly B | 11/24/16 2:05:00 PM |
|---|---|---|

Font:12 pt, Not Italic

| Page 35: [462] Formatted | Smith, Molly B | 11/24/16 2:05:00 PM |
|---|---|---|

Font:12 pt, Not Italic

| Page 35: [462] Formatted | Smith, Molly B | 11/24/16 2:05:00 PM |
|---|---|---|

Font:12 pt, Not Italic

| Page 35: [462] Formatted | Smith, Molly B | 11/24/16 2:05:00 PM |
|---|---|---|

Font:12 pt, Not Italic

| Page 35: [462] Formatted | Smith, Molly B | 11/24/16 2:05:00 PM |
|---|---|---|

Font:12 pt, Not Italic

| Page 35: [462] Formatted | Smith, Molly B | 11/24/16 2:05:00 PM |
|---|---|---|

Font:12 pt, Not Italic

| Page 35: [462] Formatted | Smith, Molly B | 11/24/16 2:05:00 PM |
|---|---|---|

Font:12 pt, Not Italic

| Page 35: [462] Formatted | Smith, Molly B | 11/24/16 2:05:00 PM |
|---|---|---|

Font:12 pt, Not Italic

| Page 35: [462] Formatted | Smith, Molly B | 11/24/16 2:05:00 PM |
|---|---|---|

Font:12 pt, Not Italic

| Page 35: [462] Formatted | Smith, Molly B | 11/24/16 2:05:00 PM |
|---|---|---|

Font:12 pt, Not Italic

| Page 35: [462] Formatted | Smith, Molly B | 11/24/16 2:05:00 PM |
|---|---|---|

Font:12 pt, Not Italic

| Page 35: [462] Formatted | Smith, Molly B | 11/24/16 2:05:00 PM |
|---|---|---|

Font:12 pt, Not Italic

| Page 35: [462] Formatted | Smith, Molly B | 11/24/16 2:05:00 PM |
|---|---|---|

Font:12 pt, Not Italic

| Page 35: [462] Formatted | Smith, Molly B | 11/24/16 2:05:00 PM |
|---|---|---|

Font:12 pt, Not Italic

| Page 35: [462] Formatted | Smith, Molly B | 11/24/16 2:05:00 PM |
|---|---|---|

Font:12 pt, Not Italic

| Page 35: [462] Formatted | Smith, Molly B | 11/24/16 2:05:00 PM |
|---|---|---|

Font:12 pt, Not Italic

| Page 35: [463] Formatted | Smith, Molly B | 11/24/16 2:05:00 PM |
|---|---|---|

Font:12 pt

| Page 35: [464] Formatted | Smith, Molly B | 11/24/16 2:05:00 PM |
|---|---|---|

Line spacing:  single

| Page 35: [465] Formatted | Smith, Molly B | 11/24/16 2:05:00 PM |
|---|---|---|

Font:12 pt

| Page 35: [465] Formatted | Smith, Molly B | 11/24/16 2:05:00 PM |
|---|---|---|

Font:12 pt

| Page 35: [465] Formatted | Smith, Molly B | 11/24/16 2:05:00 PM |
|---|---|---|

Font:12 pt

| Page 35: [465] Formatted | Smith, Molly B | 11/24/16 2:05:00 PM |
|---|---|---|

Font:12 pt

| Page 35: [465] Formatted | Smith, Molly B | 11/24/16 2:05:00 PM |
|---|---|---|

Font:12 pt

| Page 35: [465] Formatted | Smith, Molly B | 11/24/16 2:05:00 PM |
|---|---|---|

Font:12 pt

| Page 35: [465] Formatted | Smith, Molly B | 11/24/16 2:05:00 PM |
|---|---|---|

Font:12 pt

| Page 35: [465] Formatted | Smith, Molly B | 11/24/16 2:05:00 PM |
|---|---|---|

Font:12 pt

| Page 35: [465] Formatted | Smith, Molly B | 11/24/16 2:05:00 PM |
|---|---|---|

Font:12 pt

| Page 35: [465] Formatted | Smith, Molly B | 11/24/16 2:05:00 PM |
|---|---|---|

Font:12 pt

| Page 35: [465] Formatted | Smith, Molly B | 11/24/16 2:05:00 PM |
|---|---|---|

Font:12 pt

| Page 35: [465] Formatted | Smith, Molly B | 11/24/16 2:05:00 PM |
|---|---|---|

Font:12 pt

| Page 35: [465] Formatted | Smith, Molly B | 11/24/16 2:05:00 PM |
|---|---|---|

Font:12 pt

| Page 35: [465] Formatted | Smith, Molly B | 11/24/16 2:05:00 PM |
|---|---|---|

Font:12 pt

| Page 35: [465] Formatted | Smith, Molly B | 11/24/16 2:05:00 PM |
|---|---|---|

Font:12 pt

| Page 35: [465] Formatted | Smith, Molly B | 11/24/16 2:05:00 PM |
|---|---|---|

Font:12 pt

| Page 35: [465] Formatted | Smith, Molly B | 11/24/16 2:05:00 PM |
|---|---|---|

Font:12 pt

| Page 35: [465] Formatted | Smith, Molly B | 11/24/16 2:05:00 PM |
|---|---|---|

Font:12 pt

| Page 35: [465] Formatted | Smith, Molly B | 11/24/16 2:05:00 PM |
|---|---|---|

Font:12 pt

| Page 35: [465] Formatted | Smith, Molly B | 11/24/16 2:05:00 PM |
|---|---|---|

Font:12 pt

| Page 35: [465] Formatted | Smith, Molly B | 11/24/16 2:05:00 PM |
|---|---|---|

Font:12 pt

| Page 35: [465] Formatted | Smith, Molly B | 11/24/16 2:05:00 PM |
|---|---|---|

Font:12 pt

| Page 35: [465] Formatted | Smith, Molly B | 11/24/16 2:05:00 PM |
|---|---|---|

Font:12 pt

| Page 35: [465] Formatted | Smith, Molly B | 11/24/16 2:05:00 PM |
|---|---|---|

Font:12 pt

| Page 35: [465] Formatted | Smith, Molly B | 11/24/16 2:05:00 PM |
|---|---|---|

Font:12 pt

| Page 35: [465] Formatted | Smith, Molly B | 11/24/16 2:05:00 PM |
|---|---|---|

Font:12 pt

| Page 35: [465] Formatted | Smith, Molly B | 11/24/16 2:05:00 PM |
|---|---|---|

Font:12 pt

| Page 35: [465] Formatted | Smith, Molly B | 11/24/16 2:05:00 PM |
|---|---|---|

Font:12 pt

| Page 35: [465] Formatted | Smith, Molly B | 11/24/16 2:05:00 PM |
|---|---|---|

Font:12 pt

| Page 35: [465] Formatted | Smith, Molly B | 11/24/16 2:05:00 PM |
|---|---|---|

Font:12 pt

| Page 35: [465] Formatted | Smith, Molly B | 11/24/16 2:05:00 PM |
|---|---|---|

Font:12 pt

| Page 35: [465] Formatted | Smith, Molly B | 11/24/16 2:05:00 PM |
|---|---|---|

Font:12 pt

| Page 35: [466] Formatted | Smith, Molly B | 11/24/16 2:05:00 PM |
|---|---|---|

Line spacing:  single

| Page 35: [467] Formatted | Smith, Molly B | 11/24/16 2:05:00 PM |
|---|---|---|

None, Space Before:  0 pt, Line spacing:  single, Don't keep with next, Don't keep lines together

| Page 35: [468] Formatted | Smith, Molly B | 11/24/16 2:05:00 PM |
|---|---|---|

Font:12 pt, Not Bold, Not Italic, Font color: Auto

| Page 35: [469] Formatted | Smith, Molly B | 11/24/16 2:05:00 PM |
|---|---|---|

Font:12 pt

| Page 35: [469] Formatted | Smith, Molly B | 11/24/16 2:05:00 PM |
|---|---|---|

Font:12 pt

| Page 35: [470] Formatted | Smith, Molly B | 11/24/16 2:05:00 PM |
|---|---|---|

Font:12 pt

| Page 35: [470] Formatted | Smith, Molly B | 11/24/16 2:05:00 PM |
|---|---|---|

Font:12 pt

| Page 35: [471] Formatted | Smith, Molly B | 11/24/16 2:05:00 PM |
|---|---|---|

Font:12 pt

| Page 35: [472] Formatted | Smith, Molly B | 11/24/16 2:05:00 PM |
|---|---|---|

None, Space Before:  0 pt, Line spacing:  single, Don't keep with next, Don't keep lines together

| Page 35: [473] Formatted | Smith, Molly B | 11/24/16 2:05:00 PM |
|---|---|---|

Font:12 pt, Not Bold, Not Italic, Font color: Auto

| Page 35: [474] Formatted | Smith, Molly B | 11/24/16 2:05:00 PM |
|---|---|---|

Font:12 pt

| Page 35: [474] Formatted | Smith, Molly B | 11/24/16 2:05:00 PM |
|---|---|---|

Font:12 pt

| Page 35: [475] Formatted | Smith, Molly B | 11/24/16 2:05:00 PM |
|---|---|---|

Font:12 pt

| Page 35: [475] Formatted | Smith, Molly B | 11/24/16 2:05:00 PM |
|---|---|---|

Font:12 pt

| Page 35: [476] Formatted | Smith, Molly B | 11/24/16 2:05:00 PM |
|---|---|---|

Font:12 pt

| Page 35: [476] Formatted | Smith, Molly B | 11/24/16 2:05:00 PM |
|---|---|---|

Font:12 pt

| Page 35: [477] Formatted | Smith, Molly B | 11/24/16 2:05:00 PM |
|---|---|---|

Font:12 pt

| Page 35: [478] Formatted | Smith, Molly B | 11/24/16 2:05:00 PM |
|---|---|---|

None, Space Before:  0 pt, Line spacing:  single, Don't keep with next, Don't keep lines together

| Page 35: [479] Formatted | Smith, Molly B | 11/24/16 2:05:00 PM |
|---|---|---|

Font:12 pt, Not Bold, Not Italic, Font color: Auto

| Page 35: [480] Formatted | Smith, Molly B | 11/24/16 2:05:00 PM |
|---|---|---|

Font:12 pt

| Page 35: [480] Formatted | Smith, Molly B | 11/24/16 2:05:00 PM |
|---|---|---|

Font:12 pt

| Page 35: [481] Formatted | Smith, Molly B | 11/24/16 2:05:00 PM |
|---|---|---|

Font:12 pt

| Page 35: [481] Formatted | Smith, Molly B | 11/24/16 2:05:00 PM |
|---|---|---|

Font:12 pt

| Page 35: [482] Formatted | Smith, Molly B | 11/24/16 2:05:00 PM |
|---|---|---|

Font:12 pt

| Page 35: [482] Formatted | Smith, Molly B | 11/24/16 2:05:00 PM |
|---|---|---|

Font:12 pt

| Page 35: [483] Formatted | Smith, Molly B | 11/24/16 2:05:00 PM |
|---|---|---|

Font:12 pt

| Page 35: [484] Formatted | Smith, Molly B | 11/24/16 2:05:00 PM |
|---|---|---|

None, Space Before:  0 pt, Line spacing:  single, Don't keep with next, Don't keep lines together

| Page 35: [485] Formatted | Smith, Molly B | 11/24/16 2:05:00 PM |
|---|---|---|

Font:12 pt, Not Bold, Not Italic, Font color: Auto

| Page 35: [486] Formatted | Smith, Molly B | 11/24/16 2:05:00 PM |
|---|---|---|

Font:12 pt

| Page 35: [486] Formatted | Smith, Molly B | 11/24/16 2:05:00 PM |
|---|---|---|

Font:12 pt

| Page 35: [487] Formatted | Smith, Molly B | 11/24/16 2:05:00 PM |
|---|---|---|

Font:12 pt

| Page 35: [487] Formatted | Smith, Molly B | 11/24/16 2:05:00 PM |
|---|---|---|

Font:12 pt

| Page 35: [488] Formatted | Smith, Molly B | 11/24/16 2:05:00 PM |
|---|---|---|

Font:12 pt

| Page 35: [488] Formatted | Smith, Molly B | 11/24/16 2:05:00 PM |
|---|---|---|

Font:12 pt

| Page 35: [489] Formatted | Smith, Molly B | 11/24/16 2:05:00 PM |
|---|---|---|

Font:12 pt

| Page 35: [490] Formatted | Smith, Molly B | 11/24/16 2:05:00 PM |
|---|---|---|

None, Space Before:  0 pt, Line spacing:  single, Don't keep with next, Don't keep lines together

| Page 35: [491] Formatted | Smith, Molly B | 11/24/16 2:05:00 PM |
|---|---|---|

Font:12 pt, Not Bold, Not Italic, Font color: Auto

| Page 35: [492] Formatted | Smith, Molly B | 11/24/16 2:05:00 PM |
|---|---|---|

Font:12 pt

| Page 35: [492] Formatted | Smith, Molly B | 11/24/16 2:05:00 PM |
|---|---|---|

Font:12 pt

| Page 35: [493] Formatted | Smith, Molly B | 11/24/16 2:05:00 PM |
|---|---|---|

Font:12 pt

| Page 35: [493] Formatted | Smith, Molly B | 11/24/16 2:05:00 PM |
|---|---|---|

Font:12 pt

| Page 35: [494] Formatted | Smith, Molly B | 11/24/16 2:05:00 PM |
|---|---|---|

Font:12 pt

| Page 35: [494] Formatted | Smith, Molly B | 11/24/16 2:05:00 PM |
|---|---|---|

Font:12 pt

| Page 35: [495] Formatted | Smith, Molly B | 11/24/16 2:05:00 PM |
|---|---|---|

Font:12 pt

| Page 35: [496] Formatted | Smith, Molly B | 11/24/16 2:05:00 PM |
|---|---|---|

None, Space Before:  0 pt, Line spacing:  single, Don't keep with next, Don't keep lines together

| Page 35: [497] Formatted | Smith, Molly B | 11/24/16 2:05:00 PM |
|---|---|---|

Font:12 pt, Not Bold, Not Italic, Font color: Auto

| Page 35: [498] Formatted | Smith, Molly B | 11/24/16 2:05:00 PM |
|---|---|---|

Font:12 pt

| Page 35: [498] Formatted | Smith, Molly B | 11/24/16 2:05:00 PM |

Font:12 pt

| Page 35: [499] Formatted | Smith, Molly B | 11/24/16 2:05:00 PM |

Font:12 pt

| Page 35: [499] Formatted | Smith, Molly B | 11/24/16 2:05:00 PM |

Font:12 pt

| Page 35: [500] Formatted | Smith, Molly B | 11/24/16 2:05:00 PM |

Font:12 pt

| Page 35: [500] Formatted | Smith, Molly B | 11/24/16 2:05:00 PM |

Font:12 pt

| Page 35: [501] Formatted | Smith, Molly B | 11/24/16 2:05:00 PM |

Font:12 pt

| Page 35: [502] Formatted | Smith, Molly B | 11/24/16 2:05:00 PM |

None, Space Before:  0 pt, Line spacing:  single, Don't keep with next, Don't keep lines together

| Page 35: [503] Formatted | Smith, Molly B | 11/24/16 2:05:00 PM |

Font:12 pt, Not Bold, Not Italic, Font color: Auto

| Page 35: [504] Formatted | Smith, Molly B | 11/24/16 2:05:00 PM |

Font:12 pt

| Page 35: [504] Formatted | Smith, Molly B | 11/24/16 2:05:00 PM |

Font:12 pt

| Page 35: [505] Formatted | Smith, Molly B | 11/24/16 2:05:00 PM |

Font:12 pt

| Page 35: [505] Formatted | Smith, Molly B | 11/24/16 2:05:00 PM |

Font:12 pt

| Page 35: [506] Formatted | Smith, Molly B | 11/24/16 2:05:00 PM |

Font:12 pt

| Page 35: [506] Formatted | Smith, Molly B | 11/24/16 2:05:00 PM |

Font:12 pt

| Page 35: [507] Formatted | Smith, Molly B | 11/24/16 2:05:00 PM |

Font:12 pt

| Page 35: [508] Formatted | Smith, Molly B | 11/24/16 2:05:00 PM |

None, Space Before:  0 pt, Line spacing:  single, Don't keep with next, Don't keep lines together

| Page 35: [509] Formatted | Smith, Molly B | 11/24/16 2:05:00 PM |

Font:12 pt, Not Bold, Not Italic, Font color: Auto

| Page 35: [510] Formatted | Smith, Molly B | 11/24/16 2:05:00 PM |

Font:12 pt

| Page 35: [510] Formatted | Smith, Molly B | 11/24/16 2:05:00 PM |

Font:12 pt

| Page 35: [511] Formatted | Smith, Molly B | 11/24/16 2:05:00 PM |
|---|---|---|

Font:12 pt

| Page 35: [511] Formatted | Smith, Molly B | 11/24/16 2:05:00 PM |
|---|---|---|

Font:12 pt

| Page 35: [512] Formatted | Smith, Molly B | 11/24/16 2:05:00 PM |
|---|---|---|

Font:12 pt

| Page 35: [512] Formatted | Smith, Molly B | 11/24/16 2:05:00 PM |
|---|---|---|

Font:12 pt

| Page 35: [513] Formatted | Smith, Molly B | 11/24/16 2:05:00 PM |
|---|---|---|

Font:12 pt

| Page 35: [514] Formatted | Smith, Molly B | 11/24/16 2:05:00 PM |
|---|---|---|

None, Space Before:  0 pt, Line spacing:  single, Don't keep with next, Don't keep lines together

| Page 35: [515] Formatted | Smith, Molly B | 11/24/16 2:05:00 PM |
|---|---|---|

Font:12 pt, Not Bold, Not Italic, Font color: Auto

| Page 35: [516] Formatted | Smith, Molly B | 11/24/16 2:05:00 PM |
|---|---|---|

Font:12 pt

| Page 35: [516] Formatted | Smith, Molly B | 11/24/16 2:05:00 PM |
|---|---|---|

Font:12 pt

| Page 35: [517] Formatted | Smith, Molly B | 11/24/16 2:05:00 PM |
|---|---|---|

Font:12 pt

| Page 35: [517] Formatted | Smith, Molly B | 11/24/16 2:05:00 PM |
|---|---|---|

Font:12 pt

| Page 35: [518] Formatted | Smith, Molly B | 11/24/16 2:05:00 PM |
|---|---|---|

Font:12 pt

| Page 35: [518] Formatted | Smith, Molly B | 11/24/16 2:05:00 PM |
|---|---|---|

Font:12 pt

| Page 35: [519] Formatted | Smith, Molly B | 11/24/16 2:05:00 PM |
|---|---|---|

Font:12 pt

| Page 35: [520] Formatted | Smith, Molly B | 11/24/16 2:05:00 PM |
|---|---|---|

None, Space Before:  0 pt, Line spacing:  single, Don't keep with next, Don't keep lines together

| Page 35: [521] Formatted | Smith, Molly B | 11/24/16 2:05:00 PM |
|---|---|---|

Font:12 pt, Not Bold, Not Italic, Font color: Auto

| Page 35: [522] Formatted | Smith, Molly B | 11/24/16 2:05:00 PM |
|---|---|---|

Font:12 pt

| Page 35: [522] Formatted | Smith, Molly B | 11/24/16 2:05:00 PM |
|---|---|---|

Font:12 pt

| Page 35: [523] Formatted | Smith, Molly B | 11/24/16 2:05:00 PM |
| --- | --- | --- |

Font:12 pt

| Page 35: [523] Formatted | Smith, Molly B | 11/24/16 2:05:00 PM |
| --- | --- | --- |

Font:12 pt

| Page 35: [524] Formatted | Smith, Molly B | 11/24/16 2:05:00 PM |
| --- | --- | --- |

Font:12 pt

| Page 35: [524] Formatted | Smith, Molly B | 11/24/16 2:05:00 PM |
| --- | --- | --- |

Font:12 pt

| Page 35: [525] Formatted | Smith, Molly B | 11/24/16 2:05:00 PM |
| --- | --- | --- |

Font:12 pt

| Page 35: [526] Formatted | Smith, Molly B | 11/24/16 2:05:00 PM |
| --- | --- | --- |

Line spacing:  single

| Page 36: [527] Formatted | Smith, Molly B | 11/24/16 2:05:00 PM |
| --- | --- | --- |

Font:12 pt

| Page 36: [528] Formatted | Smith, Molly B | 11/24/16 2:16:00 PM |
| --- | --- | --- |

Line spacing:  single

| Page 36: [529] Formatted Table | Smith, Molly B | 11/24/16 2:15:00 PM |
| --- | --- | --- |

Formatted Table

| Page 36: [530] Formatted | Smith, Molly B | 11/24/16 2:16:00 PM |
| --- | --- | --- |

Space Before:  0 pt, Line spacing:  single

| Page 36: [531] Formatted | Smith, Molly B | 11/24/16 2:05:00 PM |
| --- | --- | --- |

Font:12 pt, Not Bold, Not Italic, Font color: Auto

| Page 36: [532] Formatted | Smith, Molly B | 11/24/16 2:05:00 PM |
| --- | --- | --- |

Font:12 pt

| Page 36: [533] Formatted | Smith, Molly B | 11/24/16 2:16:00 PM |
| --- | --- | --- |

None, Space Before:  0 pt, Line spacing:  single, Don't keep with next, Don't keep lines together

| Page 36: [534] Formatted | Smith, Molly B | 11/24/16 2:05:00 PM |
| --- | --- | --- |

Font:12 pt, Not Bold, Not Italic, Font color: Auto

| Page 36: [535] Formatted | Smith, Molly B | 11/24/16 2:05:00 PM |
| --- | --- | --- |

Font:12 pt

| Page 36: [536] Formatted | Smith, Molly B | 11/24/16 2:16:00 PM |
| --- | --- | --- |

Space Before:  0 pt, Line spacing:  single

| Page 36: [537] Formatted | Smith, Molly B | 11/24/16 2:05:00 PM |
| --- | --- | --- |

Font:12 pt, Not Bold, Not Italic, Font color: Auto

| Page 36: [538] Formatted | Smith, Molly B | 11/24/16 2:05:00 PM |
| --- | --- | --- |

Font:12 pt

| Page 36: [538] Formatted | Smith, Molly B | 11/24/16 2:05:00 PM |
| --- | --- | --- |

Font:12 pt

| Page 36: [539] Formatted | Smith, Molly B | 11/24/16 2:05:00 PM |
|---|---|---|

Font:12 pt

| Page 36: [540] Formatted | Smith, Molly B | 11/24/16 2:16:00 PM |
|---|---|---|

None, Space Before:  0 pt, Line spacing:  single, Don't keep with next, Don't keep lines together

| Page 36: [541] Formatted | Smith, Molly B | 11/24/16 2:05:00 PM |
|---|---|---|

Font:12 pt, Not Bold, Not Italic, Font color: Auto

| Page 36: [542] Formatted | Smith, Molly B | 11/24/16 2:05:00 PM |
|---|---|---|

Font:12 pt

| Page 36: [542] Formatted | Smith, Molly B | 11/24/16 2:05:00 PM |
|---|---|---|

Font:12 pt

| Page 36: [543] Formatted | Smith, Molly B | 11/24/16 2:05:00 PM |
|---|---|---|

Font:12 pt

| Page 36: [543] Formatted | Smith, Molly B | 11/24/16 2:05:00 PM |
|---|---|---|

Font:12 pt

| Page 36: [544] Formatted | Smith, Molly B | 11/24/16 2:05:00 PM |
|---|---|---|

Font:12 pt

| Page 36: [544] Formatted | Smith, Molly B | 11/24/16 2:05:00 PM |
|---|---|---|

Font:12 pt

| Page 36: [545] Formatted | Smith, Molly B | 11/24/16 2:05:00 PM |
|---|---|---|

Font:12 pt

| Page 36: [545] Formatted | Smith, Molly B | 11/24/16 2:05:00 PM |
|---|---|---|

Font:12 pt

| Page 36: [546] Formatted | Smith, Molly B | 11/24/16 2:05:00 PM |
|---|---|---|

Font:12 pt

| Page 36: [547] Formatted | Smith, Molly B | 11/24/16 2:16:00 PM |
|---|---|---|

None, Space Before:  0 pt, Line spacing:  single, Don't keep with next, Don't keep lines together

| Page 36: [548] Formatted | Smith, Molly B | 11/24/16 2:05:00 PM |
|---|---|---|

Font:12 pt, Not Bold, Not Italic, Font color: Auto

| Page 36: [549] Formatted | Smith, Molly B | 11/24/16 2:05:00 PM |
|---|---|---|

Font:12 pt

| Page 36: [549] Formatted | Smith, Molly B | 11/24/16 2:05:00 PM |
|---|---|---|

Font:12 pt

| Page 36: [550] Formatted | Smith, Molly B | 11/24/16 2:05:00 PM |
|---|---|---|

Font:12 pt

| Page 36: [550] Formatted | Smith, Molly B | 11/24/16 2:05:00 PM |
|---|---|---|

Font:12 pt

| Page 36: [551] Formatted | Smith, Molly B | 11/24/16 2:05:00 PM |
|---|---|---|

Font:12 pt

| Page 36: [551] Formatted | Smith, Molly B | 11/24/16 2:05:00 PM |
|---|---|---|

Font:12 pt

| Page 36: [552] Formatted | Smith, Molly B | 11/24/16 2:05:00 PM |
|---|---|---|

Font:12 pt

| Page 36: [552] Formatted | Smith, Molly B | 11/24/16 2:05:00 PM |
|---|---|---|

Font:12 pt

| Page 36: [553] Formatted | Smith, Molly B | 11/24/16 2:05:00 PM |
|---|---|---|

Font:12 pt

| Page 36: [554] Formatted | Smith, Molly B | 11/24/16 2:16:00 PM |
|---|---|---|

None, Space Before:  0 pt, Line spacing:  single, Don't keep with next, Don't keep lines together

| Page 36: [555] Formatted | Smith, Molly B | 11/24/16 2:05:00 PM |
|---|---|---|

Font:12 pt, Not Bold, Not Italic, Font color: Auto

| Page 36: [556] Formatted | Smith, Molly B | 11/24/16 2:05:00 PM |
|---|---|---|

Font:12 pt

| Page 36: [556] Formatted | Smith, Molly B | 11/24/16 2:05:00 PM |
|---|---|---|

Font:12 pt

| Page 36: [557] Formatted | Smith, Molly B | 11/24/16 2:05:00 PM |
|---|---|---|

Font:12 pt

| Page 36: [557] Formatted | Smith, Molly B | 11/24/16 2:05:00 PM |
|---|---|---|

Font:12 pt

| Page 36: [558] Formatted | Smith, Molly B | 11/24/16 2:05:00 PM |
|---|---|---|

Font:12 pt

| Page 36: [558] Formatted | Smith, Molly B | 11/24/16 2:05:00 PM |
|---|---|---|

Font:12 pt

| Page 36: [559] Formatted | Smith, Molly B | 11/24/16 2:05:00 PM |
|---|---|---|

Font:12 pt

| Page 36: [559] Formatted | Smith, Molly B | 11/24/16 2:05:00 PM |
|---|---|---|

Font:12 pt

| Page 36: [560] Formatted | Smith, Molly B | 11/24/16 2:05:00 PM |
|---|---|---|

Font:12 pt

| Page 36: [561] Formatted | Smith, Molly B | 11/24/16 2:16:00 PM |
|---|---|---|

None, Space Before:  0 pt, Line spacing:  single, Don't keep with next, Don't keep lines together

| Page 36: [562] Formatted | Smith, Molly B | 11/24/16 2:05:00 PM |
|---|---|---|

Font:12 pt, Not Bold, Not Italic, Font color: Auto

| Page 36: [563] Formatted | Smith, Molly B | 11/24/16 2:05:00 PM |
|---|---|---|

Font:12 pt

| Page 36: [563] Formatted | Smith, Molly B | 11/24/16 2:05:00 PM |
|---|---|---|

Font:12 pt

| Page 36: [564] Formatted | Smith, Molly B | 11/24/16 2:05:00 PM |
|---|---|---|

Font:12 pt

| Page 36: [564] Formatted | Smith, Molly B | 11/24/16 2:05:00 PM |
|---|---|---|

Font:12 pt

| Page 36: [565] Formatted | Smith, Molly B | 11/24/16 2:05:00 PM |
|---|---|---|

Font:12 pt

| Page 36: [565] Formatted | Smith, Molly B | 11/24/16 2:05:00 PM |
|---|---|---|

Font:12 pt

| Page 36: [566] Formatted | Smith, Molly B | 11/24/16 2:05:00 PM |
|---|---|---|

Font:12 pt

| Page 36: [566] Formatted | Smith, Molly B | 11/24/16 2:05:00 PM |
|---|---|---|

Font:12 pt

| Page 36: [567] Formatted | Smith, Molly B | 11/24/16 2:05:00 PM |
|---|---|---|

Font:12 pt

| Page 36: [568] Formatted | Smith, Molly B | 11/24/16 2:16:00 PM |
|---|---|---|

None, Space Before:  0 pt, Line spacing:  single, Don't keep with next, Don't keep lines together

| Page 36: [569] Formatted | Smith, Molly B | 11/24/16 2:05:00 PM |
|---|---|---|

Font:12 pt, Not Bold, Not Italic, Font color: Auto

| Page 36: [570] Formatted | Smith, Molly B | 11/24/16 2:05:00 PM |
|---|---|---|

Font:12 pt

| Page 36: [570] Formatted | Smith, Molly B | 11/24/16 2:05:00 PM |
|---|---|---|

Font:12 pt

| Page 36: [571] Formatted | Smith, Molly B | 11/24/16 2:05:00 PM |
|---|---|---|

Font:12 pt

| Page 36: [571] Formatted | Smith, Molly B | 11/24/16 2:05:00 PM |
|---|---|---|

Font:12 pt

| Page 36: [572] Formatted | Smith, Molly B | 11/24/16 2:05:00 PM |
|---|---|---|

Font:12 pt

| Page 36: [572] Formatted | Smith, Molly B | 11/24/16 2:05:00 PM |
|---|---|---|

Font:12 pt

| Page 36: [573] Formatted | Smith, Molly B | 11/24/16 2:05:00 PM |
|---|---|---|

Font:12 pt

| Page 36: [573] Formatted | Smith, Molly B | 11/24/16 2:05:00 PM |
|---|---|---|

Font:12 pt

| Page 36: [574] Formatted | Smith, Molly B | 11/24/16 2:05:00 PM |
|---|---|---|

Font:12 pt

| Page 36: [575] Formatted | Smith, Molly B | 11/24/16 2:16:00 PM |
|---|---|---|

None, Space Before:  0 pt, Line spacing:  single, Don't keep with next, Don't keep lines together

| Page 36: [576] Formatted | Smith, Molly B | 11/24/16 2:05:00 PM |
|---|---|---|

Font:12 pt, Not Bold, Not Italic, Font color: Auto

| Page 36: [577] Formatted | Smith, Molly B | 11/24/16 2:05:00 PM |
|---|---|---|

Font:12 pt

| Page 36: [577] Formatted | Smith, Molly B | 11/24/16 2:05:00 PM |
|---|---|---|

Font:12 pt

| Page 36: [578] Formatted | Smith, Molly B | 11/24/16 2:05:00 PM |
|---|---|---|

Font:12 pt

| Page 36: [578] Formatted | Smith, Molly B | 11/24/16 2:05:00 PM |
|---|---|---|

Font:12 pt

| Page 36: [579] Formatted | Smith, Molly B | 11/24/16 2:05:00 PM |
|---|---|---|

Font:12 pt

| Page 36: [579] Formatted | Smith, Molly B | 11/24/16 2:05:00 PM |
|---|---|---|

Font:12 pt

| Page 36: [580] Formatted | Smith, Molly B | 11/24/16 2:05:00 PM |
|---|---|---|

Font:12 pt

| Page 36: [580] Formatted | Smith, Molly B | 11/24/16 2:05:00 PM |
|---|---|---|

Font:12 pt

| Page 36: [581] Formatted | Smith, Molly B | 11/24/16 2:05:00 PM |
|---|---|---|

Font:12 pt

| Page 36: [582] Formatted | Smith, Molly B | 11/24/16 2:16:00 PM |
|---|---|---|

None, Space Before:  0 pt, Line spacing:  single, Don't keep with next, Don't keep lines together

| Page 36: [583] Formatted | Smith, Molly B | 11/24/16 2:05:00 PM |
|---|---|---|

Font:12 pt, Not Bold, Not Italic, Font color: Auto

| Page 36: [584] Formatted | Smith, Molly B | 11/24/16 2:05:00 PM |
|---|---|---|

Font:12 pt

| Page 36: [584] Formatted | Smith, Molly B | 11/24/16 2:05:00 PM |
|---|---|---|

Font:12 pt

| Page 36: [585] Formatted | Smith, Molly B | 11/24/16 2:05:00 PM |
|---|---|---|

Font:12 pt

| Page 36: [585] Formatted | Smith, Molly B | 11/24/16 2:05:00 PM |
|---|---|---|

Font:12 pt

| Page 36: [586] Formatted | Smith, Molly B | 11/24/16 2:05:00 PM |
|---|---|---|

Font:12 pt

| Page 36: [586] Formatted | Smith, Molly B | 11/24/16 2:05:00 PM |

Font:12 pt

| Page 36: [587] Formatted | Smith, Molly B | 11/24/16 2:05:00 PM |

Font:12 pt

| Page 36: [587] Formatted | Smith, Molly B | 11/24/16 2:05:00 PM |

Font:12 pt

| Page 36: [588] Formatted | Smith, Molly B | 11/24/16 2:05:00 PM |

Font:12 pt

| Page 36: [589] Formatted | Smith, Molly B | 11/24/16 2:16:00 PM |

None, Space Before:  0 pt, Line spacing:  single, Don't keep with next, Don't keep lines together

| Page 36: [590] Formatted | Smith, Molly B | 11/24/16 2:05:00 PM |

Font:12 pt, Not Bold, Not Italic

| Page 36: [591] Formatted | Smith, Molly B | 11/24/16 2:05:00 PM |

Font:12 pt

| Page 36: [592] Formatted | Smith, Molly B | 11/24/16 2:16:00 PM |

Space Before:  0 pt, Line spacing:  single

| Page 36: [593] Formatted | Smith, Molly B | 11/24/16 2:05:00 PM |

Font:12 pt, Not Bold, Not Italic

| Page 36: [594] Formatted | Smith, Molly B | 11/24/16 2:05:00 PM |

Font:12 pt

| Page 36: [594] Formatted | Smith, Molly B | 11/24/16 2:05:00 PM |

Font:12 pt

| Page 36: [595] Formatted | Smith, Molly B | 11/24/16 2:05:00 PM |

Font:12 pt

| Page 36: [595] Formatted | Smith, Molly B | 11/24/16 2:05:00 PM |

Font:12 pt

| Page 36: [596] Formatted | Smith, Molly B | 11/24/16 2:05:00 PM |

Font:12 pt

| Page 36: [596] Formatted | Smith, Molly B | 11/24/16 2:05:00 PM |

Font:12 pt

| Page 36: [597] Moved from page 35 (Move #3)Cornell University | 12/3/16 9:08:00 AM |

**Table 86: Surface concentration over ocean basins.  For each ocean region, the averaged correlation across time between annual mean deposition fluxes for the CAM-RE cases is shown in the second column. The third column shows the annual mean correlation with NAO, while the third column shows the annual mean correlation with the El Nino/Southern Oscillation climate index.  Regions are defined as the ocean gridboxes (not including sea ice or land boxes) in the following latitude and longitude areas as from**

(Gregg et al., 2003): North Atlantic (>30°N; 270 to 30°E); North Pacific (>30°N; 120 to 270°E); North Central Atlantic (10 to 30°N, 270 to 30°E); North Central Pacific (10 to 30N; 120 to 270°E); North Indian (10 to 30°N; 30 to 120°E); Equatorial Atlantic (-10 to 10°N; 300 to 30°E); Equatorial Pacific (-10 to 10°N; 120 to 285°E); Equatorial Indian (-10 to 10°N; 30-120°E); South Atlantic (-30 to -10°N; 30 to 300°E); South Pacific (-30 to -10°N; 120 to 295°E); South Indian (-30 to -30°N, 30 to 120°E); Antarctic (<-30°N).

| | CAM4-RE across model Correlation | NAO correlation | El Nino Correlation |
|---|---|---|---|
| North Atlantic | 0.66 | 0.10 | 0.45 |
| North Pacific | 0.51 | 0.19 | 0.62 |
| North Central Atlantic | 0.75 | 0.04 | -0.10 |
| North Central Pacific | 0.46 | -0.19 | 0.01 |
| North Indian | 0.30 | 0.13 | 0.38 |
| Equatorial Atlantic | 0.59 | -0.02 | -0.31 |
| Equatorial Pacific | 0.19 | -0.12 | 0.42 |
| Equatorial Indian | 0.31 | -0.18 | -0.15 |
| South Atlantic | 0.11 | -0.22 | -0.42 |
| South Pacific | 0.65 | 0.03 | 0.03 |
| South Indian | 0.46 | 0.29 | 0.16 |
| Antarctic | 0.28 | -0.42 | -0.63 |
| Global | 0.42 | 0.01 | -0.03 |

**Table 8: Deposition into ocean basins. For each ocean region, the averaged correlation across time between annual mean deposition fluxes for the CAM-RE cases is shown in the second column. The following columns show the climatological mean deposition flux (Tg/year) for each model simulation. Regions are defined as the ocean gridboxes (not including sea ice or land boxes) in the following latitude and longitude areas as from (Gregg et al., 2003): North Atlantic (>30°N; 270 to 30°E); North Pacific (>30°N; 120 to 270°E); North Central Atlantic (10 to 30°N, 270 to 30°E); North Central Pacific (10 to 30N; 120 to 270°E); North Indian (10 to 30°N; 30 to 120°E); Equatorial Atlantic (-10 to 10°N; 300 to 30°E); Equatorial Pacific (-10 to 10°N; 120 to 285°E); Equatorial Indian (-10 to 10°N; 30-120°E); South Atlantic (-30 to -10°N; 30 to 300°E); South Pacific (-30 to -10°N; 120 to 295°E); South Indian (-30 to -30°N, 30 to 120°E); Antarctic (<-30°N).**

| | Correlation | Mean CAM4-Re | Mean all |
|---|---|---|---|
| North Atlantic | 0.68 | 54 +/-7% | 61 +/-17% |
| North Pacific | 0.50 | 30 +/- 30% | 30 +/- 60% |
| North Central Atlantic | 0.66 | 91 +/- 8% | 120 +/- 30% |
| North Central Pacific | 0.33 | 20 +/- 90% | 20 +/- 90% |
| North Indian | 0.10 | 100 +/- 30% | 110 +/- 40% |
| Equatorial Atlantic | 0.39 | 77 +/- 20% | 80 +/- 35% |
| Equatorial Pacific | 0.33 | 4 +/- 100% | 6 +/- 130% |
| Equatorial Indian | 0.31 | 23 +/- 25% | 21 +/-23% |
| South Atlantic | 0.16 | 3+/-60% | 3 +/- 90% |
| South Pacific | 0.62 | 1 +/- 40% | 3 +/- 60% |
| South Indian | 0.38 | 2 +/- 40% | 4 +/- 75% |
| Antarctic | 0.35 | 15 +/- 100% | 30 +/- 100% |
| Global | 0.45 | 450 +/- 10% | 510 +/- 24% |

| Page 37: [600] Formatted | Smith, Molly B | 11/24/16 2:05:00 PM |
|---|---|---|

Font:12 pt

| Page 37: [600] Formatted | Smith, Molly B | 11/24/16 2:05:00 PM |
|---|---|---|

Font:12 pt

| Page 37: [600] Formatted | Smith, Molly B | 11/24/16 2:05:00 PM |
|---|---|---|

Font:12 pt

| Page 37: [600] Formatted | Smith, Molly B | 11/24/16 2:05:00 PM |
|---|---|---|

Font:12 pt

| Page 37: [600] Formatted | Smith, Molly B | 11/24/16 2:05:00 PM |
|---|---|---|

Font:12 pt

| Page 37: [600] Formatted | Smith, Molly B | 11/24/16 2:05:00 PM |
|---|---|---|

Font:12 pt

| Page 37: [600] Formatted | Smith, Molly B | 11/24/16 2:05:00 PM |
|---|---|---|

Font:12 pt

| Page 37: [600] Formatted | Smith, Molly B | 11/24/16 2:05:00 PM |
|---|---|---|

Font:12 pt

| Page 37: [600] Formatted | Smith, Molly B | 11/24/16 2:05:00 PM |
|---|---|---|

Font:12 pt

| Page 37: [600] Formatted | Smith, Molly B | 11/24/16 2:05:00 PM |
|---|---|---|

Font:12 pt

| Page 37: [600] Formatted | Smith, Molly B | 11/24/16 2:05:00 PM |
|---|---|---|

Font:12 pt

| Page 37: [600] Formatted | Smith, Molly B | 11/24/16 2:05:00 PM |
|---|---|---|

Font:12 pt

| Page 37: [600] Formatted | Smith, Molly B | 11/24/16 2:05:00 PM |
|---|---|---|

Font:12 pt

| Page 37: [600] Formatted | Smith, Molly B | 11/24/16 2:05:00 PM |
|---|---|---|

Font:12 pt

| Page 37: [600] Formatted | Smith, Molly B | 11/24/16 2:05:00 PM |
|---|---|---|

Font:12 pt

| Page 37: [600] Formatted | Smith, Molly B | 11/24/16 2:05:00 PM |
|---|---|---|

Font:12 pt

| Page 37: [600] Formatted | Smith, Molly B | 11/24/16 2:05:00 PM |
|---|---|---|

Font:12 pt

| Page 37: [600] Formatted | Smith, Molly B | 11/24/16 2:05:00 PM |
|---|---|---|

Font:12 pt

| Page 37: [600] Formatted | Smith, Molly B | 11/24/16 2:05:00 PM |
|---|---|---|

Font:12 pt

| Page 37: [600] Formatted | Smith, Molly B | 11/24/16 2:05:00 PM |
|---|---|---|

Font:12 pt

| Page 37: [600] Formatted | Smith, Molly B | 11/24/16 2:05:00 PM |
|---|---|---|

Font:12 pt

| Page 37: [600] Formatted | Smith, Molly B | 11/24/16 2:05:00 PM |
|---|---|---|

Font:12 pt

| Page 37: [600] Formatted | Smith, Molly B | 11/24/16 2:05:00 PM |
|---|---|---|

Font:12 pt

| Page 37: [600] Formatted | Smith, Molly B | 11/24/16 2:05:00 PM |
|---|---|---|

Font:12 pt

| Page 37: [600] Formatted | Smith, Molly B | 11/24/16 2:05:00 PM |
|---|---|---|

Font:12 pt

| Page 37: [600] Formatted | Smith, Molly B | 11/24/16 2:05:00 PM |
|---|---|---|

Font:12 pt

| Page 37: [600] Formatted | Smith, Molly B | 11/24/16 2:05:00 PM |
|---|---|---|

Font:12 pt

| Page 37: [600] Formatted | Smith, Molly B | 11/24/16 2:05:00 PM |
|---|---|---|

Font:12 pt

| Page 37: [600] Formatted | Smith, Molly B | 11/24/16 2:05:00 PM |
|---|---|---|

Font:12 pt

| Page 37: [600] Formatted | Smith, Molly B | 11/24/16 2:05:00 PM |
|---|---|---|

Font:12 pt

| Page 37: [600] Formatted | Smith, Molly B | 11/24/16 2:05:00 PM |
|---|---|---|

Font:12 pt

| Page 37: [600] Formatted | Smith, Molly B | 11/24/16 2:05:00 PM |
|---|---|---|

Font:12 pt

| Page 37: [600] Formatted | Smith, Molly B | 11/24/16 2:05:00 PM |
|---|---|---|

Font:12 pt

| Page 37: [600] Formatted | Smith, Molly B | 11/24/16 2:05:00 PM |
|---|---|---|

Font:12 pt

| Page 37: [600] Formatted | Smith, Molly B | 11/24/16 2:05:00 PM |
|---|---|---|

Font:12 pt

| Page 37: [600] Formatted | Smith, Molly B | 11/24/16 2:05:00 PM |
|---|---|---|

Font:12 pt

| Page 37: [600] Formatted | Smith, Molly B | 11/24/16 2:05:00 PM |
|---|---|---|

Font:12 pt

| Page 37: [600] Formatted | Smith, Molly B | 11/24/16 2:05:00 PM |
|---|---|---|

Font:12 pt

| Page 37: [600] Formatted | Smith, Molly B | 11/24/16 2:05:00 PM |
|---|---|---|

Font:12 pt

| Page 37: [600] Formatted | Smith, Molly B | 11/24/16 2:05:00 PM |
|---|---|---|

Font:12 pt

| Page 37: [600] Formatted | Smith, Molly B | 11/24/16 2:05:00 PM |
|---|---|---|

Font:12 pt

| Page 37: [601] Formatted | Smith, Molly B | 11/24/16 2:05:00 PM |
|---|---|---|

Font:12 pt, Not Bold, Not Italic, Font color: Auto

| Page 37: [601] Formatted | Smith, Molly B | 11/24/16 2:05:00 PM |
|---|---|---|

Font:12 pt, Not Bold, Not Italic, Font color: Auto

| Page 37: [601] Formatted | Smith, Molly B | 11/24/16 2:05:00 PM |
|---|---|---|

Font:12 pt, Not Bold, Not Italic, Font color: Auto

| Page 37: [601] Formatted | Smith, Molly B | 11/24/16 2:05:00 PM |
|---|---|---|

Font:12 pt, Not Bold, Not Italic, Font color: Auto

| Page 37: [601] Formatted | Smith, Molly B | 11/24/16 2:05:00 PM |
|---|---|---|

Font:12 pt, Not Bold, Not Italic, Font color: Auto

| Page 37: [601] Formatted | Smith, Molly B | 11/24/16 2:05:00 PM |
|---|---|---|

Font:12 pt, Not Bold, Not Italic, Font color: Auto

| Page 37: [601] Formatted | Smith, Molly B | 11/24/16 2:05:00 PM |
|---|---|---|

Font:12 pt, Not Bold, Not Italic, Font color: Auto

| Page 37: [601] Formatted | Smith, Molly B | 11/24/16 2:05:00 PM |
|---|---|---|

Font:12 pt, Not Bold, Not Italic, Font color: Auto

| Page 37: [601] Formatted | Smith, Molly B | 11/24/16 2:05:00 PM |
|---|---|---|

Font:12 pt, Not Bold, Not Italic, Font color: Auto

| Page 37: [601] Formatted | Smith, Molly B | 11/24/16 2:05:00 PM |
|---|---|---|

Font:12 pt, Not Bold, Not Italic, Font color: Auto

| Page 37: [601] Formatted | Smith, Molly B | 11/24/16 2:05:00 PM |
|---|---|---|

Font:12 pt, Not Bold, Not Italic, Font color: Auto

| Page 37: [601] Formatted | Smith, Molly B | 11/24/16 2:05:00 PM |
|---|---|---|

Font:12 pt, Not Bold, Not Italic, Font color: Auto

| Page 37: [601] Formatted | Smith, Molly B | 11/24/16 2:05:00 PM |
|---|---|---|

Font:12 pt, Not Bold, Not Italic, Font color: Auto

| Page 37: [601] Formatted | Smith, Molly B | 11/24/16 2:05:00 PM |
|---|---|---|

Font:12 pt, Not Bold, Not Italic, Font color: Auto

| Page 37: [601] Formatted | Smith, Molly B | 11/24/16 2:05:00 PM |
|---|---|---|

Font:12 pt, Not Bold, Not Italic, Font color: Auto

| Page 37: [601] Formatted | Smith, Molly B | 11/24/16 2:05:00 PM |
|---|---|---|

Font:12 pt, Not Bold, Not Italic, Font color: Auto

| Page 37: [601] Formatted | Smith, Molly B | 11/24/16 2:05:00 PM |
|---|---|---|

Font:12 pt, Not Bold, Not Italic, Font color: Auto

| Page 37: [601] Formatted | Smith, Molly B | 11/24/16 2:05:00 PM |
|---|---|---|

Font:12 pt, Not Bold, Not Italic, Font color: Auto

| Page 37: [601] Formatted | Smith, Molly B | 11/24/16 2:05:00 PM |
|---|---|---|

Font:12 pt, Not Bold, Not Italic, Font color: Auto

| Page 37: [601] Formatted | Smith, Molly B | 11/24/16 2:05:00 PM |
|---|---|---|

Font:12 pt, Not Bold, Not Italic, Font color: Auto

| Page 37: [601] Formatted | Smith, Molly B | 11/24/16 2:05:00 PM |
|---|---|---|

Font:12 pt, Not Bold, Not Italic, Font color: Auto

| Page 37: [601] Formatted | Smith, Molly B | 11/24/16 2:05:00 PM |
|---|---|---|

Font:12 pt, Not Bold, Not Italic, Font color: Auto

| Page 37: [601] Formatted | Smith, Molly B | 11/24/16 2:05:00 PM |
|---|---|---|

Font:12 pt, Not Bold, Not Italic, Font color: Auto

| Page 37: [601] Formatted | Smith, Molly B | 11/24/16 2:05:00 PM |
|---|---|---|

Font:12 pt, Not Bold, Not Italic, Font color: Auto

| Page 37: [601] Formatted | Smith, Molly B | 11/24/16 2:05:00 PM |
|---|---|---|

Font:12 pt, Not Bold, Not Italic, Font color: Auto

| Page 37: [601] Formatted | Smith, Molly B | 11/24/16 2:05:00 PM |
|---|---|---|

Font:12 pt, Not Bold, Not Italic, Font color: Auto

| Page 37: [601] Formatted | Smith, Molly B | 11/24/16 2:05:00 PM |
|---|---|---|

Font:12 pt, Not Bold, Not Italic, Font color: Auto

| Page 37: [601] Formatted | Smith, Molly B | 11/24/16 2:05:00 PM |
|---|---|---|

Font:12 pt, Not Bold, Not Italic, Font color: Auto

| Page 37: [601] Formatted | Smith, Molly B | 11/24/16 2:05:00 PM |
|---|---|---|

Font:12 pt, Not Bold, Not Italic, Font color: Auto

| Page 37: [601] Formatted | Smith, Molly B | 11/24/16 2:05:00 PM |
|---|---|---|

Font:12 pt, Not Bold, Not Italic, Font color: Auto

| Page 37: [601] Formatted | Smith, Molly B | 11/24/16 2:05:00 PM |
|---|---|---|

Font:12 pt, Not Bold, Not Italic, Font color: Auto

| Page 37: [601] Formatted | Smith, Molly B | 11/24/16 2:05:00 PM |
|---|---|---|

Font:12 pt, Not Bold, Not Italic, Font color: Auto

| Page 37: [601] Formatted | Smith, Molly B | 11/24/16 2:05:00 PM |
|---|---|---|

Font:12 pt, Not Bold, Not Italic, Font color: Auto

| Page 37: [601] Formatted | Smith, Molly B | 11/24/16 2:05:00 PM |
|---|---|---|

Font:12 pt, Not Bold, Not Italic, Font color: Auto

| Page 37: [601] Formatted | Smith, Molly B | 11/24/16 2:05:00 PM |
|---|---|---|

Font:12 pt, Not Bold, Not Italic, Font color: Auto

| Page 37: [601] Formatted | Smith, Molly B | 11/24/16 2:05:00 PM |
|---|---|---|

Font:12 pt, Not Bold, Not Italic, Font color: Auto

| Page 37: [601] Formatted | Smith, Molly B | 11/24/16 2:05:00 PM |
|---|---|---|

Font:12 pt, Not Bold, Not Italic, Font color: Auto

| Page 37: [601] Formatted | Smith, Molly B | 11/24/16 2:05:00 PM |
|---|---|---|

Font:12 pt, Not Bold, Not Italic, Font color: Auto

| Page 37: [601] Formatted | Smith, Molly B | 11/24/16 2:05:00 PM |
|---|---|---|

Font:12 pt, Not Bold, Not Italic, Font color: Auto

| Page 37: [601] Formatted | Smith, Molly B | 11/24/16 2:05:00 PM |
|---|---|---|

Font:12 pt, Not Bold, Not Italic, Font color: Auto

| Page 37: [601] Formatted | Smith, Molly B | 11/24/16 2:05:00 PM |
|---|---|---|

Font:12 pt, Not Bold, Not Italic, Font color: Auto

| Page 37: [601] Formatted | Smith, Molly B | 11/24/16 2:05:00 PM |
|---|---|---|

Font:12 pt, Not Bold, Not Italic, Font color: Auto

| Page 37: [601] Formatted | Smith, Molly B | 11/24/16 2:05:00 PM |
|---|---|---|

Font:12 pt, Not Bold, Not Italic, Font color: Auto

| Page 37: [601] Formatted | Smith, Molly B | 11/24/16 2:05:00 PM |
|---|---|---|

Font:12 pt, Not Bold, Not Italic, Font color: Auto

| | | |
|---|---|---|
| **Page 37: [601] Formatted** | **Smith, Molly B** | **11/24/16 2:05:00 PM** |

Font:12 pt, Not Bold, Not Italic, Font color: Auto

| | | |
|---|---|---|
| **Page 37: [601] Formatted** | **Smith, Molly B** | **11/24/16 2:05:00 PM** |

Font:12 pt, Not Bold, Not Italic, Font color: Auto

| | | |
|---|---|---|
| **Page 37: [601] Formatted** | **Smith, Molly B** | **11/24/16 2:05:00 PM** |

Font:12 pt, Not Bold, Not Italic, Font color: Auto

| | | |
|---|---|---|
| **Page 37: [601] Formatted** | **Smith, Molly B** | **11/24/16 2:05:00 PM** |

Font:12 pt, Not Bold, Not Italic, Font color: Auto

| | | |
|---|---|---|
| **Page 37: [601] Formatted** | **Smith, Molly B** | **11/24/16 2:05:00 PM** |

Font:12 pt, Not Bold, Not Italic, Font color: Auto

| | | |
|---|---|---|
| **Page 37: [601] Formatted** | **Smith, Molly B** | **11/24/16 2:05:00 PM** |

Font:12 pt, Not Bold, Not Italic, Font color: Auto

| | | |
|---|---|---|
| **Page 37: [601] Formatted** | **Smith, Molly B** | **11/24/16 2:05:00 PM** |

Font:12 pt, Not Bold, Not Italic, Font color: Auto

| | | |
|---|---|---|
| **Page 37: [601] Formatted** | **Smith, Molly B** | **11/24/16 2:05:00 PM** |

Font:12 pt, Not Bold, Not Italic, Font color: Auto

| | | |
|---|---|---|
| **Page 37: [601] Formatted** | **Smith, Molly B** | **11/24/16 2:05:00 PM** |

Font:12 pt, Not Bold, Not Italic, Font color: Auto

| | | |
|---|---|---|
| **Page 37: [601] Formatted** | **Smith, Molly B** | **11/24/16 2:05:00 PM** |

Font:12 pt, Not Bold, Not Italic, Font color: Auto

| | | |
|---|---|---|
| **Page 37: [601] Formatted** | **Smith, Molly B** | **11/24/16 2:05:00 PM** |

Font:12 pt, Not Bold, Not Italic, Font color: Auto

| | | |
|---|---|---|
| **Page 37: [601] Formatted** | **Smith, Molly B** | **11/24/16 2:05:00 PM** |

Font:12 pt, Not Bold, Not Italic, Font color: Auto

| | | |
|---|---|---|
| **Page 37: [601] Formatted** | **Smith, Molly B** | **11/24/16 2:05:00 PM** |

Font:12 pt, Not Bold, Not Italic, Font color: Auto

| | | |
|---|---|---|
| **Page 37: [601] Formatted** | **Smith, Molly B** | **11/24/16 2:05:00 PM** |

Font:12 pt, Not Bold, Not Italic, Font color: Auto

| | | |
|---|---|---|
| **Page 37: [601] Formatted** | **Smith, Molly B** | **11/24/16 2:05:00 PM** |

Font:12 pt, Not Bold, Not Italic, Font color: Auto

| | | |
|---|---|---|
| **Page 37: [601] Formatted** | **Smith, Molly B** | **11/24/16 2:05:00 PM** |

Font:12 pt, Not Bold, Not Italic, Font color: Auto

| Page 37: [601] Formatted | Smith, Molly B | 11/24/16 2:05:00 PM |

Font:12 pt, Not Bold, Not Italic, Font color: Auto

| Page 37: [601] Formatted | Smith, Molly B | 11/24/16 2:05:00 PM |

Font:12 pt, Not Bold, Not Italic, Font color: Auto

| Page 37: [601] Formatted | Smith, Molly B | 11/24/16 2:05:00 PM |

Font:12 pt, Not Bold, Not Italic, Font color: Auto

| Page 37: [601] Formatted | Smith, Molly B | 11/24/16 2:05:00 PM |

Font:12 pt, Not Bold, Not Italic, Font color: Auto

| Page 37: [601] Formatted | Smith, Molly B | 11/24/16 2:05:00 PM |

Font:12 pt, Not Bold, Not Italic, Font color: Auto

| Page 37: [601] Formatted | Smith, Molly B | 11/24/16 2:05:00 PM |

Font:12 pt, Not Bold, Not Italic, Font color: Auto

| Page 37: [601] Formatted | Smith, Molly B | 11/24/16 2:05:00 PM |

Font:12 pt, Not Bold, Not Italic, Font color: Auto

| Page 37: [601] Formatted | Smith, Molly B | 11/24/16 2:05:00 PM |

Font:12 pt, Not Bold, Not Italic, Font color: Auto

| Page 37: [601] Formatted | Smith, Molly B | 11/24/16 2:05:00 PM |

Font:12 pt, Not Bold, Not Italic, Font color: Auto

| Page 37: [601] Formatted | Smith, Molly B | 11/24/16 2:05:00 PM |

Font:12 pt, Not Bold, Not Italic, Font color: Auto

| Page 37: [601] Formatted | Smith, Molly B | 11/24/16 2:05:00 PM |

Font:12 pt, Not Bold, Not Italic, Font color: Auto

| Page 37: [601] Formatted | Smith, Molly B | 11/24/16 2:05:00 PM |

Font:12 pt, Not Bold, Not Italic, Font color: Auto

| Page 37: [601] Formatted | Smith, Molly B | 11/24/16 2:05:00 PM |

Font:12 pt, Not Bold, Not Italic, Font color: Auto

| Page 37: [601] Formatted | Smith, Molly B | 11/24/16 2:05:00 PM |

Font:12 pt, Not Bold, Not Italic, Font color: Auto

| Page 37: [601] Formatted | Smith, Molly B | 11/24/16 2:05:00 PM |

Font:12 pt, Not Bold, Not Italic, Font color: Auto

| Page 37: [601] Formatted | Smith, Molly B | 11/24/16 2:05:00 PM |

Font:12 pt, Not Bold, Not Italic, Font color: Auto

| Page 37: [601] Formatted | Smith, Molly B | 11/24/16 2:05:00 PM |

Font:12 pt, Not Bold, Not Italic, Font color: Auto

| Page 37: [601] Formatted | Smith, Molly B | 11/24/16 2:05:00 PM |
|---|---|---|

Font:12 pt, Not Bold, Not Italic, Font color: Auto

| Page 37: [601] Formatted | Smith, Molly B | 11/24/16 2:05:00 PM |
|---|---|---|

Font:12 pt, Not Bold, Not Italic, Font color: Auto

| Page 37: [601] Formatted | Smith, Molly B | 11/24/16 2:05:00 PM |
|---|---|---|

Font:12 pt, Not Bold, Not Italic, Font color: Auto

| Page 37: [601] Formatted | Smith, Molly B | 11/24/16 2:05:00 PM |
|---|---|---|

Font:12 pt, Not Bold, Not Italic, Font color: Auto

| Page 37: [601] Formatted | Smith, Molly B | 11/24/16 2:05:00 PM |
|---|---|---|

Font:12 pt, Not Bold, Not Italic, Font color: Auto

| Page 37: [601] Formatted | Smith, Molly B | 11/24/16 2:05:00 PM |
|---|---|---|

Font:12 pt, Not Bold, Not Italic, Font color: Auto

| Page 37: [601] Formatted | Smith, Molly B | 11/24/16 2:05:00 PM |
|---|---|---|

Font:12 pt, Not Bold, Not Italic, Font color: Auto

| Page 37: [601] Formatted | Smith, Molly B | 11/24/16 2:05:00 PM |
|---|---|---|

Font:12 pt, Not Bold, Not Italic, Font color: Auto

| Page 37: [601] Formatted | Smith, Molly B | 11/24/16 2:05:00 PM |
|---|---|---|

Font:12 pt, Not Bold, Not Italic, Font color: Auto

| Page 37: [601] Formatted | Smith, Molly B | 11/24/16 2:05:00 PM |
|---|---|---|

Font:12 pt, Not Bold, Not Italic, Font color: Auto

| Page 37: [601] Formatted | Smith, Molly B | 11/24/16 2:05:00 PM |
|---|---|---|

Font:12 pt, Not Bold, Not Italic, Font color: Auto

| Page 37: [601] Formatted | Smith, Molly B | 11/24/16 2:05:00 PM |
|---|---|---|

Font:12 pt, Not Bold, Not Italic, Font color: Auto

| Page 37: [601] Formatted | Smith, Molly B | 11/24/16 2:05:00 PM |
|---|---|---|

Font:12 pt, Not Bold, Not Italic, Font color: Auto

| Page 37: [601] Formatted | Smith, Molly B | 11/24/16 2:05:00 PM |
|---|---|---|

Font:12 pt, Not Bold, Not Italic, Font color: Auto

| Page 37: [601] Formatted | Smith, Molly B | 11/24/16 2:05:00 PM |
|---|---|---|

Font:12 pt, Not Bold, Not Italic, Font color: Auto

| Page 37: [601] Formatted | Smith, Molly B | 11/24/16 2:05:00 PM |
|---|---|---|

Font:12 pt, Not Bold, Not Italic, Font color: Auto

| Page 37: [601] Formatted | Smith, Molly B | 11/24/16 2:05:00 PM |
|---|---|---|

Font:12 pt, Not Bold, Not Italic, Font color: Auto

| Page 37: [601] Formatted | Smith, Molly B | 11/24/16 2:05:00 PM |
|---|---|---|

Font:12 pt, Not Bold, Not Italic, Font color: Auto

| Page 37: [601] Formatted | Smith, Molly B | 11/24/16 2:05:00 PM |
|---|---|---|

Font:12 pt, Not Bold, Not Italic, Font color: Auto

| Page 37: [601] Formatted | Smith, Molly B | 11/24/16 2:05:00 PM |
|---|---|---|

Font:12 pt, Not Bold, Not Italic, Font color: Auto

| Page 37: [601] Formatted | Smith, Molly B | 11/24/16 2:05:00 PM |
|---|---|---|

Font:12 pt, Not Bold, Not Italic, Font color: Auto

| Page 37: [601] Formatted | Smith, Molly B | 11/24/16 2:05:00 PM |
|---|---|---|

Font:12 pt, Not Bold, Not Italic, Font color: Auto

| Page 37: [602] Deleted | Cornell University | 11/15/16 2:49:00 PM |
|---|---|---|

**Figure 1: Climatological monthly average time series of dust concentration divided by the mean of each time series (unitless) at stations a) Banizoumbou, b) Barbados, c) Bermuda, d) Cinzana, e) Izana, f) Macehead, g) Mbour, h) Miami, i) Midway. Observations are in black and are from the locations and citations in Table 2. Different colors and line styles indicate the different model versions: CAM4 (MERRA) (blue solid), CAM4 (NCEP) (green solid), CAM4 (ERAI) (pink solid), CAM4 (AMIP) (orange solid), GCHEM (MERRA) (blue dashed), MATCH (NCEP) (green dashed), CAM5 (AMIP) (orange dashed). The monthly means are divided by the long term annual mean to allow comparison with interannual variability, since they are similarly normalized (Figure 3).**

[Figure]

**Figure 2: Climatological monthly average aerosol optical depth (AOD) for model simulations (based on dust only), compared with AERONET observations for: a) Bahrain b) Dahkla c) Dalanzadgad, d) Dhabi, e) Ilorin and f) Sede Boker for each of the different model versions (Colors and information are the same as in Figure 1, but for the AOD). Observational data from AERONET stations (citations and locations listed in Table 2). The monthly means are divided by the long term annual mean to allow comparison with interannual variability, since they are similarly normalized (Figure 4).**

| Page 38: [603] Deleted | Cornell University | 11/15/16 2:49:00 PM |
|---|---|---|

| Page 38: [604] Deleted | Smith, Molly B | 11/28/16 6:24:00 PM |
|---|---|---|

Dahkla c)

| Page 38: [605] Deleted | Smith, Molly B | 11/28/16 6:25:00 PM |
|---|---|---|

Dhabi, e)

| Page 38: [606] Deleted | Cornell University | 11/15/16 2:50:00 PM |
|---|---|---|

).

[Figure]

| Page 38: [607] Deleted | Cornell University | 11/15/16 2:51:00 PM |
|---|---|---|

monthly

| Page 38: [607] Deleted | Cornell University | 11/15/16 2:51:00 PM |
|---|---|---|

monthly

| Page 38: [607] Deleted | Cornell University | 11/15/16 2:51:00 PM |
|---|---|---|

monthly

| Page 38: [607] Deleted | Cornell University | 11/15/16 2:51:00 PM |
|---|---|---|

monthly

| Page 38: [608] Deleted | Smith, Molly B | 11/24/16 1:03:00 PM |
|---|---|---|

the

| Page 38: [608] Deleted | Smith, Molly B | 11/24/16 1:03:00 PM |
|---|---|---|

the

| Page 38: [609] Deleted | Cornell University | 11/15/16 2:51:00 PM |
|---|---|---|

monthly

| Page 38: [609] Deleted | Cornell University | 11/15/16 2:51:00 PM |
|---|---|---|

monthly

| Page 38: [610] Deleted | Smith, Molly B | 11/28/16 6:27:00 PM |
|---|---|---|

**Dah: Dahkla;**

| Page 38: [610] Deleted | Smith, Molly B | 11/28/16 6:27:00 PM |
|---|---|---|

**Dah: Dahkla;**

| Page 38: [611] Deleted | Cornell University | 11/15/16 2:52:00 PM |
|---|---|---|

[Figure]

[Figure]

| Page 38: [612] Deleted | Cornell University | 11/15/16 2:52:00 PM |
|---|---|---|

a. Surf. conc. variability

b. Fraction of surf. conc. variability from seasonal cycle

c. Deposition variability

d. Fraction of deposition variability from seasonal cycle

e. AOD variability

f. Fraction of AOD variability from seasonal cycle

0.3  0.6  0.9  1.2  1.5  1.8  2.1  2.4  2.7  3.0

0.1  0.2  0.3  0.4  0.5  0.6  0.7  0.8  0.9

a. Corr. Surf. Conc. CAM4 models

b. Corr. Surf. Conc. same met. models

c. Corr. Deposition CAM4 models

d. Corr. Deposition same met. models

e. Corr. AOD CAM4 models

f. Corr. AOD same met. models

0.1  0.2  0.3  0.4  0.5  0.6  0.7  0.8  0.9

| Page 39: [614] Deleted | Smith, Molly B | 11/24/16 2:18:00 PM |

| Page 40: [615] Deleted | Smith, Molly B | 12/1/16 2:27:00 PM |

**Figure 104: Time series of the annual mean deposition flux in different ocean regions as simulated in the different model versions (different colors, as in legend). All time series are normalized by the climatological mean in order to focus on interannual variability (shown**

in Table S9). The yellow highlighted area is the area encompassed by the 5 reanalysis-based simulations (CAM4 (MERRA), CAM4 (NCEP), CAM4 (ERAI), GCHEM (MERRA), MATCH (NCEP)). The ocean basin areas are defined as over ocean in the regions in Table S8 from (Gregg et al., 2003).

**Figure 115:** Time series of the annual mean Aerosol Optical Depth (AOD) in different ocean regions as simulated in the different model versions (different colors, as in legend). All time series are normalized by the climatological mean (Table S10) in order to focus on interannual variability. The yellow highlighted area is the area encompassed by the 5 reanalysis-based simulations (CAM4 (MERRA), CAM4 (NCEP), CAM4 (ERAI), GCHEM (MERRA), MATCH (NCEP)). The ocean basin areas are defined as over ocean in the regions in Table S8 from (Gregg et al., 2003Gregg et al., 2003).

| Page 40: [616] Deleted | Smith, Molly B | 11/24/16 1:06:00 PM |
|---|---|---|

Engelstaedter, S., Kohfeld, K. E., Tegen, I., and Harrison, S. P.: Controls of dust emissions by vegetation and topographic depressions: An evaluation using dust storm frequency data, Geophysical Research Letters, 30, 1294, doi:1210.1029/2002GL016471, 012003, 2003.
Huneeus, N., Schulz, M., Balkanski, Y., Griesfeller, J., Kinne, S., Prospero, J., Bauer, S., Boucher, O., Chin, M., Dentener, F., Diehl, T., Easter, R., Fillmore, D., Ghan, S., Ginoux, P., Grini, A., Horowitz, L., Koch, D., Krol, M. C., Landing, W., Liu, X., Mahowald, N., Miller, R., Morccrette, J.-J., Myhre, G., Penner, J., Perlwitz, J., Siter, P., Takemura, T., and Zender, C.: Global dust model intercomparison in AEROCOM, Atmospheric Chemistry and Physics, 11, 1-36, 2011.
Knippertz, P., and Todd, M.: Mineral dust aerosols over the Sahara: Meteorological controls on emission and transport and implications for modeling, Reviews of Geophysics, 50, doi:10.1029/2011RG000362, 2012.
Mahowald, N., Luo, C., Corral, J. d., and Zender, C.: Interannual variability in atmospheric mineral aerosols from a 22-year model simulation and observational data, Journal of Geophysical Research, 108, 4352, 4310.1029/2002JD002821, 2003.
Moulin, C., Lambert, C. E., Dulac, F., and Dayan, U.: Control of atmospheric export of dust from North Africa by the North Atlantic Oscillation, Nature, 387, 691-694, 1997.
Roe, G.: On the interpretation of Chinese loess as a paleoclimate indicator, Quaternary Research, 71, 150-161, 2008.

| Page 40: [617] Deleted | Cornell University | 12/3/16 12:14:00 PM |
|---|---|---|

Cowie, S., Knippertz, P., and Marsham, J.: Are the vegetation-related roughness changes the cause of the recent decrease in dust emission from the Sahel?, Geophysical Research Letters, 40, 1868-1872; doi:1810.1002/grl.50273, 2013.
Doney, S., Lima, I., Feeley, R., Glover, D., Lindsay, K., Mahowald, N., Moore, J. K., and Wanninkhof, R.: Mechanisms governing interannual variability in upper-ocean inorganic carbon system and air-sea CO2 fluxes: physical climate and atmospheric dust, Deep Sea Research Part II: Topical Studies in Oceanography, 56, 640-655, 2009.

Ginoux, P., Prospero, J., and Torres, O.: Long-term simulation of dust distribution with the GOCART model: Correlation with the North Atlantic Oscillation, Proceedings of ICAR5/GCTE-SEN Joint Conference, 241, 2002.

Mahowald, N., Luo, C., Corral, J. d., and Zender, C.: Interannual variability in atmospheric mineral aerosols from a 22-year model simulation and observational data, Journal of Geophysical Research, 108, 4352, 4310.1029/2002JD002821, 2003.

Marsham, J., Knippertz, P., Dixon, N., Parker, D., and Lister, G.: The importance of representation of deep convection for modeled dust-generated winds over West Africa during the summer, Geophysical Research Letters, 38, doi:10.1029/2001GL048368, 2011.

Moulin, C., Lambert, C. E., Dulac, F., and Dayan, U.: Control of atmospheric export of dust from North Africa by the North Atlantic Oscillation, Nature, 387, 691-694, 1997.

Renno, N., Nash, A. A., Lunine, J., and Murphy, J.: Martian and terrestrial dust devils: test of a scaling theory using Pathfinder data, Journal of Geophysical Research, 105, 1859-1865, 2000.

Streets, D., Yan, F., Chin, M., Diehl, T., Mahowald, N., Schultz, M., Wild, M., Wu, Y., and Yu, C.: Anthropogenic and natural contributions to regional trends in aerosol optical depth, 1980-2006, Journal of Geophysical Research, 114, doi:10.2029/2008JD011624, 2009.

---

## Author Comment (AC2) · 23 Dec 2016

We would like to thank Referee #1 for the suggestions for improvements, as we think these will improve the quality of the manuscript. We try to incorporate all the comments of the reviewer, as discussed below, and try to reconcile these comments with the sometimes contradictory comments by the other reviewer. Reviewer comments are in black, responses in blue.

Reviewer 1 feedback:

The authors compare several model frameworks, including different forcing data, different forcing methods and different models, to estimate the sensitivity in the seasonal and interannual variability of dust emission and transport. They find that the correlation in variability is roughly similar between simulations using different meteorological forcings and different models, while simulations forced by sea surface temperature only do not perform as well.

Considering the high uncertainty in dust simulations, investigating the sensitivity to different forcing data sets and models is an important contribution to the field of dust modelling. However, the manuscript presents a lot of information that is not necessarily relevant and that dilutes the actual results of the study. In addition, several parts of the methodology and results are unclear, which make the interpretation further confusing.

We have rewritten the methodology and results sections to be clearer and to focus primarily on interannual variability in surface concentration in order to streamline the paper. Our aim is to eliminate superfluous information that distracts from these core results, but we do use available deposition and AOD observations and mention the implications of these results for deposition and AOD, using the supplemental material.

Note that the recommendations of this reviewer contradict in some places, the recommendations of the other reviewer (who wants more details), so that we do add some details about NAO and ENSO to accommodate the constructive comments of the Reviewer #2.

The manuscript thus likely need to be rewritten before it can be considered for publication in Atmospheric Chemistry and Physics.

We have rewritten the manuscript for clarity, and hope that it is now acceptable for publication.

General comments

A. The model description is often confusing and strongly needs to be reorganized and clarified.

We have rewritten this section and added an introduction to improve clarity.

B. The discussion of the results jumps between a large number of figures and tables split between the manuscript and the supplement, which need to be reduced.

We have reduced the number of figures and tables in the manuscript, as part of our effort to focus on interannual variability in surface concentration and cut down on superfluous information. We had to add back in some figures in the supplement, due to questions from the other reviewers, but streamline whenever possible.

C. A significant part of the results further appears not to be significant, thus the discussion needs to be more focused to emphasize the actual contribution of the manuscript.

We have modified the discussion of our results to emphasize trends in dust interannual variability for particular regions, as well as the differences in model output produced by individual datasets.

Specific comments

1. Introduction

p.2 l.7-15: this part is disconnected from the previous one and needs to be separated into a new paragraph; the different space and time scales and their relevance are confusing and need some hierarchy; what are "4x fluctuations"?

The paragraph has been rewritten to make it clearer. We have rewritten out (4-fold fluctuations) to make the text clearer:

"with 4-fold fluctuations in surface concentration observed across a large region on decadal time scales (Prospero and Lamb, 2003)"

p.2 l.15: this is partially redundant with the following lines and should be reorganized.

This line has been moved to a more relevant position in the next paragraph.

p.2 l.21: this is obvious and could be more specific (interannual variability of dust emissions?).

We clarify that this is true across time scales:

"It is well established in the dust literature that meteorology and surface conditions play central roles in driving changes in dust emissions, primarily from changes in precipitation, winds, surface roughness or vegetation cover on daily to interannual to geological time scales (e.g. Westphal et al., 1987; Petit et al., 1999; Marticorena and Bergametti, 1996; Mahowald et al., 2002; Prospero and Lamb, 2004; Engelstadter and Washington, 2007; Engelstaedter et al., 2003; Roe, 2008; McGee et al., 2010; Knippertz and Todd, 2012; Cowie et al., 2013; Yu et al., 2015), and we do not seek to repeat or review that work here."

p.2 l.28-29: the purpose of the sentence is unclear.

The sentence has been clarified to show that it explains our choice to use the time range 1990-2005 for this study:

"The period between 1990-2005 was chosen for this study because it has more available observational and reanalysis data than other years, but it must be noted that this time range does not have as much variability as previous periods (e.g. dry 1980s versus wet 1960s in the Sahel region; Prospero and Lamb, 2003)."

p.2 l.31: a short paragraph to describe the contents of the paper would be helpful at the end of the

introduction.

A very good suggestion: this has been added:

"In order to simplify the paper, we will focus on IAV in surface concentrations, and provide information on how deposition and aerosol optical depth contrast with surface concentration variability. Section 2 describes the methods used in the study, including briefly describing the models, data and comparison metrics. Section 3 describes the results of the study, starting with comparison to observations, comparison between different model simulations, and the implication for observational needs."

2. Methodology

p.3 l.5-10: the meaning of "model" is confusing here between CESM, CAM, GCM... A description of the nature of these "models" would be helpful.

A very good point. Clarification was added to the beginning of the methods section, but the models are explained in detail farther on in the paper:

"Several models are used in this study, all of which include prognostic dust. The atmospheric component of an earth system model, the Community Atmospheric Model (CAM4) of the Community Earth System Model (CESM) (Neale et al., 2013; Hurrell et al., 2013) is capable being forced either by online-calculated dynamical winds or by reanalyses datasets and is the model used for the bulk of the analysis to test sensitivity to which reanalyses winds are used (Section 2.1.1 and 2.1.2). To contrast with this model, results from other models, including the CAM5 version of the same model, and two chemical transport models, driven by reanalyses are also used (Model of Atmospheric Transport and Chemistry or MATCH and GEOS-chem) (Section 2.1.3). More details are described below, along with a summary of the model simulations (Table 1)."

p.3 l.12: "when the winds are strong" needs to be clarified.

This sentence has been rewritten to clarify (note that we don't want to repeat the development of dust entrainment mechanisms which are well developed elsewhere in the literature, and we cite):

"Dust is entrained into the atmosphere when strong winds occur in dry, unvegetated regions with easily erodible soils (Marticorena and Bergametti, 1995)."

p.3 l.15: "as described in more detail elsewhere" needs to be rephrased.

"Elsewhere" has been clarified to mean "in previous studies".

p.3 l.25-27: the purpose of the sentence is unclear.

Good point: we add in reference to the results section, where we discuss this in more detail.

"For example, in most versions of the model the tuning reduced dust source strength over Central Asia and the Atacama Desert, and increased dust source strength over Argentina (Albani et al.,

2014), which becomes important when analyzing dust IAV, as discussed in the results sections (Section 3)."

p.3 l.31- p.4 l.7: the methodology is unclear. Which variables are nudged? What about the wind? What is the MATCH/CAM approach? Is MATCH used here?

We add more details of this approach, which is widely used in the general circulation community to drive nudged climate model versions. Note that here we are describing the CAM4, but a similar approach is used in the MATCH model, as discussed in that section of the methods.

"In this procedure the horizontal wind components, air temperature, surface temperature, surface pressure, sensible and latent heat flux and wind stress are read into the model simulation from the input meteorological dataset. These fields are subsequently used to internally generate (using the existing CAM4 parameterizations) the variables necessary for (1) calculating subgrid scale transport including boundary layer transport and convective transport; (2) the variables necessary for specifying the hydrological cycle, including cloud and water25 vapor distributions and rainfall (see Lamarque et al., 2011 for more details; developed in Rasch et al., 1997;Mahowald et al., 1997)."

p.4 l.8-11: some details about the data are needed e.g. which reanalysis is used (NCEP and ECMWF are forecast centres).

Good point: clarification has been added to specify the particular datasets (NCEP-NCAR and ERA-Interim):

"Three different reanalysis meteorological datasets were used to simulate dust entrainment, transport and removal in the CAM4 simulations: MERRA (Modern Era-Retrospective Analysis for Research and Applications version 1; Rienecker et al., 2011), NCEP (National Centers for Environmental Prediction)-NCAR (National Center for Atmospheric Research) 50- year reanalysis (Kistler et al., 2001), and ECMWF (European Center for Medium-Range Weather Forecasts) ERA-Interim (Dee et al., 2011)."

p.4 l.11-13: the purpose of the sentence is unclear; discuss conclusions of these studies?

Our goal is to point out that none of these are perfect: we modify the sentence to make some of the conclusions clearer, although it is beyond the scope of this study to review all these results:

"Within the meteorological literature there are many studies contrasting these datasets to available observations, and showing the errors in the reanalyses, especially the moisture transports and precipitation (Trenberth and Guillemot, 1998;Trenberth et al., 2000; Treberth et al., 2011;Trenberth and Fasullo, 2013). Reanalyses should not themselves be considered observations, but are the closest representation we have to observed meteorology, that can drive chemical transport models, and thus represent an important resource. Here in this paper we supplement the meteorological analysis of different reanalysis datasets by contrasting how they impact dust emission, transport and deposition."

p.4 l.18-20: more details are needed, e.g. how the model is initialized and what is computed in the atmospheric model (CAM?) exactly (winds cannot be computed alone).

Of course the reviewer is correct: this is an atmospheric model, so all fields are calculated, not just wind, as the first manuscript erroneously indicated. We clarify this:

"For AMIP simulations, the monthly mean sea surface temperatures are used to force the model online meteorology, but no atmospheric fields are used to constrain the model, in contrast to the reanalysis-driven simulations described above. Because in this case there is no inconsistency between the atmospheric model and the reanalysis, which can result in sources or sinks of water or energy, it is often considered a more robust way to simulate water vapor and thus chemistry (e.g. Hess and Mahowald, 2008;Trenberth and Guillemot, 1998;Trenberth et al., 2000). However, AMIP simulations cannot simulate exact weather events, but only interannual variability."

p.4 l.20-22: this is unclear; what is the "correct input format"?

We clarify:

"Not all of the input reanalysis data was available in the format required for CAM4 for the entire satellite era, so most of the analysis was conducted over a time period of 15 years when all the input data was accessible."

p.4 l.31 onwards: the paragraph should be moved and merged with the description of AMIP above.

This is a good point was incorporated into the manuscript.

p.4 l.34: why is there "no inconsistency between the atmospheric model and observations"?

When models are forced to follow observations, there is an inconsistency in the model simulation and real world, which is sampled in the observations. This inconsistency often results in effective heating terms, which cause vertical velocities, for example, and lack of energy conservation. "Observations" was clarified to be "observations from the reanalysis". AMIP simulations don't have a conflict because they don't use the reanalyses. We clarify:

"Because in this case there is no inconsistency between the atmospheric model and the reanalysis, which can result in sources or sinks of water or energy, it is often considered a more robust way to simulate water vapor and thus chemistry (e.g. Hess and Mahowald, 2008;Trenberth and Guillemot, 1998;Trenberth et al., 2000)."

p.5 l.9-10: the purpose of the sentence is unclear.

We wish to specify exactly what our CAM5 setup was. We hope the additional sentences at the beginning of section 2.1 will make this clearer.

p.5 l.11-21: the contrast with CAM needs to be better emphasized.

The wording was changed to highlight differences.

p.5 l.30 onwards: this should probably appear earlier, as MATCH is referred to in the previous

section.

We respectfully disagree on this point. Even though MATCH is mentioned earlier, the CAM4 simulations are the core focus of the paper, and we wish to fully discuss them before branching out into other models. It was necessary to refer to MATCH, though, because some of the CAM methodology was developed in MATCH. We hope that the new paragraph at the beginning of the modeling section will clarify this.

p.6 l.9-11: this appears disconnected from the other paragraphs.

Good point: we move this paragraph up to the CAM4 part of the text, which makes more sense.

p.7 l.5-7: the definitions are confusing.

This paragraph has been removed from the new draft, because of the emphasis on IAV.

p.8 l.1-4: this is very confusing; does it mean that the method is wrong?

Good point: we clarify:

"Note that the modeled monthly mean values are not at all Gaussian distributed, and thus normal methods for determining the number of observations would not work (e.g. Wilks, 2006). Thus, for this analysis, we use we use rank correlations, which works with non-gaussian data. To be consistent with the climate model community (Taylor, 2001; Gleckler et al., 2008), for mean and standard deviation analysis described above, we use these standard metrics, despite the fact that our datasets do have not Gaussian distribution, which will lead to some errors in our results."

3. Results and Discussion

p.8 l.8: does it mean that the simulations are taken from previous studies?

We conducted all of the CAM simulations for this study, but some of that model results was used for additional studies (with different focuses) published before this one. The MATCH and GEOS-CHEM simulations were conducted for other studies (also with different focuses), and then that data was also used here. We clarify:

"The annual mean distribution of the model simulations included here are evaluated elsewhere in more detail, since many of these model results were previously published (Luo et al., 2003;Huneeus et al., 2011;Albani et al., 2014;Ridley et al., 2013;Ridley et al., 2014)…"

p.8 l.11: should it be S3?

Yes, this error has been fixed.

p.8 l.24-25: the sentence is vague.

We rephrase:

"Thus there are large differences in the deduced AOD and uncertainties depending on

assumptions about how to include different data, as well as the details of the models and methodology used."

p.9 l.1: which data? The purpose of the sentence is unclear.

A clarification has been added: "these in situ and sun photometry data"

p.9 l.5 onwards: there is a lot of information presented in many figures and tables split between the main manuscript and the supporting information, which makes the discussion very difficult to follow; the information needs to be presented in a concise way and the discussion better structured.

We have tried to streamline the manuscript and reduce the number of figures in both the paper and the online supplement. We focus on surface concentration and IAV, instead of presenting so many different metrics.

p.10 l.17-32: this should be moved to the introduction.

We have added more information about the role of meteorology in variability in the introduction, in response to both reviewers, and try to explain better the focus of this paper. We think this material needs to be specifically in this section as well, as we discuss the Sahel here, as opposed to all global sources.

p.11: the whole discussion is confusing (is there a trend or not?) and its purpose is unclear, as it is not related neither to the monthly nor to the interannual variability.

This section focuses on IAV, especially trends in IAV, which we try to make sure is more clear. Our goal is to understand both whether the IAV seen in the previous study (Ridley et al., 2015) is robust across models, and whether the mechanisms hypothesized in the previous study looks robust across models. We think this type of hypothesis should be tested across multiple model frameworks, since, as we show in later sections, most IAV is not robust across the models. We try to explain better in the text:

"Here we can consider whether the hypothesis put forward in Ridley et al. (2014), that the decrease in winds in the surface region is responsible for the observed annually averaged decrease in surface concentration at Barbados, is consistent with the simulated trends in the multiple models included in this analysis."

p.12 l.7-19: CAM4 exhibits much higher monthly variability than the other models.

We agree that some of the versions of the CAM4 have much higher variability, but not all of them. The text has been expanded to discuss this:

"The CAM4-ERAI and CAM4-AMIP simulations especially overpredict variability. Note however, that some of the CAM4 models (CAM4-MERRA and CAM4-NCEP) have similar variability as the non-CAM4 models (GCHEM-MERRA and MATCH-NCEP), especially at Kerguelen and Tinga Tingana, suggesting that some of this variability may also be associated with the meteorological dataset."

p.12 l.19-21: models driven by MERRA and NCEP are clearly better than ERAI and AMIP.

We include this point above.

p.12 l.25-27: the statement is weak.

We agree that both the data and our statement is weak. We have included more Southern Hemisphere data to strengthen our argument, but fundamentally we are limited by the lack of available data.  We modify to indicate this:

"Because of the limited data and length of data, we cannot be sure, but the observations presented here are consistent with a stronger role of interannual variability, compared to seasonal variability, in dust sources in the Southern Hemisphere than in the Northern Hemisphere, as simulated by the models. Additional long term data in the Southern Hemisphere would allow more testing of this model result."

p.13 l.1-25: the results do not appear robust.

This section has been rewritten and modified for the new focus on IAV and surface concentration, so we hope the new section more convincingly illustrates the mentioned points.

p.13 l.28-31: this should be moved to the methods.

 This information is already contained in the method section, and we think it is required here for clarify and help interpreting the results.

p.13 l.33-p.14 l.2: are the AMIP simulations meaningful then?

Comparisons with the AMIP simulations show that using reanalyzed meteorology does produced an improvement of the results when reanalyses is used, so this is helpful to those of us who can use AMIP or reanalysis data, and also show what fraction of the dust variability is due just to ocean SSTs.  We clarify:

"There are much higher correlations between model results when we use reanalysis winds compared to forcing with only sea-surface temperatures, indicating the value of using reanalyses datasets to obtain more robust results."

p.15 l.8-10: is Figure 13 relevant then?

One of the goals of the study is to explore the coherence of the different models in terms of IAV, so we think Figure 13 (now Figure 9) is extremely relevant to show, and illustrate the variability of IAV across models.

p.16 l.28: should this be Figure 16? Are Figure 14 and 15 needed at all?

The caption should have been for Figure 16, and that has been corrected (the figure in question is now Figure 11). Figures 14 and 15 are less relevant, but still demonstrate important trends in regional deposition and AOD, and have thus been moved to the supplement.

p.16 l.28-30: isn't it obvious that a whole year of data is required to estimate the annual mean?

Yes, we agree (and add in "as expected" to the text), but in some regions we need more. Our goal is to provide 'evidence' for observationalists so they can get funding to conduct longer time series, since this is such a constraint on any understanding of dust distributions

p.17 l.11-12: this again requires the set-up of AMIP simulations to be better explained and asks if they are meaningful.

Our goal is to understand how much AMIP-style simulations can simulate dust, so the fact that they do worse is of course meaningful and helpful.  We clarify:

"The model simulations do roughly similarly well compared to observations when driven by reanalysis meteorology, but less well when driven by sea surface temperatures with meteorology being prognostically calculated, implying that using reanalyzed meteorology does improve dust simulations (Figure 3)."

p.17 l.25 onwards: the results do not appear that clear in section 3.

We have rewritten the results section to present the results more clearly. We also modify the conclusion section to make more clear our results, and the association with specific points in the results section.

[revised manuscript text omitted]

Formatted [201]
Formatted [204]
Formatted [205]
Formatted Table [202]
Formatted [207]
Formatted [206]
Formatted [208]
Formatted [209]
Formatted [210]
Formatted [211]
Formatted [212]
Formatted [213]
Formatted [214]
Formatted [216]
Formatted [217]
Formatted [218]
Formatted [219]
Formatted [215]
Formatted [220]
Formatted [221]
Formatted [222]
Formatted [223]
Formatted [224]
Formatted [225]
Formatted [226]
Formatted [227]
Formatted [228]
Formatted [229]
Formatted [230]
Formatted [231]
Formatted [232]
Formatted [233]
Formatted [235]
Formatted [236]
Formatted [237]
Formatted [238]
Formatted [239]
Formatted [234]
Formatted [240]
Formatted [241]
Formatted [242]
Formatted [243]
Formatted [244]
Formatted [245]
Formatted [246]
Formatted [247]
Formatted [248]
Formatted [249]
Formatted [250]
Formatted [251]
Formatted [252]
Formatted [253]
Formatted [254]
Formatted [255]

**Table 4: Slope of the normalized annual mean values from 1982 to 2008 (or time period available, shown in Table 1) for the Western Sahel (13 to 22°N and -20 to 13°E) and North Africa (0 to 35°N and -20 to 40°E) and model (Figure 4) (statistically significant values are in bold, standard deviation of slope in parenthesis for Barbados surf. conc. slope) in the first four columns. Values are normalized by dividing by the mean, so that slopes represent relative change per year. The last column is the correlation of interannual variability in precipitation in each model compared to observations.**

| | Slope Barbados surf. conc. | Slope Sahel source | Slope North African source | Slope Sahel Precip. | Correl. With obs. Sahel precip. |
|---|---|---|---|---|---|
| CAM4 (MERRA) | **-0.014 (0.0054)** | **-0.0065** | -0.0005 | 0.0035 | **0.43** |
| CAM4 (NCEP) | -0.0017 (0.016) | **-0.0169** | **-0.0074** | **0.0425** | 0.29 |
| CAM4 (ERAI) | **-0.0058 (0.0051)** | -0.0006 | 0.002 | 0.0008 | **0.81** |
| CAM4 (AMIP) | **-0.0079 (0.0035)** | **-0.0061** | **-0.0037** | **0.0186** | **0.42** |
| GCHEM (MERRA) | **-0.025 (0.0047)** | **-0.021** | **-0.0072** | 0.0035 | **0.43** |
| MATCH (NCEP) | **-0.0087 (0.0059)** | **-0.0047** | 0.027 | **0.0425** | 0.29 |
| CAM5 (AMIP) | -0.01 (0.01) | **0.0027** | 0.029 | | |
| Obs | **-0.016 (0.006)** | | | **0.0089** | |

**Table 5: Variability in Southern Hemisphere**
**Values for the IAV variability (annual average standard deviation divided by mean) and the ratio of the variability from the seasonal cycle over the IAV (for the surface concentration in the model cases and data from Rio Gallegos; deposition data from Kerguelen; and coarse mode AOD from Tinga Tingana (locations listed in Table 1).**

| Model/ Observations | Rio Gallegos Surface concentrations | | Kerguelen deposition | | Tinga Tingana AOD | |
|---|---|---|---|---|---|---|
| | IAV variability | Ratio variability from seasonal cycle over IAV | IAV variability | Ratio variability from seasonal cycle over IAV | IAV variability | Ratio variability from seasonal cycle over IAV |
| CAM4 (MERRA) | 1.78 | 1.06 | 0.22 | 1.21 | 0.11 | 3.61 |
| CAM4 (NCEP) | 2.13 | 0.79 | 0.27 | 1.02 | 0.23 | 2.21 |
| CAM4 (ERAI) | 2.66 | 0.84 | 1.42 | 1.32 | 0.21 | 1.81 |
| CAM4 (AMIP) | 3.86 | 0.55 | 2.06 | 0.48 | 0.17 | 2.68 |
| GCHEM (MERRA) | 0.67 | 1.26 | 0.32 | 1.03 | 0.25 | 1.87 |
| MATCH (NCEP) | 0.68 | 1.37 | 0.17 | 2.16 | 0.21 | 2.55 |
| CAM5 (AMIP) | 0.42 | 3.42 | 0.46 | 2.37 | 0.14 | 3.25 |
| Obs | 0.10 | 1.39 | 0.08 | 4.01 | 0.42 | 1.48 |

Formatted ... [456]
Formatted ... [369]
Formatted ... [372]
Formatted ... [373]
Formatted Table ... [370]
Formatted ... [371]
Formatted ... [374]
Formatted ... [375]
Formatted ... [377]
Formatted ... [378]
Formatted ... [379]
Formatted ... [380]
Formatted ... [381]
Formatted ... [382]
Formatted ... [376]
Formatted ... [383]
Formatted ... [384]
Formatted ... [386]
Formatted ... [387]
Formatted ... [388]
Formatted ... [389]
Formatted ... [390]
Formatted ... [391]
Formatted ... [385]
Formatted ... [392]
Formatted ... [393]
Formatted ... [395]
Formatted ... [396]
Formatted ... [397]
Formatted ... [398]
Formatted ... [399]
Formatted ... [400]
Formatted ... [394]
Formatted ... [401]
Formatted ... [402]
Formatted ... [404]
Formatted ... [405]
Formatted ... [406]
Formatted ... [407]
Formatted ... [408]
Formatted ... [409]
Formatted ... [403]
Formatted ... [410]
Formatted ... [411]
Formatted ... [413]
Formatted ... [414]
Formatted ... [415]
Formatted ... [416]
Formatted ... [417]
Formatted ... [418]
Formatted ... [412]
Formatted ... [419]
Formatted ... [420]
Formatted ... [422]
Formatted ... [423]
Formatted ... [424]
Formatted ... [425]
Formatted ... [426]
Formatted ... [427]
Formatted ... [421]
Formatted ... [428]
Formatted ... [431]
Formatted ... [432]
Formatted ... [433]
Formatted ... [434]
Formatted ... [435]
Formatted ... [436]

**Table 6: Regional sources of dust. For each source region, the averaged correlation across time between annual mean source strengths for the CAM-RE cases is shown in the second column. The following columns show the climatological mean source strength (Tg/year) for the mean of the 3 CAM4-RE simulations and the mean of the 7 simulations included in this study. The +/- % standard deviation is also shown, and represents the standard deviations across the models included in the averaging. The regions are defined as follows: Australia: 35 to 25°S, 130 to 150°E; East Asia: 35 to 50°N, 70 to 112°E; Middle East: 10 to 45°N, 40 to 70°E; North Africa: 10 to 35°N, 40°W to 40°E; Sahel (western): 13 to 22°N, 40°W to 13°E; South Africa: 35 to 20°S, 15 to 40°E; South America (Argentina): 55 to 35°S, 285 to 310°E.**

| | Avg. IAV Temporal Correlation | Mean CAM4-RE | Mean all |
|---|---|---|---|
| Australia | 0.73 | 25 +/-70% | 38 +/- 90% |
| East Asia | 0.58 | 230+/- 70% | 230 +/-70% |
| Middle East | 0.40 | 570+/-40% | 510 +/- 40% |
| North Africa | 0.47 | 1370 +/-23% | 1490 +/- 40% |
| North America | 0.78 | 72 +/-160% | 70 +/-130% |
| Sahel (western) | 0.13 | 460 +/-27% | 520 +/- 40% |
| South Africa | 0.46 | 9 +/- 90% | 8 +/- 70% |
| South America | 0.68 | 14 +/-160% | 34 +/- 130% |
| Globe | 0.49 | 2400 +/-26% | 2500 +/- 40% |

**Table 7:** Correlations in meteorological variables and mobilization in different regions for IAV. Time series are correlated for the annual average over 1990-2005 in each region (only including gridboxes which are active at any time in that model simulation). Values shown are the averages of the correlations across the CAM4-Reanalysis models (CAM4 (MERRA), CAM4 (NCEP) and CAM4 (ERAI)). The regions are defined as in Table 6.

| | Precipitation | Soil moisture | Leaf Area Index | Sfc. Wind |
|---|---|---|---|---|
| Australia | -0.59 | -0.61 | -0.72 | 0.10 |
| East Asia | 0.06 | -0.08 | -0.32 | 0.67 |
| Middle East | -0.32 | -0.33 | -0.28 | 0.36 |
| North Africa | -0.27 | -0.26 | -0.20 | 0.51 |
| North America | -0.57 | -0.64 | -0.53 | 0.32 |
| Sahel (western) | -0.28 | -0.26 | -0.42 | 0.81 |
| South Africa | -0.37 | -0.38 | -0.55 | 0.22 |
| Globe | -0.36 | -0.29 | -0.46 | 0.33 |

**Table 8:** Surface concentration over ocean basins. For each ocean region, the averaged correlation across time between annual mean deposition fluxes for the CAM-RE cases is shown in the second column. The third column shows the annual mean correlation with NAO, while the third column shows the annual mean correlation with the El Nino/Southern Oscillation climate index. Regions are defined as the ocean gridboxes (not including sea ice or land boxes) in the following latitude and longitude areas as from (Gregg et al., 2003): North Atlantic (>30°N; 270 to 30°E); North Pacific (>30°N; 120 to 270°E); North Central Atlantic (10 to 30°N, 270 to 30°E); North Central Pacific (10 to 30N; 120 to 270°E); North Indian (10 to 30°N; 30 to 120°E); Equatorial Atlantic (-10 to 10°N; 300 to 30°E); Equatorial Pacific (-10 to 10°N; 120 to 285°E); Equatorial Indian (-10 to 10°N; 30-120°E); South Atlantic (-30 to -10°N; 30 to 300°E); South Pacific (-30 to -10°N; 120 to 295°E); South Indian (-30 to -30°N, 30 to 120°E); Antarctic (<-30°N).

| | CAM4-RE across model Correlation | NAO correlation | El Nino Correlation |
|---|---|---|---|
| North Atlantic | 0.66 | 0.10 | 0.45 |
| North Pacific | 0.51 | 0.19 | 0.62 |
| North Central | 0.75 | 0.04 | -0.10 |

Formatted ... [527]
Formatted ... [528]
Formatted ... [532]
Formatted ... [533]
Formatted ... [535]
Formatted Table ... [529]
Formatted ... [530]
Formatted ... [536]
Formatted ... [538]
Formatted ... [531]
Formatted ... [534]
Formatted ... [537]
Formatted ... [539]
Formatted ... [540]
Formatted ... [541]
Formatted ... [542]
Formatted ... [543]
Formatted ... [544]
Formatted ... [545]
Formatted ... [546]
Formatted ... [547]
Formatted ... [548]
Formatted ... [549]
Formatted ... [550]
Formatted ... [551]
Formatted ... [552]
Formatted ... [553]
Formatted ... [554]
Formatted ... [555]
Formatted ... [556]
Formatted ... [557]
Formatted ... [558]
Formatted ... [559]
Formatted ... [560]
Formatted ... [561]
Formatted ... [562]
Formatted ... [563]
Formatted ... [564]
Formatted ... [565]
Formatted ... [566]
Formatted ... [567]
Formatted ... [568]
Formatted ... [569]
Formatted ... [570]
Formatted ... [571]
Formatted ... [572]
Formatted ... [573]
Formatted ... [574]
Formatted ... [575]
Formatted ... [576]
Formatted ... [577]
Formatted ... [578]
Formatted ... [579]
Formatted ... [580]
Formatted ... [581]
Formatted ... [582]
Formatted ... [583]
Formatted ... [584]
Formatted ... [585]
Formatted ... [586]
Formatted ... [587]
Formatted ... [588]
Formatted ... [589]
Formatted ... [590]
Formatted ... [591]
Formatted ... [592]
Formatted ... [593]
Formatted ... [594]

| | | | |
|---|---|---|---|

[revised manuscript text omitted]
 (0.42), with a strong anticorrelation over the Antarctic Ocean (-0.63). The NAO also featured a strong Antarctic anticorrelation (-0.42), and had the highest correlation coefficient over the South Indian Ocean (0.49). Again, this is similar to previous studies (e.g. Mahowald, 2003).The correlations tend to be lower near the southwestern US, and in the Southern hemisphere (Figure 10).

The correlation coefficients averaged over the CAM4-reanalysis models (Figure 11a, c and f) suggest that the surface concentration and deposition are similarly correlated, but that AOD has smoother and higher correlations. This is consistent with AOD being a more integrated quantity (Figure 8) (Huneeus et al., 2011; Albani et al., 2014). The lowest correlations occur over remote ocean regions, especially in the Southern Hemisphere, for all the variables. Note also that the correlation coefficients between the simulations using the same modeling framework but different meteorology (Figure 11a, c and f) had similar correlation coefficients, with a similar spatial structure, as the correlations between simulations using different modeling frameworks but the same meteorology (Figure 11b, d, and e). This suggests that meteorology and the modeling framework are equally important for simulating variability.

| Page 15: [9] Deleted | Cornell University | 11/15/16 2:23:00 PM |

The magnitude and the spatial structure of the correlation in the seasonal cycle is similar for all the variables as that seen using all monthly means (Figure 12a, c, e vs. 11a, c, e), suggesting that most of the correlation between the model versions is due to the correlation in the seasonal cycle (Figure 12). This is made clearer when considering just

the correlation of the annual means, showing that the models simulations are much less similar in their interannual variability (Figure 12b, d and f).  Note however, that the spatial structure in the correlations for IAV is very different than considering either the monthly mean or seasonal cycle (Figure 12b, d, f vs. 12 a, c, e).

**Page 15: [10] Moved to page 15 (Move #2)Cornell University**         **11/15/16 2:25:00 PM**

  A comparison of other aerosols in two of the CAM4-reanalysis based simulations available here (Figure S5), is consistent with the idea that dust is highly variable, and that there is some correlation between the models driven by different meteorology far from the sources as well as close, but that interannual variability can be quite different for transport of aerosols (Figure S5). We will next discuss regional averages to understand how similarly ocean basin averages are simulated, to see if these IAV correlations in some regions are large enough to provide coherent basin estimates.

**Page 15: [11] Deleted**         **Cornell University**         **12/3/16 3:04:00 PM**

, but that interannual variability can be quite different for transport of aerosols

**Page 15: [12] Deleted**         **Smith, Molly B**         **11/24/16 2:03:00 PM**

**Page 30: [13] Formatted**         **Smith, Molly B**         **11/24/16 2:05:00 PM**

Font:12 pt

**Page 30: [14] Formatted**         **Smith, Molly B**         **11/24/16 2:12:00 PM**

None, Space Before:  0 pt, Line spacing:  single, Don't keep with next, Don't keep lines together

**Page 30: [15] Formatted Table**         **Smith, Molly B**         **11/24/16 2:12:00 PM**

Formatted Table

**Page 30: [16] Formatted**         **Smith, Molly B**         **11/24/16 2:05:00 PM**

Font:12 pt, Not Bold, Not Italic, Font color: Auto

**Page 30: [17] Formatted**         **Smith, Molly B**         **11/24/16 2:05:00 PM**

Font:12 pt

**Page 30: [17] Formatted**         **Smith, Molly B**         **11/24/16 2:05:00 PM**

Font:12 pt

**Page 30: [18] Formatted**         **Smith, Molly B**         **11/24/16 2:05:00 PM**

Font:12 pt

**Page 30: [18] Formatted**         **Smith, Molly B**         **11/24/16 2:05:00 PM**

Font:12 pt

**Page 30: [19] Formatted**         **Smith, Molly B**         **11/24/16 2:05:00 PM**

Font:12 pt

**Page 30: [19] Formatted**         **Smith, Molly B**         **11/24/16 2:05:00 PM**

Font:12 pt

**Page 30: [20] Formatted**         **Smith, Molly B**         **11/24/16 2:05:00 PM**

Font:12 pt

| Page 30: [20] Formatted | Smith, Molly B | 11/24/16 2:05:00 PM |

Font:12 pt

| Page 30: [21] Formatted | Smith, Molly B | 11/24/16 2:05:00 PM |

Font:12 pt

| Page 30: [21] Formatted | Smith, Molly B | 11/24/16 2:05:00 PM |

Font:12 pt

| Page 30: [22] Formatted | Smith, Molly B | 11/24/16 2:05:00 PM |

Font:12 pt

| Page 30: [23] Formatted | Smith, Molly B | 11/24/16 2:12:00 PM |

None, Space Before:  0 pt, Line spacing:  single, Don't keep with next, Don't keep lines together

| Page 30: [24] Formatted | Smith, Molly B | 11/24/16 2:05:00 PM |

Font:12 pt, Not Bold, Not Italic, Font color: Auto

| Page 30: [25] Formatted | Smith, Molly B | 11/24/16 2:05:00 PM |

Font:12 pt

| Page 30: [25] Formatted | Smith, Molly B | 11/24/16 2:05:00 PM |

Font:12 pt

| Page 30: [26] Formatted | Smith, Molly B | 11/24/16 2:05:00 PM |

Font:12 pt

| Page 30: [26] Formatted | Smith, Molly B | 11/24/16 2:05:00 PM |

Font:12 pt

| Page 30: [27] Formatted | Smith, Molly B | 11/24/16 2:05:00 PM |

Font:12 pt

| Page 30: [27] Formatted | Smith, Molly B | 11/24/16 2:05:00 PM |

Font:12 pt

| Page 30: [28] Formatted | Smith, Molly B | 11/24/16 2:05:00 PM |

Font:12 pt

| Page 30: [28] Formatted | Smith, Molly B | 11/24/16 2:05:00 PM |

Font:12 pt

| Page 30: [29] Formatted | Smith, Molly B | 11/24/16 2:05:00 PM |

Font:12 pt

| Page 30: [29] Formatted | Smith, Molly B | 11/24/16 2:05:00 PM |

Font:12 pt

| Page 30: [30] Formatted | Smith, Molly B | 11/24/16 2:05:00 PM |

Font:12 pt

| Page 30: [31] Formatted | Smith, Molly B | 11/24/16 2:12:00 PM |

None, Space Before:  0 pt, Line spacing:  single, Don't keep with next, Don't keep lines together

| Page 30: [32] Formatted | Smith, Molly B | 11/24/16 2:05:00 PM |
|---|---|---|

Font:12 pt, Not Bold, Not Italic, Font color: Auto

| Page 30: [33] Formatted | Smith, Molly B | 11/24/16 2:05:00 PM |
|---|---|---|

Font:12 pt

| Page 30: [33] Formatted | Smith, Molly B | 11/24/16 2:05:00 PM |
|---|---|---|

Font:12 pt

| Page 30: [34] Formatted | Smith, Molly B | 11/24/16 2:05:00 PM |
|---|---|---|

Font:12 pt

| Page 30: [34] Formatted | Smith, Molly B | 11/24/16 2:05:00 PM |
|---|---|---|

Font:12 pt

| Page 30: [35] Formatted | Smith, Molly B | 11/24/16 2:05:00 PM |
|---|---|---|

Font:12 pt

| Page 30: [35] Formatted | Smith, Molly B | 11/24/16 2:05:00 PM |
|---|---|---|

Font:12 pt

| Page 30: [36] Formatted | Smith, Molly B | 11/24/16 2:05:00 PM |
|---|---|---|

Font:12 pt

| Page 30: [36] Formatted | Smith, Molly B | 11/24/16 2:05:00 PM |
|---|---|---|

Font:12 pt

| Page 30: [37] Formatted | Smith, Molly B | 11/24/16 2:05:00 PM |
|---|---|---|

Font:12 pt

| Page 30: [37] Formatted | Smith, Molly B | 11/24/16 2:05:00 PM |
|---|---|---|

Font:12 pt

| Page 30: [38] Formatted | Smith, Molly B | 11/24/16 2:05:00 PM |
|---|---|---|

Font:12 pt

| Page 30: [39] Formatted | Smith, Molly B | 11/24/16 2:12:00 PM |
|---|---|---|

None, Space Before:  0 pt, Line spacing:  single, Don't keep with next, Don't keep lines together

| Page 30: [40] Formatted | Smith, Molly B | 11/24/16 2:05:00 PM |
|---|---|---|

Font:12 pt, Not Bold, Not Italic, Font color: Auto

| Page 30: [41] Formatted | Smith, Molly B | 11/24/16 2:05:00 PM |
|---|---|---|

Font:12 pt

| Page 30: [41] Formatted | Smith, Molly B | 11/24/16 2:05:00 PM |
|---|---|---|

Font:12 pt

| Page 30: [42] Formatted | Smith, Molly B | 11/24/16 2:05:00 PM |
|---|---|---|

Font:12 pt

| Page 30: [42] Formatted | Smith, Molly B | 11/24/16 2:05:00 PM |
|---|---|---|

Font:12 pt

| Page 30: [43] Formatted | Smith, Molly B | 11/24/16 2:05:00 PM |
|---|---|---|

Font:12 pt

| Page 30: [43] Formatted | Smith, Molly B | 11/24/16 2:05:00 PM |

Font:12 pt

| Page 30: [44] Formatted | Smith, Molly B | 11/24/16 2:05:00 PM |

Font:12 pt

| Page 30: [44] Formatted | Smith, Molly B | 11/24/16 2:05:00 PM |

Font:12 pt

| Page 30: [45] Formatted | Smith, Molly B | 11/24/16 2:05:00 PM |

Font:12 pt

| Page 30: [45] Formatted | Smith, Molly B | 11/24/16 2:05:00 PM |

Font:12 pt

| Page 30: [46] Formatted | Smith, Molly B | 11/24/16 2:05:00 PM |

Font:12 pt

| Page 30: [47] Formatted | Smith, Molly B | 11/24/16 2:12:00 PM |

None, Space Before:  0 pt, Line spacing:  single, Don't keep with next, Don't keep lines together

| Page 30: [48] Formatted | Smith, Molly B | 11/24/16 2:05:00 PM |

Font:12 pt, Not Bold, Not Italic, Font color: Auto

| Page 30: [49] Formatted | Smith, Molly B | 11/24/16 2:05:00 PM |

Font:12 pt

| Page 30: [49] Formatted | Smith, Molly B | 11/24/16 2:05:00 PM |

Font:12 pt

| Page 30: [50] Formatted | Smith, Molly B | 11/24/16 2:05:00 PM |

Font:12 pt

| Page 30: [50] Formatted | Smith, Molly B | 11/24/16 2:05:00 PM |

Font:12 pt

| Page 30: [51] Formatted | Smith, Molly B | 11/24/16 2:05:00 PM |

Font:12 pt

| Page 30: [51] Formatted | Smith, Molly B | 11/24/16 2:05:00 PM |

Font:12 pt

| Page 30: [52] Formatted | Smith, Molly B | 11/24/16 2:05:00 PM |

Font:12 pt

| Page 30: [52] Formatted | Smith, Molly B | 11/24/16 2:05:00 PM |

Font:12 pt

| Page 30: [53] Formatted | Smith, Molly B | 11/24/16 2:05:00 PM |

Font:12 pt

| Page 30: [53] Formatted | Smith, Molly B | 11/24/16 2:05:00 PM |

Font:12 pt

| Page 30: [54] Formatted | Smith, Molly B | 11/24/16 2:05:00 PM |
|---|---|---|

Font:12 pt

| Page 30: [55] Formatted | Smith, Molly B | 11/24/16 2:12:00 PM |
|---|---|---|

None, Space Before:  0 pt, Line spacing:  single, Don't keep with next, Don't keep lines together

| Page 30: [56] Formatted | Smith, Molly B | 11/24/16 2:05:00 PM |
|---|---|---|

Font:12 pt, Not Bold, Not Italic, Font color: Auto

| Page 30: [57] Formatted | Smith, Molly B | 11/24/16 2:05:00 PM |
|---|---|---|

Font:12 pt

| Page 30: [57] Formatted | Smith, Molly B | 11/24/16 2:05:00 PM |
|---|---|---|

Font:12 pt

| Page 30: [58] Formatted | Smith, Molly B | 11/24/16 2:05:00 PM |
|---|---|---|

Font:12 pt

| Page 30: [58] Formatted | Smith, Molly B | 11/24/16 2:05:00 PM |
|---|---|---|

Font:12 pt

| Page 30: [59] Formatted | Smith, Molly B | 11/24/16 2:05:00 PM |
|---|---|---|

Font:12 pt

| Page 30: [59] Formatted | Smith, Molly B | 11/24/16 2:05:00 PM |
|---|---|---|

Font:12 pt

| Page 30: [60] Formatted | Smith, Molly B | 11/24/16 2:05:00 PM |
|---|---|---|

Font:12 pt

| Page 30: [60] Formatted | Smith, Molly B | 11/24/16 2:05:00 PM |
|---|---|---|

Font:12 pt

| Page 30: [61] Formatted | Smith, Molly B | 11/24/16 2:05:00 PM |
|---|---|---|

Font:12 pt

| Page 30: [61] Formatted | Smith, Molly B | 11/24/16 2:05:00 PM |
|---|---|---|

Font:12 pt

| Page 30: [62] Formatted | Smith, Molly B | 11/24/16 2:05:00 PM |
|---|---|---|

Font:12 pt

| Page 30: [63] Formatted | Smith, Molly B | 11/24/16 2:12:00 PM |
|---|---|---|

None, Space Before:  0 pt, Line spacing:  single, Don't keep with next, Don't keep lines together

| Page 30: [64] Formatted | Smith, Molly B | 11/24/16 2:05:00 PM |
|---|---|---|

Font:12 pt, Not Bold, Not Italic, Font color: Auto

| Page 30: [65] Formatted | Smith, Molly B | 11/24/16 2:05:00 PM |
|---|---|---|

Font:12 pt

| Page 30: [65] Formatted | Smith, Molly B | 11/24/16 2:05:00 PM |
|---|---|---|

Font:12 pt

| Page 30: [66] Formatted | Smith, Molly B | 11/24/16 2:05:00 PM |
|---|---|---|

Font:12 pt

| Page 30: [66] Formatted | Smith, Molly B | 11/24/16 2:05:00 PM |

Font:12 pt

| Page 30: [67] Formatted | Smith, Molly B | 11/24/16 2:05:00 PM |

Font:12 pt

| Page 30: [67] Formatted | Smith, Molly B | 11/24/16 2:05:00 PM |

Font:12 pt

| Page 30: [68] Formatted | Smith, Molly B | 11/24/16 2:05:00 PM |

Font:12 pt

| Page 30: [68] Formatted | Smith, Molly B | 11/24/16 2:05:00 PM |

Font:12 pt

| Page 30: [69] Formatted | Smith, Molly B | 11/24/16 2:05:00 PM |

Font:12 pt

| Page 30: [69] Formatted | Smith, Molly B | 11/24/16 2:05:00 PM |

Font:12 pt

| Page 30: [70] Formatted | Smith, Molly B | 11/24/16 2:05:00 PM |

Font:12 pt

| Page 30: [71] Formatted | Smith, Molly B | 11/24/16 2:12:00 PM |

None, Space Before:  0 pt, Line spacing:  single, Don't keep with next, Don't keep lines together

| Page 30: [72] Formatted | Smith, Molly B | 11/24/16 2:05:00 PM |

Font:12 pt, Not Bold, Not Italic, Font color: Auto

| Page 30: [73] Formatted | Smith, Molly B | 11/24/16 2:05:00 PM |

Font:12 pt

| Page 30: [73] Formatted | Smith, Molly B | 11/24/16 2:05:00 PM |

Font:12 pt

| Page 30: [74] Formatted | Smith, Molly B | 11/24/16 2:05:00 PM |

Font:12 pt

| Page 30: [74] Formatted | Smith, Molly B | 11/24/16 2:05:00 PM |

Font:12 pt

| Page 30: [75] Formatted | Smith, Molly B | 11/24/16 2:05:00 PM |

Font:12 pt

| Page 30: [75] Formatted | Smith, Molly B | 11/24/16 2:05:00 PM |

Font:12 pt

| Page 30: [76] Formatted | Smith, Molly B | 11/24/16 2:05:00 PM |

Font:12 pt

| Page 30: [76] Formatted | Smith, Molly B | 11/24/16 2:05:00 PM |

Font:12 pt

| Page 30: [77] Formatted | Smith, Molly B | 11/24/16 2:05:00 PM |
|---|---|---|

Font:12 pt

| Page 30: [77] Formatted | Smith, Molly B | 11/24/16 2:05:00 PM |
|---|---|---|

Font:12 pt

| Page 30: [78] Deleted | Smith, Molly B | 11/24/16 2:07:00 PM |
|---|---|---|

|  |  |  |  |  |
|---|---|---|---|---|
|  |  |  |  |  |

| Page 30: [79] Formatted | Smith, Molly B | 11/24/16 2:05:00 PM |
|---|---|---|

Font:12 pt

| Page 30: [80] Formatted | Smith, Molly B | 11/24/16 2:05:00 PM |
|---|---|---|

Line spacing:  single

| Page 30: [81] Formatted | Smith, Molly B | 11/24/16 2:05:00 PM |
|---|---|---|

Font:12 pt

| Page 31: [82] Formatted | Smith, Molly B | 11/24/16 2:05:00 PM |
|---|---|---|

Font:12 pt

| Page 31: [83] Formatted | Smith, Molly B | 11/24/16 2:12:00 PM |
|---|---|---|

None, Space Before:  0 pt, Line spacing:  single, Don't keep with next, Don't keep lines together

| Page 31: [84] Formatted Table | Smith, Molly B | 11/24/16 2:12:00 PM |
|---|---|---|

Formatted Table

| Page 31: [85] Formatted | Smith, Molly B | 11/24/16 2:05:00 PM |
|---|---|---|

Font:12 pt, Not Bold, Not Italic, Font color: Auto

| Page 31: [86] Formatted | Smith, Molly B | 11/24/16 2:05:00 PM |
|---|---|---|

Font:12 pt

| Page 31: [86] Formatted | Smith, Molly B | 11/24/16 2:05:00 PM |
|---|---|---|

Font:12 pt

| Page 31: [86] Formatted | Smith, Molly B | 11/24/16 2:05:00 PM |
|---|---|---|

Font:12 pt

| Page 31: [86] Formatted | Smith, Molly B | 11/24/16 2:05:00 PM |
|---|---|---|

Font:12 pt

| Page 31: [87] Formatted | Smith, Molly B | 11/24/16 2:05:00 PM |
|---|---|---|

Font:12 pt

| Page 31: [87] Formatted | Smith, Molly B | 11/24/16 2:05:00 PM |
|---|---|---|

Font:12 pt

| Page 31: [87] Formatted | Smith, Molly B | 11/24/16 2:05:00 PM |
|---|---|---|

Font:12 pt

| Page 31: [87] Formatted | Smith, Molly B | 11/24/16 2:05:00 PM |
|---|---|---|

Font:12 pt

| Page 31: [88] Formatted | Smith, Molly B | 11/24/16 2:05:00 PM |
|---|---|---|

Font:12 pt

| Page 31: [88] Formatted | Smith, Molly B | 11/24/16 2:05:00 PM |
|---|---|---|

Font:12 pt

| Page 31: [89] Formatted | Smith, Molly B | 11/24/16 2:05:00 PM |
|---|---|---|

Font:12 pt

| Page 31: [89] Formatted | Smith, Molly B | 11/24/16 2:05:00 PM |
|---|---|---|

Font:12 pt

| Page 31: [90] Formatted | Smith, Molly B | 11/24/16 2:05:00 PM |
|---|---|---|

Font:12 pt

| Page 31: [91] Formatted | Smith, Molly B | 11/24/16 2:12:00 PM |
|---|---|---|

None, Space Before:  0 pt, Line spacing:  single, Don't keep with next, Don't keep lines together

| Page 31: [92] Formatted | Smith, Molly B | 11/24/16 2:05:00 PM |
|---|---|---|

Font:12 pt, Not Italic, Font color: Auto

| Page 31: [93] Formatted | Smith, Molly B | 11/24/16 2:05:00 PM |
|---|---|---|

Font:12 pt

| Page 31: [94] Formatted | Smith, Molly B | 11/24/16 2:12:00 PM |
|---|---|---|

Line spacing:  single

| Page 31: [95] Formatted | Smith, Molly B | 11/24/16 2:05:00 PM |
|---|---|---|

Font:12 pt

| Page 31: [96] Formatted | Smith, Molly B | 11/24/16 2:12:00 PM |
|---|---|---|

None, Space Before:  0 pt, Line spacing:  single, Don't keep with next, Don't keep lines together

| Page 31: [97] Formatted | Smith, Molly B | 11/24/16 2:05:00 PM |
|---|---|---|

Font:12 pt, Not Bold, Not Italic, Font color: Auto

| Page 31: [98] Formatted | Smith, Molly B | 11/24/16 2:05:00 PM |
|---|---|---|

Font:12 pt

| Page 31: [98] Formatted | Smith, Molly B | 11/24/16 2:05:00 PM |
|---|---|---|

Font:12 pt

| Page 31: [99] Formatted | Smith, Molly B | 11/24/16 2:05:00 PM |
|---|---|---|

Font:12 pt

| Page 31: [99] Formatted | Smith, Molly B | 11/24/16 2:05:00 PM |
|---|---|---|

Font:12 pt

| Page 31: [100] Formatted | Smith, Molly B | 11/24/16 2:05:00 PM |
|---|---|---|

Font:12 pt

| Page 31: [100] Formatted | Smith, Molly B | 11/24/16 2:05:00 PM |
|---|---|---|

Font:12 pt

| Page 31: [101] Formatted | Smith, Molly B | 11/24/16 2:05:00 PM |
|---|---|---|

Font:12 pt

| Page 31: [101] Formatted | Smith, Molly B | 11/24/16 2:05:00 PM |
|---|---|---|

Font:12 pt

| Page 31: [102] Formatted | Smith, Molly B | 11/24/16 2:05:00 PM |
|---|---|---|

Font:12 pt

| Page 31: [103] Formatted | Smith, Molly B | 11/24/16 2:12:00 PM |
|---|---|---|

None, Space Before:  0 pt, Line spacing:  single, Don't keep with next, Don't keep lines together

| Page 31: [104] Formatted | Smith, Molly B | 11/24/16 2:05:00 PM |
|---|---|---|

Font:12 pt, Not Italic, Font color: Auto

| Page 31: [105] Formatted | Smith, Molly B | 11/24/16 2:05:00 PM |
|---|---|---|

Font:12 pt

| Page 31: [105] Formatted | Smith, Molly B | 11/24/16 2:05:00 PM |
|---|---|---|

Font:12 pt

| Page 31: [106] Formatted | Smith, Molly B | 11/24/16 2:05:00 PM |
|---|---|---|

Font:12 pt

| Page 31: [106] Formatted | Smith, Molly B | 11/24/16 2:05:00 PM |
|---|---|---|

Font:12 pt

| Page 31: [107] Formatted | Smith, Molly B | 11/24/16 2:05:00 PM |
|---|---|---|

Font:12 pt

| Page 31: [107] Formatted | Smith, Molly B | 11/24/16 2:05:00 PM |
|---|---|---|

Font:12 pt

| Page 31: [108] Formatted | Smith, Molly B | 11/24/16 2:05:00 PM |
|---|---|---|

Font:12 pt

| Page 31: [108] Formatted | Smith, Molly B | 11/24/16 2:05:00 PM |
|---|---|---|

Font:12 pt

| Page 31: [109] Formatted | Smith, Molly B | 11/24/16 2:05:00 PM |
|---|---|---|

Font:12 pt

| Page 31: [110] Formatted | Smith, Molly B | 11/24/16 2:12:00 PM |
|---|---|---|

None, Space Before:  0 pt, Line spacing:  single, Don't keep with next, Don't keep lines together

| Page 31: [111] Formatted | Smith, Molly B | 11/24/16 2:05:00 PM |
|---|---|---|

Font:12 pt, Not Bold, Not Italic, Font color: Auto

| Page 31: [112] Formatted | Smith, Molly B | 11/24/16 2:05:00 PM |
|---|---|---|

Font:12 pt

| Page 31: [112] Formatted | Smith, Molly B | 11/24/16 2:05:00 PM |
|---|---|---|

Font:12 pt

| Page 31: [113] Formatted | Smith, Molly B | 11/24/16 2:05:00 PM |
|---|---|---|

Font:12 pt

| Page 31: [113] Formatted | Smith, Molly B | 11/24/16 2:05:00 PM |
|---|---|---|

Font:12 pt

| Page 31: [114] Formatted | Smith, Molly B | 11/24/16 2:05:00 PM |

Font:12 pt

| Page 31: [114] Formatted | Smith, Molly B | 11/24/16 2:05:00 PM |

Font:12 pt

| Page 31: [115] Formatted | Smith, Molly B | 11/24/16 2:05:00 PM |

Font:12 pt

| Page 31: [115] Formatted | Smith, Molly B | 11/24/16 2:05:00 PM |

Font:12 pt

| Page 31: [116] Formatted | Smith, Molly B | 11/24/16 2:05:00 PM |

Font:12 pt

| Page 31: [117] Formatted | Smith, Molly B | 11/24/16 2:12:00 PM |

None, Space Before:  0 pt, Line spacing:  single, Don't keep with next, Don't keep lines together

| Page 31: [118] Formatted | Smith, Molly B | 11/24/16 2:05:00 PM |

Font:12 pt, Not Bold, Not Italic, Font color: Auto

| Page 31: [119] Formatted | Smith, Molly B | 11/24/16 2:05:00 PM |

Font:12 pt

| Page 31: [119] Formatted | Smith, Molly B | 11/24/16 2:05:00 PM |

Font:12 pt

| Page 31: [120] Formatted | Smith, Molly B | 11/24/16 2:05:00 PM |

Font:12 pt

| Page 31: [120] Formatted | Smith, Molly B | 11/24/16 2:05:00 PM |

Font:12 pt

| Page 31: [121] Formatted | Smith, Molly B | 11/24/16 2:05:00 PM |

Font:12 pt

| Page 31: [121] Formatted | Smith, Molly B | 11/24/16 2:05:00 PM |

Font:12 pt

| Page 31: [122] Formatted | Smith, Molly B | 11/24/16 2:05:00 PM |

Font:12 pt

| Page 31: [122] Formatted | Smith, Molly B | 11/24/16 2:05:00 PM |

Font:12 pt

| Page 31: [123] Formatted | Smith, Molly B | 11/24/16 2:05:00 PM |

Font:12 pt

| Page 31: [124] Formatted | Smith, Molly B | 11/24/16 2:12:00 PM |

None, Space Before:  0 pt, Line spacing:  single, Don't keep with next, Don't keep lines together

| Page 31: [125] Formatted | Smith, Molly B | 11/24/16 2:05:00 PM |

Font:12 pt, Not Bold, Not Italic, Font color: Auto

| Page 31: [126] Formatted | Smith, Molly B | 11/24/16 2:05:00 PM |
|---|---|---|

Font:12 pt

| Page 31: [126] Formatted | Smith, Molly B | 11/24/16 2:05:00 PM |
|---|---|---|

Font:12 pt

| Page 31: [127] Formatted | Smith, Molly B | 11/24/16 2:05:00 PM |
|---|---|---|

Font:12 pt

| Page 31: [127] Formatted | Smith, Molly B | 11/24/16 2:05:00 PM |
|---|---|---|

Font:12 pt

| Page 31: [128] Formatted | Smith, Molly B | 11/24/16 2:05:00 PM |
|---|---|---|

Font:12 pt

| Page 31: [128] Formatted | Smith, Molly B | 11/24/16 2:05:00 PM |
|---|---|---|

Font:12 pt

| Page 31: [129] Formatted | Smith, Molly B | 11/24/16 2:05:00 PM |
|---|---|---|

Font:12 pt

| Page 31: [129] Formatted | Smith, Molly B | 11/24/16 2:05:00 PM |
|---|---|---|

Font:12 pt

| Page 31: [130] Formatted | Smith, Molly B | 11/24/16 2:05:00 PM |
|---|---|---|

Font:12 pt

| Page 31: [131] Formatted | Smith, Molly B | 11/24/16 2:12:00 PM |
|---|---|---|

None, Space Before:  0 pt, Line spacing:  single, Don't keep with next, Don't keep lines together

| Page 31: [132] Formatted | Smith, Molly B | 11/24/16 2:05:00 PM |
|---|---|---|

Font:12 pt, Not Bold, Not Italic, Font color: Auto

| Page 31: [133] Formatted | Smith, Molly B | 11/24/16 2:05:00 PM |
|---|---|---|

Font:12 pt

| Page 31: [133] Formatted | Smith, Molly B | 11/24/16 2:05:00 PM |
|---|---|---|

Font:12 pt

| Page 31: [134] Formatted | Smith, Molly B | 11/24/16 2:05:00 PM |
|---|---|---|

Font:12 pt

| Page 31: [134] Formatted | Smith, Molly B | 11/24/16 2:05:00 PM |
|---|---|---|

Font:12 pt

| Page 31: [135] Formatted | Smith, Molly B | 11/24/16 2:05:00 PM |
|---|---|---|

Font:12 pt

| Page 31: [135] Formatted | Smith, Molly B | 11/24/16 2:05:00 PM |
|---|---|---|

Font:12 pt

| Page 31: [136] Formatted | Smith, Molly B | 11/24/16 2:05:00 PM |
|---|---|---|

Font:12 pt

| Page 31: [136] Formatted | Smith, Molly B | 11/24/16 2:05:00 PM |
|---|---|---|

Font:12 pt

| Page 31: [137] Formatted | Smith, Molly B | 11/24/16 2:05:00 PM |

Font:12 pt

| Page 31: [138] Formatted | Smith, Molly B | 11/24/16 2:12:00 PM |

None, Space Before:  0 pt, Line spacing:  single, Don't keep with next, Don't keep lines together

| Page 31: [139] Formatted | Smith, Molly B | 11/24/16 2:05:00 PM |

Font:12 pt, Not Bold, Not Italic, Font color: Auto

| Page 31: [140] Formatted | Smith, Molly B | 11/24/16 2:05:00 PM |

Font:12 pt

| Page 31: [140] Formatted | Smith, Molly B | 11/24/16 2:05:00 PM |

Font:12 pt

| Page 31: [141] Formatted | Smith, Molly B | 11/24/16 2:05:00 PM |

Font:12 pt

| Page 31: [141] Formatted | Smith, Molly B | 11/24/16 2:05:00 PM |

Font:12 pt

| Page 31: [142] Formatted | Smith, Molly B | 11/24/16 2:05:00 PM |

Font:12 pt

| Page 31: [142] Formatted | Smith, Molly B | 11/24/16 2:05:00 PM |

Font:12 pt

| Page 31: [143] Formatted | Smith, Molly B | 11/24/16 2:05:00 PM |

Font:12 pt

| Page 31: [143] Formatted | Smith, Molly B | 11/24/16 2:05:00 PM |

Font:12 pt

| Page 31: [144] Formatted | Smith, Molly B | 11/24/16 2:05:00 PM |

Font:12 pt

| Page 31: [145] Formatted | Smith, Molly B | 11/24/16 2:12:00 PM |

None, Space Before:  0 pt, Line spacing:  single, Don't keep with next, Don't keep lines together

| Page 31: [146] Formatted | Smith, Molly B | 11/24/16 2:05:00 PM |

Font:12 pt, Not Bold, Not Italic, Font color: Auto

| Page 31: [147] Formatted | Smith, Molly B | 11/24/16 2:05:00 PM |

Font:12 pt

| Page 31: [147] Formatted | Smith, Molly B | 11/24/16 2:05:00 PM |

Font:12 pt

| Page 31: [148] Formatted | Smith, Molly B | 11/24/16 2:05:00 PM |

Font:12 pt

| Page 31: [148] Formatted | Smith, Molly B | 11/24/16 2:05:00 PM |

Font:12 pt

| Page 31: [149] Formatted | Smith, Molly B | 11/24/16 2:05:00 PM |
|---|---|---|

Font:12 pt

| Page 31: [149] Formatted | Smith, Molly B | 11/24/16 2:05:00 PM |
|---|---|---|

Font:12 pt

| Page 31: [150] Formatted | Smith, Molly B | 11/24/16 2:05:00 PM |
|---|---|---|

Font:12 pt

| Page 31: [150] Formatted | Smith, Molly B | 11/24/16 2:05:00 PM |
|---|---|---|

Font:12 pt

| Page 31: [151] Formatted | Smith, Molly B | 11/24/16 2:05:00 PM |
|---|---|---|

Font:12 pt

| Page 31: [152] Formatted | Smith, Molly B | 11/24/16 2:12:00 PM |
|---|---|---|

None, Space Before:  0 pt, Line spacing:  single, Don't keep with next, Don't keep lines together

| Page 31: [153] Formatted | Smith, Molly B | 11/24/16 2:05:00 PM |
|---|---|---|

Font:12 pt, Not Bold, Not Italic, Font color: Auto

| Page 31: [154] Formatted | Smith, Molly B | 11/24/16 2:05:00 PM |
|---|---|---|

Font:12 pt

| Page 31: [154] Formatted | Smith, Molly B | 11/24/16 2:05:00 PM |
|---|---|---|

Font:12 pt

| Page 31: [155] Formatted | Smith, Molly B | 11/24/16 2:05:00 PM |
|---|---|---|

Font:12 pt

| Page 31: [155] Formatted | Smith, Molly B | 11/24/16 2:05:00 PM |
|---|---|---|

Font:12 pt

| Page 31: [156] Formatted | Smith, Molly B | 11/24/16 2:05:00 PM |
|---|---|---|

Font:12 pt

| Page 31: [156] Formatted | Smith, Molly B | 11/24/16 2:05:00 PM |
|---|---|---|

Font:12 pt

| Page 31: [157] Formatted | Smith, Molly B | 11/24/16 2:05:00 PM |
|---|---|---|

Font:12 pt

| Page 31: [157] Formatted | Smith, Molly B | 11/24/16 2:05:00 PM |
|---|---|---|

Font:12 pt

| Page 31: [158] Formatted | Smith, Molly B | 11/24/16 2:05:00 PM |
|---|---|---|

Font:12 pt

| Page 31: [159] Formatted | Smith, Molly B | 11/24/16 2:12:00 PM |
|---|---|---|

Line spacing:  single

| Page 31: [160] Formatted | Smith, Molly B | 11/24/16 2:12:00 PM |
|---|---|---|

Space Before:  0 pt, Line spacing:  single

| Page 31: [161] Formatted | Smith, Molly B | 11/24/16 2:05:00 PM |
|---|---|---|

Font:12 pt

| Page 31: [162] Formatted | Smith, Molly B | 11/24/16 2:12:00 PM |
|---|---|---|

None, Space Before:  0 pt, Line spacing:  single, Don't keep with next, Don't keep lines together

| Page 31: [163] Formatted | Smith, Molly B | 11/24/16 2:05:00 PM |
|---|---|---|

Font:12 pt, Not Italic, Font color: Auto

| Page 31: [164] Formatted | Smith, Molly B | 11/24/16 2:05:00 PM |
|---|---|---|

Font:12 pt

| Page 31: [165] Formatted | Smith, Molly B | 11/24/16 2:12:00 PM |
|---|---|---|

Line spacing:  single

| Page 31: [166] Formatted | Smith, Molly B | 11/24/16 2:05:00 PM |
|---|---|---|

Font:12 pt

| Page 31: [167] Formatted | Smith, Molly B | 11/24/16 2:12:00 PM |
|---|---|---|

None, Space Before:  0 pt, Line spacing:  single, Don't keep with next, Don't keep lines together

| Page 31: [168] Formatted | Smith, Molly B | 11/24/16 2:05:00 PM |
|---|---|---|

Font:12 pt, Not Bold, Not Italic, Font color: Auto

| Page 31: [169] Formatted | Smith, Molly B | 11/24/16 2:05:00 PM |
|---|---|---|

Font:12 pt

| Page 31: [169] Formatted | Smith, Molly B | 11/24/16 2:05:00 PM |
|---|---|---|

Font:12 pt

| Page 31: [170] Formatted | Smith, Molly B | 11/24/16 2:05:00 PM |
|---|---|---|

Font:12 pt

| Page 31: [170] Formatted | Smith, Molly B | 11/24/16 2:05:00 PM |
|---|---|---|

Font:12 pt

| Page 31: [171] Formatted | Smith, Molly B | 11/24/16 2:05:00 PM |
|---|---|---|

Font:12 pt

| Page 31: [171] Formatted | Smith, Molly B | 11/24/16 2:05:00 PM |
|---|---|---|

Font:12 pt

| Page 31: [172] Formatted | Smith, Molly B | 11/24/16 2:05:00 PM |
|---|---|---|

Font:12 pt

| Page 31: [172] Formatted | Smith, Molly B | 11/24/16 2:05:00 PM |
|---|---|---|

Font:12 pt

| Page 31: [173] Formatted | Smith, Molly B | 12/1/16 2:33:00 PM |
|---|---|---|

Font:12 pt

| Page 31: [174] Formatted | Smith, Molly B | 11/24/16 2:12:00 PM |
|---|---|---|

Line spacing:  single

| Page 31: [175] Deleted | Cornell University | 11/15/16 2:45:00 PM |
|---|---|---|

| Dalanzadgad | 43 | 104 | 1997-2012 | B. Holben |
|---|---|---|---|---|

| Page 31: [176] Formatted | Smith, Molly B | 12/1/16 2:33:00 PM |
|---|---|---|

Font:12 pt

| Page 31: [177] Formatted | Smith, Molly B | 11/24/16 2:12:00 PM |
|---|---|---|

Line spacing: single

| Page 31: [178] Formatted | Smith, Molly B | 12/1/16 2:33:00 PM |
|---|---|---|

Font:12 pt

| Page 31: [179] Formatted | Smith, Molly B | 12/1/16 2:33:00 PM |
|---|---|---|

Font:12 pt

| Page 31: [180] Formatted | Smith, Molly B | 11/24/16 2:12:00 PM |
|---|---|---|

Line spacing: single

| Page 31: [181] Formatted | Smith, Molly B | 12/1/16 2:33:00 PM |
|---|---|---|

Font:12 pt

| Page 31: [182] Formatted | Smith, Molly B | 11/24/16 2:12:00 PM |
|---|---|---|

Line spacing: single

| Page 31: [183] Formatted | Smith, Molly B | 12/1/16 2:33:00 PM |
|---|---|---|

Font:12 pt

| Page 31: [184] Formatted | Smith, Molly B | 11/24/16 2:12:00 PM |
|---|---|---|

Line spacing: single

| Page 31: [185] Formatted | Smith, Molly B | 12/1/16 2:33:00 PM |
|---|---|---|

Font:12 pt

| Page 31: [186] Formatted | Smith, Molly B | 12/1/16 2:32:00 PM |
|---|---|---|

Line spacing: single

| Page 31: [187] Formatted | Smith, Molly B | 12/1/16 2:32:00 PM |
|---|---|---|

Space Before: 0 pt, Line spacing: single

| Page 31: [188] Formatted | Smith, Molly B | 12/1/16 2:33:00 PM |
|---|---|---|

Font:12 pt, Not Italic, Font color: Auto

| Page 31: [189] Formatted | Smith, Molly B | 11/24/16 2:05:00 PM |
|---|---|---|

Font:12 pt

| Page 31: [190] Formatted | Smith, Molly B | 11/24/16 2:12:00 PM |
|---|---|---|

Line spacing: single

| Page 31: [191] Formatted | Smith, Molly B | 12/1/16 2:33:00 PM |
|---|---|---|

Font:12 pt

| Page 31: [192] Formatted | Smith, Molly B | 11/24/16 2:12:00 PM |
|---|---|---|

Line spacing: single

| Page 31: [193] Deleted | Cornell University | 11/15/16 2:45:00 PM |
|---|---|---|

Rio Gallegos
Surface concentrations

| Page 31: [194] Formatted | Smith, Molly B | 11/24/16 2:12:00 PM |
|---|---|---|

Line spacing:  single

| Page 31: [195] Formatted | Smith, Molly B | 12/1/16 2:33:00 PM |
|---|---|---|

Font:12 pt

| Page 31: [196] Formatted | Smith, Molly B | 11/24/16 2:05:00 PM |
|---|---|---|

Font:12 pt

| Page 31: [197] Formatted | Smith, Molly B | 11/24/16 2:05:00 PM |
|---|---|---|

Font:12 pt

| Page 31: [198] Formatted | Smith, Molly B | 11/24/16 2:05:00 PM |
|---|---|---|

Font:12 pt

| Page 31: [199] Formatted | Smith, Molly B | 11/24/16 2:05:00 PM |
|---|---|---|

Font:12 pt

| Page 31: [200] Formatted | Smith, Molly B | 11/24/16 2:05:00 PM |
|---|---|---|

Font:12 pt

| Page 32: [201] Formatted | Smith, Molly B | 11/24/16 2:13:00 PM |
|---|---|---|

None, Space Before:  0 pt, Line spacing:  single, Don't keep with next, Don't keep lines together

| Page 32: [202] Formatted Table | Smith, Molly B | 11/24/16 2:13:00 PM |
|---|---|---|

Formatted Table

| Page 32: [203] Formatted | Smith, Molly B | 11/24/16 2:05:00 PM |
|---|---|---|

Font:12 pt, Not Bold, Not Italic, Font color: Auto

| Page 32: [204] Formatted | Smith, Molly B | 11/24/16 2:05:00 PM |
|---|---|---|

Font:12 pt

| Page 32: [204] Formatted | Smith, Molly B | 11/24/16 2:05:00 PM |
|---|---|---|

Font:12 pt

| Page 32: [205] Formatted | Smith, Molly B | 11/24/16 2:05:00 PM |
|---|---|---|

Font:12 pt

| Page 32: [205] Formatted | Smith, Molly B | 11/24/16 2:05:00 PM |
|---|---|---|

Font:12 pt

| Page 32: [206] Formatted | Smith, Molly B | 11/24/16 2:05:00 PM |
|---|---|---|

Font:12 pt

| Page 32: [206] Formatted | Smith, Molly B | 11/24/16 2:05:00 PM |
|---|---|---|

Font:12 pt

| Page 32: [207] Formatted | Smith, Molly B | 11/24/16 2:05:00 PM |
|---|---|---|

Font:12 pt

| Page 32: [207] Formatted | Smith, Molly B | 11/24/16 2:05:00 PM |
|---|---|---|

Font:12 pt

| Page 32: [208] Formatted | Smith, Molly B | 11/24/16 2:05:00 PM |
|---|---|---|

Font:12 pt

| Page 32: [209] Formatted | Smith, Molly B | 11/24/16 2:13:00 PM |
|---|---|---|

None, Space Before:  0 pt, Line spacing:  single, Don't keep with next, Don't keep lines together

| Page 32: [210] Formatted | Smith, Molly B | 11/24/16 2:05:00 PM |
|---|---|---|

Font:12 pt, Not Bold, Not Italic, Font color: Auto

| Page 32: [211] Formatted | Smith, Molly B | 11/24/16 2:05:00 PM |
|---|---|---|

Font:12 pt

| Page 32: [211] Formatted | Smith, Molly B | 11/24/16 2:05:00 PM |
|---|---|---|

Font:12 pt

| Page 32: [212] Formatted | Smith, Molly B | 11/24/16 2:05:00 PM |
|---|---|---|

Font:12 pt

| Page 32: [212] Formatted | Smith, Molly B | 11/24/16 2:05:00 PM |
|---|---|---|

Font:12 pt

| Page 32: [213] Formatted | Smith, Molly B | 11/24/16 2:05:00 PM |
|---|---|---|

Font:12 pt

| Page 32: [213] Formatted | Smith, Molly B | 11/24/16 2:05:00 PM |
|---|---|---|

Font:12 pt

| Page 32: [214] Formatted | Smith, Molly B | 11/24/16 2:05:00 PM |
|---|---|---|

Font:12 pt

| Page 32: [215] Formatted | Smith, Molly B | 11/24/16 2:13:00 PM |
|---|---|---|

None, Space Before:  0 pt, Line spacing:  single, Don't keep with next, Don't keep lines together

| Page 32: [216] Formatted | Smith, Molly B | 11/24/16 2:05:00 PM |
|---|---|---|

Font:12 pt, Not Bold, Not Italic, Font color: Auto

| Page 32: [217] Formatted | Smith, Molly B | 11/24/16 2:05:00 PM |
|---|---|---|

Font:12 pt

| Page 32: [217] Formatted | Smith, Molly B | 11/24/16 2:05:00 PM |
|---|---|---|

Font:12 pt

| Page 32: [218] Formatted | Smith, Molly B | 11/24/16 2:05:00 PM |
|---|---|---|

Font:12 pt

| Page 32: [218] Formatted | Smith, Molly B | 11/24/16 2:05:00 PM |
|---|---|---|

Font:12 pt

| Page 32: [219] Formatted | Smith, Molly B | 11/24/16 2:05:00 PM |
|---|---|---|

Font:12 pt

| Page 32: [219] Formatted | Smith, Molly B | 11/24/16 2:05:00 PM |
|---|---|---|

Font:12 pt

| Page 32: [220] Formatted | Smith, Molly B | 11/24/16 2:05:00 PM |
|---|---|---|

Font:12 pt

| Page 32: [221] Formatted | Smith, Molly B | 11/24/16 2:13:00 PM |
|---|---|---|

None, Space Before:  0 pt, Line spacing:  single, Don't keep with next, Don't keep lines together

| Page 32: [222] Formatted | Smith, Molly B | 11/24/16 2:05:00 PM |
|---|---|---|

Font:12 pt, Not Italic, Font color: Auto

| Page 32: [223] Formatted | Smith, Molly B | 11/24/16 2:05:00 PM |
|---|---|---|

Font:12 pt

| Page 32: [223] Formatted | Smith, Molly B | 11/24/16 2:05:00 PM |
|---|---|---|

Font:12 pt

| Page 32: [224] Formatted | Smith, Molly B | 11/24/16 2:05:00 PM |
|---|---|---|

Font:12 pt

| Page 32: [224] Formatted | Smith, Molly B | 11/24/16 2:05:00 PM |
|---|---|---|

Font:12 pt

| Page 32: [225] Formatted | Smith, Molly B | 11/24/16 2:05:00 PM |
|---|---|---|

Font:12 pt

| Page 32: [225] Formatted | Smith, Molly B | 11/24/16 2:05:00 PM |
|---|---|---|

Font:12 pt

| Page 32: [226] Formatted | Smith, Molly B | 11/24/16 2:05:00 PM |
|---|---|---|

Font:12 pt

| Page 32: [227] Formatted | Smith, Molly B | 11/24/16 2:13:00 PM |
|---|---|---|

None, Space Before:  0 pt, Line spacing:  single, Don't keep with next, Don't keep lines together

| Page 32: [228] Formatted | Smith, Molly B | 11/24/16 2:05:00 PM |
|---|---|---|

Font:12 pt, Not Bold, Not Italic, Font color: Auto

| Page 32: [229] Formatted | Smith, Molly B | 11/24/16 2:05:00 PM |
|---|---|---|

Font:12 pt

| Page 32: [229] Formatted | Smith, Molly B | 11/24/16 2:05:00 PM |
|---|---|---|

Font:12 pt

| Page 32: [230] Formatted | Smith, Molly B | 11/24/16 2:05:00 PM |
|---|---|---|

Font:12 pt

| Page 32: [230] Formatted | Smith, Molly B | 11/24/16 2:05:00 PM |
|---|---|---|

Font:12 pt

| Page 32: [231] Formatted | Smith, Molly B | 11/24/16 2:05:00 PM |
|---|---|---|

Font:12 pt

| Page 32: [231] Formatted | Smith, Molly B | 11/24/16 2:05:00 PM |
|---|---|---|

Font:12 pt

| Page 32: [232] Formatted | Smith, Molly B | 11/24/16 2:05:00 PM |
|---|---|---|

Font:12 pt

| Page 32: [233] Formatted | Smith, Molly B | 11/24/16 2:13:00 PM |
|---|---|---|

None, Space Before:  0 pt, Line spacing:  single, Don't keep with next, Don't keep lines together

| Page 32: [234] Formatted | Smith, Molly B | 11/24/16 2:05:00 PM |
|---|---|---|

Font:12 pt, Not Bold, Not Italic, Font color: Auto

| Page 32: [235] Formatted | Smith, Molly B | 11/24/16 2:05:00 PM |
|---|---|---|

Font:12 pt

| Page 32: [236] Formatted | Smith, Molly B | 11/24/16 2:13:00 PM |
|---|---|---|

Line spacing:  single

| Page 32: [237] Formatted | Smith, Molly B | 11/24/16 2:05:00 PM |
|---|---|---|

Font:12 pt, Not Italic, Font color: Auto

| Page 32: [238] Formatted | Smith, Molly B | 11/24/16 2:05:00 PM |
|---|---|---|

Font:12 pt

| Page 32: [238] Formatted | Smith, Molly B | 11/24/16 2:05:00 PM |
|---|---|---|

Font:12 pt

| Page 32: [239] Formatted | Smith, Molly B | 11/24/16 2:05:00 PM |
|---|---|---|

Font:12 pt

| Page 32: [239] Formatted | Smith, Molly B | 11/24/16 2:05:00 PM |
|---|---|---|

Font:12 pt

| Page 32: [240] Formatted | Smith, Molly B | 11/24/16 2:05:00 PM |
|---|---|---|

Font:12 pt

| Page 32: [241] Formatted | Smith, Molly B | 11/24/16 2:13:00 PM |
|---|---|---|

None, Space Before:  0 pt, Line spacing:  single, Don't keep with next, Don't keep lines together

| Page 32: [242] Formatted | Smith, Molly B | 11/24/16 2:05:00 PM |
|---|---|---|

Font:12 pt, Not Bold, Not Italic, Font color: Auto

| Page 32: [243] Formatted | Smith, Molly B | 11/24/16 2:05:00 PM |
|---|---|---|

Font:12 pt

| Page 32: [243] Formatted | Smith, Molly B | 11/24/16 2:05:00 PM |
|---|---|---|

Font:12 pt

| Page 32: [244] Formatted | Smith, Molly B | 11/24/16 2:05:00 PM |
|---|---|---|

Font:12 pt

| Page 32: [244] Formatted | Smith, Molly B | 11/24/16 2:05:00 PM |
|---|---|---|

Font:12 pt

| Page 32: [245] Formatted | Smith, Molly B | 11/24/16 2:05:00 PM |
|---|---|---|

Font:12 pt

| Page 32: [245] Formatted | Smith, Molly B | 11/24/16 2:05:00 PM |
|---|---|---|

Font:12 pt

| Page 32: [246] Formatted | Smith, Molly B | 11/24/16 2:05:00 PM |
|---|---|---|

Font:12 pt

| Page 32: [247] Formatted | Smith, Molly B | 11/24/16 2:13:00 PM |
|---|---|---|

None, Space Before:  0 pt, Line spacing:  single, Don't keep with next, Don't keep lines together

| Page 32: [248] Formatted | Smith, Molly B | 11/24/16 2:05:00 PM |
|---|---|---|

Font:12 pt, Not Bold, Not Italic, Font color: Auto

| Page 32: [249] Formatted | Smith, Molly B | 11/24/16 2:05:00 PM |
|---|---|---|

Font:12 pt

| Page 32: [249] Formatted | Smith, Molly B | 11/24/16 2:05:00 PM |
|---|---|---|

Font:12 pt

| Page 32: [250] Formatted | Smith, Molly B | 11/24/16 2:05:00 PM |
|---|---|---|

Font:12 pt

| Page 32: [250] Formatted | Smith, Molly B | 11/24/16 2:05:00 PM |
|---|---|---|

Font:12 pt

| Page 32: [251] Formatted | Smith, Molly B | 11/24/16 2:05:00 PM |
|---|---|---|

Font:12 pt

| Page 32: [251] Formatted | Smith, Molly B | 11/24/16 2:05:00 PM |
|---|---|---|

Font:12 pt

| Page 32: [252] Formatted | Smith, Molly B | 11/24/16 2:05:00 PM |
|---|---|---|

Font:12 pt

| Page 32: [253] Formatted | Smith, Molly B | 11/24/16 2:13:00 PM |
|---|---|---|

Line spacing:  single

| Page 32: [254] Formatted | Smith, Molly B | 11/24/16 2:05:00 PM |
|---|---|---|

Font:12 pt

| Page 32: [255] Formatted | Smith, Molly B | 11/24/16 2:05:00 PM |
|---|---|---|

Line spacing:  single

| Page 33: [256] Formatted | Smith, Molly B | 11/24/16 2:05:00 PM |
|---|---|---|

Font:12 pt

| Page 33: [256] Formatted | Smith, Molly B | 11/24/16 2:05:00 PM |
|---|---|---|

Font:12 pt

| Page 33: [256] Formatted | Smith, Molly B | 11/24/16 2:05:00 PM |
|---|---|---|

Font:12 pt

| Page 33: [256] Formatted | Smith, Molly B | 11/24/16 2:05:00 PM |
|---|---|---|

Font:12 pt

| Page 33: [256] Formatted | Smith, Molly B | 11/24/16 2:05:00 PM |
|---|---|---|

Font:12 pt

| Page 33: [256] Formatted | Smith, Molly B | 11/24/16 2:05:00 PM |
|---|---|---|

Font:12 pt

| Page 33: [256] Formatted | Smith, Molly B | 11/24/16 2:05:00 PM |
|---|---|---|

Font:12 pt

| Page 33: [256] Formatted | Smith, Molly B | 11/24/16 2:05:00 PM |
|---|---|---|

Font:12 pt

| Page 33: [257] Formatted | Smith, Molly B | 11/24/16 2:13:00 PM |
|---|---|---|

Line spacing:  single

| Page 33: [258] Formatted Table | Smith, Molly B | 11/24/16 2:13:00 PM |
|---|---|---|

Formatted Table

| Page 33: [259] Formatted | Smith, Molly B | 11/24/16 2:13:00 PM |
|---|---|---|

Space Before:  0 pt, Line spacing:  single

| Page 33: [260] Formatted | Smith, Molly B | 11/24/16 2:05:00 PM |
|---|---|---|

Font:12 pt, Not Bold, Not Italic

| Page 33: [261] Formatted | Smith, Molly B | 11/24/16 2:05:00 PM |
|---|---|---|

Font:12 pt

| Page 33: [261] Formatted | Smith, Molly B | 11/24/16 2:05:00 PM |
|---|---|---|

Font:12 pt

| Page 33: [262] Formatted | Smith, Molly B | 11/24/16 2:05:00 PM |
|---|---|---|

Font:12 pt

| Page 33: [262] Formatted | Smith, Molly B | 11/24/16 2:05:00 PM |
|---|---|---|

Font:12 pt

| Page 33: [263] Formatted | Smith, Molly B | 11/24/16 2:05:00 PM |
|---|---|---|

Font:12 pt

| Page 33: [263] Formatted | Smith, Molly B | 11/24/16 2:05:00 PM |
|---|---|---|

Font:12 pt

| Page 33: [264] Formatted | Smith, Molly B | 11/24/16 2:05:00 PM |
|---|---|---|

Font:12 pt

| Page 33: [264] Formatted | Smith, Molly B | 11/24/16 2:05:00 PM |
|---|---|---|

Font:12 pt

| Page 33: [265] Formatted | Smith, Molly B | 11/24/16 2:05:00 PM |
|---|---|---|

Font:12 pt

| Page 33: [265] Formatted | Smith, Molly B | 11/24/16 2:05:00 PM |
|---|---|---|

Font:12 pt

| Page 33: [266] Formatted | Smith, Molly B | 11/24/16 2:05:00 PM |
|---|---|---|

Font:12 pt

| Page 33: [266] Formatted | Smith, Molly B | 11/24/16 2:05:00 PM |
|---|---|---|

Font:12 pt

| Page 33: [267] Formatted | Smith, Molly B | 11/24/16 2:05:00 PM |
|---|---|---|

Font:12 pt

| Page 33: [268] Formatted | Smith, Molly B | 11/24/16 2:13:00 PM |
|---|---|---|

None, Space Before:  0 pt, Line spacing:  single, Don't keep with next, Don't keep lines together

| Page 33: [269] Formatted | Smith, Molly B | 11/24/16 2:05:00 PM |
|---|---|---|

Font:12 pt, Not Bold, Not Italic

| Page 33: [270] Formatted | Smith, Molly B | 11/24/16 2:05:00 PM |
|---|---|---|

Font:12 pt

| Page 33: [271] Formatted | Smith, Molly B | 11/24/16 2:13:00 PM |
|---|---|---|

Space Before:  0 pt, Line spacing:  single

| Page 33: [272] Formatted | Smith, Molly B | 11/24/16 2:05:00 PM |
|---|---|---|

Font:12 pt, Not Bold, Not Italic, Font color: Auto

| Page 33: [273] Formatted | Smith, Molly B | 11/24/16 2:05:00 PM |
|---|---|---|

Font:12 pt

| Page 33: [273] Formatted | Smith, Molly B | 11/24/16 2:05:00 PM |
|---|---|---|

Font:12 pt

| Page 33: [274] Formatted | Smith, Molly B | 11/24/16 2:05:00 PM |
|---|---|---|

Font:12 pt

| Page 33: [274] Formatted | Smith, Molly B | 11/24/16 2:05:00 PM |
|---|---|---|

Font:12 pt

| Page 33: [275] Formatted | Smith, Molly B | 11/24/16 2:05:00 PM |
|---|---|---|

Font:12 pt

| Page 33: [276] Formatted | Smith, Molly B | 11/24/16 2:13:00 PM |
|---|---|---|

None, Space Before:  0 pt, Line spacing:  single, Don't keep with next, Don't keep lines together

| Page 33: [277] Formatted | Smith, Molly B | 11/24/16 2:05:00 PM |
|---|---|---|

Font:12 pt, Not Bold, Not Italic, Font color: Auto

| Page 33: [278] Formatted | Smith, Molly B | 11/24/16 2:05:00 PM |
|---|---|---|

Font:12 pt

| Page 33: [279] Formatted | Smith, Molly B | 11/24/16 2:13:00 PM |
|---|---|---|

Space Before:  0 pt, Line spacing:  single

| Page 33: [280] Formatted | Smith, Molly B | 11/24/16 2:05:00 PM |
|---|---|---|

Font:12 pt, Not Italic, Font color: Auto

| Page 33: [281] Formatted | Smith, Molly B | 11/24/16 2:05:00 PM |
|---|---|---|

Font:12 pt

| Page 33: [281] Formatted | Smith, Molly B | 11/24/16 2:05:00 PM |
|---|---|---|

Font:12 pt

| Page 33: [282] Formatted | Smith, Molly B | 11/24/16 2:05:00 PM |
|---|---|---|

Font:12 pt

| Page 33: [282] Formatted | Smith, Molly B | 11/24/16 2:05:00 PM |
|---|---|---|

Font:12 pt

| Page 33: [283] Formatted | Smith, Molly B | 11/24/16 2:05:00 PM |

Font:12 pt

| Page 33: [283] Formatted | Smith, Molly B | 11/24/16 2:05:00 PM |

Font:12 pt

| Page 33: [284] Formatted | Smith, Molly B | 11/24/16 2:05:00 PM |

Font:12 pt

| Page 33: [285] Formatted | Smith, Molly B | 11/24/16 2:13:00 PM |

None, Space Before:  0 pt, Line spacing:  single, Don't keep with next, Don't keep lines together

| Page 33: [286] Formatted | Smith, Molly B | 11/24/16 2:05:00 PM |

Font:12 pt, Not Italic

| Page 33: [287] Formatted | Smith, Molly B | 11/24/16 2:05:00 PM |

Font:12 pt

| Page 33: [288] Formatted | Smith, Molly B | 11/24/16 2:13:00 PM |

None, Space Before:  0 pt, Line spacing:  single, Don't keep with next, Don't keep lines together

| Page 33: [289] Formatted | Smith, Molly B | 11/24/16 2:05:00 PM |

Font:12 pt, Not Bold, Not Italic, Font color: Auto

| Page 33: [290] Formatted | Smith, Molly B | 11/24/16 2:05:00 PM |

Font:12 pt

| Page 33: [290] Formatted | Smith, Molly B | 11/24/16 2:05:00 PM |

Font:12 pt

| Page 33: [291] Formatted | Smith, Molly B | 11/24/16 2:05:00 PM |

Font:12 pt

| Page 33: [291] Formatted | Smith, Molly B | 11/24/16 2:05:00 PM |

Font:12 pt

| Page 33: [292] Formatted | Smith, Molly B | 11/24/16 2:05:00 PM |

Font:12 pt

| Page 33: [293] Formatted | Smith, Molly B | 11/24/16 2:13:00 PM |

Space Before:  0 pt, Line spacing:  single

| Page 33: [294] Formatted | Smith, Molly B | 11/24/16 2:05:00 PM |

Font:12 pt, Not Italic, Font color: Auto

| Page 33: [295] Formatted | Smith, Molly B | 11/24/16 2:05:00 PM |

Font:12 pt

| Page 33: [295] Formatted | Smith, Molly B | 11/24/16 2:05:00 PM |

Font:12 pt

| Page 33: [296] Formatted | Smith, Molly B | 11/24/16 2:05:00 PM |

Font:12 pt

| Page 33: [296] Formatted | Smith, Molly B | 11/24/16 2:05:00 PM |
|---|---|---|

Font:12 pt

| Page 33: [297] Formatted | Smith, Molly B | 11/24/16 2:05:00 PM |
|---|---|---|

Font:12 pt

| Page 33: [298] Formatted | Smith, Molly B | 11/24/16 2:13:00 PM |
|---|---|---|

None, Space Before:  0 pt, Line spacing:  single, Don't keep with next, Don't keep lines together

| Page 33: [299] Formatted | Smith, Molly B | 11/24/16 2:05:00 PM |
|---|---|---|

Font:12 pt, Not Bold, Not Italic, Font color: Auto

| Page 33: [300] Formatted | Smith, Molly B | 11/24/16 2:05:00 PM |
|---|---|---|

Font:12 pt

| Page 33: [301] Formatted | Smith, Molly B | 11/24/16 2:13:00 PM |
|---|---|---|

Space Before:  0 pt, Line spacing:  single

| Page 33: [302] Formatted | Smith, Molly B | 11/24/16 2:05:00 PM |
|---|---|---|

Font:12 pt, Not Italic, Font color: Auto

| Page 33: [303] Formatted | Smith, Molly B | 11/24/16 2:05:00 PM |
|---|---|---|

Font:12 pt

| Page 33: [303] Formatted | Smith, Molly B | 11/24/16 2:05:00 PM |
|---|---|---|

Font:12 pt

| Page 33: [304] Formatted | Smith, Molly B | 11/24/16 2:05:00 PM |
|---|---|---|

Font:12 pt

| Page 33: [305] Formatted | Smith, Molly B | 11/24/16 2:13:00 PM |
|---|---|---|

None, Space Before:  0 pt, Line spacing:  single, Don't keep with next, Don't keep lines together

| Page 33: [306] Formatted | Smith, Molly B | 11/24/16 2:05:00 PM |
|---|---|---|

Font:12 pt, Not Bold, Not Italic, Font color: Auto

| Page 33: [307] Formatted | Smith, Molly B | 11/24/16 2:05:00 PM |
|---|---|---|

Font:12 pt

| Page 33: [307] Formatted | Smith, Molly B | 11/24/16 2:05:00 PM |
|---|---|---|

Font:12 pt

| Page 33: [308] Formatted | Smith, Molly B | 11/24/16 2:05:00 PM |
|---|---|---|

Font:12 pt

| Page 33: [309] Formatted | Smith, Molly B | 11/24/16 2:13:00 PM |
|---|---|---|

Space Before:  0 pt, Line spacing:  single

| Page 33: [310] Formatted | Smith, Molly B | 11/24/16 2:05:00 PM |
|---|---|---|

Font:12 pt, Not Italic

| Page 33: [311] Formatted | Smith, Molly B | 11/24/16 2:05:00 PM |
|---|---|---|

Font:12 pt

| Page 33: [312] Formatted | Smith, Molly B | 11/24/16 2:13:00 PM |
|---|---|---|

None, Space Before:  0 pt, Line spacing:  single, Don't keep with next, Don't keep lines together

| Page 33: [313] Formatted | Smith, Molly B | 11/24/16 2:05:00 PM |
|---|---|---|

Font:12 pt, Not Bold, Not Italic, Font color: Auto

| Page 33: [314] Formatted | Smith, Molly B | 11/24/16 2:05:00 PM |
|---|---|---|

Font:12 pt

| Page 33: [315] Formatted | Smith, Molly B | 11/24/16 2:13:00 PM |
|---|---|---|

Space Before:  0 pt, Line spacing:  single

| Page 33: [316] Formatted | Smith, Molly B | 11/24/16 2:05:00 PM |
|---|---|---|

Font:12 pt, Not Italic, Font color: Auto

| Page 33: [317] Formatted | Smith, Molly B | 11/24/16 2:05:00 PM |
|---|---|---|

Font:12 pt

| Page 33: [317] Formatted | Smith, Molly B | 11/24/16 2:05:00 PM |
|---|---|---|

Font:12 pt

| Page 33: [318] Formatted | Smith, Molly B | 11/24/16 2:05:00 PM |
|---|---|---|

Font:12 pt

| Page 33: [318] Formatted | Smith, Molly B | 11/24/16 2:05:00 PM |
|---|---|---|

Font:12 pt

| Page 33: [319] Formatted | Smith, Molly B | 11/24/16 2:05:00 PM |
|---|---|---|

Font:12 pt

| Page 33: [319] Formatted | Smith, Molly B | 11/24/16 2:05:00 PM |
|---|---|---|

Font:12 pt

| Page 33: [320] Formatted | Smith, Molly B | 11/24/16 2:05:00 PM |
|---|---|---|

Font:12 pt

| Page 33: [320] Formatted | Smith, Molly B | 11/24/16 2:05:00 PM |
|---|---|---|

Font:12 pt

| Page 33: [321] Formatted | Smith, Molly B | 11/24/16 2:05:00 PM |
|---|---|---|

Font:12 pt

| Page 33: [322] Formatted | Smith, Molly B | 11/24/16 2:13:00 PM |
|---|---|---|

None, Space Before:  0 pt, Line spacing:  single, Don't keep with next, Don't keep lines together

| Page 33: [323] Formatted | Smith, Molly B | 11/24/16 2:05:00 PM |
|---|---|---|

Font:12 pt, Not Bold, Not Italic, Font color: Auto

| Page 33: [324] Formatted | Smith, Molly B | 11/24/16 2:05:00 PM |
|---|---|---|

Font:12 pt

| Page 33: [325] Formatted | Smith, Molly B | 11/24/16 2:13:00 PM |
|---|---|---|

Space Before:  0 pt, Line spacing:  single

| Page 33: [326] Formatted | Smith, Molly B | 11/24/16 2:05:00 PM |
|---|---|---|

Font:12 pt, Not Italic, Font color: Auto

| Page 33: [327] Formatted | Smith, Molly B | 11/24/16 2:05:00 PM |
|---|---|---|

Font:12 pt

| Page 33: [327] Formatted | Smith, Molly B | 11/24/16 2:05:00 PM |
|---|---|---|

Font:12 pt

| Page 33: [328] Formatted | Smith, Molly B | 11/24/16 2:05:00 PM |
|---|---|---|

Font:12 pt

| Page 33: [328] Formatted | Smith, Molly B | 11/24/16 2:05:00 PM |
|---|---|---|

Font:12 pt

| Page 33: [329] Formatted | Smith, Molly B | 11/24/16 2:05:00 PM |
|---|---|---|

Font:12 pt

| Page 33: [329] Formatted | Smith, Molly B | 11/24/16 2:05:00 PM |
|---|---|---|

Font:12 pt

| Page 33: [330] Formatted | Smith, Molly B | 11/24/16 2:05:00 PM |
|---|---|---|

Font:12 pt

| Page 33: [331] Formatted | Smith, Molly B | 11/24/16 2:13:00 PM |
|---|---|---|

None, Space Before:  0 pt, Line spacing:  single, Don't keep with next, Don't keep lines together

| Page 33: [332] Formatted | Smith, Molly B | 11/24/16 2:05:00 PM |
|---|---|---|

Font:12 pt, Not Italic

| Page 33: [333] Formatted | Smith, Molly B | 11/24/16 2:05:00 PM |
|---|---|---|

Font:12 pt

| Page 33: [334] Formatted | Smith, Molly B | 11/24/16 2:13:00 PM |
|---|---|---|

None, Space Before:  0 pt, Line spacing:  single, Don't keep with next, Don't keep lines together

| Page 33: [335] Formatted | Smith, Molly B | 11/24/16 2:05:00 PM |
|---|---|---|

Font:12 pt, Not Bold, Not Italic, Font color: Auto

| Page 33: [336] Formatted | Smith, Molly B | 11/24/16 2:05:00 PM |
|---|---|---|

Font:12 pt

| Page 33: [337] Formatted | Smith, Molly B | 11/24/16 2:13:00 PM |
|---|---|---|

Space Before:  0 pt, Line spacing:  single

| Page 33: [338] Formatted | Smith, Molly B | 11/24/16 2:05:00 PM |
|---|---|---|

Font:12 pt, Not Italic, Font color: Auto

| Page 33: [339] Formatted | Smith, Molly B | 11/24/16 2:05:00 PM |
|---|---|---|

Font:12 pt

| Page 33: [339] Formatted | Smith, Molly B | 11/24/16 2:05:00 PM |
|---|---|---|

Font:12 pt

| Page 33: [340] Formatted | Smith, Molly B | 11/24/16 2:05:00 PM |
|---|---|---|

Font:12 pt

| Page 33: [340] Formatted | Smith, Molly B | 11/24/16 2:05:00 PM |
|---|---|---|

Font:12 pt

| Page 33: [341] Formatted | Smith, Molly B | 11/24/16 2:05:00 PM |

Font:12 pt

| Page 33: [341] Formatted | Smith, Molly B | 11/24/16 2:05:00 PM |

Font:12 pt

| Page 33: [342] Formatted | Smith, Molly B | 11/24/16 2:05:00 PM |

Font:12 pt

| Page 33: [343] Formatted | Smith, Molly B | 11/24/16 2:13:00 PM |

None, Space Before:  0 pt, Line spacing:  single, Don't keep with next, Don't keep lines together

| Page 33: [344] Formatted | Smith, Molly B | 11/24/16 2:05:00 PM |

Font:12 pt, Not Italic

| Page 33: [345] Formatted | Smith, Molly B | 11/24/16 2:05:00 PM |

Font:12 pt

| Page 33: [346] Formatted | Smith, Molly B | 11/24/16 2:13:00 PM |

None, Space Before:  0 pt, Line spacing:  single, Don't keep with next, Don't keep lines together

| Page 33: [347] Formatted | Smith, Molly B | 11/24/16 2:05:00 PM |

Font:12 pt, Not Bold, Not Italic, Font color: Auto

| Page 33: [348] Formatted | Smith, Molly B | 11/24/16 2:05:00 PM |

Font:12 pt

| Page 33: [349] Formatted | Smith, Molly B | 11/24/16 2:13:00 PM |

Space Before:  0 pt, Line spacing:  single

| Page 33: [350] Formatted | Smith, Molly B | 11/24/16 2:05:00 PM |

Font:12 pt, Not Bold, Not Italic, Font color: Auto

| Page 33: [351] Formatted | Smith, Molly B | 11/24/16 2:05:00 PM |

Font:12 pt

| Page 33: [351] Formatted | Smith, Molly B | 11/24/16 2:05:00 PM |

Font:12 pt

| Page 33: [352] Formatted | Smith, Molly B | 11/24/16 2:05:00 PM |

Font:12 pt

| Page 33: [352] Formatted | Smith, Molly B | 11/24/16 2:05:00 PM |

Font:12 pt

| Page 33: [353] Formatted | Smith, Molly B | 11/24/16 2:05:00 PM |

Font:12 pt

| Page 33: [354] Formatted | Smith, Molly B | 11/24/16 2:13:00 PM |

Line spacing:  single

| Page 33: [355] Formatted | Smith, Molly B | 11/24/16 2:05:00 PM |

Font:12 pt

| Page 33: [356] Formatted | Smith, Molly B | 11/24/16 2:13:00 PM |
|---|---|---|

None, Space Before:  0 pt, Line spacing:  single, Don't keep with next, Don't keep lines together

| Page 33: [357] Formatted | Smith, Molly B | 11/24/16 2:05:00 PM |
|---|---|---|

Font:12 pt, Not Bold, Not Italic, Font color: Auto

| Page 33: [358] Formatted | Smith, Molly B | 11/24/16 2:05:00 PM |
|---|---|---|

Font:12 pt

| Page 33: [359] Formatted | Smith, Molly B | 11/24/16 2:13:00 PM |
|---|---|---|

Space Before:  0 pt, Line spacing:  single

| Page 33: [360] Formatted | Smith, Molly B | 11/24/16 2:05:00 PM |
|---|---|---|

Font:12 pt, Not Italic, Font color: Auto

| Page 33: [361] Formatted | Smith, Molly B | 11/24/16 2:05:00 PM |
|---|---|---|

Font:12 pt

| Page 33: [362] Formatted | Smith, Molly B | 11/24/16 2:13:00 PM |
|---|---|---|

Line spacing:  single

| Page 33: [363] Formatted | Smith, Molly B | 11/24/16 2:13:00 PM |
|---|---|---|

None, Space Before:  0 pt, Line spacing:  single, Don't keep with next, Don't keep lines together

| Page 33: [364] Formatted | Smith, Molly B | 11/24/16 2:05:00 PM |
|---|---|---|

Font:12 pt, Not Italic, Font color: Auto

| Page 33: [365] Formatted | Smith, Molly B | 11/24/16 2:05:00 PM |
|---|---|---|

Font:12 pt

| Page 33: [365] Formatted | Smith, Molly B | 11/24/16 2:05:00 PM |
|---|---|---|

Font:12 pt

| Page 33: [366] Formatted | Smith, Molly B | 11/24/16 2:05:00 PM |
|---|---|---|

Font:12 pt

| Page 33: [367] Formatted | Smith, Molly B | 11/24/16 2:13:00 PM |
|---|---|---|

Line spacing:  single

| Page 33: [368] Formatted | Smith, Molly B | 11/24/16 2:05:00 PM |
|---|---|---|

Font:12 pt

| Page 34: [369] Formatted | Smith, Molly B | 11/24/16 2:14:00 PM |
|---|---|---|

Line spacing:  single

| Page 34: [370] Formatted Table | Smith, Molly B | 11/24/16 2:14:00 PM |
|---|---|---|

Formatted Table

| Page 34: [371] Formatted | Smith, Molly B | 11/24/16 2:05:00 PM |
|---|---|---|

Font:12 pt, Not Italic

| Page 34: [372] Formatted | Smith, Molly B | 11/24/16 2:05:00 PM |
|---|---|---|

Font:12 pt

| Page 34: [372] Formatted | Smith, Molly B | 11/24/16 2:05:00 PM |
|---|---|---|

Font:12 pt

| Page 34: [373] Formatted | Smith, Molly B | 11/24/16 2:05:00 PM |

Font:12 pt

| Page 34: [373] Formatted | Smith, Molly B | 11/24/16 2:05:00 PM |

Font:12 pt

| Page 34: [374] Formatted | Smith, Molly B | 11/24/16 2:05:00 PM |

Font:12 pt

| Page 34: [375] Formatted | Smith, Molly B | 11/24/16 2:14:00 PM |

Line spacing:  single

| Page 34: [376] Formatted | Smith, Molly B | 11/24/16 2:05:00 PM |

Font:12 pt, Not Italic, Font color: Auto

| Page 34: [377] Formatted | Smith, Molly B | 11/24/16 2:05:00 PM |

Font:12 pt

| Page 34: [377] Formatted | Smith, Molly B | 11/24/16 2:05:00 PM |

Font:12 pt

| Page 34: [378] Formatted | Smith, Molly B | 11/24/16 2:05:00 PM |

Font:12 pt

| Page 34: [378] Formatted | Smith, Molly B | 11/24/16 2:05:00 PM |

Font:12 pt

| Page 34: [379] Formatted | Smith, Molly B | 11/24/16 2:05:00 PM |

Font:12 pt

| Page 34: [379] Formatted | Smith, Molly B | 11/24/16 2:05:00 PM |

Font:12 pt

| Page 34: [380] Formatted | Smith, Molly B | 11/24/16 2:05:00 PM |

Font:12 pt

| Page 34: [380] Formatted | Smith, Molly B | 11/24/16 2:05:00 PM |

Font:12 pt

| Page 34: [381] Formatted | Smith, Molly B | 11/24/16 2:05:00 PM |

Font:12 pt

| Page 34: [381] Formatted | Smith, Molly B | 11/24/16 2:05:00 PM |

Font:12 pt

| Page 34: [382] Formatted | Smith, Molly B | 11/24/16 2:05:00 PM |

Font:12 pt

| Page 34: [382] Formatted | Smith, Molly B | 11/24/16 2:05:00 PM |

Font:12 pt

| Page 34: [383] Formatted | Smith, Molly B | 11/24/16 2:05:00 PM |

Font:12 pt

| Page 34: [384] Formatted | Smith, Molly B | 11/24/16 2:14:00 PM |
|---|---|---|

Line spacing:  single

| Page 34: [385] Formatted | Smith, Molly B | 11/24/16 2:05:00 PM |
|---|---|---|

Font:12 pt, Not Italic, Font color: Auto

| Page 34: [386] Formatted | Smith, Molly B | 11/24/16 2:05:00 PM |
|---|---|---|

Font:12 pt

| Page 34: [386] Formatted | Smith, Molly B | 11/24/16 2:05:00 PM |
|---|---|---|

Font:12 pt

| Page 34: [387] Formatted | Smith, Molly B | 11/24/16 2:05:00 PM |
|---|---|---|

Font:12 pt

| Page 34: [387] Formatted | Smith, Molly B | 11/24/16 2:05:00 PM |
|---|---|---|

Font:12 pt

| Page 34: [388] Formatted | Smith, Molly B | 11/24/16 2:05:00 PM |
|---|---|---|

Font:12 pt

| Page 34: [388] Formatted | Smith, Molly B | 11/24/16 2:05:00 PM |
|---|---|---|

Font:12 pt

| Page 34: [389] Formatted | Smith, Molly B | 11/24/16 2:05:00 PM |
|---|---|---|

Font:12 pt

| Page 34: [389] Formatted | Smith, Molly B | 11/24/16 2:05:00 PM |
|---|---|---|

Font:12 pt

| Page 34: [390] Formatted | Smith, Molly B | 11/24/16 2:05:00 PM |
|---|---|---|

Font:12 pt

| Page 34: [390] Formatted | Smith, Molly B | 11/24/16 2:05:00 PM |
|---|---|---|

Font:12 pt

| Page 34: [391] Formatted | Smith, Molly B | 11/24/16 2:05:00 PM |
|---|---|---|

Font:12 pt

| Page 34: [391] Formatted | Smith, Molly B | 11/24/16 2:05:00 PM |
|---|---|---|

Font:12 pt

| Page 34: [392] Formatted | Smith, Molly B | 11/24/16 2:05:00 PM |
|---|---|---|

Font:12 pt

| Page 34: [393] Formatted | Smith, Molly B | 11/24/16 2:14:00 PM |
|---|---|---|

Line spacing:  single

| Page 34: [394] Formatted | Smith, Molly B | 11/24/16 2:05:00 PM |
|---|---|---|

Font:12 pt, Not Italic, Font color: Auto

| Page 34: [395] Formatted | Smith, Molly B | 11/24/16 2:05:00 PM |
|---|---|---|

Font:12 pt

| Page 34: [395] Formatted | Smith, Molly B | 11/24/16 2:05:00 PM |
|---|---|---|

Font:12 pt

| Page 34: [396] Formatted | Smith, Molly B | 11/24/16 2:05:00 PM |

Font:12 pt

| Page 34: [396] Formatted | Smith, Molly B | 11/24/16 2:05:00 PM |

Font:12 pt

| Page 34: [397] Formatted | Smith, Molly B | 11/24/16 2:05:00 PM |

Font:12 pt

| Page 34: [397] Formatted | Smith, Molly B | 11/24/16 2:05:00 PM |

Font:12 pt

| Page 34: [398] Formatted | Smith, Molly B | 11/24/16 2:05:00 PM |

Font:12 pt

| Page 34: [398] Formatted | Smith, Molly B | 11/24/16 2:05:00 PM |

Font:12 pt

| Page 34: [399] Formatted | Smith, Molly B | 11/24/16 2:05:00 PM |

Font:12 pt

| Page 34: [399] Formatted | Smith, Molly B | 11/24/16 2:05:00 PM |

Font:12 pt

| Page 34: [400] Formatted | Smith, Molly B | 11/24/16 2:05:00 PM |

Font:12 pt

| Page 34: [400] Formatted | Smith, Molly B | 11/24/16 2:05:00 PM |

Font:12 pt

| Page 34: [401] Formatted | Smith, Molly B | 11/24/16 2:05:00 PM |

Font:12 pt

| Page 34: [402] Formatted | Smith, Molly B | 11/24/16 2:14:00 PM |

Line spacing:  single

| Page 34: [403] Formatted | Smith, Molly B | 11/24/16 2:05:00 PM |

Font:12 pt, Not Italic, Font color: Auto

| Page 34: [404] Formatted | Smith, Molly B | 11/24/16 2:05:00 PM |

Font:12 pt

| Page 34: [404] Formatted | Smith, Molly B | 11/24/16 2:05:00 PM |

Font:12 pt

| Page 34: [405] Formatted | Smith, Molly B | 11/24/16 2:05:00 PM |

Font:12 pt

| Page 34: [405] Formatted | Smith, Molly B | 11/24/16 2:05:00 PM |

Font:12 pt

| Page 34: [406] Formatted | Smith, Molly B | 11/24/16 2:05:00 PM |

Font:12 pt

| Page 34: [406] Formatted | Smith, Molly B | 11/24/16 2:05:00 PM |
| --- | --- | --- |

Font:12 pt

| Page 34: [407] Formatted | Smith, Molly B | 11/24/16 2:05:00 PM |
| --- | --- | --- |

Font:12 pt

| Page 34: [407] Formatted | Smith, Molly B | 11/24/16 2:05:00 PM |
| --- | --- | --- |

Font:12 pt

| Page 34: [408] Formatted | Smith, Molly B | 11/24/16 2:05:00 PM |
| --- | --- | --- |

Font:12 pt

| Page 34: [408] Formatted | Smith, Molly B | 11/24/16 2:05:00 PM |
| --- | --- | --- |

Font:12 pt

| Page 34: [409] Formatted | Smith, Molly B | 11/24/16 2:05:00 PM |
| --- | --- | --- |

Font:12 pt

| Page 34: [409] Formatted | Smith, Molly B | 11/24/16 2:05:00 PM |
| --- | --- | --- |

Font:12 pt

| Page 34: [410] Formatted | Smith, Molly B | 11/24/16 2:05:00 PM |
| --- | --- | --- |

Font:12 pt

| Page 34: [411] Formatted | Smith, Molly B | 11/24/16 2:14:00 PM |
| --- | --- | --- |

Line spacing:  single

| Page 34: [412] Formatted | Smith, Molly B | 11/24/16 2:05:00 PM |
| --- | --- | --- |

Font:12 pt, Not Italic, Font color: Auto

| Page 34: [413] Formatted | Smith, Molly B | 11/24/16 2:05:00 PM |
| --- | --- | --- |

Font:12 pt

| Page 34: [413] Formatted | Smith, Molly B | 11/24/16 2:05:00 PM |
| --- | --- | --- |

Font:12 pt

| Page 34: [414] Formatted | Smith, Molly B | 11/24/16 2:05:00 PM |
| --- | --- | --- |

Font:12 pt

| Page 34: [414] Formatted | Smith, Molly B | 11/24/16 2:05:00 PM |
| --- | --- | --- |

Font:12 pt

| Page 34: [415] Formatted | Smith, Molly B | 11/24/16 2:05:00 PM |
| --- | --- | --- |

Font:12 pt

| Page 34: [415] Formatted | Smith, Molly B | 11/24/16 2:05:00 PM |
| --- | --- | --- |

Font:12 pt

| Page 34: [416] Formatted | Smith, Molly B | 11/24/16 2:05:00 PM |
| --- | --- | --- |

Font:12 pt

| Page 34: [416] Formatted | Smith, Molly B | 11/24/16 2:05:00 PM |
| --- | --- | --- |

Font:12 pt

| Page 34: [417] Formatted | Smith, Molly B | 11/24/16 2:05:00 PM |
| --- | --- | --- |

Font:12 pt

| Page 34: [417] Formatted | Smith, Molly B | 11/24/16 2:05:00 PM |
|---|---|---|

Font:12 pt

| Page 34: [418] Formatted | Smith, Molly B | 11/24/16 2:05:00 PM |
|---|---|---|

Font:12 pt

| Page 34: [418] Formatted | Smith, Molly B | 11/24/16 2:05:00 PM |
|---|---|---|

Font:12 pt

| Page 34: [419] Formatted | Smith, Molly B | 11/24/16 2:05:00 PM |
|---|---|---|

Font:12 pt

| Page 34: [420] Formatted | Smith, Molly B | 11/24/16 2:14:00 PM |
|---|---|---|

Line spacing:  single

| Page 34: [421] Formatted | Smith, Molly B | 11/24/16 2:05:00 PM |
|---|---|---|

Font:12 pt, Not Italic, Font color: Auto

| Page 34: [422] Formatted | Smith, Molly B | 11/24/16 2:05:00 PM |
|---|---|---|

Font:12 pt

| Page 34: [422] Formatted | Smith, Molly B | 11/24/16 2:05:00 PM |
|---|---|---|

Font:12 pt

| Page 34: [423] Formatted | Smith, Molly B | 11/24/16 2:05:00 PM |
|---|---|---|

Font:12 pt

| Page 34: [423] Formatted | Smith, Molly B | 11/24/16 2:05:00 PM |
|---|---|---|

Font:12 pt

| Page 34: [424] Formatted | Smith, Molly B | 11/24/16 2:05:00 PM |
|---|---|---|

Font:12 pt

| Page 34: [424] Formatted | Smith, Molly B | 11/24/16 2:05:00 PM |
|---|---|---|

Font:12 pt

| Page 34: [425] Formatted | Smith, Molly B | 11/24/16 2:05:00 PM |
|---|---|---|

Font:12 pt

| Page 34: [425] Formatted | Smith, Molly B | 11/24/16 2:05:00 PM |
|---|---|---|

Font:12 pt

| Page 34: [426] Formatted | Smith, Molly B | 11/24/16 2:05:00 PM |
|---|---|---|

Font:12 pt

| Page 34: [426] Formatted | Smith, Molly B | 11/24/16 2:05:00 PM |
|---|---|---|

Font:12 pt

| Page 34: [427] Formatted | Smith, Molly B | 11/24/16 2:05:00 PM |
|---|---|---|

Font:12 pt

| Page 34: [427] Formatted | Smith, Molly B | 11/24/16 2:05:00 PM |
|---|---|---|

Font:12 pt

| Page 34: [428] Formatted | Smith, Molly B | 11/24/16 2:05:00 PM |
|---|---|---|

Font:12 pt

| Page 34: [429] Formatted | Smith, Molly B | 11/24/16 2:14:00 PM |
|---|---|---|

Line spacing:  single

| Page 34: [430] Formatted | Smith, Molly B | 11/24/16 2:05:00 PM |
|---|---|---|

Font:12 pt, Not Italic, Font color: Auto

| Page 34: [431] Formatted | Smith, Molly B | 11/24/16 2:05:00 PM |
|---|---|---|

Font:12 pt

| Page 34: [431] Formatted | Smith, Molly B | 11/24/16 2:05:00 PM |
|---|---|---|

Font:12 pt

| Page 34: [432] Formatted | Smith, Molly B | 11/24/16 2:05:00 PM |
|---|---|---|

Font:12 pt

| Page 34: [432] Formatted | Smith, Molly B | 11/24/16 2:05:00 PM |
|---|---|---|

Font:12 pt

| Page 34: [433] Formatted | Smith, Molly B | 11/24/16 2:05:00 PM |
|---|---|---|

Font:12 pt

| Page 34: [433] Formatted | Smith, Molly B | 11/24/16 2:05:00 PM |
|---|---|---|

Font:12 pt

| Page 34: [434] Formatted | Smith, Molly B | 11/24/16 2:05:00 PM |
|---|---|---|

Font:12 pt

| Page 34: [434] Formatted | Smith, Molly B | 11/24/16 2:05:00 PM |
|---|---|---|

Font:12 pt

| Page 34: [435] Formatted | Smith, Molly B | 11/24/16 2:05:00 PM |
|---|---|---|

Font:12 pt

| Page 34: [435] Formatted | Smith, Molly B | 11/24/16 2:05:00 PM |
|---|---|---|

Font:12 pt

| Page 34: [436] Formatted | Smith, Molly B | 11/24/16 2:05:00 PM |
|---|---|---|

Font:12 pt

| Page 34: [436] Formatted | Smith, Molly B | 11/24/16 2:05:00 PM |
|---|---|---|

Font:12 pt

| Page 34: [437] Formatted | Smith, Molly B | 11/24/16 2:05:00 PM |
|---|---|---|

Font:12 pt

| Page 34: [438] Formatted | Smith, Molly B | 11/24/16 2:14:00 PM |
|---|---|---|

Line spacing:  single

| Page 34: [439] Formatted | Smith, Molly B | 11/24/16 2:05:00 PM |
|---|---|---|

Font:12 pt, Not Italic, Font color: Auto

| Page 34: [440] Formatted | Smith, Molly B | 11/24/16 2:05:00 PM |
|---|---|---|

Font:12 pt

| Page 34: [440] Formatted | Smith, Molly B | 11/24/16 2:05:00 PM |
|---|---|---|

Font:12 pt

| Page 34: [441] Formatted | Smith, Molly B | 11/24/16 2:05:00 PM |
|---|---|---|

Font:12 pt

| Page 34: [441] Formatted | Smith, Molly B | 11/24/16 2:05:00 PM |
|---|---|---|

Font:12 pt

| Page 34: [442] Formatted | Smith, Molly B | 11/24/16 2:05:00 PM |
|---|---|---|

Font:12 pt

| Page 34: [442] Formatted | Smith, Molly B | 11/24/16 2:05:00 PM |
|---|---|---|

Font:12 pt

| Page 34: [443] Formatted | Smith, Molly B | 11/24/16 2:05:00 PM |
|---|---|---|

Font:12 pt

| Page 34: [443] Formatted | Smith, Molly B | 11/24/16 2:05:00 PM |
|---|---|---|

Font:12 pt

| Page 34: [444] Formatted | Smith, Molly B | 11/24/16 2:05:00 PM |
|---|---|---|

Font:12 pt

| Page 34: [444] Formatted | Smith, Molly B | 11/24/16 2:05:00 PM |
|---|---|---|

Font:12 pt

| Page 34: [445] Formatted | Smith, Molly B | 11/24/16 2:05:00 PM |
|---|---|---|

Font:12 pt

| Page 34: [445] Formatted | Smith, Molly B | 11/24/16 2:05:00 PM |
|---|---|---|

Font:12 pt

| Page 34: [446] Formatted | Smith, Molly B | 11/24/16 2:05:00 PM |
|---|---|---|

Font:12 pt

| Page 34: [447] Formatted | Smith, Molly B | 11/24/16 2:14:00 PM |
|---|---|---|

Line spacing:  single

| Page 34: [448] Formatted | Smith, Molly B | 11/24/16 2:05:00 PM |
|---|---|---|

Font:12 pt, Not Italic, Font color: Auto

| Page 34: [449] Formatted | Smith, Molly B | 11/24/16 2:05:00 PM |
|---|---|---|

Font:12 pt

| Page 34: [449] Formatted | Smith, Molly B | 11/24/16 2:05:00 PM |
|---|---|---|

Font:12 pt

| Page 34: [450] Formatted | Smith, Molly B | 11/24/16 2:05:00 PM |
|---|---|---|

Font:12 pt

| Page 34: [450] Formatted | Smith, Molly B | 11/24/16 2:05:00 PM |
|---|---|---|

Font:12 pt

| Page 34: [451] Formatted | Smith, Molly B | 11/24/16 2:05:00 PM |
|---|---|---|

Font:12 pt

| Page 34: [451] Formatted | Smith, Molly B | 11/24/16 2:05:00 PM |
|---|---|---|

Font:12 pt

| Page 34: [452] Formatted | Smith, Molly B | 11/24/16 2:05:00 PM |
|---|---|---|

Font:12 pt

| Page 34: [452] Formatted | Smith, Molly B | 11/24/16 2:05:00 PM |
|---|---|---|

Font:12 pt

| Page 34: [453] Formatted | Smith, Molly B | 11/24/16 2:05:00 PM |
|---|---|---|

Font:12 pt

| Page 34: [453] Formatted | Smith, Molly B | 11/24/16 2:05:00 PM |
|---|---|---|

Font:12 pt

| Page 34: [454] Formatted | Smith, Molly B | 11/24/16 2:05:00 PM |
|---|---|---|

Font:12 pt

| Page 34: [454] Formatted | Smith, Molly B | 11/24/16 2:05:00 PM |
|---|---|---|

Font:12 pt

| Page 34: [455] Deleted | Cornell University | 11/14/16 5:26:00 PM |
|---|---|---|

**Table 5: Variability in Southern Hemisphere**
**Values for the monthly variability (monthly average standard deviation divided by mean) and the fraction of the variability from the seasonal cycle (climatological monthly average standard deviation, divided by monthly mean standard deviation) for the surface concentration in the model cases and data from Rio Gallego and deposition data from Kerguelen (locations listed in Table 1).**

| Model/Observations | Rio Gallego Surface concentrations | | Kerguelen deposition | |
|---|---|---|---|---|
| | Monthly variability | Fraction variability from seasonal cycle | Monthly variability | Fraction variability from seasonal cycle |
| CAM4 (MERRA) | 5.67 | 0.33 | 0.64 | 0.41 |
| CAM4 (NCEP) | 5.89 | 0.29 | 0.82 | 0.33 |
| CAM4 (ERAI) | 7.52 | 0.30 | 4.39 | 0.43 |
| CAM4 (AMIP) | 8.98 | 0.24 | 4.47 | 0.22 |
| GCHEM (MERRA) | 1.94 | 0.43 | 0.70 | 0.47 |
| MATCH (NCEP) | 2.18 | 0.42 | 0.62 | 0.59 |
| CAM5 (AMIP) | 2.01 | 0.71 | 1.94 | 0.56 |
| Obs | 0.70 | 0.20 | 0.71 | 0.45 |

| Page 34: [456] Formatted | Smith, Molly B | 11/24/16 2:05:00 PM |
|---|---|---|

Font:12 pt

| Page 34: [457] Formatted | Smith, Molly B | 11/24/16 2:05:00 PM |
|---|---|---|

Font:12 pt, Not Italic

| Page 34: [457] Formatted | Smith, Molly B | 11/24/16 2:05:00 PM |
|---|---|---|

Font:12 pt, Not Italic

| Page 34: [457] Formatted | Smith, Molly B | 11/24/16 2:05:00 PM |
|---|---|---|

Font:12 pt, Not Italic

| Page 34: [457] Formatted | Smith, Molly B | 11/24/16 2:05:00 PM |
|---|---|---|

Font:12 pt, Not Italic

| Page 34: [457] Formatted | Smith, Molly B | 11/24/16 2:05:00 PM |
|---|---|---|

Font:12 pt, Not Italic

| Page 34: [457] Formatted | Smith, Molly B | 11/24/16 2:05:00 PM |
|---|---|---|

Font:12 pt, Not Italic

| Page 34: [457] Formatted | Smith, Molly B | 11/24/16 2:05:00 PM |
|---|---|---|

Font:12 pt, Not Italic

| Page 34: [457] Formatted | Smith, Molly B | 11/24/16 2:05:00 PM |
|---|---|---|

Font:12 pt, Not Italic

| Page 34: [457] Formatted | Smith, Molly B | 11/24/16 2:05:00 PM |
|---|---|---|

Font:12 pt, Not Italic

| Page 34: [457] Formatted | Smith, Molly B | 11/24/16 2:05:00 PM |
|---|---|---|

Font:12 pt, Not Italic

| Page 34: [457] Formatted | Smith, Molly B | 11/24/16 2:05:00 PM |
|---|---|---|

Font:12 pt, Not Italic

| Page 34: [457] Formatted | Smith, Molly B | 11/24/16 2:05:00 PM |
|---|---|---|

Font:12 pt, Not Italic

| Page 34: [457] Formatted | Smith, Molly B | 11/24/16 2:05:00 PM |
|---|---|---|

Font:12 pt, Not Italic

| Page 34: [457] Formatted | Smith, Molly B | 11/24/16 2:05:00 PM |
|---|---|---|

Font:12 pt, Not Italic

| Page 34: [457] Formatted | Smith, Molly B | 11/24/16 2:05:00 PM |
|---|---|---|

Font:12 pt, Not Italic

| Page 34: [457] Formatted | Smith, Molly B | 11/24/16 2:05:00 PM |
|---|---|---|

Font:12 pt, Not Italic

| Page 34: [457] Formatted | Smith, Molly B | 11/24/16 2:05:00 PM |
|---|---|---|

Font:12 pt, Not Italic

| Page 34: [457] Formatted | Smith, Molly B | 11/24/16 2:05:00 PM |
|---|---|---|

Font:12 pt, Not Italic

| Page 34: [457] Formatted | Smith, Molly B | 11/24/16 2:05:00 PM |
|---|---|---|

Font:12 pt, Not Italic

| Page 34: [457] Formatted | Smith, Molly B | 11/24/16 2:05:00 PM |
|---|---|---|

Font:12 pt, Not Italic

| Page 34: [457] Formatted | Smith, Molly B | 11/24/16 2:05:00 PM |
|---|---|---|

Font:12 pt, Not Italic

| Page 34: [457] Formatted | Smith, Molly B | 11/24/16 2:05:00 PM |
|---|---|---|

Font:12 pt, Not Italic

| Page 34: [457] Formatted | Smith, Molly B | 11/24/16 2:05:00 PM |
|---|---|---|

Font:12 pt, Not Italic

| Page 34: [457] Formatted | Smith, Molly B | 11/24/16 2:05:00 PM |
|---|---|---|

Font:12 pt, Not Italic

| Page 34: [457] Formatted | Smith, Molly B | 11/24/16 2:05:00 PM |
|---|---|---|

Font:12 pt, Not Italic

| Page 34: [457] Formatted | Smith, Molly B | 11/24/16 2:05:00 PM |
|---|---|---|

Font:12 pt, Not Italic

| Page 34: [457] Formatted | Smith, Molly B | 11/24/16 2:05:00 PM |
|---|---|---|

Font:12 pt, Not Italic

| Page 34: [457] Formatted | Smith, Molly B | 11/24/16 2:05:00 PM |
|---|---|---|

Font:12 pt, Not Italic

| Page 34: [457] Formatted | Smith, Molly B | 11/24/16 2:05:00 PM |
|---|---|---|

Font:12 pt, Not Italic

| Page 34: [457] Formatted | Smith, Molly B | 11/24/16 2:05:00 PM |
|---|---|---|

Font:12 pt, Not Italic

| Page 34: [457] Formatted | Smith, Molly B | 11/24/16 2:05:00 PM |
|---|---|---|

Font:12 pt, Not Italic

| Page 34: [457] Formatted | Smith, Molly B | 11/24/16 2:05:00 PM |
|---|---|---|

Font:12 pt, Not Italic

| Page 34: [457] Formatted | Smith, Molly B | 11/24/16 2:05:00 PM |
|---|---|---|

Font:12 pt, Not Italic

| Page 34: [457] Formatted | Smith, Molly B | 11/24/16 2:05:00 PM |
|---|---|---|

Font:12 pt, Not Italic

| Page 34: [457] Formatted | Smith, Molly B | 11/24/16 2:05:00 PM |
|---|---|---|

Font:12 pt, Not Italic

| Page 34: [457] Formatted | Smith, Molly B | 11/24/16 2:05:00 PM |
|---|---|---|

Font:12 pt, Not Italic

| Page 34: [457] Formatted | Smith, Molly B | 11/24/16 2:05:00 PM |
|---|---|---|

Font:12 pt, Not Italic

| Page 34: [457] Formatted | Smith, Molly B | 11/24/16 2:05:00 PM |
|---|---|---|

Font:12 pt, Not Italic

| Page 34: [457] Formatted | Smith, Molly B | 11/24/16 2:05:00 PM |
|---|---|---|

Font:12 pt, Not Italic

| Page 34: [457] Formatted | Smith, Molly B | 11/24/16 2:05:00 PM |
|---|---|---|

Font:12 pt, Not Italic

| Page 34: [457] Formatted | Smith, Molly B | 11/24/16 2:05:00 PM |
|---|---|---|

Font:12 pt, Not Italic

| Page 34: [457] Formatted | Smith, Molly B | 11/24/16 2:05:00 PM |
|---|---|---|

Font:12 pt, Not Italic

| Page 34: [457] Formatted | Smith, Molly B | 11/24/16 2:05:00 PM |
|---|---|---|

Font:12 pt, Not Italic

| Page 34: [457] Formatted | Smith, Molly B | 11/24/16 2:05:00 PM |
|---|---|---|

Font:12 pt, Not Italic

| Page 34: [457] Formatted | Smith, Molly B | 11/24/16 2:05:00 PM |
|---|---|---|

Font:12 pt, Not Italic

| Page 34: [457] Formatted | Smith, Molly B | 11/24/16 2:05:00 PM |
|---|---|---|

Font:12 pt, Not Italic

| Page 34: [457] Formatted | Smith, Molly B | 11/24/16 2:05:00 PM |
|---|---|---|

Font:12 pt, Not Italic

| Page 34: [457] Formatted | Smith, Molly B | 11/24/16 2:05:00 PM |
|---|---|---|

Font:12 pt, Not Italic

| Page 34: [457] Formatted | Smith, Molly B | 11/24/16 2:05:00 PM |
|---|---|---|

Font:12 pt, Not Italic

| Page 34: [457] Formatted | Smith, Molly B | 11/24/16 2:05:00 PM |
|---|---|---|

Font:12 pt, Not Italic

| Page 34: [457] Formatted | Smith, Molly B | 11/24/16 2:05:00 PM |
|---|---|---|

Font:12 pt, Not Italic

| Page 34: [457] Formatted | Smith, Molly B | 11/24/16 2:05:00 PM |
|---|---|---|

Font:12 pt, Not Italic

| Page 34: [457] Formatted | Smith, Molly B | 11/24/16 2:05:00 PM |
|---|---|---|

Font:12 pt, Not Italic

| Page 34: [457] Formatted | Smith, Molly B | 11/24/16 2:05:00 PM |
|---|---|---|

Font:12 pt, Not Italic

| Page 34: [457] Formatted | Smith, Molly B | 11/24/16 2:05:00 PM |
|---|---|---|

Font:12 pt, Not Italic

| Page 34: [457] Formatted | Smith, Molly B | 11/24/16 2:05:00 PM |
|---|---|---|

Font:12 pt, Not Italic

| Page 34: [457] Formatted | Smith, Molly B | 11/24/16 2:05:00 PM |
|---|---|---|

Font:12 pt, Not Italic

| Page 34: [457] Formatted | Smith, Molly B | 11/24/16 2:05:00 PM |
|---|---|---|

Font:12 pt, Not Italic

| Page 34: [457] Formatted | Smith, Molly B | 11/24/16 2:05:00 PM |
|---|---|---|

Font:12 pt, Not Italic

| Page 34: [457] Formatted | Smith, Molly B | 11/24/16 2:05:00 PM |
|---|---|---|

Font:12 pt, Not Italic

| Page 34: [457] Formatted | Smith, Molly B | 11/24/16 2:05:00 PM |
|---|---|---|

Font:12 pt, Not Italic

| Page 34: [457] Formatted | Smith, Molly B | 11/24/16 2:05:00 PM |
|---|---|---|

Font:12 pt, Not Italic

| Page 34: [457] Formatted | Smith, Molly B | 11/24/16 2:05:00 PM |
|---|---|---|

Font:12 pt, Not Italic

| Page 34: [457] Formatted | Smith, Molly B | 11/24/16 2:05:00 PM |
|---|---|---|

Font:12 pt, Not Italic

| Page 34: [457] Formatted | Smith, Molly B | 11/24/16 2:05:00 PM |
|---|---|---|

Font:12 pt, Not Italic

| Page 34: [457] Formatted | Smith, Molly B | 11/24/16 2:05:00 PM |
|---|---|---|

Font:12 pt, Not Italic

| Page 34: [457] Formatted | Smith, Molly B | 11/24/16 2:05:00 PM |
|---|---|---|

Font:12 pt, Not Italic

| Page 34: [457] Formatted | Smith, Molly B | 11/24/16 2:05:00 PM |
|---|---|---|

Font:12 pt, Not Italic

| Page 34: [457] Formatted | Smith, Molly B | 11/24/16 2:05:00 PM |
|---|---|---|

Font:12 pt, Not Italic

| Page 34: [457] Formatted | Smith, Molly B | 11/24/16 2:05:00 PM |
|---|---|---|

Font:12 pt, Not Italic

| Page 34: [457] Formatted | Smith, Molly B | 11/24/16 2:05:00 PM |
|---|---|---|

Font:12 pt, Not Italic

| Page 34: [457] Formatted | Smith, Molly B | 11/24/16 2:05:00 PM |
|---|---|---|

Font:12 pt, Not Italic

| Page 34: [457] Formatted | Smith, Molly B | 11/24/16 2:05:00 PM |
|---|---|---|

Font:12 pt, Not Italic

| Page 34: [457] Formatted | Smith, Molly B | 11/24/16 2:05:00 PM |
|---|---|---|

Font:12 pt, Not Italic

| Page 34: [457] Formatted | Smith, Molly B | 11/24/16 2:05:00 PM |
|---|---|---|

Font:12 pt, Not Italic

| Page 34: [457] Formatted | Smith, Molly B | 11/24/16 2:05:00 PM |
|---|---|---|

Font:12 pt, Not Italic

| Page 34: [457] Formatted | Smith, Molly B | 11/24/16 2:05:00 PM |
|---|---|---|

Font:12 pt, Not Italic

| Page 34: [457] Formatted | Smith, Molly B | 11/24/16 2:05:00 PM |
|---|---|---|

Font:12 pt, Not Italic

| Page 34: [457] Formatted | Smith, Molly B | 11/24/16 2:05:00 PM |
|---|---|---|

Font:12 pt, Not Italic

| Page 34: [457] Formatted | Smith, Molly B | 11/24/16 2:05:00 PM |
|---|---|---|

Font:12 pt, Not Italic

| Page 34: [457] Formatted | Smith, Molly B | 11/24/16 2:05:00 PM |
|---|---|---|

Font:12 pt, Not Italic

| Page 34: [457] Formatted | Smith, Molly B | 11/24/16 2:05:00 PM |
|---|---|---|

Font:12 pt, Not Italic

| Page 34: [457] Formatted | Smith, Molly B | 11/24/16 2:05:00 PM |
|---|---|---|

Font:12 pt, Not Italic

| Page 34: [457] Formatted | Smith, Molly B | 11/24/16 2:05:00 PM |
|---|---|---|

Font:12 pt, Not Italic

| Page 34: [457] Formatted | Smith, Molly B | 11/24/16 2:05:00 PM |
|---|---|---|

Font:12 pt, Not Italic

| Page 34: [457] Formatted | Smith, Molly B | 11/24/16 2:05:00 PM |
|---|---|---|

Font:12 pt, Not Italic

| Page 34: [457] Formatted | Smith, Molly B | 11/24/16 2:05:00 PM |
|---|---|---|

Font:12 pt, Not Italic

| Page 34: [458] Formatted | Smith, Molly B | 11/24/16 2:05:00 PM |
|---|---|---|

Font:12 pt

| Page 35: [459] Moved to page 36 (Move #3)Cornell University | 12/3/16 9:08:00 AM |
|---|---|

**Table 6: Surface concentration over ocean basins. For each ocean region, the averaged correlation across time between annual mean deposition fluxes for the CAM-RE cases is shown in the second column. The third column shows the annual mean correlation with NAO, while the third column shows the annual mean correlation with the El Nino/Southern Oscillation climate index. Regions are defined as the ocean gridboxes (not including sea ice or land boxes) in the following latitude and longitude areas as from (Gregg et al., 2003): North Atlantic (>30°N; 270 to 30°E); North Pacific (>30°N; 120 to 270°E); North Central Atlantic (10 to 30°N, 270 to 30°E); North Central Pacific (10 to 30N; 120 to 270°E); North Indian (10 to 30°N; 30 to 120°E); Equatorial Atlantic (-10 to 10°N; 300 to 30°E); Equatorial Pacific (-10 to 10°N; 120 to 285°E); Equatorial Indian (-10 to 10°N; 30-120°E); South Atlantic (-30 to -10°N; 30 to 300°E); South Pacific (-30 to -10°N; 120 to 295°E); South Indian (-30 to -30°N, 30 to 120°E); Antarctic (<-30°N).**

| | CAM4-RE across model Correlation | NAO correlation | El Nino Correlation |
|---|---|---|---|
| North Atlantic | 0.66 | 0.10 | 0.45 |
| North Pacific | 0.51 | 0.19 | 0.62 |

| | | | |
|---|---|---|---|
| North Central Atlantic | 0.75 | 0.04 | -0.10 |
| North Central Pacific | 0.46 | -0.19 | 0.01 |
| North Indian | 0.30 | 0.13 | 0.38 |
| Equatorial Atlantic | 0.59 | -0.02 | -0.31 |
| Equatorial Pacific | 0.19 | -0.12 | 0.42 |
| Equatorial Indian | 0.31 | -0.18 | -0.15 |
| South Atlantic | 0.11 | -0.22 | -0.42 |
| South Pacific | 0.65 | 0.03 | 0.03 |
| South Indian | 0.46 | 0.29 | 0.16 |
| Antarctic | 0.28 | -0.42 | -0.63 |
| Global | 0.42 | 0.01 | -0.03 |

| Page 35: [460] Formatted | Smith, Molly B | 11/24/16 2:05:00 PM |
|---|---|---|

Font:12 pt

| Page 35: [460] Formatted | Smith, Molly B | 11/24/16 2:05:00 PM |
|---|---|---|

Font:12 pt

| Page 35: [460] Formatted | Smith, Molly B | 11/24/16 2:05:00 PM |
|---|---|---|

Font:12 pt

| Page 35: [460] Formatted | Smith, Molly B | 11/24/16 2:05:00 PM |
|---|---|---|

Font:12 pt

| Page 35: [460] Formatted | Smith, Molly B | 11/24/16 2:05:00 PM |
|---|---|---|

Font:12 pt

| Page 35: [460] Formatted | Smith, Molly B | 11/24/16 2:05:00 PM |
|---|---|---|

Font:12 pt

| Page 35: [460] Formatted | Smith, Molly B | 11/24/16 2:05:00 PM |
|---|---|---|

Font:12 pt

| Page 35: [460] Formatted | Smith, Molly B | 11/24/16 2:05:00 PM |
|---|---|---|

Font:12 pt

| Page 35: [460] Formatted | Smith, Molly B | 11/24/16 2:05:00 PM |
|---|---|---|

Font:12 pt

| Page 35: [460] Formatted | Smith, Molly B | 11/24/16 2:05:00 PM |
|---|---|---|

Font:12 pt

| Page 35: [460] Formatted | Smith, Molly B | 11/24/16 2:05:00 PM |
|---|---|---|

Font:12 pt

| Page 35: [460] Formatted | Smith, Molly B | 11/24/16 2:05:00 PM |
|---|---|---|

Font:12 pt

| Page 35: [460] Formatted | Smith, Molly B | 11/24/16 2:05:00 PM |
|---|---|---|

Font:12 pt

| Page 35: [460] Formatted | Smith, Molly B | 11/24/16 2:05:00 PM |
|---|---|---|

Font:12 pt

| Page 35: [460] Formatted | Smith, Molly B | 11/24/16 2:05:00 PM |
|---|---|---|

Font:12 pt

| Page 35: [460] Formatted | Smith, Molly B | 11/24/16 2:05:00 PM |
|---|---|---|

Font:12 pt

| Page 35: [460] Formatted | Smith, Molly B | 11/24/16 2:05:00 PM |
|---|---|---|

Font:12 pt

| Page 35: [460] Formatted | Smith, Molly B | 11/24/16 2:05:00 PM |
|---|---|---|

Font:12 pt

| Page 35: [460] Formatted | Smith, Molly B | 11/24/16 2:05:00 PM |
|---|---|---|

Font:12 pt

| Page 35: [460] Formatted | Smith, Molly B | 11/24/16 2:05:00 PM |
|---|---|---|

Font:12 pt

| Page 35: [460] Formatted | Smith, Molly B | 11/24/16 2:05:00 PM |
|---|---|---|

Font:12 pt

| Page 35: [460] Formatted | Smith, Molly B | 11/24/16 2:05:00 PM |
|---|---|---|

Font:12 pt

| Page 35: [460] Formatted | Smith, Molly B | 11/24/16 2:05:00 PM |
|---|---|---|

Font:12 pt

| Page 35: [460] Formatted | Smith, Molly B | 11/24/16 2:05:00 PM |
|---|---|---|

Font:12 pt

| Page 35: [460] Formatted | Smith, Molly B | 11/24/16 2:05:00 PM |
|---|---|---|

Font:12 pt

| Page 35: [460] Formatted | Smith, Molly B | 11/24/16 2:05:00 PM |
|---|---|---|

Font:12 pt

| Page 35: [460] Formatted | Smith, Molly B | 11/24/16 2:05:00 PM |
|---|---|---|

Font:12 pt

| Page 35: [460] Formatted | Smith, Molly B | 11/24/16 2:05:00 PM |
|---|---|---|

Font:12 pt

| Page 35: [460] Formatted | Smith, Molly B | 11/24/16 2:05:00 PM |
|---|---|---|

Font:12 pt

| Page 35: [460] Formatted | Smith, Molly B | 11/24/16 2:05:00 PM |
|---|---|---|

Font:12 pt

| Page 35: [460] Formatted | Smith, Molly B | 11/24/16 2:05:00 PM |
|---|---|---|

Font:12 pt

| Page 35: [460] Formatted | Smith, Molly B | 11/24/16 2:05:00 PM |
|---|---|---|

Font:12 pt

| Page 35: [460] Formatted | Smith, Molly B | 11/24/16 2:05:00 PM |
|---|---|---|

Font:12 pt

| Page 35: [460] Formatted | Smith, Molly B | 11/24/16 2:05:00 PM |
|---|---|---|

Font:12 pt

| Page 35: [460] Formatted | Smith, Molly B | 11/24/16 2:05:00 PM |
|---|---|---|

Font:12 pt

| Page 35: [460] Formatted | Smith, Molly B | 11/24/16 2:05:00 PM |
|---|---|---|

Font:12 pt

| Page 35: [460] Formatted | Smith, Molly B | 11/24/16 2:05:00 PM |
|---|---|---|

Font:12 pt

| Page 35: [460] Formatted | Smith, Molly B | 11/24/16 2:05:00 PM |
|---|---|---|

Font:12 pt

| Page 35: [460] Formatted | Smith, Molly B | 11/24/16 2:05:00 PM |
|---|---|---|

Font:12 pt

| Page 35: [460] Formatted | Smith, Molly B | 11/24/16 2:05:00 PM |
|---|---|---|

Font:12 pt

| Page 35: [460] Formatted | Smith, Molly B | 11/24/16 2:05:00 PM |
|---|---|---|

Font:12 pt

| Page 35: [460] Formatted | Smith, Molly B | 11/24/16 2:05:00 PM |
|---|---|---|

Font:12 pt

| Page 35: [461] Deleted | Cornell University | 12/3/16 12:16:00 PM |
|---|---|---|

|  | CAM4-RE across model Correlation | NAO correlation | El Nino Correlation |
|---|---|---|---|
| North Atlantic | 0.66 | 0.10 | 0.45 |
| North Pacific | 0.51 | 0.19 | 0.62 |
| North Central Atlantic | 0.75 | 0.04 | -0.10 |
| North Central Pacific | 0.46 | -0.19 | 0.01 |
| North Indian | 0.30 | 0.13 | 0.38 |
| Equatorial Atlantic | 0.59 | -0.02 | -0.31 |
| Equatorial Pacific | 0.19 | -0.12 | 0.42 |
| Equatorial Indian | 0.31 | -0.18 | -0.15 |
| South Atlantic | 0.11 | -0.22 | -0.42 |
| South Pacific | 0.65 | 0.03 | 0.03 |
| South Indian | 0.46 | 0.29 | 0.16 |
| Antarctic | 0.28 | -0.42 | -0.63 |
| Global | 0.42 | 0.01 | -0.03 |

| Page 35: [462] Formatted | Smith, Molly B | 11/24/16 2:05:00 PM |
|---|---|---|

Font:12 pt, Not Italic

| Page 35: [462] Formatted | Smith, Molly B | 11/24/16 2:05:00 PM |
|---|---|---|

Font:12 pt, Not Italic

| Page 35: [462] Formatted | Smith, Molly B | 11/24/16 2:05:00 PM |
|---|---|---|

Font:12 pt, Not Italic

| Page 35: [462] Formatted | Smith, Molly B | 11/24/16 2:05:00 PM |
|---|---|---|

Font:12 pt, Not Italic

| Page 35: [462] Formatted | Smith, Molly B | 11/24/16 2:05:00 PM |
|---|---|---|

Font:12 pt, Not Italic

| Page 35: [462] Formatted | Smith, Molly B | 11/24/16 2:05:00 PM |
|---|---|---|

Font:12 pt, Not Italic

| Page 35: [462] Formatted | Smith, Molly B | 11/24/16 2:05:00 PM |
|---|---|---|

Font:12 pt, Not Italic

| Page 35: [462] Formatted | Smith, Molly B | 11/24/16 2:05:00 PM |
|---|---|---|

Font:12 pt, Not Italic

| Page 35: [462] Formatted | Smith, Molly B | 11/24/16 2:05:00 PM |
|---|---|---|

Font:12 pt, Not Italic

| Page 35: [462] Formatted | Smith, Molly B | 11/24/16 2:05:00 PM |
|---|---|---|

Font:12 pt, Not Italic

| Page 35: [462] Formatted | Smith, Molly B | 11/24/16 2:05:00 PM |
|---|---|---|

Font:12 pt, Not Italic

| Page 35: [462] Formatted | Smith, Molly B | 11/24/16 2:05:00 PM |
|---|---|---|

Font:12 pt, Not Italic

| Page 35: [462] Formatted | Smith, Molly B | 11/24/16 2:05:00 PM |
|---|---|---|

Font:12 pt, Not Italic

| Page 35: [462] Formatted | Smith, Molly B | 11/24/16 2:05:00 PM |
|---|---|---|

Font:12 pt, Not Italic

| Page 35: [462] Formatted | Smith, Molly B | 11/24/16 2:05:00 PM |
|---|---|---|

Font:12 pt, Not Italic

| Page 35: [462] Formatted | Smith, Molly B | 11/24/16 2:05:00 PM |
|---|---|---|

Font:12 pt, Not Italic

| Page 35: [462] Formatted | Smith, Molly B | 11/24/16 2:05:00 PM |
|---|---|---|

Font:12 pt, Not Italic

| Page 35: [462] Formatted | Smith, Molly B | 11/24/16 2:05:00 PM |
|---|---|---|

Font:12 pt, Not Italic

| Page 35: [462] Formatted | Smith, Molly B | 11/24/16 2:05:00 PM |
|---|---|---|

Font:12 pt, Not Italic

| Page 35: [462] Formatted | Smith, Molly B | 11/24/16 2:05:00 PM |
|---|---|---|

Font:12 pt, Not Italic

| Page 35: [462] Formatted | Smith, Molly B | 11/24/16 2:05:00 PM |
|---|---|---|

Font:12 pt, Not Italic

| Page 35: [462] Formatted | Smith, Molly B | 11/24/16 2:05:00 PM |
|---|---|---|

Font:12 pt, Not Italic

| Page 35: [462] Formatted | Smith, Molly B | 11/24/16 2:05:00 PM |
|---|---|---|

Font:12 pt, Not Italic

| Page 35: [462] Formatted | Smith, Molly B | 11/24/16 2:05:00 PM |
|---|---|---|

Font:12 pt, Not Italic

| Page 35: [462] Formatted | Smith, Molly B | 11/24/16 2:05:00 PM |
|---|---|---|

Font:12 pt, Not Italic

| Page 35: [462] Formatted | Smith, Molly B | 11/24/16 2:05:00 PM |
|---|---|---|

Font:12 pt, Not Italic

| Page 35: [462] Formatted | Smith, Molly B | 11/24/16 2:05:00 PM |
|---|---|---|

Font:12 pt, Not Italic

| Page 35: [462] Formatted | Smith, Molly B | 11/24/16 2:05:00 PM |
|---|---|---|

Font:12 pt, Not Italic

| Page 35: [462] Formatted | Smith, Molly B | 11/24/16 2:05:00 PM |
|---|---|---|

Font:12 pt, Not Italic

| Page 35: [462] Formatted | Smith, Molly B | 11/24/16 2:05:00 PM |
|---|---|---|

Font:12 pt, Not Italic

| Page 35: [462] Formatted | Smith, Molly B | 11/24/16 2:05:00 PM |
|---|---|---|

Font:12 pt, Not Italic

| Page 35: [462] Formatted | Smith, Molly B | 11/24/16 2:05:00 PM |
|---|---|---|

Font:12 pt, Not Italic

| Page 35: [462] Formatted | Smith, Molly B | 11/24/16 2:05:00 PM |
|---|---|---|

Font:12 pt, Not Italic

| Page 35: [462] Formatted | Smith, Molly B | 11/24/16 2:05:00 PM |
|---|---|---|

Font:12 pt, Not Italic

| Page 35: [462] Formatted | Smith, Molly B | 11/24/16 2:05:00 PM |
|---|---|---|

Font:12 pt, Not Italic

| Page 35: [462] Formatted | Smith, Molly B | 11/24/16 2:05:00 PM |
|---|---|---|

Font:12 pt, Not Italic

| Page 35: [462] Formatted | Smith, Molly B | 11/24/16 2:05:00 PM |
|---|---|---|

Font:12 pt, Not Italic

| Page 35: [462] Formatted | Smith, Molly B | 11/24/16 2:05:00 PM |
|---|---|---|

Font:12 pt, Not Italic

| Page 35: [462] Formatted | Smith, Molly B | 11/24/16 2:05:00 PM |
|---|---|---|

Font:12 pt, Not Italic

| Page 35: [462] Formatted | Smith, Molly B | 11/24/16 2:05:00 PM |
|---|---|---|

Font:12 pt, Not Italic

| Page 35: [462] Formatted | Smith, Molly B | 11/24/16 2:05:00 PM |
|---|---|---|

Font:12 pt, Not Italic

| Page 35: [462] Formatted | Smith, Molly B | 11/24/16 2:05:00 PM |
|---|---|---|

Font:12 pt, Not Italic

| Page 35: [462] Formatted | Smith, Molly B | 11/24/16 2:05:00 PM |
|---|---|---|

Font:12 pt, Not Italic

| Page 35: [462] Formatted | Smith, Molly B | 11/24/16 2:05:00 PM |
|---|---|---|

Font:12 pt, Not Italic

| Page 35: [462] Formatted | Smith, Molly B | 11/24/16 2:05:00 PM |
|---|---|---|

Font:12 pt, Not Italic

| Page 35: [462] Formatted | Smith, Molly B | 11/24/16 2:05:00 PM |
|---|---|---|

Font:12 pt, Not Italic

| Page 35: [462] Formatted | Smith, Molly B | 11/24/16 2:05:00 PM |
|---|---|---|

Font:12 pt, Not Italic

| Page 35: [462] Formatted | Smith, Molly B | 11/24/16 2:05:00 PM |
|---|---|---|

Font:12 pt, Not Italic

| Page 35: [462] Formatted | Smith, Molly B | 11/24/16 2:05:00 PM |
|---|---|---|

Font:12 pt, Not Italic

| Page 35: [462] Formatted | Smith, Molly B | 11/24/16 2:05:00 PM |
|---|---|---|

Font:12 pt, Not Italic

| Page 35: [462] Formatted | Smith, Molly B | 11/24/16 2:05:00 PM |
|---|---|---|

Font:12 pt, Not Italic

| Page 35: [462] Formatted | Smith, Molly B | 11/24/16 2:05:00 PM |
|---|---|---|

Font:12 pt, Not Italic

| Page 35: [462] Formatted | Smith, Molly B | 11/24/16 2:05:00 PM |
|---|---|---|

Font:12 pt, Not Italic

| Page 35: [462] Formatted | Smith, Molly B | 11/24/16 2:05:00 PM |
|---|---|---|

Font:12 pt, Not Italic

| Page 35: [462] Formatted | Smith, Molly B | 11/24/16 2:05:00 PM |
|---|---|---|

Font:12 pt, Not Italic

| Page 35: [462] Formatted | Smith, Molly B | 11/24/16 2:05:00 PM |
|---|---|---|

Font:12 pt, Not Italic

| Page 35: [462] Formatted | Smith, Molly B | 11/24/16 2:05:00 PM |
|---|---|---|

Font:12 pt, Not Italic

| Page 35: [462] Formatted | Smith, Molly B | 11/24/16 2:05:00 PM |
|---|---|---|

Font:12 pt, Not Italic

| Page 35: [462] Formatted | Smith, Molly B | 11/24/16 2:05:00 PM |
|---|---|---|

Font:12 pt, Not Italic

| Page 35: [462] Formatted | Smith, Molly B | 11/24/16 2:05:00 PM |
|---|---|---|

Font:12 pt, Not Italic

| Page 35: [462] Formatted | Smith, Molly B | 11/24/16 2:05:00 PM |
|---|---|---|

Font:12 pt, Not Italic

| Page 35: [462] Formatted | Smith, Molly B | 11/24/16 2:05:00 PM |
|---|---|---|

Font:12 pt, Not Italic

| Page 35: [462] Formatted | Smith, Molly B | 11/24/16 2:05:00 PM |
|---|---|---|

Font:12 pt, Not Italic

| Page 35: [462] Formatted | Smith, Molly B | 11/24/16 2:05:00 PM |
|---|---|---|

Font:12 pt, Not Italic

| Page 35: [462] Formatted | Smith, Molly B | 11/24/16 2:05:00 PM |
|---|---|---|

Font:12 pt, Not Italic

| Page 35: [462] Formatted | Smith, Molly B | 11/24/16 2:05:00 PM |
|---|---|---|

Font:12 pt, Not Italic

| Page 35: [462] Formatted | Smith, Molly B | 11/24/16 2:05:00 PM |
|---|---|---|

Font:12 pt, Not Italic

| Page 35: [462] Formatted | Smith, Molly B | 11/24/16 2:05:00 PM |
|---|---|---|

Font:12 pt, Not Italic

| Page 35: [462] Formatted | Smith, Molly B | 11/24/16 2:05:00 PM |
|---|---|---|

Font:12 pt, Not Italic

| Page 35: [462] Formatted | Smith, Molly B | 11/24/16 2:05:00 PM |
|---|---|---|

Font:12 pt, Not Italic

| Page 35: [462] Formatted | Smith, Molly B | 11/24/16 2:05:00 PM |
|---|---|---|

Font:12 pt, Not Italic

| Page 35: [462] Formatted | Smith, Molly B | 11/24/16 2:05:00 PM |
|---|---|---|

Font:12 pt, Not Italic

| Page 35: [462] Formatted | Smith, Molly B | 11/24/16 2:05:00 PM |
|---|---|---|

Font:12 pt, Not Italic

| Page 35: [462] Formatted | Smith, Molly B | 11/24/16 2:05:00 PM |
|---|---|---|

Font:12 pt, Not Italic

| Page 35: [462] Formatted | Smith, Molly B | 11/24/16 2:05:00 PM |
|---|---|---|

Font:12 pt, Not Italic

| Page 35: [462] Formatted | Smith, Molly B | 11/24/16 2:05:00 PM |
|---|---|---|

Font:12 pt, Not Italic

| Page 35: [462] Formatted | Smith, Molly B | 11/24/16 2:05:00 PM |
|---|---|---|

Font:12 pt, Not Italic

| Page 35: [462] Formatted | Smith, Molly B | 11/24/16 2:05:00 PM |
|---|---|---|

Font:12 pt, Not Italic

| Page 35: [462] Formatted | Smith, Molly B | 11/24/16 2:05:00 PM |
|---|---|---|

Font:12 pt, Not Italic

| Page 35: [462] Formatted | Smith, Molly B | 11/24/16 2:05:00 PM |
|---|---|---|

Font:12 pt, Not Italic

| Page 35: [462] Formatted | Smith, Molly B | 11/24/16 2:05:00 PM |
|---|---|---|

Font:12 pt, Not Italic

| Page 35: [462] Formatted | Smith, Molly B | 11/24/16 2:05:00 PM |
|---|---|---|

Font:12 pt, Not Italic

| Page 35: [462] Formatted | Smith, Molly B | 11/24/16 2:05:00 PM |
|---|---|---|

Font:12 pt, Not Italic

| Page 35: [462] Formatted | Smith, Molly B | 11/24/16 2:05:00 PM |
|---|---|---|

Font:12 pt, Not Italic

| Page 35: [462] Formatted | Smith, Molly B | 11/24/16 2:05:00 PM |
|---|---|---|

Font:12 pt, Not Italic

| Page 35: [462] Formatted | Smith, Molly B | 11/24/16 2:05:00 PM |
|---|---|---|

Font:12 pt, Not Italic

| Page 35: [462] Formatted | Smith, Molly B | 11/24/16 2:05:00 PM |
|---|---|---|

Font:12 pt, Not Italic

| Page 35: [462] Formatted | Smith, Molly B | 11/24/16 2:05:00 PM |
|---|---|---|

Font:12 pt, Not Italic

| Page 35: [462] Formatted | Smith, Molly B | 11/24/16 2:05:00 PM |
|---|---|---|

Font:12 pt, Not Italic

| Page 35: [462] Formatted | Smith, Molly B | 11/24/16 2:05:00 PM |
|---|---|---|

Font:12 pt, Not Italic

| Page 35: [462] Formatted | Smith, Molly B | 11/24/16 2:05:00 PM |
|---|---|---|

Font:12 pt, Not Italic

| Page 35: [462] Formatted | Smith, Molly B | 11/24/16 2:05:00 PM |
|---|---|---|

Font:12 pt, Not Italic

| Page 35: [462] Formatted | Smith, Molly B | 11/24/16 2:05:00 PM |
|---|---|---|

Font:12 pt, Not Italic

| Page 35: [462] Formatted | Smith, Molly B | 11/24/16 2:05:00 PM |
|---|---|---|

Font:12 pt, Not Italic

| Page 35: [462] Formatted | Smith, Molly B | 11/24/16 2:05:00 PM |
|---|---|---|

Font:12 pt, Not Italic

| Page 35: [462] Formatted | Smith, Molly B | 11/24/16 2:05:00 PM |
|---|---|---|

Font:12 pt, Not Italic

| Page 35: [462] Formatted | Smith, Molly B | 11/24/16 2:05:00 PM |
|---|---|---|

Font:12 pt, Not Italic

| Page 35: [462] Formatted | Smith, Molly B | 11/24/16 2:05:00 PM |
|---|---|---|

Font:12 pt, Not Italic

| Page 35: [462] Formatted | Smith, Molly B | 11/24/16 2:05:00 PM |
|---|---|---|

Font:12 pt, Not Italic

| Page 35: [463] Formatted | Smith, Molly B | 11/24/16 2:05:00 PM |
|---|---|---|

Font:12 pt

| Page 35: [464] Formatted | Smith, Molly B | 11/24/16 2:05:00 PM |
|---|---|---|

Line spacing:  single

| Page 35: [465] Formatted | Smith, Molly B | 11/24/16 2:05:00 PM |
|---|---|---|

Font:12 pt

| Page 35: [465] Formatted | Smith, Molly B | 11/24/16 2:05:00 PM |
|---|---|---|

Font:12 pt

| Page 35: [465] Formatted | Smith, Molly B | 11/24/16 2:05:00 PM |
|---|---|---|

Font:12 pt

| Page 35: [465] Formatted | Smith, Molly B | 11/24/16 2:05:00 PM |
|---|---|---|

Font:12 pt

| Page 35: [465] Formatted | Smith, Molly B | 11/24/16 2:05:00 PM |
|---|---|---|

Font:12 pt

| Page 35: [465] Formatted | Smith, Molly B | 11/24/16 2:05:00 PM |
|---|---|---|

Font:12 pt

| Page 35: [465] Formatted | Smith, Molly B | 11/24/16 2:05:00 PM |
|---|---|---|

Font:12 pt

| Page 35: [465] Formatted | Smith, Molly B | 11/24/16 2:05:00 PM |
|---|---|---|

Font:12 pt

| Page 35: [465] Formatted | Smith, Molly B | 11/24/16 2:05:00 PM |
|---|---|---|

Font:12 pt

| Page 35: [465] Formatted | Smith, Molly B | 11/24/16 2:05:00 PM |
|---|---|---|

Font:12 pt

| Page 35: [465] Formatted | Smith, Molly B | 11/24/16 2:05:00 PM |
|---|---|---|

Font:12 pt

| Page 35: [465] Formatted | Smith, Molly B | 11/24/16 2:05:00 PM |
|---|---|---|

Font:12 pt

| Page 35: [465] Formatted | Smith, Molly B | 11/24/16 2:05:00 PM |
|---|---|---|

Font:12 pt

| Page 35: [465] Formatted | Smith, Molly B | 11/24/16 2:05:00 PM |
|---|---|---|

Font:12 pt

| Page 35: [465] Formatted | Smith, Molly B | 11/24/16 2:05:00 PM |
|---|---|---|

Font:12 pt

| Page 35: [465] Formatted | Smith, Molly B | 11/24/16 2:05:00 PM |
|---|---|---|

Font:12 pt

| Page 35: [465] Formatted | Smith, Molly B | 11/24/16 2:05:00 PM |
|---|---|---|

Font:12 pt

| Page 35: [465] Formatted | Smith, Molly B | 11/24/16 2:05:00 PM |
|---|---|---|

Font:12 pt

| Page 35: [465] Formatted | Smith, Molly B | 11/24/16 2:05:00 PM |
|---|---|---|

Font:12 pt

| Page 35: [465] Formatted | Smith, Molly B | 11/24/16 2:05:00 PM |
|---|---|---|

Font:12 pt

| Page 35: [465] Formatted | Smith, Molly B | 11/24/16 2:05:00 PM |
|---|---|---|

Font:12 pt

| Page 35: [465] Formatted | Smith, Molly B | 11/24/16 2:05:00 PM |
|---|---|---|

Font:12 pt

| Page 35: [465] Formatted | Smith, Molly B | 11/24/16 2:05:00 PM |
|---|---|---|

Font:12 pt

| Page 35: [465] Formatted | Smith, Molly B | 11/24/16 2:05:00 PM |
|---|---|---|

Font:12 pt

| Page 35: [465] Formatted | Smith, Molly B | 11/24/16 2:05:00 PM |
|---|---|---|

Font:12 pt

| Page 35: [465] Formatted | Smith, Molly B | 11/24/16 2:05:00 PM |
|---|---|---|

Font:12 pt

| Page 35: [465] Formatted | Smith, Molly B | 11/24/16 2:05:00 PM |
|---|---|---|

Font:12 pt

| Page 35: [465] Formatted | Smith, Molly B | 11/24/16 2:05:00 PM |
|---|---|---|

Font:12 pt

| Page 35: [465] Formatted | Smith, Molly B | 11/24/16 2:05:00 PM |
|---|---|---|

Font:12 pt

| Page 35: [465] Formatted | Smith, Molly B | 11/24/16 2:05:00 PM |
|---|---|---|

Font:12 pt

| Page 35: [465] Formatted | Smith, Molly B | 11/24/16 2:05:00 PM |
|---|---|---|

Font:12 pt

| Page 35: [465] Formatted | Smith, Molly B | 11/24/16 2:05:00 PM |
|---|---|---|

Font:12 pt

| Page 35: [466] Formatted | Smith, Molly B | 11/24/16 2:05:00 PM |
|---|---|---|

Line spacing:  single

| Page 35: [467] Formatted | Smith, Molly B | 11/24/16 2:05:00 PM |
|---|---|---|

None, Space Before:  0 pt, Line spacing:  single, Don't keep with next, Don't keep lines together

| Page 35: [468] Formatted | Smith, Molly B | 11/24/16 2:05:00 PM |
|---|---|---|

Font:12 pt, Not Bold, Not Italic, Font color: Auto

| Page 35: [469] Formatted | Smith, Molly B | 11/24/16 2:05:00 PM |
|---|---|---|

Font:12 pt

| Page 35: [469] Formatted | Smith, Molly B | 11/24/16 2:05:00 PM |
|---|---|---|

Font:12 pt

| Page 35: [470] Formatted | Smith, Molly B | 11/24/16 2:05:00 PM |
|---|---|---|

Font:12 pt

| Page 35: [470] Formatted | Smith, Molly B | 11/24/16 2:05:00 PM |
|---|---|---|

Font:12 pt

| Page 35: [471] Formatted | Smith, Molly B | 11/24/16 2:05:00 PM |
|---|---|---|

Font:12 pt

| Page 35: [472] Formatted | Smith, Molly B | 11/24/16 2:05:00 PM |
|---|---|---|

None, Space Before:  0 pt, Line spacing:  single, Don't keep with next, Don't keep lines together

| Page 35: [473] Formatted | Smith, Molly B | 11/24/16 2:05:00 PM |
|---|---|---|

Font:12 pt, Not Bold, Not Italic, Font color: Auto

| Page 35: [474] Formatted | Smith, Molly B | 11/24/16 2:05:00 PM |
|---|---|---|

Font:12 pt

| Page 35: [474] Formatted | Smith, Molly B | 11/24/16 2:05:00 PM |
|---|---|---|

Font:12 pt

| Page 35: [475] Formatted | Smith, Molly B | 11/24/16 2:05:00 PM |
|---|---|---|

Font:12 pt

| Page 35: [475] Formatted | Smith, Molly B | 11/24/16 2:05:00 PM |
|---|---|---|

Font:12 pt

| Page 35: [476] Formatted | Smith, Molly B | 11/24/16 2:05:00 PM |
|---|---|---|

Font:12 pt

| Page 35: [476] Formatted | Smith, Molly B | 11/24/16 2:05:00 PM |
|---|---|---|

Font:12 pt

| Page 35: [477] Formatted | Smith, Molly B | 11/24/16 2:05:00 PM |
|---|---|---|

Font:12 pt

| Page 35: [478] Formatted | Smith, Molly B | 11/24/16 2:05:00 PM |
|---|---|---|

None, Space Before:  0 pt, Line spacing:  single, Don't keep with next, Don't keep lines together

| Page 35: [479] Formatted | Smith, Molly B | 11/24/16 2:05:00 PM |
|---|---|---|

Font:12 pt, Not Bold, Not Italic, Font color: Auto

| Page 35: [480] Formatted | Smith, Molly B | 11/24/16 2:05:00 PM |
|---|---|---|

Font:12 pt

| Page 35: [480] Formatted | Smith, Molly B | 11/24/16 2:05:00 PM |
|---|---|---|

Font:12 pt

| Page 35: [481] Formatted | Smith, Molly B | 11/24/16 2:05:00 PM |
|---|---|---|

Font:12 pt

| Page 35: [481] Formatted | Smith, Molly B | 11/24/16 2:05:00 PM |
|---|---|---|

Font:12 pt

| Page 35: [482] Formatted | Smith, Molly B | 11/24/16 2:05:00 PM |
|---|---|---|

Font:12 pt

| Page 35: [482] Formatted | Smith, Molly B | 11/24/16 2:05:00 PM |
|---|---|---|

Font:12 pt

| Page 35: [483] Formatted | Smith, Molly B | 11/24/16 2:05:00 PM |
|---|---|---|

Font:12 pt

| Page 35: [484] Formatted | Smith, Molly B | 11/24/16 2:05:00 PM |
|---|---|---|

None, Space Before:  0 pt, Line spacing:  single, Don't keep with next, Don't keep lines together

| Page 35: [485] Formatted | Smith, Molly B | 11/24/16 2:05:00 PM |
|---|---|---|

Font:12 pt, Not Bold, Not Italic, Font color: Auto

| Page 35: [486] Formatted | Smith, Molly B | 11/24/16 2:05:00 PM |
|---|---|---|

Font:12 pt

| Page 35: [486] Formatted | Smith, Molly B | 11/24/16 2:05:00 PM |
|---|---|---|

Font:12 pt

| Page 35: [487] Formatted | Smith, Molly B | 11/24/16 2:05:00 PM |
|---|---|---|

Font:12 pt

| Page 35: [487] Formatted | Smith, Molly B | 11/24/16 2:05:00 PM |
|---|---|---|

Font:12 pt

| Page 35: [488] Formatted | Smith, Molly B | 11/24/16 2:05:00 PM |
|---|---|---|

Font:12 pt

| Page 35: [488] Formatted | Smith, Molly B | 11/24/16 2:05:00 PM |
|---|---|---|

Font:12 pt

| Page 35: [489] Formatted | Smith, Molly B | 11/24/16 2:05:00 PM |
|---|---|---|

Font:12 pt

| Page 35: [490] Formatted | Smith, Molly B | 11/24/16 2:05:00 PM |
|---|---|---|

None, Space Before:  0 pt, Line spacing:  single, Don't keep with next, Don't keep lines together

| Page 35: [491] Formatted | Smith, Molly B | 11/24/16 2:05:00 PM |
|---|---|---|

Font:12 pt, Not Bold, Not Italic, Font color: Auto

| Page 35: [492] Formatted | Smith, Molly B | 11/24/16 2:05:00 PM |
|---|---|---|

Font:12 pt

| Page 35: [492] Formatted | Smith, Molly B | 11/24/16 2:05:00 PM |
|---|---|---|

Font:12 pt

| Page 35: [493] Formatted | Smith, Molly B | 11/24/16 2:05:00 PM |
|---|---|---|

Font:12 pt

| Page 35: [493] Formatted | Smith, Molly B | 11/24/16 2:05:00 PM |
|---|---|---|

Font:12 pt

| Page 35: [494] Formatted | Smith, Molly B | 11/24/16 2:05:00 PM |
|---|---|---|

Font:12 pt

| Page 35: [494] Formatted | Smith, Molly B | 11/24/16 2:05:00 PM |
|---|---|---|

Font:12 pt

| Page 35: [495] Formatted | Smith, Molly B | 11/24/16 2:05:00 PM |
|---|---|---|

Font:12 pt

| Page 35: [496] Formatted | Smith, Molly B | 11/24/16 2:05:00 PM |
|---|---|---|

None, Space Before:  0 pt, Line spacing:  single, Don't keep with next, Don't keep lines together

| Page 35: [497] Formatted | Smith, Molly B | 11/24/16 2:05:00 PM |
|---|---|---|

Font:12 pt, Not Bold, Not Italic, Font color: Auto

| Page 35: [498] Formatted | Smith, Molly B | 11/24/16 2:05:00 PM |
|---|---|---|

Font:12 pt

| Page 35: [498] Formatted | Smith, Molly B | 11/24/16 2:05:00 PM |
|---|---|---|

Font:12 pt

| Page 35: [499] Formatted | Smith, Molly B | 11/24/16 2:05:00 PM |
|---|---|---|

Font:12 pt

| Page 35: [499] Formatted | Smith, Molly B | 11/24/16 2:05:00 PM |
|---|---|---|

Font:12 pt

| Page 35: [500] Formatted | Smith, Molly B | 11/24/16 2:05:00 PM |
|---|---|---|

Font:12 pt

| Page 35: [500] Formatted | Smith, Molly B | 11/24/16 2:05:00 PM |
|---|---|---|

Font:12 pt

| Page 35: [501] Formatted | Smith, Molly B | 11/24/16 2:05:00 PM |
|---|---|---|

Font:12 pt

| Page 35: [502] Formatted | Smith, Molly B | 11/24/16 2:05:00 PM |
|---|---|---|

None, Space Before:  0 pt, Line spacing:  single, Don't keep with next, Don't keep lines together

| Page 35: [503] Formatted | Smith, Molly B | 11/24/16 2:05:00 PM |
|---|---|---|

Font:12 pt, Not Bold, Not Italic, Font color: Auto

| Page 35: [504] Formatted | Smith, Molly B | 11/24/16 2:05:00 PM |
|---|---|---|

Font:12 pt

| Page 35: [504] Formatted | Smith, Molly B | 11/24/16 2:05:00 PM |
|---|---|---|

Font:12 pt

| Page 35: [505] Formatted | Smith, Molly B | 11/24/16 2:05:00 PM |
|---|---|---|

Font:12 pt

| Page 35: [505] Formatted | Smith, Molly B | 11/24/16 2:05:00 PM |
|---|---|---|

Font:12 pt

| Page 35: [506] Formatted | Smith, Molly B | 11/24/16 2:05:00 PM |
|---|---|---|

Font:12 pt

| Page 35: [506] Formatted | Smith, Molly B | 11/24/16 2:05:00 PM |
|---|---|---|

Font:12 pt

| Page 35: [507] Formatted | Smith, Molly B | 11/24/16 2:05:00 PM |
|---|---|---|

Font:12 pt

| Page 35: [508] Formatted | Smith, Molly B | 11/24/16 2:05:00 PM |
|---|---|---|

None, Space Before:  0 pt, Line spacing:  single, Don't keep with next, Don't keep lines together

| Page 35: [509] Formatted | Smith, Molly B | 11/24/16 2:05:00 PM |
|---|---|---|

Font:12 pt, Not Bold, Not Italic, Font color: Auto

| Page 35: [510] Formatted | Smith, Molly B | 11/24/16 2:05:00 PM |
|---|---|---|

Font:12 pt

| Page 35: [510] Formatted | Smith, Molly B | 11/24/16 2:05:00 PM |
|---|---|---|

Font:12 pt

| Page 35: [511] Formatted | Smith, Molly B | 11/24/16 2:05:00 PM |
|---|---|---|

Font:12 pt

| Page 35: [511] Formatted | Smith, Molly B | 11/24/16 2:05:00 PM |
|---|---|---|

Font:12 pt

| Page 35: [512] Formatted | Smith, Molly B | 11/24/16 2:05:00 PM |
|---|---|---|

Font:12 pt

| Page 35: [512] Formatted | Smith, Molly B | 11/24/16 2:05:00 PM |
|---|---|---|

Font:12 pt

| Page 35: [513] Formatted | Smith, Molly B | 11/24/16 2:05:00 PM |
|---|---|---|

Font:12 pt

| Page 35: [514] Formatted | Smith, Molly B | 11/24/16 2:05:00 PM |
|---|---|---|

None, Space Before:  0 pt, Line spacing:  single, Don't keep with next, Don't keep lines together

| Page 35: [515] Formatted | Smith, Molly B | 11/24/16 2:05:00 PM |
|---|---|---|

Font:12 pt, Not Bold, Not Italic, Font color: Auto

| Page 35: [516] Formatted | Smith, Molly B | 11/24/16 2:05:00 PM |
|---|---|---|

Font:12 pt

| Page 35: [516] Formatted | Smith, Molly B | 11/24/16 2:05:00 PM |
|---|---|---|

Font:12 pt

| Page 35: [517] Formatted | Smith, Molly B | 11/24/16 2:05:00 PM |
|---|---|---|

Font:12 pt

| Page 35: [517] Formatted | Smith, Molly B | 11/24/16 2:05:00 PM |
|---|---|---|

Font:12 pt

| Page 35: [518] Formatted | Smith, Molly B | 11/24/16 2:05:00 PM |
|---|---|---|

Font:12 pt

| Page 35: [518] Formatted | Smith, Molly B | 11/24/16 2:05:00 PM |
|---|---|---|

Font:12 pt

| Page 35: [519] Formatted | Smith, Molly B | 11/24/16 2:05:00 PM |
|---|---|---|

Font:12 pt

| Page 35: [520] Formatted | Smith, Molly B | 11/24/16 2:05:00 PM |
|---|---|---|

None, Space Before:  0 pt, Line spacing:  single, Don't keep with next, Don't keep lines together

| Page 35: [521] Formatted | Smith, Molly B | 11/24/16 2:05:00 PM |
|---|---|---|

Font:12 pt, Not Bold, Not Italic, Font color: Auto

| Page 35: [522] Formatted | Smith, Molly B | 11/24/16 2:05:00 PM |
|---|---|---|

Font:12 pt

| Page 35: [522] Formatted | Smith, Molly B | 11/24/16 2:05:00 PM |
|---|---|---|

Font:12 pt

| Page 35: [523] Formatted | Smith, Molly B | 11/24/16 2:05:00 PM |
|---|---|---|

Font:12 pt

| Page 35: [523] Formatted | Smith, Molly B | 11/24/16 2:05:00 PM |
|---|---|---|

Font:12 pt

| Page 35: [524] Formatted | Smith, Molly B | 11/24/16 2:05:00 PM |
|---|---|---|

Font:12 pt

| Page 35: [524] Formatted | Smith, Molly B | 11/24/16 2:05:00 PM |
|---|---|---|

Font:12 pt

| Page 35: [525] Formatted | Smith, Molly B | 11/24/16 2:05:00 PM |
|---|---|---|

Font:12 pt

| Page 35: [526] Formatted | Smith, Molly B | 11/24/16 2:05:00 PM |
|---|---|---|

Line spacing:  single

| Page 36: [527] Formatted | Smith, Molly B | 11/24/16 2:05:00 PM |
|---|---|---|

Font:12 pt

| Page 36: [528] Formatted | Smith, Molly B | 11/24/16 2:16:00 PM |
|---|---|---|

Line spacing:  single

| Page 36: [529] Formatted Table | Smith, Molly B | 11/24/16 2:15:00 PM |
|---|---|---|

Formatted Table

| Page 36: [530] Formatted | Smith, Molly B | 11/24/16 2:16:00 PM |
|---|---|---|

Space Before:  0 pt, Line spacing:  single

| Page 36: [531] Formatted | Smith, Molly B | 11/24/16 2:05:00 PM |
|---|---|---|

Font:12 pt, Not Bold, Not Italic, Font color: Auto

| Page 36: [532] Formatted | Smith, Molly B | 11/24/16 2:05:00 PM |
|---|---|---|

Font:12 pt

| Page 36: [533] Formatted | Smith, Molly B | 11/24/16 2:16:00 PM |
|---|---|---|

None, Space Before:  0 pt, Line spacing:  single, Don't keep with next, Don't keep lines together

| Page 36: [534] Formatted | Smith, Molly B | 11/24/16 2:05:00 PM |
|---|---|---|

Font:12 pt, Not Bold, Not Italic, Font color: Auto

| Page 36: [535] Formatted | Smith, Molly B | 11/24/16 2:05:00 PM |
|---|---|---|

Font:12 pt

| Page 36: [536] Formatted | Smith, Molly B | 11/24/16 2:16:00 PM |
|---|---|---|

Space Before:  0 pt, Line spacing:  single

| Page 36: [537] Formatted | Smith, Molly B | 11/24/16 2:05:00 PM |
|---|---|---|

Font:12 pt, Not Bold, Not Italic, Font color: Auto

| Page 36: [538] Formatted | Smith, Molly B | 11/24/16 2:05:00 PM |
|---|---|---|

Font:12 pt

| Page 36: [538] Formatted | Smith, Molly B | 11/24/16 2:05:00 PM |
|---|---|---|

Font:12 pt

| **Page 36: [539] Formatted** | **Smith, Molly B** | **11/24/16 2:05:00 PM** |

Font:12 pt

| **Page 36: [540] Formatted** | **Smith, Molly B** | **11/24/16 2:16:00 PM** |

None, Space Before:  0 pt, Line spacing:  single, Don't keep with next, Don't keep lines together

| **Page 36: [541] Formatted** | **Smith, Molly B** | **11/24/16 2:05:00 PM** |

Font:12 pt, Not Bold, Not Italic, Font color: Auto

| **Page 36: [542] Formatted** | **Smith, Molly B** | **11/24/16 2:05:00 PM** |

Font:12 pt

| **Page 36: [542] Formatted** | **Smith, Molly B** | **11/24/16 2:05:00 PM** |

Font:12 pt

| **Page 36: [543] Formatted** | **Smith, Molly B** | **11/24/16 2:05:00 PM** |

Font:12 pt

| **Page 36: [543] Formatted** | **Smith, Molly B** | **11/24/16 2:05:00 PM** |

Font:12 pt

| **Page 36: [544] Formatted** | **Smith, Molly B** | **11/24/16 2:05:00 PM** |

Font:12 pt

| **Page 36: [544] Formatted** | **Smith, Molly B** | **11/24/16 2:05:00 PM** |

Font:12 pt

| **Page 36: [545] Formatted** | **Smith, Molly B** | **11/24/16 2:05:00 PM** |

Font:12 pt

| **Page 36: [545] Formatted** | **Smith, Molly B** | **11/24/16 2:05:00 PM** |

Font:12 pt

| **Page 36: [546] Formatted** | **Smith, Molly B** | **11/24/16 2:05:00 PM** |

Font:12 pt

| **Page 36: [547] Formatted** | **Smith, Molly B** | **11/24/16 2:16:00 PM** |

None, Space Before:  0 pt, Line spacing:  single, Don't keep with next, Don't keep lines together

| **Page 36: [548] Formatted** | **Smith, Molly B** | **11/24/16 2:05:00 PM** |

Font:12 pt, Not Bold, Not Italic, Font color: Auto

| **Page 36: [549] Formatted** | **Smith, Molly B** | **11/24/16 2:05:00 PM** |

Font:12 pt

| **Page 36: [549] Formatted** | **Smith, Molly B** | **11/24/16 2:05:00 PM** |

Font:12 pt

| **Page 36: [550] Formatted** | **Smith, Molly B** | **11/24/16 2:05:00 PM** |

Font:12 pt

| **Page 36: [550] Formatted** | **Smith, Molly B** | **11/24/16 2:05:00 PM** |

Font:12 pt

| Page 36: [551] Formatted | Smith, Molly B | 11/24/16 2:05:00 PM |
|---|---|---|

Font:12 pt

| Page 36: [551] Formatted | Smith, Molly B | 11/24/16 2:05:00 PM |
|---|---|---|

Font:12 pt

| Page 36: [552] Formatted | Smith, Molly B | 11/24/16 2:05:00 PM |
|---|---|---|

Font:12 pt

| Page 36: [552] Formatted | Smith, Molly B | 11/24/16 2:05:00 PM |
|---|---|---|

Font:12 pt

| Page 36: [553] Formatted | Smith, Molly B | 11/24/16 2:05:00 PM |
|---|---|---|

Font:12 pt

| Page 36: [554] Formatted | Smith, Molly B | 11/24/16 2:16:00 PM |
|---|---|---|

None, Space Before:  0 pt, Line spacing:  single, Don't keep with next, Don't keep lines together

| Page 36: [555] Formatted | Smith, Molly B | 11/24/16 2:05:00 PM |
|---|---|---|

Font:12 pt, Not Bold, Not Italic, Font color: Auto

| Page 36: [556] Formatted | Smith, Molly B | 11/24/16 2:05:00 PM |
|---|---|---|

Font:12 pt

| Page 36: [556] Formatted | Smith, Molly B | 11/24/16 2:05:00 PM |
|---|---|---|

Font:12 pt

| Page 36: [557] Formatted | Smith, Molly B | 11/24/16 2:05:00 PM |
|---|---|---|

Font:12 pt

| Page 36: [557] Formatted | Smith, Molly B | 11/24/16 2:05:00 PM |
|---|---|---|

Font:12 pt

| Page 36: [558] Formatted | Smith, Molly B | 11/24/16 2:05:00 PM |
|---|---|---|

Font:12 pt

| Page 36: [558] Formatted | Smith, Molly B | 11/24/16 2:05:00 PM |
|---|---|---|

Font:12 pt

| Page 36: [559] Formatted | Smith, Molly B | 11/24/16 2:05:00 PM |
|---|---|---|

Font:12 pt

| Page 36: [559] Formatted | Smith, Molly B | 11/24/16 2:05:00 PM |
|---|---|---|

Font:12 pt

| Page 36: [560] Formatted | Smith, Molly B | 11/24/16 2:05:00 PM |
|---|---|---|

Font:12 pt

| Page 36: [561] Formatted | Smith, Molly B | 11/24/16 2:16:00 PM |
|---|---|---|

None, Space Before:  0 pt, Line spacing:  single, Don't keep with next, Don't keep lines together

| Page 36: [562] Formatted | Smith, Molly B | 11/24/16 2:05:00 PM |
|---|---|---|

Font:12 pt, Not Bold, Not Italic, Font color: Auto

| Page 36: [563] Formatted | Smith, Molly B | 11/24/16 2:05:00 PM |
|---|---|---|

Font:12 pt

| Page 36: [563] Formatted | Smith, Molly B | 11/24/16 2:05:00 PM |
|---|---|---|

Font:12 pt

| Page 36: [564] Formatted | Smith, Molly B | 11/24/16 2:05:00 PM |
|---|---|---|

Font:12 pt

| Page 36: [564] Formatted | Smith, Molly B | 11/24/16 2:05:00 PM |
|---|---|---|

Font:12 pt

| Page 36: [565] Formatted | Smith, Molly B | 11/24/16 2:05:00 PM |
|---|---|---|

Font:12 pt

| Page 36: [565] Formatted | Smith, Molly B | 11/24/16 2:05:00 PM |
|---|---|---|

Font:12 pt

| Page 36: [566] Formatted | Smith, Molly B | 11/24/16 2:05:00 PM |
|---|---|---|

Font:12 pt

| Page 36: [566] Formatted | Smith, Molly B | 11/24/16 2:05:00 PM |
|---|---|---|

Font:12 pt

| Page 36: [567] Formatted | Smith, Molly B | 11/24/16 2:05:00 PM |
|---|---|---|

Font:12 pt

| Page 36: [568] Formatted | Smith, Molly B | 11/24/16 2:16:00 PM |
|---|---|---|

None, Space Before:  0 pt, Line spacing:  single, Don't keep with next, Don't keep lines together

| Page 36: [569] Formatted | Smith, Molly B | 11/24/16 2:05:00 PM |
|---|---|---|

Font:12 pt, Not Bold, Not Italic, Font color: Auto

| Page 36: [570] Formatted | Smith, Molly B | 11/24/16 2:05:00 PM |
|---|---|---|

Font:12 pt

| Page 36: [570] Formatted | Smith, Molly B | 11/24/16 2:05:00 PM |
|---|---|---|

Font:12 pt

| Page 36: [571] Formatted | Smith, Molly B | 11/24/16 2:05:00 PM |
|---|---|---|

Font:12 pt

| Page 36: [571] Formatted | Smith, Molly B | 11/24/16 2:05:00 PM |
|---|---|---|

Font:12 pt

| Page 36: [572] Formatted | Smith, Molly B | 11/24/16 2:05:00 PM |
|---|---|---|

Font:12 pt

| Page 36: [572] Formatted | Smith, Molly B | 11/24/16 2:05:00 PM |
|---|---|---|

Font:12 pt

| Page 36: [573] Formatted | Smith, Molly B | 11/24/16 2:05:00 PM |
|---|---|---|

Font:12 pt

| Page 36: [573] Formatted | Smith, Molly B | 11/24/16 2:05:00 PM |
|---|---|---|

Font:12 pt

| Page 36: [574] Formatted | Smith, Molly B | 11/24/16 2:05:00 PM |
|---|---|---|

Font:12 pt

| Page 36: [575] Formatted | Smith, Molly B | 11/24/16 2:16:00 PM |
|---|---|---|

None, Space Before:  0 pt, Line spacing:  single, Don't keep with next, Don't keep lines together

| Page 36: [576] Formatted | Smith, Molly B | 11/24/16 2:05:00 PM |
|---|---|---|

Font:12 pt, Not Bold, Not Italic, Font color: Auto

| Page 36: [577] Formatted | Smith, Molly B | 11/24/16 2:05:00 PM |
|---|---|---|

Font:12 pt

| Page 36: [577] Formatted | Smith, Molly B | 11/24/16 2:05:00 PM |
|---|---|---|

Font:12 pt

| Page 36: [578] Formatted | Smith, Molly B | 11/24/16 2:05:00 PM |
|---|---|---|

Font:12 pt

| Page 36: [578] Formatted | Smith, Molly B | 11/24/16 2:05:00 PM |
|---|---|---|

Font:12 pt

| Page 36: [579] Formatted | Smith, Molly B | 11/24/16 2:05:00 PM |
|---|---|---|

Font:12 pt

| Page 36: [579] Formatted | Smith, Molly B | 11/24/16 2:05:00 PM |
|---|---|---|

Font:12 pt

| Page 36: [580] Formatted | Smith, Molly B | 11/24/16 2:05:00 PM |
|---|---|---|

Font:12 pt

| Page 36: [580] Formatted | Smith, Molly B | 11/24/16 2:05:00 PM |
|---|---|---|

Font:12 pt

| Page 36: [581] Formatted | Smith, Molly B | 11/24/16 2:05:00 PM |
|---|---|---|

Font:12 pt

| Page 36: [582] Formatted | Smith, Molly B | 11/24/16 2:16:00 PM |
|---|---|---|

None, Space Before:  0 pt, Line spacing:  single, Don't keep with next, Don't keep lines together

| Page 36: [583] Formatted | Smith, Molly B | 11/24/16 2:05:00 PM |
|---|---|---|

Font:12 pt, Not Bold, Not Italic, Font color: Auto

| Page 36: [584] Formatted | Smith, Molly B | 11/24/16 2:05:00 PM |
|---|---|---|

Font:12 pt

| Page 36: [584] Formatted | Smith, Molly B | 11/24/16 2:05:00 PM |
|---|---|---|

Font:12 pt

| Page 36: [585] Formatted | Smith, Molly B | 11/24/16 2:05:00 PM |
|---|---|---|

Font:12 pt

| Page 36: [585] Formatted | Smith, Molly B | 11/24/16 2:05:00 PM |
|---|---|---|

Font:12 pt

| Page 36: [586] Formatted | Smith, Molly B | 11/24/16 2:05:00 PM |
|---|---|---|

Font:12 pt

| Page 36: [586] Formatted | Smith, Molly B | 11/24/16 2:05:00 PM |

Font:12 pt

| Page 36: [587] Formatted | Smith, Molly B | 11/24/16 2:05:00 PM |

Font:12 pt

| Page 36: [587] Formatted | Smith, Molly B | 11/24/16 2:05:00 PM |

Font:12 pt

| Page 36: [588] Formatted | Smith, Molly B | 11/24/16 2:05:00 PM |

Font:12 pt

| Page 36: [589] Formatted | Smith, Molly B | 11/24/16 2:16:00 PM |

None, Space Before: 0 pt, Line spacing: single, Don't keep with next, Don't keep lines together

| Page 36: [590] Formatted | Smith, Molly B | 11/24/16 2:05:00 PM |

Font:12 pt, Not Bold, Not Italic

| Page 36: [591] Formatted | Smith, Molly B | 11/24/16 2:05:00 PM |

Font:12 pt

| Page 36: [592] Formatted | Smith, Molly B | 11/24/16 2:16:00 PM |

Space Before: 0 pt, Line spacing: single

| Page 36: [593] Formatted | Smith, Molly B | 11/24/16 2:05:00 PM |

Font:12 pt, Not Bold, Not Italic

| Page 36: [594] Formatted | Smith, Molly B | 11/24/16 2:05:00 PM |

Font:12 pt

| Page 36: [594] Formatted | Smith, Molly B | 11/24/16 2:05:00 PM |

Font:12 pt

| Page 36: [595] Formatted | Smith, Molly B | 11/24/16 2:05:00 PM |

Font:12 pt

| Page 36: [595] Formatted | Smith, Molly B | 11/24/16 2:05:00 PM |

Font:12 pt

| Page 36: [596] Formatted | Smith, Molly B | 11/24/16 2:05:00 PM |

Font:12 pt

| Page 36: [596] Formatted | Smith, Molly B | 11/24/16 2:05:00 PM |

Font:12 pt

| Page 36: [597] Moved from page 35 (Move #3)Cornell University | 12/3/16 9:08:00 AM |

**Table 86: Surface concentration over ocean basins. For each ocean region, the averaged correlation across time between annual mean deposition fluxes for the CAM-RE cases is shown in the second column. The third column shows the annual mean correlation with NAO, while the third column shows the annual mean correlation with the El Nino/Southern Oscillation climate index. Regions are defined as the ocean gridboxes (not including sea ice or land boxes) in the following latitude and longitude areas as from**

(Gregg et al., 2003): North Atlantic (>30°N; 270 to 30°E); North Pacific (>30°N; 120 to 270°E); North Central Atlantic (10 to 30°N, 270 to 30°E); North Central Pacific (10 to 30N; 120 to 270°E); North Indian (10 to 30°N; 30 to 120°E); Equatorial Atlantic (-10 to 10°N; 300 to 30°E); Equatorial Pacific (-10 to 10°N; 120 to 285°E); Equatorial Indian (-10 to 10°N; 30-120°E); South Atlantic (-30 to -10°N; 30 to 300°E); South Pacific (-30 to -10°N; 120 to 295°E); South Indian (-30 to -30°N, 30 to 120°E); Antarctic (<-30°N).

| | CAM4-RE across model Correlation | NAO correlation | El Nino Correlation |
|---|---|---|---|
| North Atlantic | 0.66 | 0.10 | 0.45 |
| North Pacific | 0.51 | 0.19 | 0.62 |
| North Central Atlantic | 0.75 | 0.04 | -0.10 |
| North Central Pacific | 0.46 | -0.19 | 0.01 |
| North Indian | 0.30 | 0.13 | 0.38 |
| Equatorial Atlantic | 0.59 | -0.02 | -0.31 |
| Equatorial Pacific | 0.19 | -0.12 | 0.42 |
| Equatorial Indian | 0.31 | -0.18 | -0.15 |
| South Atlantic | 0.11 | -0.22 | -0.42 |
| South Pacific | 0.65 | 0.03 | 0.03 |
| South Indian | 0.46 | 0.29 | 0.16 |
| Antarctic | 0.28 | -0.42 | -0.63 |
| Global | 0.42 | 0.01 | -0.03 |

**Table 8: Deposition into ocean basins. For each ocean region, the averaged correlation across time between annual mean deposition fluxes for the CAM-RE cases is shown in the second column. The following columns show the climatological mean deposition flux (Tg/year) for each model simulation. Regions are defined as the ocean gridboxes (not including sea ice or land boxes) in the following latitude and longitude areas as from (Gregg et al., 2003): North Atlantic (>30°N; 270 to 30°E); North Pacific (>30°N; 120 to 270°E); North Central Atlantic (10 to 30°N, 270 to 30°E); North Central Pacific (10 to 30N; 120 to 270°E); North Indian (10 to 30°N; 30 to 120°E); Equatorial Atlantic (-10 to 10°N; 300 to 30°E); Equatorial Pacific (-10 to 10°N; 120 to 285°E); Equatorial Indian (-10 to 10°N; 30-120°E); South Atlantic (-30 to -10°N; 30 to 300°E); South Pacific (-30 to -10°N; 120 to 295°E); South Indian (-30 to -30°N, 30 to 120°E); Antarctic (<-30°N).**

|  | Correlation | Mean CAM4-Re | Mean all |
|---|---|---|---|
| North Atlantic | 0.68 | 54 +/-7% | 61 +/-17% |
| North Pacific | 0.50 | 30 +/- 30% | 30 +/- 60% |
| North Central Atlantic | 0.66 | 91 +/- 8% | 120 +/- 30% |
| North Central Pacific | 0.33 | 20 +/- 90% | 20 +/- 90% |
| North Indian | 0.10 | 100 +/- 30% | 110 +/- 40% |
| Equatorial Atlantic | 0.39 | 77 +/- 20% | 80 +/- 35% |
| Equatorial Pacific | 0.33 | 4 +/- 100% | 6 +/- 130% |
| Equatorial Indian | 0.31 | 23 +/- 25% | 21 +/-23% |
| South Atlantic | 0.16 | 3+/-60% | 3 +/- 90% |
| South Pacific | 0.62 | 1 +/- 40% | 3 +/- 60% |
| South Indian | 0.38 | 2 +/- 40% | 4 +/- 75% |
| Antarctic | 0.35 | 15 +/- 100% | 30 +/- 100% |
| Global | 0.45 | 450 +/- 10% | 510 +/- 24% |

| **Page 37: [600] Formatted** | **Smith, Molly B** | **11/24/16 2:05:00 PM** |

Font:12 pt

| **Page 37: [600] Formatted** | **Smith, Molly B** | **11/24/16 2:05:00 PM** |

Font:12 pt

| **Page 37: [600] Formatted** | **Smith, Molly B** | **11/24/16 2:05:00 PM** |

Font:12 pt

| **Page 37: [600] Formatted** | **Smith, Molly B** | **11/24/16 2:05:00 PM** |

Font:12 pt

| **Page 37: [600] Formatted** | **Smith, Molly B** | **11/24/16 2:05:00 PM** |

Font:12 pt

| **Page 37: [600] Formatted** | **Smith, Molly B** | **11/24/16 2:05:00 PM** |

Font:12 pt

| **Page 37: [600] Formatted** | **Smith, Molly B** | **11/24/16 2:05:00 PM** |

Font:12 pt

| **Page 37: [600] Formatted** | **Smith, Molly B** | **11/24/16 2:05:00 PM** |

Font:12 pt

| **Page 37: [600] Formatted** | **Smith, Molly B** | **11/24/16 2:05:00 PM** |

Font:12 pt

| **Page 37: [600] Formatted** | **Smith, Molly B** | **11/24/16 2:05:00 PM** |

Font:12 pt

| **Page 37: [600] Formatted** | **Smith, Molly B** | **11/24/16 2:05:00 PM** |

Font:12 pt

| **Page 37: [600] Formatted** | **Smith, Molly B** | **11/24/16 2:05:00 PM** |

Font:12 pt

| **Page 37: [600] Formatted** | **Smith, Molly B** | **11/24/16 2:05:00 PM** |

Font:12 pt

| **Page 37: [600] Formatted** | **Smith, Molly B** | **11/24/16 2:05:00 PM** |

Font:12 pt

| **Page 37: [600] Formatted** | **Smith, Molly B** | **11/24/16 2:05:00 PM** |

Font:12 pt

| **Page 37: [600] Formatted** | **Smith, Molly B** | **11/24/16 2:05:00 PM** |

Font:12 pt

| **Page 37: [600] Formatted** | **Smith, Molly B** | **11/24/16 2:05:00 PM** |

Font:12 pt

| **Page 37: [600] Formatted** | **Smith, Molly B** | **11/24/16 2:05:00 PM** |

Font:12 pt

| Page 37: [600] Formatted | Smith, Molly B | 11/24/16 2:05:00 PM |
|---|---|---|

Font:12 pt

| Page 37: [600] Formatted | Smith, Molly B | 11/24/16 2:05:00 PM |
|---|---|---|

Font:12 pt

| Page 37: [600] Formatted | Smith, Molly B | 11/24/16 2:05:00 PM |
|---|---|---|

Font:12 pt

| Page 37: [600] Formatted | Smith, Molly B | 11/24/16 2:05:00 PM |
|---|---|---|

Font:12 pt

| Page 37: [600] Formatted | Smith, Molly B | 11/24/16 2:05:00 PM |
|---|---|---|

Font:12 pt

| Page 37: [600] Formatted | Smith, Molly B | 11/24/16 2:05:00 PM |
|---|---|---|

Font:12 pt

| Page 37: [600] Formatted | Smith, Molly B | 11/24/16 2:05:00 PM |
|---|---|---|

Font:12 pt

| Page 37: [600] Formatted | Smith, Molly B | 11/24/16 2:05:00 PM |
|---|---|---|

Font:12 pt

| Page 37: [600] Formatted | Smith, Molly B | 11/24/16 2:05:00 PM |
|---|---|---|

Font:12 pt

| Page 37: [600] Formatted | Smith, Molly B | 11/24/16 2:05:00 PM |
|---|---|---|

Font:12 pt

| Page 37: [600] Formatted | Smith, Molly B | 11/24/16 2:05:00 PM |
|---|---|---|

Font:12 pt

| Page 37: [600] Formatted | Smith, Molly B | 11/24/16 2:05:00 PM |
|---|---|---|

Font:12 pt

| Page 37: [600] Formatted | Smith, Molly B | 11/24/16 2:05:00 PM |
|---|---|---|

Font:12 pt

| Page 37: [600] Formatted | Smith, Molly B | 11/24/16 2:05:00 PM |
|---|---|---|

Font:12 pt

| Page 37: [600] Formatted | Smith, Molly B | 11/24/16 2:05:00 PM |
|---|---|---|

Font:12 pt

| Page 37: [600] Formatted | Smith, Molly B | 11/24/16 2:05:00 PM |
|---|---|---|

Font:12 pt

| Page 37: [600] Formatted | Smith, Molly B | 11/24/16 2:05:00 PM |
|---|---|---|

Font:12 pt

| | | |
|---|---|---|
| **Page 37: [600] Formatted** | **Smith, Molly B** | **11/24/16 2:05:00 PM** |

Font:12 pt

| | | |
|---|---|---|
| **Page 37: [600] Formatted** | **Smith, Molly B** | **11/24/16 2:05:00 PM** |

Font:12 pt

| | | |
|---|---|---|
| **Page 37: [600] Formatted** | **Smith, Molly B** | **11/24/16 2:05:00 PM** |

Font:12 pt

| | | |
|---|---|---|
| **Page 37: [600] Formatted** | **Smith, Molly B** | **11/24/16 2:05:00 PM** |

Font:12 pt

| | | |
|---|---|---|
| **Page 37: [600] Formatted** | **Smith, Molly B** | **11/24/16 2:05:00 PM** |

Font:12 pt

| | | |
|---|---|---|
| **Page 37: [600] Formatted** | **Smith, Molly B** | **11/24/16 2:05:00 PM** |

Font:12 pt

| | | |
|---|---|---|
| **Page 37: [600] Formatted** | **Smith, Molly B** | **11/24/16 2:05:00 PM** |

Font:12 pt

| | | |
|---|---|---|
| **Page 37: [601] Formatted** | **Smith, Molly B** | **11/24/16 2:05:00 PM** |

Font:12 pt, Not Bold, Not Italic, Font color: Auto

| | | |
|---|---|---|
| **Page 37: [601] Formatted** | **Smith, Molly B** | **11/24/16 2:05:00 PM** |

Font:12 pt, Not Bold, Not Italic, Font color: Auto

| | | |
|---|---|---|
| **Page 37: [601] Formatted** | **Smith, Molly B** | **11/24/16 2:05:00 PM** |

Font:12 pt, Not Bold, Not Italic, Font color: Auto

| | | |
|---|---|---|
| **Page 37: [601] Formatted** | **Smith, Molly B** | **11/24/16 2:05:00 PM** |

Font:12 pt, Not Bold, Not Italic, Font color: Auto

| | | |
|---|---|---|
| **Page 37: [601] Formatted** | **Smith, Molly B** | **11/24/16 2:05:00 PM** |

Font:12 pt, Not Bold, Not Italic, Font color: Auto

| | | |
|---|---|---|
| **Page 37: [601] Formatted** | **Smith, Molly B** | **11/24/16 2:05:00 PM** |

Font:12 pt, Not Bold, Not Italic, Font color: Auto

| | | |
|---|---|---|
| **Page 37: [601] Formatted** | **Smith, Molly B** | **11/24/16 2:05:00 PM** |

Font:12 pt, Not Bold, Not Italic, Font color: Auto

| | | |
|---|---|---|
| **Page 37: [601] Formatted** | **Smith, Molly B** | **11/24/16 2:05:00 PM** |

Font:12 pt, Not Bold, Not Italic, Font color: Auto

| | | |
|---|---|---|
| **Page 37: [601] Formatted** | **Smith, Molly B** | **11/24/16 2:05:00 PM** |

Font:12 pt, Not Bold, Not Italic, Font color: Auto

| | | |
|---|---|---|
| **Page 37: [601] Formatted** | **Smith, Molly B** | **11/24/16 2:05:00 PM** |

Font:12 pt, Not Bold, Not Italic, Font color: Auto

| | | |
|---|---|---|
| **Page 37: [601] Formatted** | **Smith, Molly B** | **11/24/16 2:05:00 PM** |

Font:12 pt, Not Bold, Not Italic, Font color: Auto

| Page 37: [601] Formatted | Smith, Molly B | 11/24/16 2:05:00 PM |
|---|---|---|

Font:12 pt, Not Bold, Not Italic, Font color: Auto

| Page 37: [601] Formatted | Smith, Molly B | 11/24/16 2:05:00 PM |
|---|---|---|

Font:12 pt, Not Bold, Not Italic, Font color: Auto

| Page 37: [601] Formatted | Smith, Molly B | 11/24/16 2:05:00 PM |
|---|---|---|

Font:12 pt, Not Bold, Not Italic, Font color: Auto

| Page 37: [601] Formatted | Smith, Molly B | 11/24/16 2:05:00 PM |
|---|---|---|

Font:12 pt, Not Bold, Not Italic, Font color: Auto

| Page 37: [601] Formatted | Smith, Molly B | 11/24/16 2:05:00 PM |
|---|---|---|

Font:12 pt, Not Bold, Not Italic, Font color: Auto

| Page 37: [601] Formatted | Smith, Molly B | 11/24/16 2:05:00 PM |
|---|---|---|

Font:12 pt, Not Bold, Not Italic, Font color: Auto

| Page 37: [601] Formatted | Smith, Molly B | 11/24/16 2:05:00 PM |
|---|---|---|

Font:12 pt, Not Bold, Not Italic, Font color: Auto

| Page 37: [601] Formatted | Smith, Molly B | 11/24/16 2:05:00 PM |
|---|---|---|

Font:12 pt, Not Bold, Not Italic, Font color: Auto

| Page 37: [601] Formatted | Smith, Molly B | 11/24/16 2:05:00 PM |
|---|---|---|

Font:12 pt, Not Bold, Not Italic, Font color: Auto

| Page 37: [601] Formatted | Smith, Molly B | 11/24/16 2:05:00 PM |
|---|---|---|

Font:12 pt, Not Bold, Not Italic, Font color: Auto

| Page 37: [601] Formatted | Smith, Molly B | 11/24/16 2:05:00 PM |
|---|---|---|

Font:12 pt, Not Bold, Not Italic, Font color: Auto

| Page 37: [601] Formatted | Smith, Molly B | 11/24/16 2:05:00 PM |
|---|---|---|

Font:12 pt, Not Bold, Not Italic, Font color: Auto

| Page 37: [601] Formatted | Smith, Molly B | 11/24/16 2:05:00 PM |
|---|---|---|

Font:12 pt, Not Bold, Not Italic, Font color: Auto

| Page 37: [601] Formatted | Smith, Molly B | 11/24/16 2:05:00 PM |
|---|---|---|

Font:12 pt, Not Bold, Not Italic, Font color: Auto

| Page 37: [601] Formatted | Smith, Molly B | 11/24/16 2:05:00 PM |
|---|---|---|

Font:12 pt, Not Bold, Not Italic, Font color: Auto

| Page 37: [601] Formatted | Smith, Molly B | 11/24/16 2:05:00 PM |
|---|---|---|

Font:12 pt, Not Bold, Not Italic, Font color: Auto

| Page 37: [601] Formatted | Smith, Molly B | 11/24/16 2:05:00 PM |
|---|---|---|

Font:12 pt, Not Bold, Not Italic, Font color: Auto

| Page 37: [601] Formatted | Smith, Molly B | 11/24/16 2:05:00 PM |
|---|---|---|

Font:12 pt, Not Bold, Not Italic, Font color: Auto

| Page 37: [601] Formatted | Smith, Molly B | 11/24/16 2:05:00 PM |
|---|---|---|

Font:12 pt, Not Bold, Not Italic, Font color: Auto

| Page 37: [601] Formatted | Smith, Molly B | 11/24/16 2:05:00 PM |
|---|---|---|

Font:12 pt, Not Bold, Not Italic, Font color: Auto

| Page 37: [601] Formatted | Smith, Molly B | 11/24/16 2:05:00 PM |
|---|---|---|

Font:12 pt, Not Bold, Not Italic, Font color: Auto

| Page 37: [601] Formatted | Smith, Molly B | 11/24/16 2:05:00 PM |
|---|---|---|

Font:12 pt, Not Bold, Not Italic, Font color: Auto

| Page 37: [601] Formatted | Smith, Molly B | 11/24/16 2:05:00 PM |
|---|---|---|

Font:12 pt, Not Bold, Not Italic, Font color: Auto

| Page 37: [601] Formatted | Smith, Molly B | 11/24/16 2:05:00 PM |
|---|---|---|

Font:12 pt, Not Bold, Not Italic, Font color: Auto

| Page 37: [601] Formatted | Smith, Molly B | 11/24/16 2:05:00 PM |
|---|---|---|

Font:12 pt, Not Bold, Not Italic, Font color: Auto

| Page 37: [601] Formatted | Smith, Molly B | 11/24/16 2:05:00 PM |
|---|---|---|

Font:12 pt, Not Bold, Not Italic, Font color: Auto

| Page 37: [601] Formatted | Smith, Molly B | 11/24/16 2:05:00 PM |
|---|---|---|

Font:12 pt, Not Bold, Not Italic, Font color: Auto

| Page 37: [601] Formatted | Smith, Molly B | 11/24/16 2:05:00 PM |
|---|---|---|

Font:12 pt, Not Bold, Not Italic, Font color: Auto

| Page 37: [601] Formatted | Smith, Molly B | 11/24/16 2:05:00 PM |
|---|---|---|

Font:12 pt, Not Bold, Not Italic, Font color: Auto

| Page 37: [601] Formatted | Smith, Molly B | 11/24/16 2:05:00 PM |
|---|---|---|

Font:12 pt, Not Bold, Not Italic, Font color: Auto

| Page 37: [601] Formatted | Smith, Molly B | 11/24/16 2:05:00 PM |
|---|---|---|

Font:12 pt, Not Bold, Not Italic, Font color: Auto

| Page 37: [601] Formatted | Smith, Molly B | 11/24/16 2:05:00 PM |
|---|---|---|

Font:12 pt, Not Bold, Not Italic, Font color: Auto

| Page 37: [601] Formatted | Smith, Molly B | 11/24/16 2:05:00 PM |
|---|---|---|

Font:12 pt, Not Bold, Not Italic, Font color: Auto

| Page 37: [601] Formatted | Smith, Molly B | 11/24/16 2:05:00 PM |
|---|---|---|

Font:12 pt, Not Bold, Not Italic, Font color: Auto

| | | |
|---|---|---|
| **Page 37: [601] Formatted** | **Smith, Molly B** | **11/24/16 2:05:00 PM** |

Font:12 pt, Not Bold, Not Italic, Font color: Auto

| | | |
|---|---|---|
| **Page 37: [601] Formatted** | **Smith, Molly B** | **11/24/16 2:05:00 PM** |

Font:12 pt, Not Bold, Not Italic, Font color: Auto

| | | |
|---|---|---|
| **Page 37: [601] Formatted** | **Smith, Molly B** | **11/24/16 2:05:00 PM** |

Font:12 pt, Not Bold, Not Italic, Font color: Auto

| | | |
|---|---|---|
| **Page 37: [601] Formatted** | **Smith, Molly B** | **11/24/16 2:05:00 PM** |

Font:12 pt, Not Bold, Not Italic, Font color: Auto

| | | |
|---|---|---|
| **Page 37: [601] Formatted** | **Smith, Molly B** | **11/24/16 2:05:00 PM** |

Font:12 pt, Not Bold, Not Italic, Font color: Auto

| | | |
|---|---|---|
| **Page 37: [601] Formatted** | **Smith, Molly B** | **11/24/16 2:05:00 PM** |

Font:12 pt, Not Bold, Not Italic, Font color: Auto

| | | |
|---|---|---|
| **Page 37: [601] Formatted** | **Smith, Molly B** | **11/24/16 2:05:00 PM** |

Font:12 pt, Not Bold, Not Italic, Font color: Auto

| | | |
|---|---|---|
| **Page 37: [601] Formatted** | **Smith, Molly B** | **11/24/16 2:05:00 PM** |

Font:12 pt, Not Bold, Not Italic, Font color: Auto

| | | |
|---|---|---|
| **Page 37: [601] Formatted** | **Smith, Molly B** | **11/24/16 2:05:00 PM** |

Font:12 pt, Not Bold, Not Italic, Font color: Auto

| | | |
|---|---|---|
| **Page 37: [601] Formatted** | **Smith, Molly B** | **11/24/16 2:05:00 PM** |

Font:12 pt, Not Bold, Not Italic, Font color: Auto

| | | |
|---|---|---|
| **Page 37: [601] Formatted** | **Smith, Molly B** | **11/24/16 2:05:00 PM** |

Font:12 pt, Not Bold, Not Italic, Font color: Auto

| | | |
|---|---|---|
| **Page 37: [601] Formatted** | **Smith, Molly B** | **11/24/16 2:05:00 PM** |

Font:12 pt, Not Bold, Not Italic, Font color: Auto

| | | |
|---|---|---|
| **Page 37: [601] Formatted** | **Smith, Molly B** | **11/24/16 2:05:00 PM** |

Font:12 pt, Not Bold, Not Italic, Font color: Auto

| | | |
|---|---|---|
| **Page 37: [601] Formatted** | **Smith, Molly B** | **11/24/16 2:05:00 PM** |

Font:12 pt, Not Bold, Not Italic, Font color: Auto

| | | |
|---|---|---|
| **Page 37: [601] Formatted** | **Smith, Molly B** | **11/24/16 2:05:00 PM** |

Font:12 pt, Not Bold, Not Italic, Font color: Auto

| | | |
|---|---|---|
| **Page 37: [601] Formatted** | **Smith, Molly B** | **11/24/16 2:05:00 PM** |

Font:12 pt, Not Bold, Not Italic, Font color: Auto

| Page 37: [601] Formatted | Smith, Molly B | 11/24/16 2:05:00 PM |
|---|---|---|

Font:12 pt, Not Bold, Not Italic, Font color: Auto

| Page 37: [601] Formatted | Smith, Molly B | 11/24/16 2:05:00 PM |
|---|---|---|

Font:12 pt, Not Bold, Not Italic, Font color: Auto

| Page 37: [601] Formatted | Smith, Molly B | 11/24/16 2:05:00 PM |
|---|---|---|

Font:12 pt, Not Bold, Not Italic, Font color: Auto

| Page 37: [601] Formatted | Smith, Molly B | 11/24/16 2:05:00 PM |
|---|---|---|

Font:12 pt, Not Bold, Not Italic, Font color: Auto

| Page 37: [601] Formatted | Smith, Molly B | 11/24/16 2:05:00 PM |
|---|---|---|

Font:12 pt, Not Bold, Not Italic, Font color: Auto

| Page 37: [601] Formatted | Smith, Molly B | 11/24/16 2:05:00 PM |
|---|---|---|

Font:12 pt, Not Bold, Not Italic, Font color: Auto

| Page 37: [601] Formatted | Smith, Molly B | 11/24/16 2:05:00 PM |
|---|---|---|

Font:12 pt, Not Bold, Not Italic, Font color: Auto

| Page 37: [601] Formatted | Smith, Molly B | 11/24/16 2:05:00 PM |
|---|---|---|

Font:12 pt, Not Bold, Not Italic, Font color: Auto

| Page 37: [601] Formatted | Smith, Molly B | 11/24/16 2:05:00 PM |
|---|---|---|

Font:12 pt, Not Bold, Not Italic, Font color: Auto

| Page 37: [601] Formatted | Smith, Molly B | 11/24/16 2:05:00 PM |
|---|---|---|

Font:12 pt, Not Bold, Not Italic, Font color: Auto

| Page 37: [601] Formatted | Smith, Molly B | 11/24/16 2:05:00 PM |
|---|---|---|

Font:12 pt, Not Bold, Not Italic, Font color: Auto

| Page 37: [601] Formatted | Smith, Molly B | 11/24/16 2:05:00 PM |
|---|---|---|

Font:12 pt, Not Bold, Not Italic, Font color: Auto

| Page 37: [601] Formatted | Smith, Molly B | 11/24/16 2:05:00 PM |
|---|---|---|

Font:12 pt, Not Bold, Not Italic, Font color: Auto

| Page 37: [601] Formatted | Smith, Molly B | 11/24/16 2:05:00 PM |
|---|---|---|

Font:12 pt, Not Bold, Not Italic, Font color: Auto

| Page 37: [601] Formatted | Smith, Molly B | 11/24/16 2:05:00 PM |
|---|---|---|

Font:12 pt, Not Bold, Not Italic, Font color: Auto

| Page 37: [601] Formatted | Smith, Molly B | 11/24/16 2:05:00 PM |
|---|---|---|

Font:12 pt, Not Bold, Not Italic, Font color: Auto

| Page 37: [601] Formatted | Smith, Molly B | 11/24/16 2:05:00 PM |
|---|---|---|

Font:12 pt, Not Bold, Not Italic, Font color: Auto

| | | |
|---|---|---|
| **Page 37: [601] Formatted** | **Smith, Molly B** | **11/24/16 2:05:00 PM** |

Font:12 pt, Not Bold, Not Italic, Font color: Auto

| | | |
|---|---|---|
| **Page 37: [601] Formatted** | **Smith, Molly B** | **11/24/16 2:05:00 PM** |

Font:12 pt, Not Bold, Not Italic, Font color: Auto

| | | |
|---|---|---|
| **Page 37: [601] Formatted** | **Smith, Molly B** | **11/24/16 2:05:00 PM** |

Font:12 pt, Not Bold, Not Italic, Font color: Auto

| | | |
|---|---|---|
| **Page 37: [601] Formatted** | **Smith, Molly B** | **11/24/16 2:05:00 PM** |

Font:12 pt, Not Bold, Not Italic, Font color: Auto

| | | |
|---|---|---|
| **Page 37: [601] Formatted** | **Smith, Molly B** | **11/24/16 2:05:00 PM** |

Font:12 pt, Not Bold, Not Italic, Font color: Auto

| | | |
|---|---|---|
| **Page 37: [601] Formatted** | **Smith, Molly B** | **11/24/16 2:05:00 PM** |

Font:12 pt, Not Bold, Not Italic, Font color: Auto

| | | |
|---|---|---|
| **Page 37: [601] Formatted** | **Smith, Molly B** | **11/24/16 2:05:00 PM** |

Font:12 pt, Not Bold, Not Italic, Font color: Auto

| | | |
|---|---|---|
| **Page 37: [601] Formatted** | **Smith, Molly B** | **11/24/16 2:05:00 PM** |

Font:12 pt, Not Bold, Not Italic, Font color: Auto

| | | |
|---|---|---|
| **Page 37: [601] Formatted** | **Smith, Molly B** | **11/24/16 2:05:00 PM** |

Font:12 pt, Not Bold, Not Italic, Font color: Auto

| | | |
|---|---|---|
| **Page 37: [601] Formatted** | **Smith, Molly B** | **11/24/16 2:05:00 PM** |

Font:12 pt, Not Bold, Not Italic, Font color: Auto

| | | |
|---|---|---|
| **Page 37: [601] Formatted** | **Smith, Molly B** | **11/24/16 2:05:00 PM** |

Font:12 pt, Not Bold, Not Italic, Font color: Auto

| | | |
|---|---|---|
| **Page 37: [601] Formatted** | **Smith, Molly B** | **11/24/16 2:05:00 PM** |

Font:12 pt, Not Bold, Not Italic, Font color: Auto

| | | |
|---|---|---|
| **Page 37: [601] Formatted** | **Smith, Molly B** | **11/24/16 2:05:00 PM** |

Font:12 pt, Not Bold, Not Italic, Font color: Auto

| | | |
|---|---|---|
| **Page 37: [601] Formatted** | **Smith, Molly B** | **11/24/16 2:05:00 PM** |

Font:12 pt, Not Bold, Not Italic, Font color: Auto

| | | |
|---|---|---|
| **Page 37: [601] Formatted** | **Smith, Molly B** | **11/24/16 2:05:00 PM** |

Font:12 pt, Not Bold, Not Italic, Font color: Auto

| | | |
|---|---|---|
| **Page 37: [601] Formatted** | **Smith, Molly B** | **11/24/16 2:05:00 PM** |

Font:12 pt, Not Bold, Not Italic, Font color: Auto

| | | |
|---|---|---|
| **Page 37: [601] Formatted** | **Smith, Molly B** | **11/24/16 2:05:00 PM** |

Font:12 pt, Not Bold, Not Italic, Font color: Auto

| Page 37: [601] Formatted | Smith, Molly B | 11/24/16 2:05:00 PM |
|---|---|---|

Font:12 pt, Not Bold, Not Italic, Font color: Auto

| Page 37: [601] Formatted | Smith, Molly B | 11/24/16 2:05:00 PM |
|---|---|---|

Font:12 pt, Not Bold, Not Italic, Font color: Auto

| Page 37: [601] Formatted | Smith, Molly B | 11/24/16 2:05:00 PM |
|---|---|---|

Font:12 pt, Not Bold, Not Italic, Font color: Auto

| Page 37: [601] Formatted | Smith, Molly B | 11/24/16 2:05:00 PM |
|---|---|---|

Font:12 pt, Not Bold, Not Italic, Font color: Auto

| Page 37: [601] Formatted | Smith, Molly B | 11/24/16 2:05:00 PM |
|---|---|---|

Font:12 pt, Not Bold, Not Italic, Font color: Auto

| Page 37: [602] Deleted | Cornell University | 11/15/16 2:49:00 PM |
|---|---|---|

**Figure 1: Climatological monthly average time series of dust concentration divided by the mean of each time series (unitless) at stations a) Banizoumbou, b) Barbados, c) Bermuda, d) Cinzana, e) Izana, f) Macehead, g) Mbour, h) Miami, i) Midway.  Observations are in black and are from the locations and citations in Table 2.  Different colors and line styles indicate the different model versions: CAM4 (MERRA) (blue solid), CAM4 (NCEP) (green solid), CAM4 (ERAI) (pink solid), CAM4 (AMIP) (orange solid), GCHEM (MERRA) (blue dashed), MATCH (NCEP) (green dashed), CAM5 (AMIP) (orange dashed).  The monthly means are divided by the long term annual mean to allow comparison with interannual variability, since they are similarly normalized (Figure 3).**

[Figure]

**Figure 2: Climatological monthly average aerosol optical depth (AOD) for model simulations (based on dust only), compared with AERONET observations   for: a) Bahrain b) Dahkla c) Dalanzadgad, d) Dhabi, e) Ilorin and f) Sede Boker for each of the different model versions (Colors and information are the same as in Figure 1, but for the AOD). Observational data from AERONET stations (citations and locations listed in Table 2). The monthly means are divided by the long term annual mean to allow comparison with interannual variability, since they are similarly normalized (Figure 4).**

| Page 38: [603] Deleted | Cornell University | 11/15/16 2:49:00 PM |
|---|---|---|

| Page 38: [604] Deleted | Smith, Molly B | 11/28/16 6:24:00 PM |
|---|---|---|

Dahkla c)

| Page 38: [605] Deleted | Smith, Molly B | 11/28/16 6:25:00 PM |
|---|---|---|

Dhabi, e)

| Page 38: [606] Deleted | Cornell University | 11/15/16 2:50:00 PM |
|---|---|---|

).

[Figure]

| Page 38: [607] Deleted | Cornell University | 11/15/16 2:51:00 PM |
| --- | --- | --- |

monthly

| Page 38: [607] Deleted | Cornell University | 11/15/16 2:51:00 PM |
| --- | --- | --- |

monthly

| Page 38: [607] Deleted | Cornell University | 11/15/16 2:51:00 PM |
| --- | --- | --- |

monthly

| Page 38: [607] Deleted | Cornell University | 11/15/16 2:51:00 PM |
| --- | --- | --- |

monthly

| Page 38: [608] Deleted | Smith, Molly B | 11/24/16 1:03:00 PM |
| --- | --- | --- |

the

| Page 38: [608] Deleted | Smith, Molly B | 11/24/16 1:03:00 PM |
|---|---|---|

the

| Page 38: [609] Deleted | Cornell University | 11/15/16 2:51:00 PM |
|---|---|---|

monthly

| Page 38: [609] Deleted | Cornell University | 11/15/16 2:51:00 PM |
|---|---|---|

monthly

| Page 38: [610] Deleted | Smith, Molly B | 11/28/16 6:27:00 PM |
|---|---|---|

**Dah: Dahkla;**

| Page 38: [610] Deleted | Smith, Molly B | 11/28/16 6:27:00 PM |
|---|---|---|

**Dah: Dahkla;**

| Page 38: [611] Deleted | Cornell University | 11/15/16 2:52:00 PM |
|---|---|---|

[Figure]

[Figure]

| Page 38: [612] Deleted | Cornell University | 11/15/16 2:52:00 PM |
|---|---|---|

a. Corr. Surf. Conc. CAM4 models

b. Corr. Surf. Conc. same met. models

c. Corr. Deposition CAM4 models

d. Corr. Deposition same met. models

e. Corr. AOD CAM4 models

f. Corr. AOD same met. models

0.1  0.2  0.3  0.4  0.5  0.6  0.7  0.8  0.9

Page 39: [614] Deleted                    Smith, Molly B                    11/24/16 2:18:00 PM

Page 40: [615] Deleted                    Smith, Molly B                    12/1/16 2:27:00 PM

[revised manuscript text omitted]